

# Annotated checklist of the beetles (Coleoptera) of the California Channel Islands

Matthew L. Gimmel[1], M. Andrew Johnston[2] and Michael S. Caterino[3]

[1] Department of Invertebrate Zoology, Santa Barbara Museum of Natural History, Santa Barbara, California, United States
[2] Biodiversity Knowledge Integration Center, Arizona State University, Tempe, Arizona, United States
[3] Department of Plant & Environmental Sciences, Clemson University, Clemson, South Carolina, United States

## ABSTRACT

The beetle fauna of the California Channel Islands is here enumerated for the first time in over 120 years. We provide an annotated checklist documenting species-by-island diversity from an exhaustive literature review and analysis of a compiled dataset of 26,609 digitized specimen records to which were added over 3,000 individual specimen determinations. We report 825 unique species from 514 genera and 71 families (including 17 new family records) comprising 1,829 species-by-island records. Species totals for each island are as follows: Anacapa (74); San Clemente (197); San Miguel (138); San Nicolas (146); Santa Barbara (64); Santa Catalina (370); Santa Cruz (503); and Santa Rosa (337). This represents the largest list of species published to date for any taxonomic group of animals on the Channel Islands; despite this, we consider the checklist to be preliminary. We present evidence that both inventory and taxonomic efforts on Channel Islands beetles are far from complete. Rarefaction estimates indicate there are at least several hundred more species of beetles yet to be recorded from the islands. Despite the incomplete nature of existing records, we found that species diversity is highly correlated with island area.

We report 56 species which are putatively geographically restricted (endemic) to the Channel Islands, with two additional species of questionable endemic status. We also report 52 species from the islands which do not natively occur in the southern California region.

# INTRODUCTION

The California Channel Islands are an archipelago of eight main islands between 20 and 98 km off the coast of southern California, USA. Often referred to as "North America's Galápagos", the biological diversity of the California Channel Islands has long captured the attention of natural historians of western North America. Detailed information about the islands' geography, geologic history, natural history, and history of human activity may be found in many other publications, especially *Schoenherr, Feldmeth & Emerson (1999)* and

Corresponding authors
Matthew L. Gimmel,
phalacrid@gmail.com
M. Andrew Johnston,
ajohnston@asu.edu

*Moody (2000)*; *Miller (1985a)* provided an introduction to the history of entomology of the islands.

Attempts to document species diversity of the islands have been scattered and unequal across taxonomic groups, with most organismal groups not having a reliable checklist or taxonomic treatment completed and made available. The largest published list is that of *Ratay, Vanderplank & Wilder (2014)* which cited 976 vascular plant taxa from the California Channel Islands, including species (922), subspecies, varieties, and forms, of which 278 taxa are nonnative; *Carter (2015)* provided a list of 157 bryophyte species from the islands. Within animals, vertebrates are best documented: Mammalia—34 species, including 14 native and 20 nonnative (*von Bloeker, 1965*); Amphibia—eight species (*Nafis, 2022*); Reptilia—16 species and subspecies (*Nafis, 2022*); Aves—422 species (*Collins & Jones, 2015*). Several non-beetle insect groups have published inventories for all California Channel Islands: Dermaptera—two species, both nonnative (*Miller, 1984*); Orthoptera—54 species (*Weissman & Rentz, 1976*; *Weissman, 1985*), list/guide published in *Rentz & Weissman (1982)*; Hemiptera: Sternorrhyncha: Pseudococcidae—43 species (*Rust, Menke & Miller, 1985*), list published in same; Lepidoptera—purportedly >800 species (*Powell, 1994*, *2005*), but no species list published; Hymenoptera: Apoidea—243 species (*Rust, Menke & Miller, 1985*), list published in same.

Prior efforts at inventorying the beetles of the California Channel Islands have been significant yet have not resulted in a published list of species for over a century. *Fall (1897)* made the first attempt at a comprehensive list of the species of Coleoptera of the southern California islands, including the Channel Islands and Guadalupe Island, Mexico. He recorded 212 species from the Channel Islands. *Fall & Davis (1934)* recorded 51 species of Coleoptera from Santa Cruz Island. *Darlington (1943*: 54–55) made general observations about the carabid beetle fauna of the "Santa Barbara Islands" but presented no new taxon or island records. *Miller (1985a)* recorded 36 endemic beetle species from the Channel Islands (not including the 10 endemic species of *Trigonoscuta*), plus two additional endemic subspecies. *Miller & Miller (1985)* reported 52 species of Coleoptera from Santa Barbara Island. Elbert L. Sleeper purportedly had an unpublished beetle list for Santa Catalina Island, with 330 species (see *Caterino & Chandler, 2010*). Scott E. Miller had an unpublished checklist of 225 beetle species prior to the efforts of the California Beetle Project (*Caterino, Chatzimanolis & Richmond, 2015*: 279). *Caterino, Chatzimanolis & Richmond (2015)* reported on an unpublished list of "over 640 named species of beetles" from the Channel Islands.

We here report, for the first time since 1897, a comprehensive list of Coleoptera species from the Channel Islands, annotated with full supporting citations, specimen records, nomenclatural authority, and notes.

## MATERIALS AND METHODS

To generate a comprehensive list of the Coleoptera known from the Channel Islands, we utilized two primary data streams: published literature and digitized specimen records. We also physically examined specimens from several institutions to complement and refine the specimen-level data delivered by them. Our methods generally follow those

suggested by *Johnston, Aalbu & Franz (2018)* to create a thoroughly traceable checklist to encourage a verified, reproducible, and readily updatable product. Specific methods for each data type are discussed below with an assessment of their strengths and limitations.

## Harvesting published literature records

The taxonomic and faunistic literature for Coleoptera of North America is incredibly expansive and intractable for a single research team to fully scour. Nevertheless, a rigorous review of the literature was attempted to identify Channel Islands distribution records. The comprehensive bibliographies for California Island entomology by *Miller & Menke (1981)*, *Miller (1985b)*, and *Miller (1993)* were used as a baseline. We then examined literature sources we were familiar with, particularly those published after the last (1993) supplement, and finally broadened the literature search by searching for keywords of "Channel Islands" and "Coleoptera" as well as by examining modern works that treated species and genera of beetles known to occur in California. Gray literature was generally not consulted for additional species or island records, although specimens resulting from such works made available as voucher specimens in institutional collections were frequently encountered. The result, we believe, is a thorough baseline literature review across all relevant publications. Every publication was vetted by us and a full citation with page number was generated for every island record of a taxon the publication presented.

Literature records, especially those from authoritative taxonomic revisions, can provide some of the best information available for Channel Island Coleoptera. In particular, revisionary studies are often based upon borrowed material from many institutions and likely report on specimens that have not yet been digitized and made available by the owning institutions. Conversely, historical literature records may often be doubtful as taxonomic names and concepts have shifted through time. Additionally, some publications do not cite particular specimens so a proper vetting of the island records may never be fully possible.

Future revisionary works and taxon-specific studies may overturn some of the records reported in the literature. However, we have here reported all such taxa and literature citations in order to make them transparent for future researchers. In the event of a publication explicitly discounting earlier published records or ascribing them to new taxa, the original citations are included in the notes under each taxon, while the island records presented have been adjusted according to more recent authorities. Notes under each taxon detail any perceived ambiguities, irregularities, or importance for each literature citation.

## Harvesting digitized specimen records

Natural history collections house the primary distributional data for insects. Each collection has idiosyncratic strengths which are often a result of the activities of its workers through time. For the Channel Islands, material is scattered throughout the world's collections and even collections with limited holdings from the region may contain valuable species records that correspond to taxonomic expertise of its staff. Visiting all collections and examining every island beetle within them is impractical and inefficient for
building a checklist. Therefore, we have focused on publicly available digitized specimen records for Channel Islands Coleoptera.

*Data sources*. Our dataset is built upon three primary groups of specimen records: (1) the beetle holdings of the Natural History Museum of Los Angeles County (LACM), which contains vast amounts of historical island survey material; (2) the beetle holdings of the Santa Barbara Museum of Natural History (SBMNH), which has a focus on both Coleoptera and the Channel Islands; and (3) digitized records available from the Symbiota Collections of Arthropods Network (SCAN, https://scan-bugs.org). The holdings from LACM and SBMNH are fully (or very nearly) digitized to the specimen level. The records from SCAN were compiled by performing two searches of the portal (in November 2020): (1) taxon "Coleoptera" within a polygon drawn around all eight Channel Islands, and (2) taxon "Coleoptera" with state "California" and "island" contained in the locality. Specimens from the following institutional or personal collections appear in our checklist:

| | |
|---|---|
| **ASUHIC** | Arizona State University Hasbrouck Insect Collection, Tempe, AZ, USA |
| **AUMNH** | Auburn University Museum of Natural History, Auburn, AL, USA |
| **BYUC** | Brigham Young University Arthropod Collection, Provo, UT, USA |
| **CASC** | California Academy of Sciences, San Francisco, CA, USA |
| **CSCA** | California State Collection of Arthropods, Sacramento, CA, USA |
| **CSUC** | Colorado State University Insect Collection, Fort Collins, CO, USA |
| **DMNS** | Denver Museum of Nature and Science, Denver, CO, USA |
| **EMEC** | Essig Museum, University of California Berkeley, Berkeley, CA, USA |
| **JNRC** | Jacques N. Rifkind Collection, Sacramento, CA, USA |
| **LACM** | Natural History Museum of Los Angeles County, Los Angeles, CA, USA |
| **MAJC** | M. Andrew Johnston Research Collection, Tempe, AZ, USA |
| **SBMNH** | Santa Barbara Museum of Natural History, Santa Barbara, CA, USA |
| **SDNHM** | San Diego Natural History Museum, San Diego, CA, USA |
| **SEMC** | Snow Entomological Museum, University of Kansas, Lawrence, KS, USA |
| **TAMU** | Texas A&M University Insect Collection, College Station, TX, USA |
| **UCMC** | University of Colorado Museum of Natural History, Boulder, CO, USA |
| **UCRC** | University of California Riverside Insect Collection, Riverside, CA, USA |
| **UCSB** | University of California Santa Barbara, Santa Barbara, CA, USA |
| **USNM** | United States National Museum of Natural History (Smithsonian Institution), Washington, DC, USA |
| **UTCI** | University of Tennessee Chattanooga, Chattanooga, TN, USA |
| **UASM** | University of Alberta Strickland Entomology Museum, Edmonton, AB, Canada |
| **YPMC** | Yale Peabody Museum of Natural History, New Haven, CT, USA |
| **iNat** | iNaturalist Research Grade Observations (https://inaturalist.org) |

*Specimen determinations*. Specimen records came with determination information from the original data providers. In addition, 3,309 taxonomic redeterminations/annotations were made by us. Most of these (2,352) were performed on specimens examined in person

at SBMNH, LACM, and the UCSB collections. Additional nomenclatural adjustments (957) were made for records that had obvious misspellings or old combinations.

*Data cleaning.* All data were imported into the Symbiota portal Ecdysis (https://serv.biokic.asu.edu/ecdysis/) utilizing best practices according to the Darwin Core data standard and FAIR (Findable, Accessible, Interoperable, Reusable) data principles. All records were examined and georeferenced (where not previously done) and added to island-specific datasets. California "island" records not located in the Channel Islands (*e.g.*, "Farallon Islands") were excluded. Throughout this process, the owner institution and metadata were preserved with each record. Records that were deemed untrustworthy, typically due to a mismatch in locality data and provided GPS coordinates or records lacking any data, were pruned from the dataset. The final set of records numbered 26,609; these are fully available in their final, cleaned form (*Johnston & Gimmel, 2022*).

*Excluded specimen data.* Digitized data for insects is not yet as complete or mature as for other groups of organisms (*e.g.*, vertebrates and plants). Many collections have no specimens digitized and most are only partially digitized. This issue is compounded by the fact that not all museums share their data publicly or do not frequently refresh their data to online aggregators. In addition, online taxonomic resources are woefully incomplete for insects, particularly beetles, so many publicly available records are not appropriately indexed to family or order level. With each of these hurdles, otherwise valuable records are in effect made unavailable to research projects like this one. Other websites (*e.g.*, California Beetle Project) occasionally provided additional species or specimen information, but these were ignored as unverifiable since they do not have a unique identifier to relocate the presumed specimen(s) anchoring the record. We hope to see increased focus on the Channel Islands by coleopterists in the future where taxonomic experts can continue to add to and refine the knowledge aggregated and summarized in this work.

## Checklist validation

Each taxon listed from the islands was critically examined as part of our literature review. In addition, we examined any records of taxa not known from southern California and, where possible, confirmed the identification of the physical specimen. The most modern and reliable treatments for all taxa were used for determining taxon validity, and were cited in full.

Non-unique order, family, or genus records in the literature were generally ignored for purposes of this checklist. For example, if a record was identified only to the genus level where a species from that genus was already known from the Channel Islands, that was not considered a new taxon for the tally. In cases where only genus-level or higher records are known for a given taxon, then that taxon was included in the species count as an undetermined species of the genus (or "undetermined genus and species" for a family in one case). All digitized and literature records are included for the genus level; many of these represent unique island records for that group. Subspecies were not counted as separate taxa in our checklist; instead, subspecies are discussed under each species where relevant.

Numbers quoted in the family accounts for California beetle diversity are mostly derived from an unpublished checklist of author MLG with other sources being cited when

used. We do not include general biological information except in special cases and except as it relates specifically to island-collected or island-observed specimens; such biological information can be found in more general guides and references.

We cite the most relevant work where we derived our taxon name, combination, and authority from as the nomenclatural authority. This is often the most recent catalog, revision, or book chapter known to us. Many of the groups represented across the California Channel Islands are in desperate need of revision and have a long and complicated taxonomic history. We anticipate that taxonomic experts will come to conclusions different than those of historical workers who have published on the Channel Islands and identified material in collections. It is with this in mind that we strive to explicitly document all name usages such that they can be tracked, validated, and updated by future generations of coleopterists (see also *Johnston, Aalbu & Franz, 2018*).

## Biogeographic and diversity analyses

Our final species list and dataset of digitized specimen records were analyzed to explore trends and correlations using R (*R Core Team, 2022*). Species diversity from each island was plotted against geographic and rainfall data for each island (taken from *Miller, 1985a*) using linear models in the R ggplot2 software package (*Wickham, 2016*). The final digitized record dataset (*i.e.*, not counting literature records but including identifications to higher taxon ranks as a single unique taxon) was used to generate specimen totals for each unique taxon on each island and totals for each taxon pooled across all of the islands. Totals for each species by island were further pooled into collecting events where all records with the same collector (recordedBy field) and collection date (eventDate field) were considered to belong to the same collecting event to examine possible limitations of the dataset. Total specimen counts were analyzed in a rarefaction and extrapolation species diversity analysis using the R iNEXT software package (*Chao et al., 2014*) to generate both rarefaction curves and an estimate of actual species diversity given the observed data. An annotated R script with raw data is available *via* Zenodo (*Johnston, 2022*).

## RESULTS AND DISCUSSION

We here provide an annotated checklist of 825 unique species (= taxa) comprising 1,829 species-by-island records (see checklist below). Individual island species counts and geographic data are given in Table 1. Curated digitized specimen records are archived and available on Zenodo (*Johnston & Gimmel, 2022*).

We report 56 species putatively restricted to the Channel Islands along with two more that are questionably so. This represents a 22% increase in the number of known endemic species since *Miller (1985a)*, which is mostly accounted for by recognition of newly described and still-undescribed endemic species. An additional 52 species in the fauna are not native to the southern California region.

Beetle diversity on individual islands was plotted against island and dataset characteristics to elucidate potential driving factors and biases in our results. Beetle diversity is extremely strongly correlated with island land area with a linear relationship (Fig. 1A) but not with island distance to mainland (Fig. 1B). Interestingly, species richness

**Table 1  California Channel Island statistics.**

| Islands | Species | DigRecords | Area (km$^2$) | Distance (km) |
|---|---|---|---|---|
| Anacapa | 74 | 814 | 2.9 | 61 |
| San Clemente | 197 | 5,092 | 145 | 79 |
| San Miguel | 138 | 2,795 | 37 | 42 |
| San Nicolas | 146 | 2,946 | 58 | 98 |
| Santa Barbara | 64 | 751 | 2.6 | 61 |
| Santa Catalina | 370 | 3,068 | 194 | 32 |
| Santa Cruz | 503 | 7,341 | 249 | 30 |
| Santa Rosa | 337 | 3,802 | 217 | 44 |

Note:
Species and DigRecords (digitized records) are counts from the checklist and dataset reported in this study. Area and distance from mainland for each island are taken from *Miller (1985a)*.

was linearly, and not logarithmically, correlated with island area, which is counter to Darlington's rule hypothesizing a doubling of species for each ten-fold increase in island area (*Darlington, 1957*). Our finding is similar to that of *Moody (2000)* for native vascular plants on the Channel Islands; however, *Powell (1994)* found only a weak species-area relationship in Channel Islands Lepidoptera. While observed species richness could be the result of bias in sampling effort, the number of species on an island did not strongly correlate with the number of digitized records (Fig. 2A). The number of records showed a similar relationship with the size of the island (Fig. 2B), perhaps indicating a somewhat even sampling per island area.

The distribution of the number of islands a single species inhabits (Fig. 3A) was strongly left-skewed with over 50% being recorded from just a single island. However, numbers of specimens and collecting events per species across all islands were also highly left-skewed (Figs. 3B and 3C); in fact, 154 species (19% of the fauna) in our list are represented by a single digitized *specimen* (Fig. 3B) while 248 species (30% of the fauna) are represented by a single digitized collecting event (*i.e.*, a series of specimens which had identical values for collector and date) (Fig. 3C). The observed distribution patterns may therefore not be a true reflection of biological diversity but are likely subject to bias from insufficient sampling.

Species richness on each island is likely still far from fully documented (Table 2; Fig. 4). The entire island beetle fauna seems to have at least several hundred species still undocumented (Fig. 4C). Interestingly, the smallest islands, Anacapa (Fig. 4A) and Santa Barbara (Fig. 4B), appear to be the least completely inventoried (62.1% and 63.1% estimated complete, respectively; Table 2). Santa Cruz, at 87.7% estimated complete, appears to be the most completely inventoried island (Fig. 4A; Table 2).

The ten most species-rich families on the islands are: Staphylinidae (105 species), Carabidae (87 species), Curculionidae (65 species), Tenebrionidae (61 species), Coccinellidae (43 species), Scarabaeidae (41 species), Chrysomelidae (38 species), Cerambycidae (34 species), Hydrophilidae (32 species), and Melyridae (27 species). The three most species-rich genera on the islands are, with 11 species each: *Eleodes* (Tenebrionidae), *Scymnus* (Coccinellidae), and *Trigonoscuta* (Curculionidae); *Bembidion*

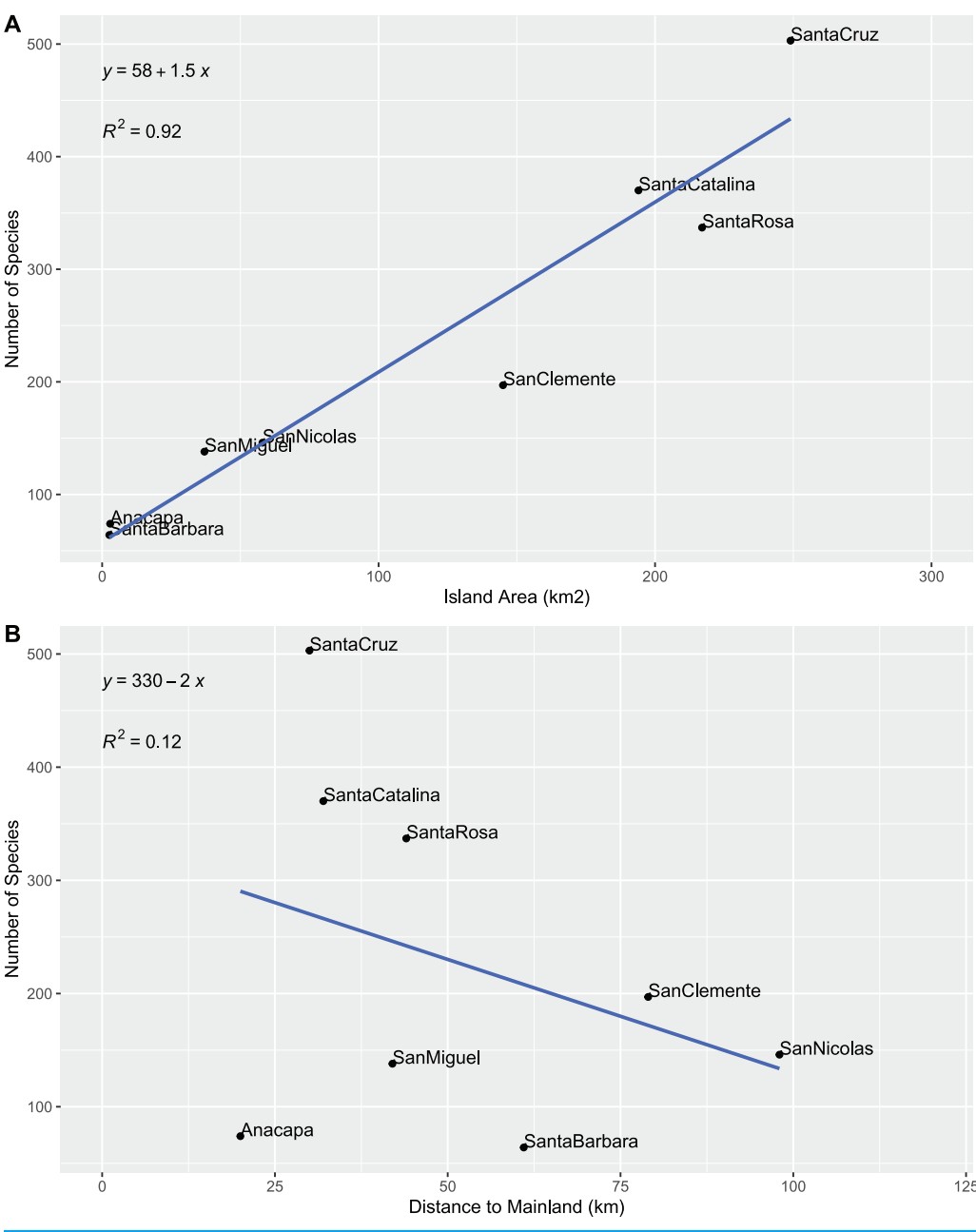

**Figure 1 Species by island size and position.** (A) Plotted against total area (km²). (B) Plotted against distance to mainland (km).               

(Carabidae) follows closely behind with 10 species. Table 3 lists all species reported in the annotated checklist below and summarizes their status (native *vs* adventive) and known island-level distribution.

## Notable taxa

In this section we highlight some island-specific findings notable for their taxonomic or biogeographic implications, and highlight taxa in need of further investigation. For specific island records and other details, please refer to the taxon entries in the main checklist.

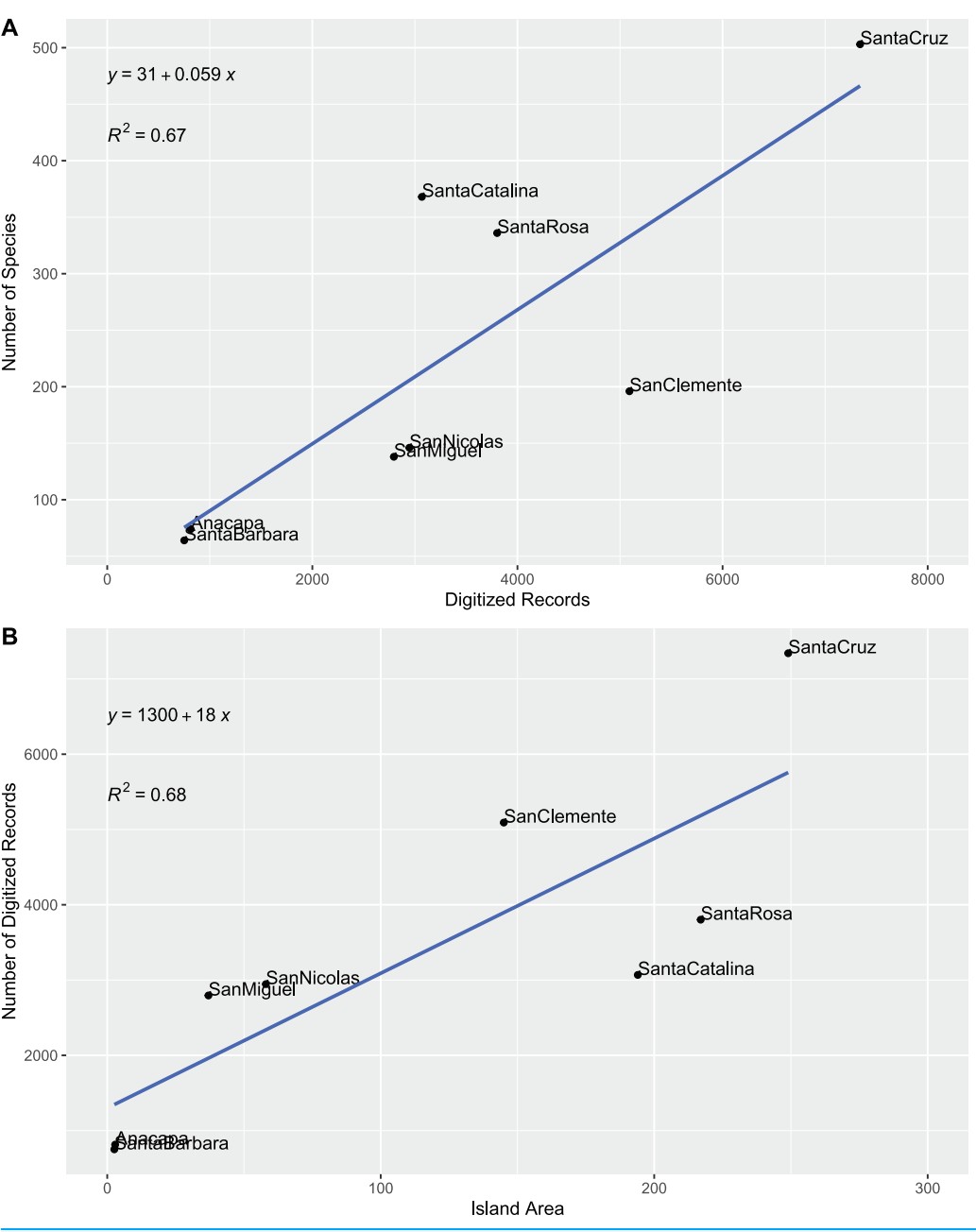

**Figure 2 Correlation of digitized specimen records to number of species and island size.** (A) Species by digitized records. (B) Digitized records by island area.

Among Carabidae, the subgenus *Pterostichus* (*Hypherpes*), a California-centric subgenus, is in serious need of investigation; currently there are six "known" species from the islands, including one putative endemic, but their taxonomy needs revision. *Amara insularis* is currently considered endemic and occurring on all islands, but based on investigation of hundreds of specimens housed in SBMNH (M. L. Gimmel, 2021, personal observation) this species is questionably distinct from mainland (and island) *A. insignis*.
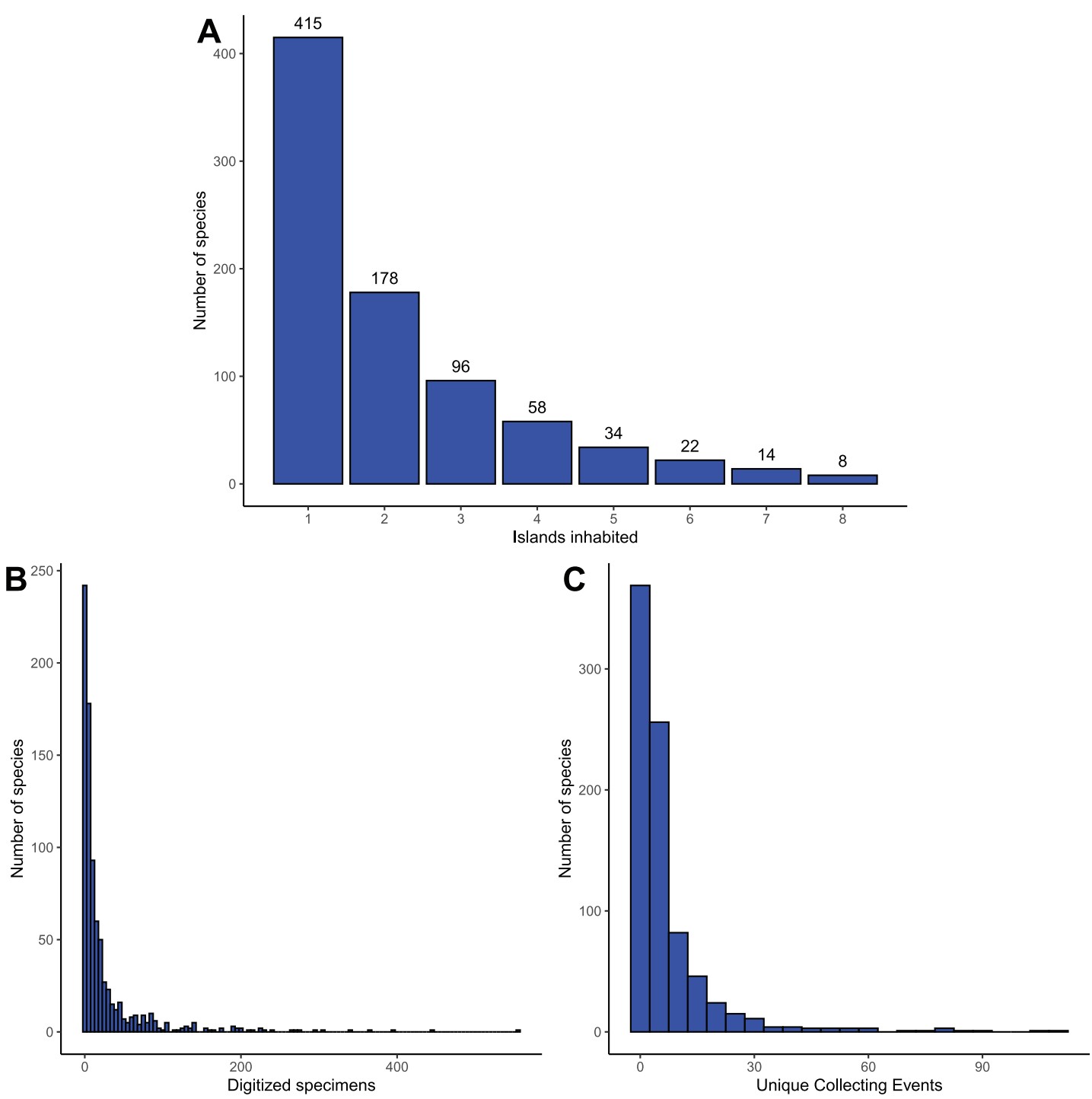

**Figure 3 Species distribution and collection frequency.** (A) Number of islands each species inhabits. (B) Raw number of digitized records per species. (C) Unique collecting events per species.

Among non-endemic taxa, a few are notable for having flightless females, yet have clearly dispersed from source populations on the mainland: *Anorus piceus* (Dascillidae), *Pterotus obscuripennis* (Lampyridae), and *Zarhipis integripennis* (Phengodidae).

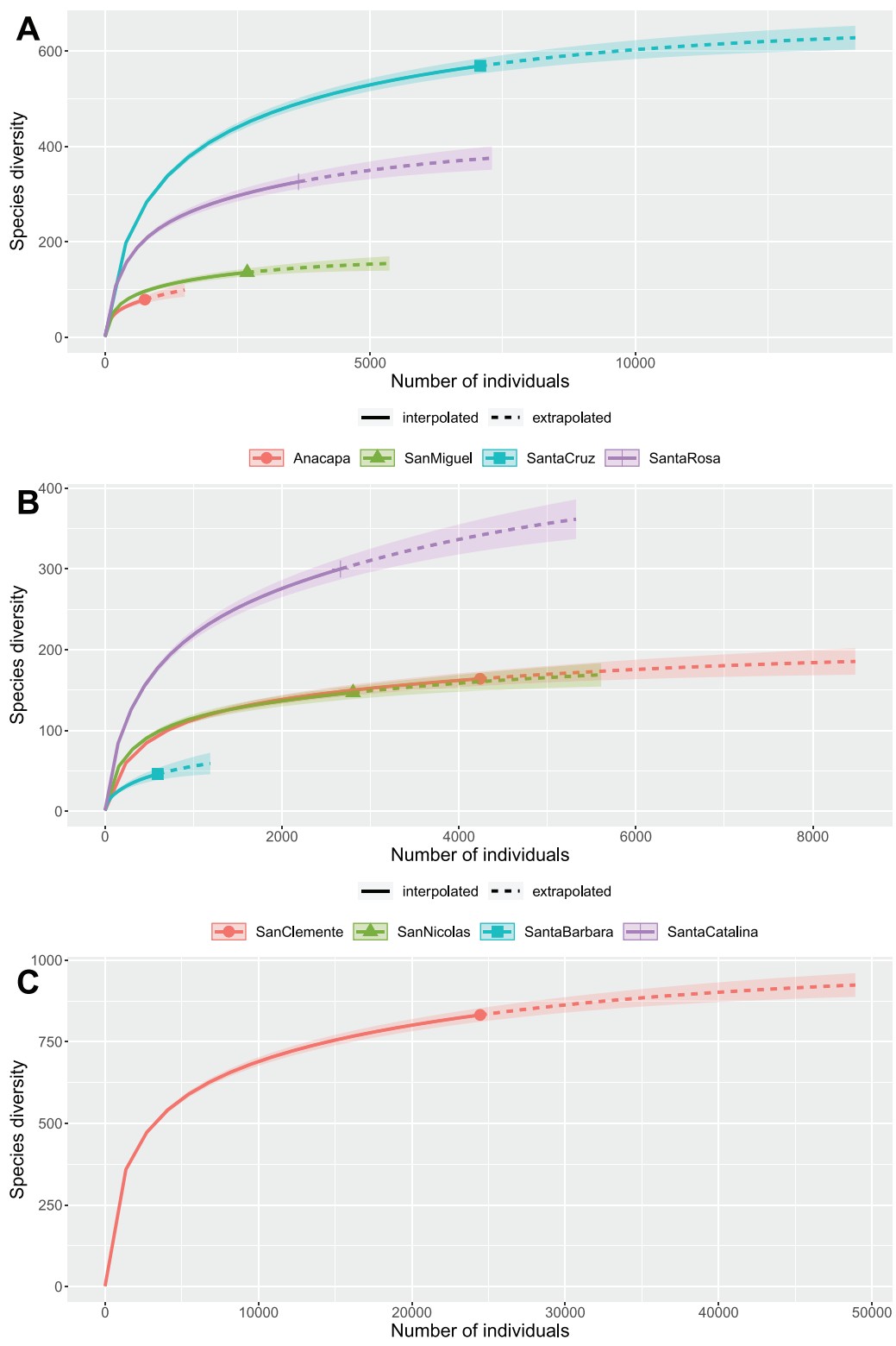

**Figure 4 Species diversity rarefaction and estimation curves.** (A) North islands. (B) South islands. (C) All islands combined.

**Table 2 Species richness estimates using rarefaction and estimation.**

| Islands | Observed taxa | Estimated taxa | Estimated % complete | s.e. | 95% LCL | 95% UCL |
|---|---|---|---|---|---|---|
| San Clemente | 164 | 197.992 | 82.8% | 15.424 | 178.557 | 243.374 |
| San Nicolas | 147 | 186.371 | 78.9% | 18.771 | 163.215 | 242.59 |
| Santa Barbara | 46 | 72.955 | 63.1% | 17.593 | 54.386 | 132.636 |
| Santa Catalina | 300 | 418.221 | 71.7% | 34.606 | 367.396 | 507.375 |
| Anacapa | 79 | 127.221 | 62.1% | 27.169 | 96.231 | 213.95 |
| San Miguel | 136 | 164.254 | 82.8% | 13.35 | 147.721 | 204.106 |
| Santa Cruz | 569 | 648.879 | 87.7% | 20.09 | 618.162 | 698.79 |
| Santa Rosa | 326 | 406.031 | 80.3% | 24.022 | 371.004 | 468.318 |
| All combined | 832 | 966.744 | 86.1% | 28.507 | 921.412 | 1,035.061 |

**Note:**
Observed and estimated taxa are from rarefaction analyses based upon digitized records (excludes taxa only known from literature records). Estimated percent complete is the proportion of the estimated taxa already observed. s.e. = standard error; LCL and UCL are lower and upper 95% confidence levels, respectively.

Apparently undescribed, and possibly endemic, island species exist in the genera *Dalopius* (Elateridae), *Mordellistena* (Mordellidae), *Carinodulinka* (Coccinellidae), *Fuchsina* (Latridiidae), *Dacne* (Erotylidae), *Longitarsus* (Chrysomelidae; at least two species), *Anthonomus* (Curculionidae), *Gilbertiola* (Curculionidae) (all M. L. Gimmel, 2021, personal observation), *Phobetus* (Scarabaeidae; at least two species; A. Evans, 2021, personal communication), and *Carphobius* (A. Cognato & S. Smith, 2022, personal communication), as well as the subfamily Leptotyphlinae (Staphylinidae). Interestingly, the *Fuchsina*, *Gilbertiola*, and Leptotyphlinae are eyeless and flightless (as is *Pinodytes gibbosus*; Leiodidae), while the two *Longitarsus* taxa are both flightless with abbreviated elytra; one undescribed *Mordellistena* has vestigial hind wings.

Taxa needing taxonomic investigation that will almost certainly reveal additional species, possibly including endemics, are as follows: *Bacanius* species (Histeridae), *Plegaderus* species (Histeridae), *Anthaxia* species (Buprestidae), *Hyperaspis* species (Coccinellidae), and *Dienerella* species (Latridiidae).

At least 63 known, unique taxa (not including known, undescribed species) still lack species determinations, predominantly in groups that lack modern taxonomic treatments, including all or most genera in the Scirtidae, Ptiliidae, Staphylinidae (Aleocharinae, Paederinae, Scydmaeninae), Salpingidae, Latridiidae, and Brentidae (Apioninae).

Island endemic taxa are distributed across many families of Coleoptera; the family with the highest proportion of endemics was found to be Melyridae, with 13 out of 28 species, or 46% of the island fauna. This family is the current research focus for MLG, and several of these island endemics are undescribed and will be receiving treatment in the near future; based on much recent fieldwork and museum work, the island endemism in this family is believed to be genuine rather than artefactual. Other families with significant proportions of endemics, either real or artefactual, include Scarabaeidae (eight out of 41 species, or 20%), Curculionidae (13 out of 65 species, or 20%), Cleridae (two out of 10 species, or 20%), Tenebrionidae (11 out of 61 species, or 18%), Latridiidae (three out of 19 species, or 16%), Zopheridae (one out of eight species, or 13%), and Ptinidae (two out of 25 species, or 8%). Notable genera with endemics include *Coenonycha* (Scarabaeidae), in which all four
**Table 3 List of California Channel Islands beetles.**

| Scientific name | Duplicate genus record | Endemic | Adventive | Anacapa | San Clemente | San Miguel | San Nicolas | Santa Barbara | Santa Catalina | Santa Cruz | Santa Rosa |
|---|---|---|---|---|---|---|---|---|---|---|---|
| **ADEPHAGA** | | | | | | | | | | | |
| **Carabidae** | | | | | | | | | | | |
| *Agonum decorum* | | | | | | | | | | D,L | |
| *Agonum limbatum* | | | | | | D | | | D,L | D | D |
| *Agonum piceolum* | | | | | | | | | | D | |
| *Agonum punctiforme* | | | | | D | D | D | | | | |
| *Akephorus marinus* | | | | | | D | D | | | D | D,L |
| *Amara aurata* | | | | | D,L | | | | | D | |
| *Amara californica* | | | | D | D,L | D | D | | | D,L | D,L |
| *Amara conflata* | | | | | | | | | | D | |
| *Amara insignis* | | | | | | | | | D,L | D | L |
| *Amara insularis* | | end | | D | D,L | D | D,L | D,L | D | D | D,L |
| *Amara pomona* | | | | | | | | | | L | L |
| *Amara scitula* | | | | | | D | | | | | |
| *Anchomenus funebris* | | | | | L | | | | L | D,L | |
| *Anisodactylus californicus* | | | | | D | D,L | D | | D,L | D,L | D,L |
| *Anisodactylus consobrinus* | | | | | | | | | L | D,L | D,L |
| *Anisodactylus similis* | | | | | | | | | | D,L | D |
| *Apristus* | dup | | | | | | | | | | D |
| *Apristus pugetanus* | | | | | | | | | | D | |
| *Axinopalpus biplagiatus* | | | | | | | | | | D | D |
| *Bembidion corgenoma* | | | | | | D | | | | D,L | D,L |
| *Bembidion ephippigerum* | | | | | | | D | | L | | |
| *Bembidion indistinctum* | | | | | | | L | | | | L |
| *Bembidion insulatum* | | | | | L | | | | | | |
| *Bembidion iridescens* | | | | | | | | | L | | |
| *Bembidion laticeps* | | | | | D | | | | | | |
| *Bembidion palosverdes* | | | | | | | | | D,L | | |
| *Bembidion platynoides* | | | | | | | | | | | L |
| *Bembidion striola* | | | | | L | | | | L | | |
| *Bembidion versicolor* | | | | | L | | | | | | |
| *Brachinus costipennis* | | | | | | | | | L | D,L | D |
| *Brachinus gebhardis* | | | | | | | | | D | D,L | D |
| *Brachinus mexicanus* | | | | | | | | | | D,L | |
| *Brachinus quadripennis* | | | | | | | | | | L | |
| *Bradycellus californicus* | | | | | D | D | D | | D | D | D,L |
| *Bradycellus nitidus* | | | | | | D | D | | D,L | D,L | D,L |
| *Bradycellus rupestris* | | | | | D | | D | | L | D | D |
| *Bradycellus sejunctus* | | | | | D | | | | | | |
| *Calathus ruficollis* | | | | D,L | L | D,L | | | D,L | D,L | D,L |
| *Calosoma eremicola* | | | | | D,L | | | | D,L | | |

(Continued)

| Scientific name | Duplicate genus record | Endemic | Adventive | Anacapa | San Clemente | San Miguel | San Nicolas | Santa Barbara | Santa Catalina | Santa Cruz | Santa Rosa |
|---|---|---|---|---|---|---|---|---|---|---|---|
| *Calosoma parvicolle* | | | | | | | | | L | | |
| *Calosoma semilaeve* | | | | | | | L | L | D,L | D | L |
| *Chlaenius cumatilis* | | | | | | | | | | D,L | D |
| *Chlaenius obsoletus* | | | | | | | | | L | D | |
| *Chlaenius tricolor* | | | | | | | | | | D | |
| *Chlaenius variabilipes* | | | | | | | | | D | D,L | D |
| *Dicheirus dilatatus* | | | | | D,L | D | | | D,L | D,L | D,L |
| *Dicheirus piceus* | | | | D | D,L | | | | D,L | D | D,L |
| *Dromius piceus* | | | | D | | | | | | D | |
| *Dyschirius aratus* | | | | | | | | | | | D |
| *Dyschirius consobrinus* | | | | | | | | | | | D |
| *Dyschirius gibbipennis* | | | | | | D | | | | D | D,L |
| *Dyschirius varidens* | | | | | | | | | | D | |
| *Elaphropus* undet. sp. | | | | | | | | | | D | |
| *Harpalus* | dup | | | D | | | | | | | |
| *Harpalus caliginosus* | | | | | | | | | | D | D |
| *Harpalus pensylvanicus* | | | | | | | | | | D | |
| *Lachnophorus elegantulus* | | | | | | | | | | D | |
| *Laemostenus complanatus* | | | adv | | | | D | | D | | |
| *Lebia cyanipennis* | | | | | | | | | | D | D |
| *Lebia perita* | | | | | | | | | D | D | |
| *Microlestes* undet. sp. | | | | | | | | | | | D |
| *Notiophilus semiopacus* | | | | | | | | | L | D | |
| *Omophron dentatum* | | | | | | | | | D | D,L | D,L |
| *Phrypeus rickseckeri* | | | | | | | | | | D | |
| *Platynus brunneomarginatus* | | | | | D | D | | | D,L | D,L | D,L |
| *Poecilus laetulus* | | | | | D | | | | D,L | L | L |
| *Pterostichus gliscans* | | end | | | L | D | | | | | |
| *Pterostichus illustris* | | | | | | | | | D | | |
| *Pterostichus inermis* | | | | | | | | | | D | |
| *Pterostichus isabellae* | | | | | L | | | | L | | |
| *Pterostichus jacobinus* | | | | | | | | | D | | |
| *Pterostichus lustrans* | | | | | | | | | | D,L | |
| *Pterostichus menetriesii* | | | | | | D | | | | D | L |
| *Scaphinotus crenatus* | | | | | | D | | | D | D | D |
| *Scaphinotus punctatus* | | | | | | | | | D,L | | |
| *Scaphinotus ventricosus* | | | | | | | | | D,L | | |
| *Schizogenius depressus* | | | | | | | | | | D | L |
| *Stenolophus anceps* | | | | | | D | | | | D | D |
| *Stenolophus flavipes* | | | | | | | | | D | D | D |

| Scientific name | Duplicate genus record | Endemic | Adventive | Anacapa | San Clemente | San Miguel | San Nicolas | Santa Barbara | Santa Catalina | Santa Cruz | Santa Rosa |
|---|---|---|---|---|---|---|---|---|---|---|---|
| *Stenolophus limbalis* | | | | | | | | | L | | |
| *Stenolophus lineola* | | | | D | | | | | | D | D,L |
| *Stenolophus ochropezus* | | | | | | | | | | D | |
| *Stenolophus rugicollis* | | | | | | D | | | | | |
| *Tachys corax* | | | | | L | | D | | | | |
| *Tachys vittiger* | | | | | | | | | L | | |
| *Tachys vorax* | | | | | | | | | | D | |
| *Tanystoma cuyama* | | | | | | | | | | | D |
| *Tanystoma maculicolle* | | | | L | D,L | D,L | D,L | | D,L | D,L | D,L |
| *Thalassotrechus barbarae* | | | | | D | | | | D | | |
| **Cicindelidae** | | | | | | | | | | | |
| *Cicindela hirticollis* | | | | | | | | | L | | L |
| *Cicindela oregona* | | | | D,L | | D,L | D,L | | L | D,L | D,L |
| *Cicindela senilis* | | | | | D,L | | | | | | |
| *Cicindelidia hemorrhagica* | | | | | | | D,L | | | D,L | |
| *Cicindelidia trifasciata* | | | | | | | | | D,L | | |
| **Dytiscidae** | | | | | | | | | | | |
| *Agabinus glabrellus* | | | | | | | | | D,L | D,L | D |
| *Agabinus sculpturellus* | | | | | | | | | | L | |
| *Agabus obsoletus* | | | | | | D | | | | | |
| *Dytiscus marginicollis* | | | | | D | | | | D,L | | |
| *Eretes sticticus* | | | | | | | | | D | | |
| *Hydrovatus brevipes* | | | | | | | | | | L | |
| *Hygrotus lutescens* | | | | | D | | D | | D,L | D | |
| *Ilybiosoma lugens* | | | | | | | D | | D,L | D | D,L |
| *Ilybiosoma regulare* | | | | | | | | | | D | D |
| *Ilybiosoma seriatum* | | | | | | | | | | D | D |
| *Ilybius discors* | | | | | | | | | | L | |
| *Ilybius lineellus* | | | | | | | | | | D | |
| *Ilybius walsinghami* | | | | | L | | | | | | D |
| *Laccophilus fasciatus* | | | | | D | | | | D,L | | |
| *Laccophilus maculosus* | | | | | | | | | D,L | | |
| *Leconectes striatellus* | | | | | D | D | | | D,L | D,L | D,L |
| *Liodessus obscurellus* | | | | | | D | | | D | D | D |
| *Neoclypeodytes pictodes* | | | | | | | | | | | D |
| *Rhantus gutticollis* | | | | | D,L | | D | | D,L | D,L | D |
| *Sanfilippodytes barbarensis* | | | | | | | | | D | D | D |
| *Sanfilippodytes latebrosus* | | | | | | | | | D | D | D |
| *Sanfilippodytes vilis* | | | | | | D | D | | L | L | |
| *Sanfilippodytes williami* | | | | | | | | | L | D | L |
| *Uvarus subtilis* | | | | | | | | | | D | D |

(Continued)

| Scientific name | Duplicate genus record | Endemic | Adventive | Anacapa | San Clemente | San Miguel | San Nicolas | Santa Barbara | Santa Catalina | Santa Cruz | Santa Rosa |
|---|---|---|---|---|---|---|---|---|---|---|---|
| **Gyrinidae** | | | | | | | | | | | |
| *Gyrinus plicifer* | | | | | | | | | D | D,L | D |
| **Haliplidae** | | | | | | | | | | | |
| *Haliplus* undet. sp. | | | | | | | | | D | | |
| *Peltodytes simplex* | | | | | | | | | D,L | D,L | D |
| **MYXOPHAGA** | | | | | | | | | | | |
| **Hydroscaphidae** | | | | | | | | | | | |
| *Hydroscapha natans* | | | | | | | | | | D,L | D |
| **Sphaeriusidae** | | | | | | | | | | | |
| *Sphaerius politus* | | | | | | | | | D | D | |
| **SCIRTOIDEA** **Scirtidae** | | | | | | | | | | | |
| Undet. genus, undet. sp. | | | | | | | | | | L | |
| **CLAMBOIDEA** **Clambidae** | | | | | | | | | | | |
| *Clambus* undet. sp. | | | | | | | | | | D | |
| *Loricaster rotundus* | | | | | D | | | | D | | |
| **DASCILLOIDEA** **Dascillidae** | | | | | | | | | | | |
| *Anorus piceus* | | | | | D | | | | D | D | D |
| **Rhipiceridae** | | | | | | | | | | | |
| *Sandalus cribricollis* | | | | | | | | | D | | |
| **BUPRESTOIDEA** **Buprestidae** | | | | | | | | | | | |
| *Acmaeodera hepburnii* | | | | | | | | | D,L | D,L | D,L |
| *Acmaeodera mariposa* | | | | | | | | | | D | |
| *Acmaeodera prorsa* | | | | | | | | | D | D,L | |
| *Agrilus quadriguttatus* | | | | | | | | | | D | |
| *Anthaxia aeneogaster* | | | | | | | | | | D | |
| *Buprestis aurulenta* | | | | | | | | | | D,L | |
| *Chrysobothris mali* | | | | | | | | | D | D | |
| *Melanophila consputa* | | | | | | | | | D | | |
| **DRYOPOIDEA** **Dryopidae** | | | | | | | | | | | |
| *Postelichus productus* | | | | | | | | | L | | |
| **Elmidae** | | | | | | | | | | | |
| *Ordobrevia nubifera* | | | | | | | | | | L | |
| **Heteroceridae** | | | | | | | | | | | |
| *Heterocerus* | dup | | | | D | | D | | D | | D |
| *Heterocerus mexicanus* | | | | | | | | | | D,L | |
| **Limnichidae** | | | | | | | | | | | |
| *Limnichites nebulosus* | | | | | | | | | | D | |

| Scientific name | Duplicate genus record | Endemic | Adventive | Anacapa | San Clemente | San Miguel | San Nicolas | Santa Barbara | Santa Catalina | Santa Cruz | Santa Rosa |
|---|---|---|---|---|---|---|---|---|---|---|---|
| **ELATEROIDEA** | | | | | | | | | | | |
| **Cantharidae** | | | | | | | | | | | |
| *Cultellunguis americanus* | | | | | | | | | L | | |
| *Cultellunguis hatchi* | | | | | | | | | L | D,L | |
| *Frostia laticollis* | | | | | | | | | | D,L | |
| *Pacificanthia consors* | | | | | | | | | D | D,L | D |
| *Podabrus pruinosus* | | | | | | | | | | D | |
| *Podabrus* undet. sp. | | | | | | | | | | D | |
| *Silis* | dup | | | | | | | | | D | |
| *Silis carmelita* | | | | | | | | | | | D |
| **Elateridae** | | | | | | | | | | | |
| *Ampedus longicornis* | | | | | | | | | D,L | | |
| *Ampedus rhodopus* | | | | | | | | | | | D |
| *Anchastus cinereipennis* | | | | | D | | D,L | L | D | | D |
| *Athous axillaris* | | | | | | | | | | D | D |
| *Athous nigropilis* | | | | | | | | | D | | |
| *Athous rufiventris* | | | | | | | | | D | D | |
| *Cardiophorus* | dup | | | | | | | | | | D |
| *Cardiophorus tenebrosus* | | | | | | | | | | D | |
| *Dalopius* | dup | | | | | | | | D | | |
| *Dalopius luteolus* | | | | | | | | | | D | |
| *Dalopius* undet. sp. | | | | | | | | | | D | D |
| *Elater lecontei* | | | | | | | | | | D | |
| *Euthysanius lautus* | | | | | | | | | | D | D |
| *Hemicrepidius californicus* | | | | | | D | D | | | | |
| *Hemicrepidius tumescens* | | | | | | | | | | L | |
| *Heteroderes amplicollis* | | | adv | | | | | | | | D |
| *Horistonotus inanus* | | | | | | | | | D | D | |
| *Limonius canus* | | | | | D | | | | | D | |
| *Melanactes densus* | | | | | | | | | D | | |
| *Melanotus longulus* | | | | | | | | | D,L | D | D |
| *Octinodes frater* | | | | | | | | | | D | |
| *Paradonus inops* | | | | | | | | | | D | |
| **Eucnemidae** | | | | | | | | | | | |
| *Asiocnemis hospitalis* | | | | | | | | | | | D,L |
| **Lampyridae** | | | | | | | | | | | |
| *Pterotus obscuripennis* | | | | | | | | | D | | |
| *Pyropyga nigricans* | | | | | | | | | | D | D |
| **Phengodidae** | | | | | | | | | | | |
| *Zarhipis integripennis* | | | | | | | | | D | | |
| **Throscidae** | | | | | | | | | | | |

(Continued)

| Scientific name | Duplicate genus record | Endemic | Adventive | Anacapa | San Clemente | San Miguel | San Nicolas | Santa Barbara | Santa Catalina | Santa Cruz | Santa Rosa |
|---|---|---|---|---|---|---|---|---|---|---|---|
| *Trixagus sericeus* | | | | | | | | | | D | |
| **HISTEROIDEA** **Histeridae** | | | | | | | | | | | |
| *Aphelosternus interstitialis* | | | | | | | | | L | | |
| *Bacanius* undet. sp. | | | | | | | | | D,L | | |
| *Carcinops opuntiae* | | | | | | | | | D | | |
| *Euspilotus scissus* | | | | | | D | D | | | D | D |
| *Euspilotus* sp. near *laridus* | | | | | | | | | L | | |
| *Geomysaprinus* undet. sp. | | | | | | | | | D | | D |
| *Halacritus maritimus* | | | | | D | | D | | | | |
| *Hololepta vicina* | | | | | | | | | L | | |
| *Hypocaccus bigemmeus* | | | | | D | D | D | | | D | D |
| *Hypocaccus gaudens* | | | | | | D | D | | D | D | D |
| *Hypocaccus lucidulus* | | | | | D,L | D,L | D,L | | | D,L | D,L |
| *Hypocaccus serrulatus* | | | | | | | | | D | | |
| *Iliotona cacti* | | | | | | D | | | | | |
| *Margarinotus sexstriatus* | | | | | | | | | | D,L | L |
| *Neopachylopus sulcifrons* | | | | | D | D | D,L | | D | D | D |
| *Plegaderus* undet. sp. | | | | | | | | | D | | |
| *Saprinus lugens* | | | | | L | D | D,L | D,L | D | D,L | D,L |
| *Saprinus oregonensis* | | | | | | | | | | L | |
| *Xerosaprinus fimbriatus* | | | | | | | | | L | | |
| *Xerosaprinus lubricus* | | | | | L | | | | D,L | D,L | D,L |
| *Xerosaprinus vitiosus* | | | | | | | | | L | | |
| **HYDROPHILOIDEA** **Helophoridae** | | | | | | | | | | | |
| *Helophorus linearis* | | | | | D | | | | | | |
| **Hydrophilidae** | | | | | | | | | | | |
| *Agna capillata* | | | | | | | | | D | | |
| *Anacaena signaticollis* | | | | | | D | | | | D,L | D |
| *Berosus fraternus* | | | | | | | | | L | | |
| *Berosus hatchi* | | | | | | | | | L | | |
| *Berosus infuscatus* | | | | | D | | | | | | |
| *Berosus punctatissimus* | | | | | | | | | D | D,L | D |
| *Cercyon fimbriatus* | | | | | D | D,L | D | | D,L | D | D |
| *Cercyon haemorrhoidalis* | | | adv | | | | | L | | | D |
| *Cercyon luniger* | | | | | D | L | | | L | D,L | |
| *Cercyon quisquilius* | | | adv | | | | | | D | D | |
| *Chaetarthria hespera* | | | | | | | | | D,L | D | |
| *Chaetarthria nigrella* | | | | | | | | | | D | D |
| *Chaetarthria punctulata* | | | | | | | | | | D | |

| Scientific name | Duplicate genus record | Endemic | Adventive | Anacapa | San Clemente | San Miguel | San Nicolas | Santa Barbara | Santa Catalina | Santa Cruz | Santa Rosa |
|---|---|---|---|---|---|---|---|---|---|---|---|
| Chaetarthria pusilla | | | | | | | | | | D | |
| Cymbiodyta columbiana | | | | | | | | | | D | |
| Cymbiodyta dorsalis | | | | | | D,L | D | | | D,L | D,L | D,L |
| Cymbiodyta punctatostriata | | | | | | | | | | | D,L | |
| Enochrus carinatus | | | | | | | L | | | | D | |
| Enochrus cristatus | | | | | | | | | | | D | |
| Enochrus hamiltoni | | | | | | | D | | | | | |
| Enochrus piceus | | | | | | | | D | | D | D | D |
| Enochrus pygmaeus | | | | | | | | | | D | | |
| Helochares normatus | | | | | | | | | | | D,L | D |
| Hydrobius fuscipes | | | | | | | | | | | L | |
| Hydrochara lineata | | | | | | | | | | | D,L | L |
| Hydrophilus triangularis | | | | | D | | | | | | D,L | |
| Laccobius californicus | | | | | | | | | | | D,L | |
| Laccobius ellipticus | | | | | | | | | | D,L | D,L | D,L |
| Laccobius insolitus | | | | | | | | D | | | | D |
| Sphaeridium scarabaeoides | | | adv | | | | | | | D | D | D |
| Tropisternus affinis | | | | | | | | | | D,L | D | D |
| Tropisternus californicus | | | | | | | | | | L | L | |

**SCARABAEOIDEA**
**Geotrupidae**

| Scientific name | Duplicate genus record | Endemic | Adventive | Anacapa | San Clemente | San Miguel | San Nicolas | Santa Barbara | Santa Catalina | Santa Cruz | Santa Rosa |
|---|---|---|---|---|---|---|---|---|---|---|---|
| Bolbocerastes regalis | | | | | L | | | | | | |
| Odonteus obesus | | | | | | | | | | | D |

**Scarabaeidae**

| Scientific name | Duplicate genus record | Endemic | Adventive | Anacapa | San Clemente | San Miguel | San Nicolas | Santa Barbara | Santa Catalina | Santa Cruz | Santa Rosa |
|---|---|---|---|---|---|---|---|---|---|---|---|
| Aegialia convexa | | | | | L | | | | | | |
| Aegialia crassa | | | | | L | | | | | | |
| Aegialia nigrella | | | | | | | L | | | | |
| Aegialia punctata | | | | | | | L | | | | |
| Amblonoxia palpalis | | | | D | L | | D,L | L | | | |
| Aphodius fimetarius | | | adv | D | | | | | D | D | |
| Calamosternus granarius | | | adv | D | | | D | | D | | D,L |
| Canthon simplex | | | | | L | | | | | | |
| Cinacanthus militaris | | | | | | | L | | | | |
| Coenonycha clementina | | end | | | D,L | | | | | | |
| Coenonycha clypeata | | end | | | | | | | D,L | | |
| Coenonycha fulva | | end | | D | | | | | D,L | | |
| Coenonycha santacruzae | | end | | | | | | | | D,L | |
| Cotinis mutabilis | | | | | | | | | D | | |
| Cremastocheilus schaumii | | | | | | | | | D,L | | |
| Cyclocephala borealis | | | | | L | | | | L | | |

(Continued)

| Scientific name | Duplicate genus record | Endemic | Adventive | Anacapa | San Clemente | San Miguel | San Nicolas | Santa Barbara | Santa Catalina | Santa Cruz | Santa Rosa |
|---|---|---|---|---|---|---|---|---|---|---|---|
| *Cyclocephala hirta* | | | | | | | | | D | | |
| *Cyclocephala longula* | | | | | L | | L | | | L | L |
| *Cyclocephala melanocephala* | | | | | | L | | | | L | L |
| *Cyclocephala pasadenae* | | | | | | L | | | | L | L |
| *Dichelonyx backii* | | | | | | | | | | D | |
| *Dichelonyx fulgida* | | | | | | | | | | L | |
| *Dichelonyx pusilla* | | | | | | D,L | | | | D,L | D,L |
| *Diplotaxis fimbriata* | | | | | L | | | | | L | |
| *Diplotaxis subangulata* | | | | | D,L | | | | | D,L | D,L |
| *Hoplia callipyge* | | | | | | | | | | D,L | |
| *Labarrus pseudolividus* | | | | D | D,L | D,L | D | | L | D,L | D,L |
| *Ligyrus gibbosus* | | | | | D,L | D,L | D,L | | | D | D,L |
| *Otophorus haemorrhoidalis* | | | adv | | | | | | | D | |
| *Phobetus* | dup | | | | D,L | | D | L | | | D |
| *Phobetus ciliatus* | | end | | | | | | | D,L | | |
| *Phobetus testaceus* | | end | | | | | | | | D,L | |
| *Phyllophaga mucorea* | | | | | L | | | | | | |
| *Planolinellus vittatus* | | | | | | | D,L | | | | D,L |
| *Polyphylla* | dup | | | | | L | | | | | |
| *Polyphylla crinita* | | | | | | | | | D | D,L | L |
| *Polyphylla nigra* | | | | | | | | | D | D,L | L |
| *Rugaphodius rugatus* | | | | | | | D,L | | | | D,L |
| *Serica alternata* | | | | | L | | L | L | | | |
| *Serica catalina* | | end | | | | | | | L | | |
| *Serica cruzi* | | end | | | | | | | | D,L | |
| *Serica mixta* | | | | | L | L | L | | | L | L |
| *Tesarius mcclayi* | | | | | | | D | | | | D |
| **Trogidae** | | | | | | | | | | | |
| *Trox atrox* | | | | | D,L | | | | | | |
| *Trox gemmulatus* | | | | | D,L | | | | | | |
| **STAPHYLINOIDEA** **Colonidae** | | | | | | | | | | | |
| *Colon forceps* | | | | | | | | | | D | |
| **Hydraenidae** | | | | | | | | | | | |
| *Hydraena* | dup | | | | D | | | | | | D |
| *Hydraena arenicola* | | | | | | | | | | L | |
| *Hydraena circulata* | | | | | | | | | | L | |
| *Hydraena vandykei* | | | | | | | | | | L | |
| *Ochthebius* | dup | | | | D | D | D | | | | D |
| *Ochthebius discretus* | | | | | | | | | L | | |

| Scientific name | Duplicate genus record | Endemic | Adventive | Anacapa | San Clemente | San Miguel | San Nicolas | Santa Barbara | Santa Catalina | Santa Cruz | Santa Rosa |
|---|---|---|---|---|---|---|---|---|---|---|---|
| *Ochthebius interruptus* | | | | | | | | | | L | |
| *Ochthebius puncticollis* | | | | | | | | | | D,L | |
| **Leiodidae** | | | | | | | | | | | |
| *Agathidium pulchrum* | | | | | | | | | | D | |
| *Agathidium virile* | | | | | D | | | | D | | |
| *Leiodes antennata* | | | | | | | | | D | | |
| *Leiodes paludicola* | | | | | | | | | D | | |
| *Pinodytes gibbosus* | | | | | | | | | D,L | D,L | D,L |
| **Ptiliidae** | | | | | | | | | | | |
| *Acrotrichis* undet. sp. | | | | | | | | | | D | D |
| *Actidium* undet. sp. | | | | | | | | | | D | |
| *Motschulskium sinuatocolle* | | | | | D | | D | | D | | |
| *Ptenidium* undet. sp. | | | | | | | | | | | D |
| *Pteryx* undet. sp. | | | | | | | | | | D | D |
| *Ptiliolum* undet. sp. | | | | | D | | | | D | | D |
| **Staphylinidae** | | | | | | | | | | | |
| *Acrotona* | dup | | | | | | D | | D | | |
| *Acrotona recondita* | | | | | | | | | L | | |
| *Acrotona sonomana* | | | | | | | | | L | | |
| *Actium californicum* | | | | | | | | | | L | |
| *Actium vestigialis* | | end | | | | | | | D,L | | |
| *Adota maritima* | | | | | | | | | L | | |
| *Aleochara bimaculata* | | | | | L | | | | L | | |
| *Aleochara curtidens* | | | | | | | | L | | | |
| *Aleochara densissima* | | | | | | | | | L | | |
| *Aleochara fumata* | | | adv | | | | | | L | | |
| *Aleochara lanuginosa* | | | adv | | | | | | | | D,L |
| *Aleochara littoralis* | | | | | | | | L | L | | |
| *Aleochara sulcicollis* | | | | | D | D,L | D,L | | D | D,L | D,L |
| *Aleochara valida* | | | | | D | D | | | L | | D |
| *Aploderus* | dup | | | | | | | | | D | D |
| *Aploderus flavipennis* | | | | | | | | | L | | |
| *Apocellus analis* | | | | | | | | | L | | |
| *Astenus* undet. sp. | | | | | D | | | | | D | |
| *Atheta hampshirensis* | | | | | | | D | | | | |
| *Belonuchus ephippiatus* | | | | | | D | | | D | | |
| *Bisnius albionicus* | | | | | | D | | | | | D |
| *Bisnius sordidus* | | | adv | | | | D | | | | D |
| *Bledius albonotatus* | | | | | | D,L | D,L | | D | D | D |
| *Bledius fenyesi* | | | | | D | D,L | D,L | | D | D,L | D |
| *Bledius opacifrons* | | | | | | | | | | D | D |

(Continued)

| Scientific name | Duplicate genus record | Endemic | Adventive | Anacapa | San Clemente | San Miguel | San Nicolas | Santa Barbara | Santa Catalina | Santa Cruz | Santa Rosa |
|---|---|---|---|---|---|---|---|---|---|---|---|
| *Bledius ruficornis* | | | | | D,L | | | | | D | D |
| *Blepharhymenus* undet. sp. | | | | | D | | | | | D | D |
| *Brachycepsis* undet. sp. | | | | | | | | | L | D,L | D |
| *Bryoporus rufescens* | | | | | D | | | | | D,L | D |
| *Bryothinusa catalinae* | | | | | | | | | D,L | | |
| *Cafius canescens* | | | | | | D | D,L | | L | D | D |
| *Cafius lithocharinus* | | | | | D | D | D,L | | D | D | D,L |
| *Cafius luteipennis* | | | | | D | D | D | | D,L | D | D,L |
| *Cafius opacus* | | | | | | | | | L | | |
| *Cafius seminitens* | | | | | D | D,L | D,L | | | D | D |
| *Cafius sulcicollis* | | | | | D | | | | | L | D,L |
| *Carpelimus* undet. sp. | | | | | D | | D | | D | D | D |
| *Cephennium urbanum* | | | | | | | | | D,L | | |
| *Creophilus maxillosus* | | | | | D,L | | | | L | D | D |
| *Diaulota fulviventris* | | | | | | | | | | D | D |
| *Diestota* undet. sp. | | | | | D | D | D | | D | D | D |
| *Erichsonius puncticeps* | | | | | | D | | | L | D | D |
| *Euconnus* undet. sp. | | | | | D | | | | D | | D |
| *Falagriota occidua* | | | | | | | | | | D,L | D |
| *Gabrius* | dup | | | | | | | | | D | |
| *Gabrius nigritulus* | | | adv | | | | D | | L | | |
| *Gnypeta* undet. sp. | | | | | | | | | | D | D |
| *Habrocerus capillaricornis* | | | adv | | | | | | D | | |
| *Hadrotes crassus* | | | | D | D,L | D | D,L | | D,L | D,L | D,L |
| *Hesperotychus* undet. sp. | | | | | | | | | D,L | | |
| *Heterosilpha ramosa* | | | | | D | L | | | | D,L | D,L |
| *Heterothops conformis* | | | | | | | | | D | D,L | D |
| *Heterothops fusculus* | | | | | | | | | L | | |
| *Holobus* undet. sp. | | | | | | | | | D | | |
| *Hydrosmecta* undet. sp. | | | | | | | | | | D | |
| Leptotyphlinae undet. sp. | | | | | | | | | | D | |
| *Linohesperus* | dup | | | | D | | | | D | | |
| *Linohesperus borealis* | | | | | | | | | | L | D |
| *Linohesperus cuspifer* | | | | | | | | | | L | |
| *Lobrathium* | dup | | | | | | | | | D | |
| *Lobrathium jacobinum* | | | | | | | | | | | L |
| *Lordithon thoracicus* | | | | | | | | | D | D | |
| *Medon* undet. sp. | | | | | D | D | | | D | D,L | D |
| Meoticina undet. sp. | | | | | | | | | | D | |
| *Mycetoporus neotomae* | | | | | | | | | D | D | |
| *Myllaena* undet. sp. | | | | | | D | | | D | D | D |

| Scientific name | Duplicate genus record | Endemic | Adventive | Anacapa | San Clemente | San Miguel | San Nicolas | Santa Barbara | Santa Catalina | Santa Cruz | Santa Rosa |
|---|---|---|---|---|---|---|---|---|---|---|---|
| *Neobisnius occidentoides* | | | | | D,L | | | | D | D | D |
| *Neobisnius sobrinus* | | | | | | | | | | D | |
| *Neobisnius terminalis* | | | | | | | | | | D | |
| *Nicrophorus guttula* | | | | | D,L | | | | D | | D |
| *Nicrophorus marginatus* | | | | | | D | | | | | |
| *Nicrophorus nigrita* | | | | L | L | | | L | D,L | D,L | D,L |
| *Nitidotachinus agilis* | | | | | | | | | | D | D |
| *Nudobius pugetanus* | | | | | | | | | | D | |
| *Oligota* undet. sp. | | | | | | | D | | | D | D |
| *Omalium algarum* | | | | | | | D | | | | |
| *Oropus* undet. sp. | | | | | | | | | L | D,L | |
| *Orus* undet. sp. | | | | | | | | | | D | |
| *Oxypoda* undet. sp. | | | | | | | | | D | D | D |
| *Palporus nitidulus* | | | | | | | D | | | D | |
| *Philonthus cruentatus* | | | adv | | | | | D | D,L | D | D |
| *Philonthus davus* | | | | | | | | | | D | D |
| *Philonthus flavolimbatus* | | | | | | | | | D | | |
| *Philonthus hepaticus* | | | | | | | | | D | | |
| *Philonthus lecontei* | | | | | | | | | | | L |
| *Philonthus longicornis* | | | adv | | | | | | L | | |
| *Philonthus quadrulus* | | | | | | | | | | D | |
| *Philonthus triangulum* | | | | | | | | | L | | |
| *Phloeopora* undet. sp. | | | | | | | | | | D | |
| *Platystethus americanus* | | | | | | | | | | | D |
| *Pontomalota opaca* | | | | | | D,L | D | | | | D |
| *Pseudopsis* | dup | | | | | | | | L | | |
| *Pseudopsis minuta* | | | | | | | | | | D,L | |
| *Quedius limbifer* | | | | | | | | | | D,L | D |
| *Sepedophilus castaneus* | | | | | | | | | | D | D |
| *Sonoma* | dup | | | | | | | | | D,L | |
| *Sonoma isabellae* | | | | | D,L | | | | D,L | | |
| *Sonomota* undet. sp. | | | | | D | | | | | D | |
| *Stenichnus* undet. sp. | | | | | | | | | D,L | | |
| *Stictalia* undet. sp. | | | | | D | | | | | D | D |
| *Sunius* | dup | | | | | | | L | | | |
| *Sunius mobilis* | | | | | | | | | L | | |
| *Sunius reductus* | | | | | | | | | L | | |
| *Tachinus debilis* | | | | | | | | | | D | |
| *Tachyporus* | dup | | | | | L | | | | | |
| *Tachyporus californicus* | | | | | D | | D | | L | D | D,L |
| *Tarphiota fucicola* | | | | | | D | D | | | D | D |

(Continued)

| Scientific name | Duplicate genus record | Endemic | Adventive | Anacapa | San Clemente | San Miguel | San Nicolas | Santa Barbara | Santa Catalina | Santa Cruz | Santa Rosa |
|---|---|---|---|---|---|---|---|---|---|---|---|
| *Tarphiota geniculata* | | | | | D | D | D | | D | D | D |
| *Tasgius ater* | | | adv | | | D,L | D | L | | | D |
| *Thanatophilus lapponicus* | | | | | | | | | | | L |
| *Thinobius* undet. sp. | | | | | | | D | | | D | D |
| *Thinopinus pictus* | | | | | | D,L | D,L | | D,L | D,L | D,L |
| *Thinusa fletcheri* | | | | | D | | D | | D | | D |
| *Thinusa maritima* | | | | | | | | | | D,L | |
| **BOSTRICHOIDEA** **Bostrichidae** | | | | | | | | | | | |
| *Amphicerus cornutus* | | | | | | | | | D | D | |
| *Lyctus cavicollis* | | | | | | | | | | D | |
| *Lyctus linearis* | | | adv | | | | | | | D | |
| *Lyctus planicollis* | | | | | | | | | | D | |
| *Melalgus confertus* | | | | | | | | | D | | |
| *Polycaon stoutii* | | | | | | | | | D | L | |
| *Psoa maculata* | | | | | | | | | D | | |
| *Psoa quadrisignata* | | | | | | | | | L | | |
| *Scobicia declivis* | | | | | | | L | | | D | |
| *Scobicia suturalis* | | | | D | | | | | D | D | |
| *Stephanopachys substriatus* | | | | | | | | | D | | |
| **Dermestidae** | | | | | | | | | | | |
| *Anthrenus lepidus* | | | | | | | | | | D | D |
| *Anthrenus verbasci* | | | adv | | | | | | D,L | | |
| *Cryptorhopalum apicale* | | | | | | | | | | D,L | |
| *Cryptorhopalum triste* | | | | | | | | | | | D |
| *Dermestes* | dup | | | D | | | | | | | |
| *Dermestes caninus* | | | | | L | | L | D,L | | | L |
| *Dermestes frischi* | | | adv | | D,L | D,L | D | D,L | | D,L | |
| *Dermestes marmoratus* | | | | | D,L | | L | | L | | L |
| *Dermestes rattus* | | | | | | | | | | D | L |
| *Dermestes talpinus* | | | | | | | | | | L | D |
| *Megatoma variegata* | | | | | | | | | | D | D |
| *Trogoderma sternale* | | | | | | | | D,L | D,L | D | D |
| **Ptinidae** | | | | | | | | | | | |
| *Actenobius pleuralis* | | | | | | | | | | D | |
| *Byrrhodes ulkei* | | | | | | | | | D | | |
| *Colposternus tenuilineatus* | | | | | | | | | L | D | |
| *Ernobius debilis* | | | | | | | | | | D,L | |
| *Ernobius punctulatus* | | | | | | | | | | L | |
| *Euceratocerus hornii* | | | | | | | | | L | | |
| *Euvrilletta catalinae* | | end | | | | | | | L | | |

| Scientific name | Duplicate genus record | Endemic | Adventive | Anacapa | San Clemente | San Miguel | San Nicolas | Santa Barbara | Santa Catalina | Santa Cruz | Santa Rosa |
|---|---|---|---|---|---|---|---|---|---|---|---|
| *Euvrilletta occidentalis* | | | | | | | | | | D | |
| *Hemicoelus nelsoni* | | | | | | | | | | D | D |
| *Lasioderma serricorne* | | | adv | | | | | | | D | |
| *Oligomerus delicatulus* | | | | D | | | | | | | |
| *Ozognathus cornutus* | | | | | | | | | D | D | D |
| *Petalium californicum* | | | | | | | | | | D | |
| *Priobium punctatum* | | | | | | | | | D | D,L | |
| *Ptilinus basalis* | | | | | | | | | | D | |
| *Ptinomorphus granosus* | | | | | | | | | | | D |
| *Ptinus agnatus* | | | | | | | | | | D | D |
| *Ptinus fallax* | | | | | | | | | D | | |
| *Stegobium paniceum* | | | adv | | | | | | | | D |
| *Tricorynus* | dup | | | D | D | | D | D | | D,L | |
| *Tricorynus nubilus* | | | | | | | | | L | | |
| *Tricorynus obsoletus* | | | | | | | | | L | | |
| *Vrilletta blaisdelli* | | | | | | | | | L | D | D |
| *Xarifa insularis* | | end | | | L | | | | D,L | D | D |
| *Xestobium marginicolle* | | | | | | | | L | | D | |
| *Xyletinus* undet. sp. | | | | | | | | | | | D |
| **CLEROIDEA** | | | | | | | | | | | |
| **Byturidae** | | | | | | | | | | | |
| *Xerasia grisescens* | | | | | | D | | | D,L | D,L | D |
| **Cleridae** | | | | | | | | | | | |
| *Cymatodera angustata* | | | | | | | | | | | L |
| *Cymatodera caterinoi* | | end | | L | | | | | | D,L | D,L |
| *Cymatodera insularis* | | end | | | L | | D | | D,L | | |
| *Cymatodera ovipennis* | | | | | | | | | L | | |
| *Loedelia maculicollis* | | | | | | | | | | D | |
| *Necrobia ruficollis* | | | adv | D | L | | | | | D | |
| *Necrobia rufipes* | | | adv | D | D,L | D | D | | D,L | D | D,L |
| *Phyllobaenus* | dup | | | | | | | | D | | |
| *Phyllobaenus funebris* | | | | | | D | | | | D | |
| *Phyllobaenus scaber* | | | | | | | | | | D | D |
| *Trichodes ornatus* | | | | | | | | | | D | |
| **Melyridae** | | | | | | | | | | | |
| *Attalus transmarinus* | | end | | | L | | | | | | |
| *Attalus* undesc. sp. | | end | | | D | | | | | | |
| *Charopus* undesc. sp. | | | | | | | | | L | D | |
| *Collops cribrosus* | | | | | | D | | | | D | D,L |
| *Collops crusoe* | | end | | | | D | D,L | | | D,L | D,L |
| *Collops vittatus* | | | | | | | | | D | | |

(Continued)

| Scientific name | Duplicate genus record | Endemic | Adventive | Anacapa | San Clemente | San Miguel | San Nicolas | Santa Barbara | Santa Catalina | Santa Cruz | Santa Rosa |
|---|---|---|---|---|---|---|---|---|---|---|---|
| *Dasytastes* | dup | | | D | D | | | D | | D | D |
| *Dasytastes catalinae* | | end | | | | | | | D,L | | |
| *Dasytastes insularis* | | end | | | | | | | L | | |
| *Dasytes* | dup | | | | | | | | | D | |
| *Dasytes clementae* | | end | | | L | | | | | | |
| *Endeodes basalis* | | | | | D | D | D | | L | D | D |
| *Endeodes collaris* | | | | | | | | | L | | D,L |
| *Endeodes insularis* | | | | | | L | D | | L | | D |
| *Eschatocrepis constrictus* | | | | D | | D | | | L | D,L | D |
| *Listrus* | dup | | | | D,L | D | | L | D | | D,L |
| *Listrus anacapaensis* | | end | | L | | | | | | | |
| *Listrus interruptus* | | end | | | | | | | | | D | |
| *Malachius* undet. sp. | | | | | | | | | | | | L |
| *Microasydates punctipennis* | | end | | | | | | | D,L | | |
| *Microasydates sanclemente* | | end | | | D,L | | | | | | |
| *Microasydates santabarbara* | | | | D,L | | | | | | D,L | D,L |
| *Microlipus laticeps* | | | | | | D | | | D | D | D |
| *Pseudasydates explanatus* | | | | | | | | | D | | |
| *Trichochrous brevicornis* | | | | | | | | | | D | D |
| *Trichochrous calcaratus* | | end | | D,L | | D,L | | | | D,L | D,L |
| *Trichochrous pedalis* | | | | | | | | | D,L | | |
| *Trichochrous* undesc. sp. 1 near *brevicornis* | | end | | D | | D | | | | D | D |
| *Trichochrous* undesc. sp. 2 near *brevicornis* | | end | | | | | D | | | | |
| *Trichochrous* undesc. sp. near *pedalis* | | end | | | D,L | | D | D,L | | | |
| **Trogossitidae** | | | | | | | | | | | |
| *Temnoscheila chlorodia* | | | | | | | | | D,L | | |
| *Tenebroides crassicornis* | | | | | | | | | L | | |
| *Tenebroides occidentalis* | | | | | | | | | D | | |
| **TENEBRIONOIDEA** **Anthicidae** | | | | | | | | | | | |
| *Amblyderus obesus* | | | | | | D | D,L | | | | |
| *Amblyderus parviceps* | | | | | | | | | | D | D |
| *Anthicus cribratus* | | | | | | | | | D,L | | |
| *Anthicus maritimus* | | | | | | | D | | | | |
| *Anthicus nanus* | | | | | | | | | D | D | |
| *Anthicus punctulatus* | | | | | D | | | | D | D | |
| *Anthicus rufulus* | | | | | | | | | D | | |
| *Cyclodinus annectens* | | | | | L | | | | D,L | | |

| Scientific name | Duplicate genus record | Endemic | Adventive | Anacapa | San Clemente | San Miguel | San Nicolas | Santa Barbara | Santa Catalina | Santa Cruz | Santa Rosa |
|---|---|---|---|---|---|---|---|---|---|---|---|
| *Ischyropalpus nitidulus* | | | | D | L | | | | D,L | D,L | |
| *Notoxus desertus* | | | | | | | | | D,L | D | D |
| *Notoxus sparsus* | | | | | | | | | | L | |
| *Omonadus floralis* | | | adv | | | | | | L | | |
| **Ciidae** | | | | | | | | | | | |
| *Ceracis californicus* | | | | | | | | | | | D |
| *Cis* undet. sp. | | | | | | | | | | L | |
| *Hadreule blaisdelli* | | | | | | | | | | D | |
| *Orthocis punctatus* | | | | | | | | | D | | D |
| *Sulcacis curtulus* | | | | | | | | | | D | |
| **Meloidae** | | | | | | | | | | | |
| *Cordylospasta opaca* | | | | | | | | | | L | |
| *Epicauta puncticollis* | | | | | | | | | | | D,L |
| *Lytta stygica* | | | | | D | | | | | | |
| *Meloe barbarus* | | | | | D,L | | D,L | L | D,L | D,L | D |
| *Meloe strigulosus* | | | | | | D,L | | | | | |
| **Mordellidae** | | | | | | | | | | | |
| *Mordella albosuturalis* | | | | | | | | | | D | |
| *Mordella hubbsi* | | | | | | | | | | D,L | |
| *Mordellina* undet. sp. | | | | | D | D | D | D | D | D | D |
| *Mordellistena* undet. sp. | | | | D | D | | D | D,L | D,L | D,L | D |
| **Mycetophagidae** | | | | | | | | | | | |
| *Litargus balteatus* | | | | | | | | | | D | D |
| *Mycetophagus pluriguttatus* | | | | | | | | | | D | D |
| *Typhaea stercorea* | | | adv | | | | | | D | | |
| **Mycteridae** | | | | | | | | | | | |
| *Lacconotus pinicola* | | | | | | | | | D,L | D,L | |
| **Oedemeridae** | | | | | | | | | | | |
| *Copidita quadrimaculata* | | | | | D | D | D | | D | D | D |
| *Nacerdes melanura* | | | adv | | | | | | D | | |
| *Xanthochroa marina* | | | | | | | | | D | | |
| **Pyrochroidae** | | | | | | | | | | | |
| *Pedilus bardii* | | | | | | | | | | D | |
| **Salpingidae** | | | | | | | | | | | |
| *Rhinosimus* undet. sp. | | | | | | | | | | D | |
| **Scraptiidae** | | | | | | | | | | | |
| *Anaspis atrata* | | | | | | | | | | D | |
| *Anaspis collaris* | | | | | | | | | D,L | | |
| *Pentaria trifasciata* | | | | | | | D | | D,L | D | |
| **Tenebrionidae** | | | | | | | | | | | |
| *Alaudes singularis* | | | | | D,L | | L | | | | |

| Scientific name | Duplicate genus record | Endemic | Adventive | Anacapa | San Clemente | San Miguel | San Nicolas | Santa Barbara | Santa Catalina | Santa Cruz | Santa Rosa |
|---|---|---|---|---|---|---|---|---|---|---|---|
| *Apocrypha anthicoides* | | | | D | | | | | | D | |
| *Apsena barbarae* | | | | | | | | | L | L | |
| *Apsena grossa* | | end | | D,L | D,L | | D,L | D,L | D,L | | D,L |
| *Apsena pubescens* | | | | | D | | | | D,L | D | D |
| *Apsena rufipes* | | | | | | | | | | D | D |
| *Batuliodes rotundicollis* | | | | | D | | | | | | |
| *Blapstinus angustus* | | | | | D | | | | | | |
| *Blapstinus brevicollis* | | | | | | | | | D | D,L | D,L |
| *Blapstinus discolor* | | | | | | D | | | L | D | D |
| *Cibdelis bachei* | | end | | | L | | | L | D,L | D,L | D |
| *Coelocnemis magna* | | | | | | | | | D,L | | |
| *Coelus ciliatus* | | | | L | | | D | | | D | D |
| *Coelus globosus* | | | | D,L | | D,L | D,L | L | L | D,L | D,L |
| *Coelus pacificus* | | end | | L | D,L | D,L | D,L | D,L | D,L | D,L | D,L |
| *Conibius seriatus* | | | | | D | | | | D | D | |
| *Coniontis elliptica* | | | | | | | | | L | | L |
| *Coniontis lamentabilis* | | | | | | | | | L | | |
| *Coniontis lata* | | end | | D,L | D,L | D,L | D,L | D,L | | D,L | L |
| *Coniontis microsticta* | | | | | | | | | | D | |
| *Coniontis nemoralis* | | | | | | | | | | D | |
| *Coniontis santarosae* | | end | | | | D,L | | | | D,L | D,L |
| *Coniontis subpubescens* | | | | | | | | | L | L | |
| *Coniontis viatica* | | | | | | | | | L | | |
| *Corticeus opaculus* | | | | | | | | | | D,L | |
| *Cryptadius inflatus* | | | | | | | | | | D,L | D |
| *Eleodes acuticauda* | | | | D,L | D,L | D,L | D,L | D,L | D | D,L | D |
| *Eleodes carbonaria* | | | | | D | | | | D,L | | D |
| *Eleodes clavicornis* | | | | D | | | | | | | |
| *Eleodes dentipes* | | | | L | L | | L | | | L | L |
| *Eleodes gigantea* | | | | | | D | | | | | D |
| *Eleodes inculta* | | end | | D,L | | D,L | | L | D | D,L | D,L |
| *Eleodes littoralis* | | | | D | D | D | | | D,L | D,L | D,L |
| *Eleodes nigropilosa* | | | | | | | | | D,L | D | D |
| *Eleodes osculans* | | | | | | D | | | D | D,L | D,L |
| *Eleodes scabripennis* | | | | | | | | L | | | L |
| *Eleodes subvestita* | | end | | | | | D,L | | | | |
| *Epantius obscurus* | | | | D,L | D | D | D,L | | D,L | D,L | D,L |
| *Eusattus difficilis* | | | | | L | | | | | | |
| *Eusattus politus* | | end | | | | D,L | | | | L | D,L |
| *Eusattus robustus* | | end | | | D,L | D | D,L | D | | | D |
| *Helops bachei* | | ?end | | D | D,L | D | D | D,L | D | D | D |

| Scientific name | Duplicate genus record | Endemic | Adventive | Anacapa | San Clemente | San Miguel | San Nicolas | Santa Barbara | Santa Catalina | Santa Cruz | Santa Rosa |
|---|---|---|---|---|---|---|---|---|---|---|---|
| *Helops blaisdelli* | | | | | | | L | | | | |
| *Helops rugicollis* | | | | | | | | | D | | |
| *Hylocrinus longulus* | | | | | | | D | | | | |
| *Hymenorus* | dup | | | | | | | | | L | |
| *Hymenorus infuscatus* | | | | | | | | | L | | |
| *Isomira* | dup | | | | | | D | | | | |
| *Isomira comstocki* | | | | | | | | D | | D,L | |
| *Isomira damnata* | | | | | | | | | D | | |
| *Isomira luscitiosa* | | | | | | | | | | D | D |
| *Isomira variabilis* | | | | | L | | | | | | |
| *Lepidocnemeplatia sericea* | | | | | | | | | | D | |
| *Metoponium* | dup | | | | D | | | | | | |
| *Metoponium convexicolle* | | | | | | | | | L | | |
| *Metoponium insulare* | | end | | | | | | | D,L | | |
| *Mycetochara* | dup | | | | | | | | | D | |
| *Mycetochara pubipennis* | | | | | | | | | L | | |
| *Nyctoporis carinata* | | | | D | | | D,L | | D,L | D,L | D,L |
| *Phaleria rotundata* | | | | D,L | D | | D,L | | D,L | D,L | D |
| *Platydema oregonensis* | | | | | | | | | | D | |
| *Telabis serratus* | | | | | | | | | D | | |
| *Tonibius sulcatus* | | | | | D,L | | | | | | |
| *Tribolium castaneum* | | | adv | | | | | | D | | |
| *Ulus crassus* | | | | | | | | | | L | |
| **Zopheridae** | | | | | | | | | | | |
| *Lasconotus linearis* | | | | | | | | | | D | D |
| *Megataphrus tenuicornis* | | | | | | | | | | | D |
| *Phloeodes diabolicus* | | | | | | | | | | L | |
| *Phloeodes plicatus* | | | | | | | | | D,L | D,L | |
| *Rhagodera costaefragmenta* | | end | | | L | | | | | | |
| *Rhagodera interrupta* | | | | | | | D,L | | | | |
| *Rhagodera tuberculata* | | | | | L | | | L | | D,L | |
| *Synchita lecontei* | | | | | | | | | | D | |
| **COCCINELLOIDEA** **Akalyptoischiidae** | | | | | | | | | | | |
| *Akalyptoischion heterotrichos* | | | | | | | | | D | | |
| *Akalyptoischion hormathos* | | | | | D | | | L | D | D | D |
| **Cerylonidae** | | | | | | | | | | | |
| *Cerylon unicolor* | | | | | | | | | | D | |
| **Coccinellidae** | | | | | | | | | | | |
| *Axion plagiatum* | | | | | | | | | | D | |

(Continued)

| Scientific name | Duplicate genus record | Endemic | Adventive | Anacapa | San Clemente | San Miguel | San Nicolas | Santa Barbara | Santa Catalina | Santa Cruz | Santa Rosa |
|---|---|---|---|---|---|---|---|---|---|---|---|
| *Carinodulinka* undesc. sp. near *baja* | | | | | D | | | | | | |
| *Cephaloscymnus occidentalis* | | | | | | | | | L | | |
| *Chilocorus* undet. sp. | | | | | | | | | L | D | |
| *Coccidophilus atronitens* | | | | | | | | | | D | D |
| *Coccinella californica* | | | | D,L | D,L | D | D,L | D,L | D,L | D,L | D,L |
| *Coccinella johnsoni* | | | | | D,L | | D,L | D,L | | | |
| *Coccinella novemnotata* | | | | | | D | | | | | |
| *Coccinella septempunctata* | | | adv | D | D | D | D | D | D | D | D |
| *Cycloneda polita* | | | | | | | | | L | D,L | D |
| *Cycloneda sanguinea* | | | | D | | | | | D,L | D | |
| *Delphastus catalinae* | | | | | L | | | | D,L | D,L | |
| *Diomus debilis* | | | | D | | | | | D | | |
| *Hippodamia convergens* | | | | D,L | D | D,L | D | D,L | D,L | D,L | D,L |
| *Hippodamia quinquesignata* | | | | D | D,L | D | D,L | | D,L | D,L | D,L |
| *Hyperaspidius* | dup | | | | | | D | | | | |
| *Hyperaspidius comparatus* | | | | | | L | | | | | |
| *Hyperaspis* | dup | | | | D | | | | | | D |
| *Hyperaspis lateralis* | | | | | | | | | D,L | D,L | |
| *Hyperaspis* sp. near *annexa* | | | | | | | | | D,L | | |
| *Hyperaspis taeniata* | | | | | | | | | D,L | | |
| *Microweisea* undet. sp. | | | | | | | | | D | | |
| *Nephus binaevatus* | | | adv | | | | | | D,L | | |
| *Nephus guttulatus* | | | | D | | | | D | L | D | |
| *Nephus sordidus* | | | | D | | | D | | D | | D |
| *Nipus niger* | | | | | | | | | | | D |
| *Olla v-nigrum* | | | | D | | | | | D | | |
| *Paranaemia vittigera* | | | | | | | | | | L | |
| *Psyllobora renifer* | | | | | | | | | | D | |
| *Psyllobora vigintimaculata* | | | | | | D | D | | D,L | D,L | D |
| *Rhyzobius forestieri* | | | adv | | | | | | L | D,L | |
| *Rhyzobius lophanthae* | | | adv | | L | | D | | | D | D |
| *Scymnus ardelio* | | | | | L | | | | L | | |
| *Scymnus cervicalis* | | | | | | | | | L | D | D |
| *Scymnus coniferarum* | | | | | | | | | D | | |
| *Scymnus difficilis* | | | | | | D | | | | | D |
| *Scymnus falli* | | end | | | | D | | L | | D,L | D,L |
| *Scymnus fenderi* | | | | | | | | | | D | |
| *Scymnus jacobianus* | | | | | D | D | | D | | | |

| Scientific name | Duplicate genus record | Endemic | Adventive | Anacapa | San Clemente | San Miguel | San Nicolas | Santa Barbara | Santa Catalina | Santa Cruz | Santa Rosa |
|---|---|---|---|---|---|---|---|---|---|---|---|
| *Scymnus loewii* | | | | | D | | | | | D,L | |
| *Scymnus marginicollis* | | | | D | | | D | | D,L | D,L | D |
| *Scymnus nebulosus* | | | | | | L | | | D,L | D | D |
| *Scymnus pallens* | | | | | | | | | L | D,L | D |
| *Stethorus punctum* | | | | | | | | | D | D | |
| *Zagloba ornata* | | | | | | D | | | L | D | |
| **Corylophidae** | | | | | | | | | | | |
| *Aenigmaticum californicum* | | | | D,L | | D | D | D,L | | | |
| *Orthoperus* undet. sp. | | | | | | | | | | D | |
| *Sericoderus* undet. sp. | | | | | | | D | | | D | D |
| **Endomychidae** | | | | | | | | | | | |
| *Aphorista morosa* | | | | | | | | | D | D | D,L |
| **Latridiidae** | | | | | | | | | | | |
| *Cartodere australica* | | | adv | | | | | | | D | |
| *Corticaria* undet. sp. | | | | | | | | | L | | D |
| *Corticarina* | dup | | | | D | | | | D | | |
| *Corticarina cavicollis* | | | | | | | | | | D | |
| *Corticarina herbivagans* | | | | | | L | | D,L | | | |
| *Corticarina milleri* | | end | | D | | L | L | L | | D | L |
| *Corticarina minuta* | | | | D | | | | | | D | |
| *Dienerella* undet. sp. | | | | | | | | | D,L | | |
| *Enicmus aterrimus* | | | | | | | | | | D | D |
| *Fuchsina* undesc. sp. | | end | | | D | | | | D | D,L | D |
| *Melanophthalma* | dup | | | D | | | | | | | D |
| *Melanophthalma americana* | | | | | L | L | | | L | D | |
| *Melanophthalma casta* | | | | | | | L | D,L | | | |
| *Melanophthalma insularis* | | end | | | L | | | | | | |
| *Metophthalmus haigi* | | | | | D | | | | D | D,L | |
| *Metophthalmus rudis* | | | | | D | | | | D | D,L | D |
| *Metophthalmus trux* | | | | | D | | | | D | D,L | D |
| *Revelieria californica* | | | | | | | | | | D | D |
| *Stephostethus armatulus* | | | | | | | | | L | | |
| *Stephostethus costicollis* | | | | | D | | | | D,L | | |
| *Stephostethus liratus* | | | | | | | | | | | D |
| **EROTYLOIDEA** **Erotylidae** | | | | | | | | | | | |
| *Cryptophilus angustus* | | | adv | | | | | | | D | |
| *Dacne californica* | | | | | D | | | | D,L | D,L | D |
| **NITIDULOIDEA** **Kateretidae** | | | | | | | | | | | |

(Continued)

| Scientific name | Duplicate genus record | Endemic | Adventive | Anacapa | San Clemente | San Miguel | San Nicolas | Santa Barbara | Santa Catalina | Santa Cruz | Santa Rosa |
|---|---|---|---|---|---|---|---|---|---|---|---|
| *Amartus tinctus* | | | | | D,L | L | | | | | D,L |
| *Heterhelus sericans* | | | | | | | | | L | | |
| **Monotomidae** | | | | | | | | | | | |
| *Hesperobaenus abbreviatus* | | | | D | | | | | | D,L | D |
| *Macreurops longicollis* | | | | | | | | | | D | |
| *Phyconomus marinus* | | | | | | D | | | | D | |
| **Nitidulidae** | | | | | | | | | | | |
| *Brassicogethes aeneus* | | | | | | | | | D | | |
| *Carpophilus* | dup | | | | | | | | | | D |
| *Carpophilus discoideus* | | | | | | | | | D | | |
| *Carpophilus ligneus* | | | | D | | D | D | | | | |
| *Cryptarcha gila* | | | | | | | | | D | L | D |
| *Glischrochilus quadrisignatus* | | | adv | | | | | | D,L | | |
| *Glischrochilus sanguinolentus* | | | adv | | | | | | D,L | | |
| *Nitidula flavomaculata* | | | adv | | | | | | D | | |
| *Nitops pallipennis* | | | | | D,L | D | D | D | D,L | D,L | D |
| *Thalycra* undet. sp. | | | | | | | | | | | D |
| **CUCUJOIDEA** **Cryptophagidae** | | | | | | | | | | | |
| *Atomaria* | dup | | | | | | | | | | L |
| *Atomaria lewisi* | | | adv | | | | | | | D | |
| *Atomaria nubipennis* | | | | | D | | | | | | |
| *Atomaria puella* | | | | | | | | | | D | |
| *Cryptophagus tuberculosus* | | | | | D,L | | | | L | D | |
| **Laemophloeidae** | | | | | | | | | | | |
| *Narthecius striaticeps* | | | | | | | | | | D | |
| **Phalacridae** | | | | | | | | | | | |
| *Phalacrus* undet. sp. 1 | | | | | | | D | | | | |
| *Phalacrus* undet. sp. 2 | | | | | | | | | | D,L | |
| **Silvanidae** | | | | | | | | | | | |
| *Silvanoprus angusticollis* | | | adv | | | | | | | D | |
| **CHRYSOMELOIDEA** **Cerambycidae** | | | | | | | | | | | |
| *Anastrangalia laetifica* | | | | | | | | | | D | |
| *Arhopalus asperatus* | | | | | | | | | L | | |
| *Arhopalus productus* | | | | | | | | | L | | |
| *Asemum nitidum* | | | | | | | | | | D | |
| *Brachysomida californica* | | | | | | | | | | | D |
| *Brothylus gemmulatus* | | | | | | | | | L | | |
| *Callidiellum rufipenne* | | | adv | | | | | L | | | |

| Scientific name | Duplicate genus record | Endemic | Adventive | Anacapa | San Clemente | San Miguel | San Nicolas | Santa Barbara | Santa Catalina | Santa Cruz | Santa Rosa |
|---|---|---|---|---|---|---|---|---|---|---|---|
| *Callimus ruficollis* | | | | | | | | | D | D | |
| *Centrodera autumnata* | | | | | | | | | | D | |
| *Centrodera spurca* | | | | | | | | | | D | |
| *Desmocerus californicus* | | | | | | | | | | D | D |
| *Enaphalodes hispicornis* | | | | | | | | | D,L | | |
| *Holopleura marginata* | | | | | | | | | D | | |
| *Ipochus fasciatus* | | | | D,L | L | D,L | D | D,L | D,L | D,L | D,L |
| *Lophopogonius crinitus* | | | | | | | | | | D | D |
| *Megobrium edwardsi* | | | | | | | | | D | | L |
| *Nathrius brevipennis* | | | adv | | | | | | | D | |
| *Necydalis laevicollis* | | | | | | | | | | | D |
| *Oberea quadricallosa* | | | | | | | | | | D | |
| *Paranoplium gracile* | | | | | | | | | D,L | | |
| *Phoracantha recurva* | | | adv | | | | | | D | D | |
| *Phoracantha semipunctata* | | | adv | | | | | | D | D | |
| *Phymatodes decussatus* | | | | | | | | | | D | D,L |
| *Phymatodes grandis* | | | | | | | | | D | D | |
| *Prionus californicus* | | | | | | | | | | D | |
| *Saperda horni* | | | | | | | | | | D | |
| *Stenocorus vestitus* | | | | | | | | | | D | |
| *Sternidocinus barbarus* | | | | | | | | | | D,L | |
| *Strophiona tigrina* | | | | | | | | | D | D | |
| *Styloxus fulleri* | | | | | | | | | | D | |
| *Trichocnemis spiculatus* | | | | | | | | | | | D |
| *Xestoleptura crassipes* | | | | | | | | | | D | |
| *Xylotrechus insignis* | | | | | | | | | D,L | | |
| *Xylotrechus nauticus* | | | | | | | | | D | D,L | |
| **Chrysomelidae** | | | | | | | | | | | |
| *Acanthoscelides margaretae* | | | | | | D | | | D | D | D |
| *Acanthoscelides napensis* | | | | | D | L | | | | D | D |
| *Acanthoscelides pauperculus* | | | | | | L | | | L | | |
| *Acanthoscelides pullus* | | | | D | D | D | D | | L | D | D |
| *Altica* undet. sp. | | | | D | | | | | | D | |
| *Aulacothorax recticollis* | | | | | | | | | D | D | |
| *Calligrapha sigmoidea* | | | | | | | | | | | D |
| *Charidotella sexpunctata* | | | | D | | | | | | D | |
| *Colaspidea smaragdula* | | | | | D,L | | | | D,L | D | |
| *Cryptocephalus sanguinicollis* | | | | | | | | | D | | |

(Continued)

| Scientific name | Duplicate genus record | Endemic | Adventive | Anacapa | San Clemente | San Miguel | San Nicolas | Santa Barbara | Santa Catalina | Santa Cruz | Santa Rosa |
|---|---|---|---|---|---|---|---|---|---|---|---|
| *Diabrotica undecimpunctata* | | | | | | | | D,L | L | D,L | D |
| *Diachus auratus* | | | | D | D,L | D,L | D | | D,L | D,L | D,L |
| *Dibolia californica* | | | | | | | | | | D | |
| *Disonycha latiovittata* | | | | | | | | | | D | D |
| *Epitrix similaris* | | | | | | | | | L | | |
| *Epitrix subcrinita* | | | | | | | | | | D,L | |
| *Erynephala morosa* | | | | | | | | | | | L |
| *Gastrophysa cyanea* | | | | | | | D | | | D | D |
| *Lema daturaphila* | | | | | | | | | | D,L | D |
| *Longitarsus* undet. sp. 1 | | | | | D | | | | | | |
| *Longitarsus* undet. sp. 2 | | | | | | | D | | | | |
| *Megacerus impiger* | | | | | | | D | | | D,L | |
| *Monoxia* undet. sp. | | | | D | | | | | | | |
| *Pachybrachis melanostictus* | | | | | | | | | | D | |
| *Pachybrachis mobilis* | | | | | | | | | D | | |
| *Pachybrachis pluripunctatus* | | | | | | | | | | D | |
| *Pachybrachis punctatus* | | | | | | | | | L | D | |
| *Pachybrachis quadratus* | | | | | | | | | L | | |
| *Phaedon prasinellus* | | | | | | | | D,L | | | |
| *Phyllotreta* | dup | | | | | | | | | D | |
| *Phyllotreta pusilla* | | | | | | | | | L | | |
| *Plagiodera californica* | | | | | | | | | | D,L | |
| *Scelolyperus torquatus* | | | | | | | | | L | | |
| *Spintherophyta punctum* | | end | | | | | | | | | L |
| *Stator limbatus* | | | | | | | | | D,L | | |
| *Trachymela sloanei* | | | adv | | | | | | | D | |
| *Trirhabda confusa* | | | | | | | | | | D | |
| *Trirhabda sericotrachyla* | | | | | | | | | | D | D |
| *Yingabruxia sordida* | | | | | L | | D | D,L | | | |
| **CURCULIONOIDEA** **Attelabidae** | | | | | | | | | | | |
| *Deporaus glastinus* | | | | | | | | | D | D,L | |
| *Temnocerus aeratoides* | | | | | | | | | | D | |
| *Temnocerus aureus* | | | | | L | | | | | D | |
| *Temnocerus insularis* | | | | | L | | | | L | | |
| *Temnocerus naso* | | | | | | | | | | D | |
| **Brentidae** | | | | | | | | | | | |
| Apioninae | dup | | | | D | D | | | | | D |
| *Coelocephalapion antennatum* | | | | | | | | | L | L | |

| Scientific name | Duplicate genus record | Endemic | Adventive | Anacapa | San Clemente | San Miguel | San Nicolas | Santa Barbara | Santa Catalina | Santa Cruz | Santa Rosa |
|---|---|---|---|---|---|---|---|---|---|---|---|
| *Coelocephalapion californicum* | | | | | | | | | | L | |
| *Coelocephalapion oedorhynchum* | | | | | | | | | L | | |
| **Curculionidae** | | | | | | | | | | | |
| *Anthonomus* | dup | | | D | | | | | | | |
| *Anthonomus inermis* | | | | | | L | | D,L | | | L |
| *Anthonomus pauperculus* | | | | | | | | | L | | |
| *Anthonomus* undesc. sp. | | ?end | | | | | | D | | | |
| *Apleurus jacobinus* | | | | | | D,L | | | | | |
| *Carphobius* undesc. sp. | | end | | | D | | | | | | |
| *Carphoborus declivis* | | | | | | | | | | | D |
| *Ceutorhynchus assimilis* | | | adv | | | | | | | D | |
| *Coccotrypes dactyliperda* | | | adv | | | | | | L | | |
| *Curculio aurivestis* | | | | | | | | | D,L | | |
| *Curculio uniformis* | | | | | | | | | L | D,L | |
| *Dendrocranulus cucurbitae* | | | | D | D | | | | D | D,L | D |
| *Dendroctonus valens* | | | | | | | | | D | | |
| *Elassoptes marinus* | | | | | D | D | D | | D | D | |
| *Emphyastes fucicola* | | | | | D | | D | | D | D | |
| *Geodercodes latipennis* | | | | | D | | | | | D,L | D,L |
| *Gilbertiola* undet. sp. | | | | | | | | | D,L | | |
| *Gnathotrichus pilosus* | | | | | | | | | D | | |
| *Hypera postica* | | | adv | | D | | D | | D | D | |
| *Hypothenemus eruditus* | | | | | D | | | | D | | D |
| *Ips paraconfusus* | | | | | | | | | D,L | | |
| *Listroderes* | dup | | | | | | | | D | | |
| *Listroderes costirostris* | | | adv | | | D | D | | D | | D |
| *Listronotus* | dup | | | | | | | | D | | |
| *Listronotus sordidus* | | | | | | | L | | | | |
| *Micromastus gracilis* | | | | | | | | | D | | |
| *Monarthrum scutellare* | | | | | | | | | D | | |
| *Naupactus cervinus* | | | adv | | | | | | D,L | D | |
| *Nemocestes* undet. sp. | | | | | | | | | D | D,L | D |
| *Notiodes aeratus* | | | | | D | | | | | | |
| *Otiorhynchus cribricollis* | | | adv | | | | D | | | | |
| *Peritelinus* undet. sp. | | | | D | | | | | | | |
| *Pityophthorus carmeli* | | | | | | | | | | D | D |
| *Procryphalus utahensis* | | | | | | D | | | | | |
| *Pselactus spadix* | | | adv | | | | | | | | D |
| *Pseudips mexicanus* | | | | | | | | | | L | |

(Continued)

| Scientific name | Duplicate genus record | Endemic | Adventive | Anacapa | San Clemente | San Miguel | San Nicolas | Santa Barbara | Santa Catalina | Santa Cruz | Santa Rosa |
|---|---|---|---|---|---|---|---|---|---|---|---|
| *Pseudopityophthorus agrifoliae* | | | | | | | | | | D | |
| *Pseudopityophthorus pubipennis* | | | | | | | | | | D | D |
| *Rhinocyllus conicus* | | | adv | | | | | | | D,L | L |
| *Rhyncolus* | dup | | | | | D | | L | | | |
| *Rhyncolus cylindricollis* | | | | | | | | | | D | |
| *Scaphomorphus americanus* | | | | | L | | | | | | |
| *Sciopithes insularis* | | end | | | L | | | | | | |
| *Sciopithes setosus* | | | | | L | | | L | | | |
| *Scyphophorus yuccae* | | | | | | | | | | D | |
| *Sibinia maculata* | | | | | | L | L | D,L | | | |
| *Sitona californius* | | | | D | | D,L | | | D | D,L | D |
| *Smicronyx* | dup | | | | D | | | | | | |
| *Smicronyx cinereus* | | | | | | | | | | | L |
| *Sphenophorus graminis* | | | | | | | | | | | D |
| *Sphenophorus phoeniciensis* | | | | | | | | | | D | |
| *Sphenophorus simplex* | | | | | | | D | | | | D |
| *Sphenophorus vomerinus* | | | | | | | | | | | L |
| *Stenoclyptus sulcatus* | | | | | | | | | | | D |
| *Stenoptochus* undet. sp. | | | | | | | | | | D | |
| *Thalasselephas testaceus* | | | | | D | | D | | | | D |
| *Trichobaris compacta* | | | | | | | | | | D | |
| *Trigonoscuta anacapensis* | | end | | D,L | | | | | | | |
| *Trigonoscuta catalina* | | end | | | | | | | D,L | | |
| *Trigonoscuta clemente* | | end | | | D,L | | | D | | | |
| *Trigonoscuta curviscroba* | | end | | | | | | L | | | |
| *Trigonoscuta miguelensis* | | end | | | | D,L | | | | | |
| *Trigonoscuta nesiotis* | | end | | D,L | | | | | | | |
| *Trigonoscuta nicolana* | | end | | | | | D,L | D | | | |
| *Trigonoscuta pilosa* | | | | | L | | | | | | L |
| *Trigonoscuta sanctabarbarae* | | end | | | | | | D,L | | | |
| *Trigonoscuta sanctarosae* | | end | | | | | | | | | D,L |
| *Trigonoscuta stantoni* | | end | | | | | | | | D,L | |
| *Tychius* | dup | | | | | | L | | | | |
| *Tychius lineellus* | | | | | | | | | | D,L | D,L |
| *Xyleborinus saxesenii* | | | adv | | | | | | | D | D |

**Note:**
The sequence of suborders, superfamilies, and families corresponds to the sequence in the Annotated Checklist; the sequence of genera and species is alphabetical within families. Taxa are marked as "undet. sp." for undetermined species that simply have not been identified to lower rank and "undesc. sp." for undescribed species where the specimens have been specifically identified to an unnamed new taxon. "Duplicate genus record" ("dup") means that the line is not counted as a unique taxon but contains additional island records for the given genus. "Endemic" ("end") = nominal species that are purportedly restricted to the Channel Islands. "Adventive" ("adv") marks species whose native ranges do not include southern California. In the columns representing each of the eight Channel Islands, "D" = digitized records anchoring the species-by-island presence, "L" = literature records.

species from the islands are endemic; *Serica* (Scarabaeidae), which contains two endemic species; *Xarifa* (Ptinidae), a genus with one island endemic species and one rarely collected mainland species, neither of which have been studied since their original description, or illustrated; and *Trigonoscuta* (Curculionidae), with 10 putative endemic species that desperately need a taxonomic reassessment.

## Missing taxa

The balance of available evidence suggests that the Channel Islands have never been connected to the mainland *via* a land bridge (*Miller, 1985a*). Consequently, the entire beetle fauna of the islands was acquired *via* over-water dispersal events or human-aided transport. As a result, the fauna is notably depauperate compared to that of the mainland (*Miller, 1985a*). While we do not provide a comprehensive faunal comparison to the mainland here, we hope that the following family-level assessment of missing taxa will aid in emphasizing this conclusion, or perhaps serve to encourage further sampling and scouring of collections to discover the existence on the islands of these "missing taxa".

Not all families occurring in California are considered to be candidates for missing taxa from the Channel Islands. These include: Amphizoidae, Archeocrypticidae, Biphyllidae, Bothrideridae, Brachypsectridae, Cerophytidae, Derodontidae, Diphyllostomatidae, Eulichadidae, Hybosoridae, Ischaliidae, Mauroniscidae, Megalopodidae, Nosodendridae, Noteridae, Prostomidae, Ptilodactylidae, Smicripidae, Sphaeritidae, Stenotrachelidae, Teredidae, Thaneroceridae, and Trachypachidae. These all occur in distant regions of the state and/or are restricted to elevations or habitat types not present on the Channel Islands, and are not likely to occur there.

## Families absent from the Channel Islands but present on nearby mainland

Cupedidae (Archostemata). This family of two genera and two species in California (M. L. Gimmel, 2022, unpublished data) contains a species, *Prolixocupes lobiceps* (LeConte, 1874), widely distributed across dry areas of southern California that may eventually be found on the Channel Islands.

Eucinetidae (Clamboidea). This family of two genera and three species in California (M. L. Gimmel, 2022, unpublished data) contains at least one widespread coastal species, *Nycteus infumatus* (LeConte, 1853), that may yet be discovered on the Channel Islands.

Schizopodidae (Buprestoidea). This family of three genera and seven species in California (*Nelson et al., 2008*) contains species of *Dystaxia* LeConte, 1866 and *Glyptoscelimorpha* Horn, 1893 present on the nearby mainland.

Byrrhidae (Byrrhoidea). This family of three subfamilies, eight genera, and 10 species in California (M. L. Gimmel, 2022, unpublished data), although primarily boreal and montane in distribution, contains a few taxa occurring along California's Central Coast south into Santa Barbara and Ventura counties (SBMNH specimen data).

Psephenidae (Dryopoidea). This family of three subfamilies and as many genera and species in California (*Shepard, 1993*) contains a widely distributed and abundant coastal

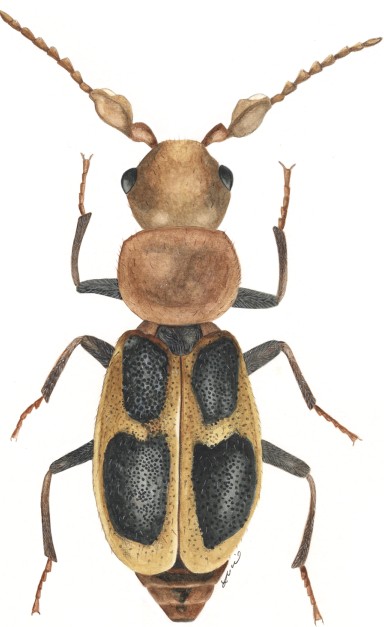

**Figure 5** *Collops crusoe* **Fall, 1910 (Melyridae).** Painting of endemic California Channel Islands species. Painting by Lucie Gimmel.

species, *Eubrianax edwardsii* (LeConte, 1874), that is conspicuously absent from the Channel Islands. Lack of suitable microhabitat may explain this absence.

Artematopodidae (Elateroidea). This family of two subfamilies, four genera, and six species in California (M. L. Gimmel, 2022, unpublished data) contains a species, *Brevipogon confusus* (Fall, 1901), widely distributed in the southern half of California (*Lawrence, 2005*).

Lycidae (Elateroidea). This family of two subfamilies, five genera, and seven species in California (M. L. Gimmel, 2022, unpublished data), contains at least a couple of species occurring in the Coast Ranges.

Omethidae (Elateroidea). This family of two subfamilies, five genera, and seven species in California (M. L. Gimmel, 2022, unpublished data), including at least one, *Ginglymocladus luteicollis* Van Dyke, 1918, occurring in coastal Santa Barbara County (SBMNH specimen data).

Georissidae (Hydrophiloidea). This family of one species in California, *Georissus californicus* LeConte, 1874 (*Hansen, 1999*), occurs at lower elevations within the Transverse Ranges (SBMNH specimen data).

Hydrochidae (Hydrophiloidea). This family of one genus, *Hydrochus* Leach, 1817, and four species in California (*Hansen, 1999*), has species that occur in the Coast Ranges (SBMNH specimen data).

Glaphyridae (Scarabaeoidea). This family of one genus, *Lichnanthe* Burmeister, 1844, and six species in California, contains species occurring in coastal portions of the state, including Santa Barbara, Ventura, and Los Angeles counties (*Carlson, 1980*).

Glaresidae (Scarabaeoidea). This family of one genus, *Glaresis* Erichson, 1848, and 13 species in California (*Gordon & Hanley, 2014*), has species that occur within the Transverse Ranges of California (SBMNH specimen data).

Lucanidae (Scarabaeoidea). This family consists of two subfamilies, four genera, and 18 species in California (M. L. Gimmel, 2022, unpublished data). Species of *Platycerus* Geoffroy, 1792 and *Sinodendron* Hellwig, 1894 occur in southern coastal California (SBMNH specimen data).

Ochodaeidae (Scarabaeoidea). This family of two subfamilies, four genera, and five species in California (*Paulsen, 2007*) has at least one species, *Parochodaeus californicus* (Horn, 1895), occurring in coastal southern California (SBMNH specimen data).

Pleocomidae (Scarabaeoidea). This family contains one genus, *Pleocoma* LeConte, 1856, and about 23 species in California (M. L. Gimmel, 2022, unpublished data), many of them occurring in coastal California.

Agyrtidae (Staphylinoidea). This family of three subfamilies, four genera, and seven species in California (*Newton, 1997*) contains species, notably *Necrophilus hydrophiloides* Guérin-Méneville, 1835, occurring in nearby coastal California (SBMNH specimen data).

Lophocateridae (Cleroidea). This family of three genera and five species in California (M. L. Gimmel, 2022, unpublished data) contains species of the genus *Eronyxa* Reitter, 1876 occurring in nearby coastal California (*Barron, 1971*; SBMNH specimen data).

Peltidae (Cleroidea). This family of one genus, *Peltis* Müller, 1764, and three species in California (*Barron, 1971*, as *Ostoma* Laicharting, 1781) contains species occurring at lower elevations within the Transverse Ranges (SBMNH specimen data).

Rhadalidae (Cleroidea). This family of two genera and three species in California (M. L. Gimmel, 2022, unpublished data) contains species of both *Rhadalus* LeConte, 1852 and *Semijlulistus* Schilsky, 1894 occurring at lower elevations within the Transverse Ranges (SBMNH specimen data).

Aderidae (Tenebrionoidea). This family of four genera and five species in California (M. L. Gimmel, 2022, unpublished data) contains species occurring in nearby coastal California (SBMNH specimen data).

Melandryidae (Tenebrionoidea). This family of two subfamilies, 12 genera, and 15 species in California (M. L. Gimmel, 2022, unpublished data) contains several genera and species in coastal southern regions of California (SBMNH specimen data). *Osphya lutea* (Horn, 1879) is a particularly abundant and well-collected species we expect might occur on the islands.

Pythidae (Tenebrionoidea). This family of three genera and three species in California (M. L. Gimmel, 2022, unpublished data) contains at least one species, *Sphalma quadricollis* Horn, 1888, occurring at lower elevations in the Transverse Ranges (SBMNH specimen data).

Ripiphoridae (Tenebrionoidea). This family of two genera and 20 species in California (*Linsley & MacSwain, 1951*; M. L. Gimmel, 2022, unpublished data) contains species occurring, but rarely collected, in coastal California (*Linsley & MacSwain, 1951*; SBMNH specimen data).

Tetratomidae (Tenebrionoidea). This family of four subfamilies, seven genera, and nine species in California (M. L. Gimmel, 2022, unpublished data) contains species occurring at lower elevations in the Transverse Ranges (SBMNH specimen data).
Anamorphidae (Coccinelloidea). This family contains a single, introduced species, *Symbiotes gibberosus* (Lucas, 1846), occurring in California (*Shockley, Tomaszewska & McHugh, 2009*), including at lower elevations across the southern coastal portion of the state (SBMNH specimen data).

Murmidiidae (Coccinelloidea). This family contains a single, introduced species in California, *Murmidius ovalis* (Beck, 1817), which is a cosmopolitan stored product associate (*Lawrence & Stephan, 1975*).

Mycetaeidae (Coccinelloidea). This family contains a single, introduced species in California, *Mycetaea subterranea* (Fabricius, 1801), which is a cosmopolitan species (*Shockley, Tomaszewska & McHugh, 2009*).

Sphindidae (Nitiduloidea). This family contains two subfamilies, two genera, and three species in California (M. L. Gimmel, 2022, unpublished data), at least one of which, *Sphindus crassulus* Casey, 1898, occurs in nearby coastal California (SBMNH specimen data).

Cucujidae (Cucujoidea). This family contains two genera and six species in California (M. L. Gimmel, 2022, unpublished data) and is widely distributed in forested areas of California (SBMNH specimen data).

Orsodacnidae (Chrysomeloidea). This family contains a single species in California, *Orsodacne atra* (Ahrens, 1810), which occurs on the nearby coastal mainland (SBMNH specimen data).

Anthribidae (Curculionoidea). This family of two subfamilies, five genera, and 11 species in California (M. L. Gimmel, 2022, unpublished data) contains at least two species occurring on the nearby coastal mainland (SBMNH specimen data).

Cimberididae (Curculionoidea). This family of four genera and 10 species in California (*Kuschel, 1989*, as Nemonychidae) contains species occurring at lower elevations in the Transverse Ranges (SBMNH specimen data). Their life histories are closely tied with *Pinus* species (Pinaceae); consequently, they should be searched for during spring in the pine groves occurring on the Channel Islands.

## ANNOTATED CHECKLIST

The format of this annotated checklist is structured to provide a foundation for future research on the taxa included. Higher classification is arranged phylogenetically by suborder and superfamily, adopting the higher groupings of *Cai et al. (2022)*. Taxa of family rank and lower are arranged alphabetically within higher taxa. Notes on the taxonomy and diversity within California are given for all higher taxa. The standardized sections for each taxon are briefly defined below. We did not attempt to provide an overview of the biology of the taxa in the checklist; this information can be gleaned from general works on beetle biology and as well as taxon-specific references cited herein.

In order to keep close accounting of the actual number of unique taxa known from the islands, the following system of presentation is employed: (1) Most family-level digital and literature records are excluded, except for select groups presenting taxonomic challenges (*e.g.*, Apioninae), and for certain notable higher-taxon literature records, which are included in the respective Notes field; (2) Island records of supraspecific taxa that have no

records at lower levels (*e.g.*, a genus record with no identified species reported from the islands) are treated with a unique header as "(Taxon) undetermined species" to identify it as a unique taxon within the checklist; (3) When a genus-level taxon *does* have lower-level representation in our list, then any records determined only to the higher level are included merely under that higher-level heading and not listed as an additional taxon.

**Nomenclatural Authority:** This field contains a reference, or set of references, from which the valid name and taxon authorship were derived for this list. This is typically a recent revision or catalog and is intended to validate the use of the name here and serve as an anchor and starting point for future use of this list as taxonomic names and concepts continue to change (see *Johnston, Aalbu & Franz, 2018*).

**Literature Records:** In this field, every published island record for the taxon is cited with a page number. In the Notes field under the taxon we provide additional comments on such records, including previous nomenclatural combinations used in cited works, discounted or ambiguous references, *etc.*

**Digitized Records:** In this field, all digitized specimen records included in our final dataset are tallied by island for each taxon, and are listed by collection. For detailed information on each specimen record see the recordset discussed and referenced above.

**Range:** This field denotes whether the taxon is known only from the Channel Islands or also from the mainland. References are given to support the claim of endemicity (only known from the Channel Islands) or presence on the mainland. Full ranges of species are not included or covered in this checklist.

**Notes:** This field contains a wide variety of information about the taxon. For entries within supraspecific taxa, we include information on diversity in California, and discuss recent taxonomic revisions or catalogs. For species-level entries, we include any information about subspecies classification of island specimens. Any additional information deemed relevant, including island-specific natural history notes, as well as decisions on taxon validity and discrepancies or issues relating to literature or digitized specimens, are included under this section.

## ADEPHAGA

### Carabidae

Notes. There are 103 genera and 647 species of this family known from California, placed in 38 tribes (*Bousquet, 2012*; M. L. Gimmel, 2022, unpublished data). A subfamily classification is not widely agreed upon in the literature; we use only tribes below, following the arrangement of *Bousquet (2012)*, with the exclusion of Cicindelidae, which has been recently recognized at the family level (*e.g.*, *Cai et al., 2022*). *Bousquet (2012)* provided an extensively annotated distributional catalog and bibliography for the North American taxa.

### Bembidiini

Notes. Thirteen genera and 148 species of Bembidiini are known to occur in California (*Bousquet, 2012*; M. L. Gimmel, 2022, unpublished data).

***Bembidion*** **Latreille, 1802**

Nomenclatural Authority: *Bousquet (2012)*

Digitized Records (genus-only): San Clemente (1 EMEC; 2 LACM; 8 SBMNH), San Miguel (8 LACM; 14 SBMNH), San Nicolas (4 LACM; 28 SBMNH), Santa Catalina (1 LACM; 23 SBMNH), Santa Cruz (18 LACM; 105 SBMNH), Santa Rosa (9 LACM; 18 SBMNH). Notes. This genus was widely known in the earlier literature as *Bembidium*. Fully 115 species have been reported to occur in California (M. L. Gimmel, 2022, unpublished data). No complete keys exist for North American or Californian species, but *Lindroth's (1963)* key included about 75% of the North American fauna (*Bousquet, 2012*). A record of *Bembidion* (*Hirmoplataphus*) *recticolle* LeConte, 1863 from Santa Cruz Island provided to the California Beetle Database was deemed unverifiable and needs to be substantiated.

***Bembidion*** **(*Furcacampa*) *versicolor*** **(LeConte, 1847)**

Nomenclatural Authority: *Bousquet (2012)*

Literature Records: San Clemente (*Cockerell, 1940*: 284)

Digitized Records: none

Range: Also known from mainland (*Bousquet, 2012*).

***Bembidion*** **(*Lymneops*) *laticeps*** **(LeConte, 1858)**

Nomenclatural Authority: *Bousquet (2012)*

Literature Records: none

Digitized Records: San Clemente (1 SBMNH)

Range: Also known from mainland (*Bousquet, 2012*). This species was transferred from the subgenus *Lymnaeum* Stephens, 1828 to the subgenus *Lymneops* Casey, 1918 by *Maddison & Maruyama (2019)*.

***Bembidion*** **(*Lymneops*) *palosverdes*** **Kavanaugh & Erwin, 1992**

Nomenclatural Authority: *Bousquet (2012)*

Literature Records: Santa Catalina (*Caterino, Caterino & Maddison, 2015*: 410; *Maddison & Maruyama, 2019*: 46)

Digitized Records: Santa Catalina (2 SBMNH)

Range: Also known from mainland, but possibly extinct there (*Caterino, Caterino & Maddison, 2015*).

Notes. This species was thought to be extinct since its original description from the Palos Verdes Peninsula in greater Los Angeles, but was rediscovered by M. and K. Caterino on Santa Catalina in 2010 (*Caterino, Caterino & Maddison, 2015*). This species was transferred from the subgenus *Cillenus* Samouelle, 1819 to the subgenus *Lymneops* Casey, 1918 by *Maddison & Maruyama (2019)*.

***Bembidion*** **(*Notaphus*) *indistinctum*** **Dejean, 1831**

Nomenclatural Authority: *Bousquet (2012)*

Literature Records: San Nicolas (*Cockerell, 1940*: 285), Santa Rosa (*Fall, 1897*: 236)

Digitized Records: none

Range: Also known from mainland (*Bousquet, 2012*).

Notes. Reported from the "Channel Islands" by *Bousquet (2012*: 613).

### Bembidion (Notaphus) insulatum (LeConte, 1852)

Nomenclatural Authority: *Bousquet (2012)*

Literature Records: San Clemente (*Cockerell, 1940*: 284)

Digitized Records: none

Range: Also known from mainland (*Bousquet, 2012*).

### Bembidion (Peryphanes) platynoides Hayward, 1897

Nomenclatural Authority: *Bousquet (2012)*

Literature Records: Santa Rosa (*Fall, 1897*: 236)

Digitized Records: none

Range: Also known from mainland (*Bousquet, 2012*).

Notes. Reported from the "Channel Islands" by *Bousquet (2012*: 579).

### Bembidion (Peryphodes) ephippigerum (LeConte, 1852)

Nomenclatural Authority: *Bousquet (2012)*

Literature Records: Santa Catalina (*Fall, 1897*: 236)

Digitized Records: San Nicolas (9 SBMNH)

Range: Also known from mainland (*Bousquet, 2012*).

Notes. Reported from the "Channel Islands" by *Bousquet (2012*: 629).

### Bembidion (Peryphus) corgenoma Maddison, 2020

Nomenclatural Authority: *Maddison (2020)*

Literature Records: Santa Cruz (*LeConte, 1876*: 298; *Fall, 1897*: 236; *Fall & Davis, 1934*: 143), Santa Rosa (*Fall, 1897*: 236)

Digitized Records: San Miguel (1 SBMNH), Santa Cruz (19 SBMNH), Santa Rosa (10 SBMNH)

Range: Also known from mainland (*Maddison, 2020*).

Notes. *LeConte (1876)* recorded this species as *Bembidium mannerheimii* (LeConte, 1852), subsequently considered a junior synonym of *Bembidion transversale* Dejean, 1831 (see *Bousquet, 2012*). *Fall (1897)* and *Fall & Davis (1934)* reported this species as *B. transversale*. *Maddison (2020)*, however, concluded based on morphology that the Pacific coast species in the *transversale* species group represented a new species and provided a revised version of the relevant portion of the key in *Lindroth (1963)*.

### Bembidion (Peryphus) striola (LeConte, 1852)

Nomenclatural Authority: *Bousquet (2012)*

Literature Records: San Clemente (*Cockerell, 1940*: 285), Santa Catalina (*Fall, 1897*: 236)

Digitized Records: none

Range: Also known from mainland (*Bousquet, 2012*).

Notes. Reported from the "Channel Islands" by *Bousquet (2012*: 564).

***Bembidion*** (***Trechonepha***) ***iridescens*** **(LeConte, 1852)**
Nomenclatural Authority: *Bousquet (2012)*
Literature Records: Santa Catalina (*Fall, 1897*: 236)
Digitized Records: none
Range: Also known from mainland (*Bousquet, 2012*).
Notes. Reported from the "Channel Islands" by *Bousquet (2012*: 654).

***Elaphropus*** **Motschulsky, 1839**
Nomenclatural Authority: *Bousquet (2012)*
Notes. Seven species of this genus have been reported from California (*Bousquet, 2012*).
The native North American members of *Elaphropus* are in need of revision (*Bousquet, 2012*).

***Elaphropus*** **undetermined species**
Literature Records: none
Digitized Records: Santa Cruz (10 SBMNH)

***Phrypeus*** **Casey, 1924**
Nomenclatural Authority: *Bousquet (2012)*
Notes. Only one species of *Phrypeus* occurs in North America (*Bousquet, 2012*).

***Phrypeus rickseckeri*** **(Hayward, 1897)**
Nomenclatural Authority: *Bousquet (2012)*
Literature Records: none
Digitized Records: Santa Cruz (6 SBMNH)
Range: Also known from mainland (*Bousquet, 2012*).

***Tachys*** **Dejean, 1821**
Nomenclatural Authority: *Bousquet (2012)*
Notes. Six species of *Tachys* have been reported from California (*Bousquet, 2012*).
The subgenus *Paratachys* Casey, 1918, with two species in California, has frequently been treated as a separate genus (*Bousquet, 2012*). Both this and the subgenus *Tachys* need revision (*Bousquet, 2012*).

***Tachys*** (***Paratachys***) ***vorax*** **LeConte, 1852**
Nomenclatural Authority: *Bousquet (2012)*
Literature Records: none
Digitized Records: Santa Cruz (1 SBMNH)
Range: Also known from mainland (*Bousquet, 2012*).

***Tachys*** (***Tachys***) ***corax*** **LeConte, 1852**
Nomenclatural Authority: *Bousquet (2012)*
Literature Records: San Clemente (*Fall, 1897*: 236)
Digitized Records: San Nicolas (5 SBMNH)
Range: Also known from mainland (*Bousquet, 2012*).

Notes. *Fall (1897*: 239) indicated that the "two specimens from San Clemente are closely allied to *corax*, Lec., but seem distinct by the obviously less transverse thorax." This is likely identical to what we have identified as *T. corax*, and we have included these records together above.

### *Tachys* (*Tachys*) *vittiger* LeConte, 1852
Nomenclatural Authority: *Bousquet (2012)*
Literature Records: Santa Catalina (*Fall, 1897*: 236; *Fall, 1901*: 43; *Bousquet, 2012*: 689)
Digitized Records: none
Range: Also known from mainland (*Fall, 1901*; *Bousquet, 2012*).

### Brachinini
Notes. One genus and 12 species of Brachinini occur in California (*Bousquet, 2012*).

### *Brachinus* Weber, 1801
Nomenclatural Authority: *Bousquet (2012)*
Digitized Records (genus-only): Santa Catalina (1 CASC), Santa Cruz (50 CASC; 31 EMEC; 50 SBMNH; 11 UCSB; 35 UASM), Santa Rosa (5 SBMNH)
Notes. There are 12 species of *Brachinus* reported from California (*Bousquet, 2012*), which were treated by *Erwin (1965*, *1970*). This genus appeared in some early literature as *Brachynus*.

### *Brachinus costipennis* Motschulsky, 1859
Nomenclatural Authority: *Bousquet (2012)*
Literature Records: Santa Catalina (*Fall, 1897*: 236), Santa Cruz (*Erwin, 1965*: 6; *Erwin, 1970*: 88)
Digitized Records: Santa Cruz (1 BYUC; 7 CASC; 4 LACM; 4 SBMNH; 1 iNat), Santa Rosa (15 SBMNH)
Range: Also known from mainland (*Erwin, 1965*, *1970*).
Notes. This species was reported as *Brachinus carinulatus* Motschulsky, 1859 by *Fall (1897)*, which was synonymized with *B. costipennis* by *Erwin (1965)*.

### *Brachinus gebhardis* Erwin, 1965
Nomenclatural Authority: *Bousquet (2012)*
Literature Records: Santa Cruz (*Erwin, 1965*: 7; *Erwin, 1970*: 132)
Digitized Records: Santa Catalina (6 LACM), Santa Cruz (1 CASC; 23 LACM; 9 SBMNH), Santa Rosa (10 SBMNH)
Range: Also known from mainland (*Erwin, 1965*, *1970*).

### *Brachinus mexicanus* Dejean, 1831
Nomenclatural Authority: *Bousquet (2012)*
Literature Records: Santa Cruz (*Erwin, 1965*: 11; *Erwin, 1970*: 107)
Digitized Records: Santa Cruz (40 CASC)
Range: Also known from mainland (*Erwin, 1965*, *1970*).

Notes. This species was recorded as *Brachinus fidelis* LeConte, 1863 by *Erwin (1965)*, which was synonymized with *B. mexicanus* by *Erwin (1970)*.

### *Brachinus quadripennis* Dejean, 1825
Nomenclatural Authority: *Bousquet (2012)*
Literature Records: Santa Cruz (*Fall & Davis, 1934*: 144)
Digitized Records: none
Range: Also known from mainland (*Erwin, 1970*).
Notes. *Fall & Davis (1934)* recorded this species as *Brachynus tschernikhii* Mannerheim, 1843, which was synonymized with *B. quadripennis* by *Erwin (1970*: 99).

### Carabini
Notes. Two genera and 28 species of Carabini are known to occur in California (*Bousquet, 2012*).

### *Calosoma* Weber, 1801
Nomenclatural Authority: *Bousquet (2012)*
Notes. The North American species of *Calosoma* were revised by *Gidaspow (1959)*. Twenty-six species have been reported as occurring in California (*Bousquet, 2012*).

### *Calosoma* (*Camegonia*) *parvicolle* Fall, 1910
Nomenclatural Authority: *Bousquet (2012)*
Literature Records: Santa Catalina (*Gidaspow, 1959*: 256)
Digitized Records: none
Range: Also known from mainland (*Gidaspow, 1959*; *Bousquet, 2012*).

### *Calosoma* (*Carabosoma*) *eremicola* Fall, 1910
Nomenclatural Authority: *Bousquet (2012)*
Literature Records: San Clemente (*Fall, 1910*: 91; *Cockerell, 1940*: 284; *Jeannel, 1940*: 206; *Gidaspow, 1959*: 259; *Bousquet, 2012*: 235), Santa Catalina (*Cockerell, 1940*: 284; *Gidaspow, 1959*: 259)
Digitized Records: San Clemente (1 CASC; 18 LACM; 5 SBMNH), Santa Catalina (2 CASC; 1 SBMNH)
Range: Also known from mainland (*Cockerell, 1940*; *Jeannel, 1940*; *Gidaspow, 1959*; *Bousquet, 2012*).
Notes. This species was synonymized with *Calosoma glabratum sponsum* Casey, 1897 by *Breuning (1928*: 103), then re-validated by *Jeannel (1940*: 206). *Calosoma eremicola* was originally described as endemic to San Clemente Island by *Fall (1910)*.

### *Calosoma* (*Chrysostigma*) *semilaeve* LeConte, 1852
Nomenclatural Authority: *Bousquet (2012)*
Literature Records: San Nicolas (*Miller & Miller, 1985*: 123), Santa Barbara (*Miller & Miller, 1985*: 123), Santa Catalina (*Miller & Miller, 1985*: 123), Santa Rosa (*Fall, 1897*: 236; *Miller & Miller, 1985*: 123)
Digitized Records: Santa Catalina (5 LACM), Santa Cruz (2 SBMNH; 1 UCSB)

Range: Also known from mainland (*Gidaspow, 1959*; *Bousquet, 2012*).

Notes. Recorded from the "Channel Islands" by *Bousquet (2012)*: 243).

### Chlaeniini

Notes. One genus and 13 species of Chlaeniini are known from California (*Bousquet, 2012*).

### *Chlaenius* Brullé, 1834

Nomenclatural Authority: *Bousquet (2012)*

Digitized Records (genus-only): Santa Cruz (4 UCSB; 13 UASM)

Notes. Thirteen species of *Chlaenius* have been reported to occur in California (*Bousquet, 2012*). The genus was revised for North America by *Bell (1960)*.

### *Chlaenius* (*Chlaeniellus*) *obsoletus* LeConte, 1851

Nomenclatural Authority: *Bousquet (2012)*

Literature Records: Santa Catalina (*Fall, 1897*: 236; *Bell, 1960*: 150)

Digitized Records: Santa Cruz (3 CASC; 1 SBMNH)

Range: Also known from mainland (*Bell, 1960*; *Bousquet, 2012*).

Notes. Reported from the "Channel Islands" by *Bousquet (2012)*: 984).

### *Chlaenius* (*Chlaeniellus*) *tricolor* Dejean, 1826

Nomenclatural Authority: *Bousquet (2012)*

Literature Records: none

Digitized Records: Santa Cruz (1 SBMNH)

Range: Also known from mainland (*Bell, 1960*; *Bousquet, 2012*).

Notes. All *C. tricolor* from California belong to the subspecies *C. t. vigilans* Say, 1830 (*Bell, 1960*; *Bousquet, 2012*).

### *Chlaenius* (*Chlaeniellus*) *variabilipes* Eschscholtz, 1833

Nomenclatural Authority: *Bousquet (2012)*

Literature Records: Santa Cruz (*Bell, 1960*: 150)

Digitized Records: Santa Catalina (1 LACM), Santa Cruz (4 LACM; 2 SBMNH), Santa Rosa (2 SBMNH)

Range: Also known from mainland (*Bell, 1960*; *Bousquet, 2012*).

### *Chlaenius* (*Chlaenius*) *cumatilis* LeConte, 1851

Nomenclatural Authority: *Bousquet (2012)*

Literature Records: Santa Cruz (*Fall & Davis, 1934*: 144)

Digitized Records: Santa Cruz (52 CASC; 11 LACM; 19 SBMNH; 7 TAMU), Santa Rosa (6 SBMNH)

Range: Also known from mainland (*Bell, 1960*; *Bousquet, 2012*).

### Clivinini

Notes. Three genera and 11 species of Clivinini are known to occur in California (*Bousquet, 2012*).

***Schizogenius*** **Putzeys, 1846**
Nomenclatural Authority: *Bousquet (2012)*
Digitized Records (genus-only): Santa Cruz (5 CASC; 14 SBMNH)
Notes. Seven species of *Schizogenius* have been recorded from California (*Bousquet, 2012*), belonging to two subgenera, *Genioschizus* Whitehead, 1972 and *Schizogenius* (*s.str.*). *Whitehead (1972)* revised the North American species.

***Schizogenius*** (***Schizogenius***) ***depressus*** **LeConte, 1852**
Nomenclatural Authority: *Bousquet (2012)*
Literature Records: Santa Rosa (*Fall, 1897*: 236; *Whitehead, 1972*: 294)
Digitized Records: Santa Cruz (2 CASC)
Range: Also known from mainland (*Whitehead, 1972*; *Bousquet, 2012*).
Notes. Reported from the "Channel Islands" by *Bousquet (2012*: 419).

**Cychrini**
Notes. One genus and 18 species of Cychrini are known to occur in California (*Bousquet, 2012*).

***Scaphinotus*** **Dejean, 1826**
Nomenclatural Authority: *Bousquet (2012)*
Digitized Records (genus-only): Santa Catalina (3 EMEC), Santa Cruz (2 EMEC; 1 UASM), Santa Rosa (1 EMEC)
Notes. Eighteen species of *Scaphinotus* are known from California (*Bousquet, 2012*). Most of these species belong to the subgenus *Brennus* Motschulsky, 1866, which was revised by *Gidaspow (1968)*.

***Scaphinotus*** (***Brennus***) ***crenatus*** **(Motschulsky, 1859)**
Nomenclatural Authority: *Bousquet (2012)*
Literature Records: none
Digitized Records: San Miguel (1 CASC; 1 SBMNH), Santa Catalina (1 CASC), Santa Cruz (8 CASC; 1 LACM; 4 SBMNH), Santa Rosa (1 CASC; 9 SBMNH)
Range: Also known from mainland (*Gidaspow, 1968*).

***Scaphinotus*** (***Brennus***) ***punctatus*** **(LeConte, 1859)**
Nomenclatural Authority: *Bousquet (2012)*
Literature Records: Santa Catalina (*Fall, 1897*: 236; *Baker, 1905*: 59; *Gidaspow, 1968*: 167; *Bousquet, 2012*: 223)
Digitized Records: Santa Catalina (5 CASC)
Range: Also known from mainland (*Gidaspow, 1968*).
Notes. This species was previously recorded as *Cychrus mimus* Horn, 1874 by *Fall (1897)* and *Baker (1905*: "*Cuchrus mimus*"), which was later synonymized with *S. punctatus* (see *Gidaspow, 1968*).

***Scaphinotus*** (***Brennus***) ***ventricosus*** **(Dejean, 1831)**
Nomenclatural Authority: *Bousquet (2012)*

Literature Records: Santa Catalina (*Gidaspow, 1968*: 171)
Digitized Records: Santa Catalina (1 CASC)
Range: Also known from mainland (*Gidaspow, 1968*).

**Dyschiriini**
Notes. Two genera and 21 species of Dyschiriini are known to occur in California (*Bousquet, 2012*).

*Akephorus* **LeConte, 1852**
Nomenclatural Authority: *Bousquet (2012)*
Notes. Two species of this genus occur in California (*Bousquet, 2012*). They were keyed (as species of *Dyschirius*) by *Bousquet (1988)*.

*Akephorus marinus* **LeConte, 1852**
Nomenclatural Authority: *Bousquet (2012)*
Literature Records: Santa Rosa (*Fall, 1897*: 236)
Digitized Records: San Miguel (33 LACM; 5 SBMNH; 18 CASC), San Nicolas (1 LACM; 5 SBMNH), Santa Cruz (34 LACM; 18 SBMNH; 8 CASC), Santa Rosa (7 SBMNH)
Range: Also known from mainland (*Bousquet, 2012*).
Notes. Reported from the "Channel Islands" by *Bousquet (2012*: 431).

*Dyschirius* **Bonelli, 1810**
Nomenclatural Authority: *Bousquet (2012)*
Digitized Records (genus-only): Santa Cruz (12 CASC)
Notes. Nineteen species of *Dyschirius* have been recorded from California (*Bousquet, 2012*). A key to most North American species of this genus was provided by *Bousquet (1988)*.

*Dyschirius aratus* **LeConte, 1852**
Nomenclatural Authority: *Bousquet (2012)*
Literature Records: none
Digitized Records: Santa Rosa (7 SBMNH)
Range: Also known from mainland (*Bousquet, 2012*).

*Dyschirius consobrinus* **LeConte, 1852**
Nomenclatural Authority: *Bousquet (2012)*
Literature Records: none
Digitized Records: Santa Rosa (4 SBMNH)
Range: Also known from mainland (*Bousquet, 2012*).

*Dyschirius gibbipennis* **LeConte, 1857**
Nomenclatural Authority: *Bousquet (2012)*
Literature Records: Santa Rosa (*Fall, 1897*: 236)
Digitized Records: San Miguel (1 CASC), Santa Cruz (2 SBMNH), Santa Rosa (6 SBMNH)
Range: Also known from mainland (*Bousquet, 2012*).
Notes. Reported from the "Channel Islands" by *Bousquet (2012*: 439).

*Dyschirius varidens* Fall, 1910
Nomenclatural Authority: *Bousquet (2012)*
Literature Records: none
Digitized Records: Santa Cruz (2 SBMNH)
Range: Also known from mainland (*Bousquet, 2012*).

**Harpalini**
Notes. Thirteen genera and 78 species of Harpalini are known to occur in California (*Bousquet, 2012*; M. L. Gimmel, 2022, unpublished data).

*Anisodactylus* Dejean, 1829
Nomenclatural Authority: *Bousquet (2012)*
Digitized Records (genus-only): San Miguel (1 LACM), Santa Cruz (19 UASM)
Notes. Thirteen species of *Anisodactylus* are known to occur in California (*Bousquet, 2012*). *Noonan (1996)* revised the subgenus *Anisodactylus* (*Anisodactylus*), the only subgenus known to occur in the Channel Islands.

*Anisodactylus* (*Anisodactylus*) *californicus* Dejean, 1829
Nomenclatural Authority: *Bousquet (2012)*
Literature Records: San Miguel (*Noonan, 1996*: 126), Santa Catalina (*Fall, 1897*: 236), Santa Cruz (*Noonan, 1996*: 126), Santa Rosa (*Fall, 1897*: 236)
Digitized Records: San Clemente (7 SBMNH), San Miguel (1 CASC; 2 SBMNH), San Nicolas (8 SBMNH), Santa Catalina (1 SBMNH), Santa Cruz (3 CASC), Santa Rosa (5 SBMNH)
Range: Also known from mainland (*Noonan, 1996*; *Bousquet, 2012*).
Notes. Reported from the "Channel Islands" by *Bousquet (2012*: 1022).

*Anisodactylus* (*Anisodactylus*) *consobrinus* LeConte, 1851
Nomenclatural Authority: *Bousquet (2012)*
Literature Records: Santa Catalina (*Fall, 1897*: 236), Santa Cruz (*LeConte, 1876*: 298; *Fall, 1897*: 236; *Fall & Davis, 1934*: 144; *Noonan, 1996*: 107), Santa Rosa (*Fall, 1897*: 236)
Digitized Records: Santa Cruz (3 SBMNH), Santa Rosa (8 SBMNH)
Range: Also known from mainland (*Noonan, 1996*; *Bousquet, 2012*).
Notes. Reported from the "Channel Islands" by *Bousquet (2012*: 1021).

*Anisodactylus* (*Anisodactylus*) *similis* LeConte, 1851
Nomenclatural Authority: *Bousquet (2012)*
Literature Records: Santa Cruz (*Fall & Davis, 1934*: 144)
Digitized Records: Santa Cruz (2 SBMNH), Santa Rosa (2 SBMNH)
Range: Also known from mainland (*Noonan, 1996*; *Bousquet, 2012*).
Notes. This species was recorded as *Anisodactylus semipunctatus* LeConte, 1859, a current junior synonym of *A. similis*, by *Fall & Davis (1934)*. Reported from the "Channel Islands" by *Bousquet (2012*: 1023).

*Bradycellus* Erichson, 1837
Nomenclatural Authority: *Bousquet (2012)*

Digitized Records (genus-only): San Clemente (4 SBMNH), San Miguel (3 CASC), San Nicolas (3 LACM), Santa Cruz (5 CASC; 2 EMEC; 1 SBMNH; 3 UASM)

Notes. Four subgenera and 25 species of *Bradycellus* are known to occur in California (*Bousquet, 2012*). The two most species-rich subgenera in California, *Liocellus* Motschulsky, 1864 and *Stenocellus* Casey, 1914, both need revision (*Bousquet, 2012*).

### *Bradycellus* (*Liocellus*) *nitidus* (Dejean, 1829)

Nomenclatural Authority: *Bousquet (2012)*

Literature Records: Santa Catalina (*Fall, 1897*: 236), Santa Cruz (*Fall & Davis, 1934*: 144), Santa Rosa (*Fall, 1897*: 236)

Digitized Records: San Miguel (2 CASC; 2 SBMNH), San Nicolas (15 LACM; 11 SBMNH), Santa Catalina (1 SBMNH), Santa Cruz (4 CASC; 10 SBMNH; 1 UCSB), Santa Rosa (1 LACM; 20 SBMNH)

Range: Also known from mainland (*Bousquet, 2012*).

Notes. *Fall (1897)* recorded this species as *Tachycellus nitidus*; *Fall & Davis (1934)* recorded this species as *Glycerius nitidus*. Reported from the "Channel Islands" by *Bousquet (2012*: 1061).

### *Bradycellus* (*Stenocellus*) *californicus* (LeConte, 1857)

Nomenclatural Authority: *Bousquet (2012)*

Literature Records: Santa Rosa (*Fall, 1897*: 236)

Digitized Records: San Clemente (16 SBMNH), San Miguel (1 SBMNH), San Nicolas (17 SBMNH), Santa Catalina (4 SBMNH), Santa Cruz (8 SBMNH), Santa Rosa (12 SBMNH)

Range: Also known from mainland (*Bousquet, 2012*).

Notes. Reported from the "Channel Islands" by *Bousquet (2012*: 1068).

### *Bradycellus* (*Stenocellus*) *rupestris* (Say, 1823)

Nomenclatural Authority: *Bousquet (2012)*

Literature Records: Santa Catalina (*Fall, 1897*: 236)

Digitized Records: San Clemente (6 SBMNH), San Nicolas (12 SBMNH), Santa Cruz (3 SBMNH), Santa Rosa (1 SBMNH)

Range: Also known from mainland (*Bousquet, 2012*).

### *Bradycellus* (*Stenocellus*) *sejunctus* (Casey, 1914)

Nomenclatural Authority: *Bousquet (2012)*

Literature Records: none

Digitized Records: San Clemente (14 EMEC)

Range: Also known from mainland (*Bousquet, 2012*).

### *Dicheirus* Mannerheim, 1843

Nomenclatural Authority: *Bousquet (2012)*

Digitized Records (genus-only): Santa Cruz (3 CASC)

Notes. Five species of *Dicheirus* are known to occur in California (*Bousquet, 2012*). They were revised and keyed by *Noonan (1968)*.

***Dicheirus dilatatus*** (Dejean, 1829)

Nomenclatural Authority: *Bousquet (2012)*

Literature Records: San Clemente (*Casey, 1914*: 201; *Noonan, 1968*: 298; *Noonan, 1975*: 7 [map]), Santa Catalina (*Noonan, 1968*: 298), Santa Cruz (*Fall & Davis, 1934*: 144), Santa Rosa (*Fall, 1897*: 236)

Digitized Records: San Clemente (29 LACM; 7 SBMNH), San Miguel (2 SBMNH), Santa Catalina (10 LACM; 5 SBMNH), Santa Cruz (2 SBMNH), Santa Rosa (10 SBMNH)

Range: Also known from mainland (*Noonan, 1975*; *Bousquet, 2012*).

Notes. This species was recorded as *Anisodactylus dilatatus* by *Fall (1897)*, and as *Dicheirus australinus* Casey, 1914 by *Casey (1914)*. The latter was synonymized with *D. dilatatus* by *Noonan (1968)*. All island records of *D. dilatatus* refer to the nominate subspecies, *D. d. dilatatus* (Dejean, 1829) (*Noonan, 1968*). Reported from the "Channel Islands" by *Bousquet (2012*: 1042).

***Dicheirus piceus*** (Ménétriés, 1843)

Nomenclatural Authority: *Bousquet (2012)*

Literature Records: San Clemente (*Noonan, 1968*: 290; *Noonan, 1975*: 7 [map]), Santa Catalina (*Noonan, 1968*: 290; *Noonan, 1975*: 7 [map]), Santa Rosa (*Fall, 1897*: 236)

Digitized Records: Anacapa (3 SBMNH), San Clemente (11 CASC; 5 LACM; 3 SBMNH), Santa Catalina (26 LACM; 6 SBMNH), Santa Cruz (4 CASC; 18 SBMNH), Santa Rosa (9 SBMNH)

Range: Also known from mainland (*Noonan, 1975*; *Bousquet, 2012*).

Notes. This species was recorded as *Anisodactylus piceus* by *Fall (1897)*. Reported from the "Channel Islands" by *Bousquet (2012*: 1043).

***Harpalus*** Latreille, 1802

Nomenclatural Authority: *Bousquet (2012)*

Digitized Records (genus-only): Anacapa (1 ASUHIC)

Notes. Thirteen species of *Harpalus* have been reported from California (*Bousquet, 2012*; M. L. Gimmel, 2022, unpublished data). Most species of the genus were treated for North America by *Noonan (1991)*, with treatments of additional subgenera by *Ball & Anderson (1962)* and *Will (1997)*.

***Harpalus*** (***Megapangus***) ***caliginosus*** (Fabricius, 1775)

Nomenclatural Authority: *Bousquet (2012)*

Literature Records: none

Digitized Records: Santa Cruz (1 SBMNH), Santa Rosa (1 SBMNH)

Range: Also known from mainland (*Will, 1997*).

Notes. *Will (1997)* provided a taxonomic review of the subgenus *Harpalus* (*Megapangus*), including a shaded range map showing *H. caliginosus* ranging partially into the Channel Islands (*Will, 1997*: 46); however, no Channel Island records were given.

***Harpalus*** (***Pseudoophonus***) ***pensylvanicus*** (DeGeer, 1774)

Nomenclatural Authority: *Bousquet (2012)*

Literature Records: none

Digitized Records: Santa Cruz (1 UCSB)

Range: Also known from mainland (*Ball & Anderson, 1962*; *Bousquet, 2012*).

Notes. *Ball & Anderson (1962)* provided a key to the species of the subgenus *Pseudoophonus* Motschulsky, 1844 (as *Pseudophonus*). Only one species of the subgenus has been recorded from California (*Bousquet, 2012*).

### *Stenolophus* Dejean, 1821

Nomenclatural Authority: *Bousquet (2012)*

Digitized Records (genus-only): Santa Cruz (2 UASM)

Notes. Eleven species of *Stenolophus* occur in California, four in subgenus *Agonoderus* Dejean, 1829 and seven in subgenus *Stenolophus* (*Bousquet, 2012*). *Lindroth's (1968)* key treated all members of the latter, but a revision of the former is needed (*Bousquet, 2012*).

### *Stenolophus* (*Agonoderus*) *lineola* (Fabricius, 1775)

Nomenclatural Authority: *Bousquet (2012)*

Literature Records: Santa Rosa (*Fall, 1897*: 236)

Digitized Records: Anacapa (2 LACM), Santa Cruz (2 LACM), Santa Rosa (1 LACM)

Range: Also known from mainland (*Bousquet, 2012*).

Notes. *Fall (1897)* recorded this species as *Agonoderus lineola*.

### *Stenolophus* (*Agonoderus*) *rugicollis* (LeConte, 1859)

Nomenclatural Authority: *Bousquet (2012)*

Literature Records: none

Digitized Records: San Miguel (1 LACM)

Range: Also known from mainland (*Bousquet, 2012*).

### *Stenolophus* (*Stenolophus*) *anceps* LeConte, 1857

Nomenclatural Authority: *Bousquet (2012)*

Literature Records: none

Digitized Records: San Miguel (2 SBMNH), Santa Cruz (7 SBMNH), Santa Rosa (2 SBMNH)

Range: Also known from mainland (*Bousquet, 2012*).

### *Stenolophus* (*Stenolophus*) *flavipes* LeConte, 1858

Nomenclatural Authority: *Bousquet (2012)*

Literature Records: none

Digitized Records: Santa Catalina (1 SBMNH), Santa Cruz (1 LACM; 5 SBMNH), Santa Rosa (14 SBMNH)

Range: Also known from mainland (*Bousquet, 2012*).

### *Stenolophus* (*Stenolophus*) *limbalis* LeConte, 1857

Nomenclatural Authority: *Bousquet (2012)*

Literature Records: Santa Catalina (*Fall, 1897*: 236)

Digitized Records: none

Range: Also known from mainland (*Bousquet, 2012*).

Notes. Reported from the "Channel Islands" by *Bousquet (2012*: 1050).

### *Stenolophus* (*Stenolophus*) *ochropezus* (Say, 1823)
Nomenclatural Authority: *Bousquet (2012)*
Literature Records: none
Digitized Records: Santa Cruz (11 EMEC)
Range: Also known from mainland (*Bousquet, 2012*).

### Lachnophorini
Notes. Three genera and three species of Lachnophorini are known to occur in California (*Bousquet, 2012*).

### *Lachnophorus* Dejean, 1831
Nomenclatural Authority: *Bousquet (2012)*
Notes. Only one species of this genus occurs in North America (*Bousquet, 2012*).

### *Lachnophorus elegantulus* Mannerheim, 1843
Nomenclatural Authority: *Bousquet (2012)*
Literature Records: none
Digitized Records: Santa Cruz (2 CASC; 1 LACM; 1 SBMNH)
Range: Also known from mainland (*Bousquet, 2012*).

### Lebiini
Notes. Fourteen genera and 39 species of Lebiini are known to occur in California (*Bousquet, 2012*; M. L. Gimmel, 2022, unpublished data).

### *Apristus* Chaudoir, 1846
Nomenclatural Authority: *Bousquet (2012)*
Digitized Records (genus-only): Santa Cruz (1 SBMNH), Santa Rosa (4 LACM)
Notes. Seven species of *Apristus* are known from California (*Bousquet, 2012*). The genus needs revision (*Bousquet, 2012*).

### *Apristus pugetanus* Casey, 1920
Nomenclatural Authority: *Bousquet (2012)*
Literature Records: none
Digitized Records: Santa Cruz (11 CASC)
Range: Also known from mainland (*Bousquet, 2012*).

### *Axinopalpus* LeConte, 1846
Nomenclatural Authority: *Bousquet (2012)*
Notes. Three species of *Axinopalpus* are known from California (*Bousquet, 2012*). The genus needs revision (*Bousquet, 2012*).

### *Axinopalpus biplagiatus* (Dejean, 1825)
Nomenclatural Authority: *Bousquet (2012)*

Literature Records: none

Digitized Records: Santa Cruz (1 CASC; 1 SBMNH), Santa Rosa (1 SBMNH)

Range: Also known from mainland (*Bousquet, 2012*).

### *Dromius* Bonelli, 1810

Nomenclatural Authority: *Bousquet (2012)*

Notes. One species of *Dromius* occurs in California (*Bousquet, 2012*).

### *Dromius piceus* Dejean, 1831

Nomenclatural Authority: *Bousquet (2012)*

Literature Records: none

Digitized Records: Anacapa (1 SBMNH), Santa Cruz (3 SBMNH)

Range: Also known from mainland (*Bousquet, 2012*).

### *Lebia* Latreille, 1802

Nomenclatural Authority: *Bousquet (2012)*

Notes. Ten species of *Lebia* have been recorded from California (*Bousquet, 2012*). The North American species were revised by *Madge (1967)*.

### *Lebia* (*Lebia*) *cyanipennis* Dejean, 1831

Nomenclatural Authority: *Bousquet (2012)*

Literature Records: none

Digitized Records: Santa Cruz (2 SBMNH), Santa Rosa (2 SBMNH)

Range: Also known from mainland (*Madge, 1967*).

### *Lebia* (*Lebia*) *perita* Casey, 1920

Nomenclatural Authority: *Bousquet (2012)*

Literature Records: none

Digitized Records: Santa Catalina (1 SBMNH), Santa Cruz (1 SBMNH)

Range: Also known from mainland (*Madge, 1967*).

### *Microlestes* Schmidt-Göbel, 1846

Nomenclatural Authority: *Bousquet (2012)*

Notes. Five species of *Microlestes* have been recorded from California (*Bousquet, 2012*). *Lindroth (1969)* reviewed the North American species known at the time.

### *Microlestes* undetermined species

Literature Records: none

Digitized Records: Santa Rosa (1 LACM)

### Notiophilini

Notes. One genus and four species of Notiophilini have been recorded from California (*Bousquet, 2012*).

### *Notiophilus* Duméril, 1805

Nomenclatural Authority: *Bousquet (2012)*

Notes. Four species of *Notiophilus* have been recorded from California (*Bousquet, 2012*). These were reviewed and keyed by *Lindroth (1961)*.

### *Notiophilus semiopacus* Eschscholtz, 1833

Nomenclatural Authority: *Bousquet (2012)*
Literature Records: Santa Catalina (*Fall, 1906*: 91)
Digitized Records: Santa Cruz (1 SBMNH)
Range: Also known from mainland (*Fall, 1906*; *Bousquet, 2012*).
Notes. Based on examination of the voucher specimen, the Santa Cruz Island record of "*Notiophilus* sp." in *Naughton et al. (2014*: 303) refers to this species (M. L. Gimmel, 2021, personal observation).

### Omophronini

Notes. One genus and six species of Omophronini have been recorded from California (*Bousquet, 2012*).

### *Omophron* Latreille, 1802

Nomenclatural Authority: *Bousquet (2012)*
Notes. Six species of *Omophron* have been recorded from California (*Bousquet, 2012*). The genus was revised and keyed for North America by *Benschoter & Cook (1956)*.

### *Omophron dentatum* LeConte, 1852

Nomenclatural Authority: *Bousquet (2012)*
Literature Records: Santa Cruz (*LeConte, 1876*: 298; *Fall, 1897*: 236; *Fall & Davis, 1934*: 144), Santa Rosa (*Fall, 1897*: 236; *Benschoter & Cook, 1956*: 422)
Digitized Records: Santa Catalina (4 CASC; 67 LACM; 1 SBMNH), Santa Cruz (22 CASC; 1 LACM; 9 SBMNH; 8 UASM), Santa Rosa (3 CASC; 12 SBMNH)
Range: Also known from mainland (*Benschoter & Cook, 1956*; *Bousquet, 2012*).
Notes. This species was reported from the "Channel Islands" by *Bousquet (2012*: 389).

### Platynini

Notes. Six genera and 33 species of Platynini are known to occur in California (*Bousquet, 2012*).

### *Agonum* Bonelli, 1810

Nomenclatural Authority: *Bousquet (2012)*
Digitized Records (genus-only): Santa Catalina (14 LACM), Santa Cruz (5 LACM; 9 UASM)
Notes. Twenty-one species of *Agonum* have been recorded from California (*Bousquet, 2012*). *Liebherr (1994)* provided a key to the North American species.

### *Agonum* (*Agonum*) *piceolum* (LeConte, 1879)

Nomenclatural Authority: *Bousquet (2012)*
Literature Records: none
Digitized Records: Santa Cruz (1 EMEC)
Range: Also known from mainland (*Bousquet, 2012*).

**Agonum (Europhilus) limbatum Motschulsky, 1845**
Nomenclatural Authority: *Bousquet (2012)*
Literature Records: Santa Catalina (*Fall, 1897*: 236)
Digitized Records: San Miguel (1 LACM), Santa Catalina (2 SBMNH), Santa Cruz (1 LACM; 7 SBMNH), Santa Rosa (5 LACM)
Range: Also known from mainland (*Bousquet, 2012*).
Notes. This species was recorded by *Fall (1897)* as *Platynus variolatus* LeConte, 1851. It has been recorded in the recent literature as *Agonum variolatum* (LeConte, 1851), which is now considered a junior synonym of *A. limbatum*. Reported from the "Channel Islands" by *Bousquet (2012*: 1202).

**Agonum (Olisares) decorum (Say, 1823)**
Nomenclatural Authority: *Bousquet (2012)*
Literature Records: Santa Cruz (*Liebherr, 1983*: 350 [map]; *Liebherr, 1986*: 130 [map])
Digitized Records: Santa Cruz (1 CASC)
Range: Also known from mainland (*Liebherr, 1983, 1986*).

**Agonum (Olisares) punctiforme (Say, 1823)**
Nomenclatural Authority: *Bousquet (2012)*
Literature Records: none
Digitized Records: San Clemente (9 SBMNH), San Miguel (1 SBMNH), San Nicolas (4 SBMNH)
Range: Also known from mainland (*Bousquet, 2012*).

**Anchomenus Bonelli, 1810**
Nomenclatural Authority: *Bousquet (2012)*
Notes. Two species of *Anchomenus* have been recorded from California (*Bousquet, 2012*). The genus was revised by *Liebherr (1991)*.

**Anchomenus funebris (LeConte, 1854)**
Nomenclatural Authority: *Bousquet (2012)*
Literature Records: San Clemente (*Liebherr, 1991*: 55), Santa Catalina (*Fall, 1897*: 236), Santa Cruz (*Fall & Davis, 1934*: 143; *Liebherr, 1991*: 55)
Digitized Records: Santa Cruz (33 CASC; 10 SBMNH; 16 TAMU; 1 UCSB)
Range: Also known from mainland (*Liebherr, 1991*; *Bousquet, 2012*).
Notes. This species was recorded as *Platynus funebris* by *Fall (1897)* and *Fall & Davis (1934)*. Reported from the "Channel Islands" by *Bousquet (2012*: 1183).

**Platynus Bonelli, 1810**
Nomenclatural Authority: *Bousquet (2012)*
Notes. Three species of *Platynus* have been recorded from California (*Bousquet, 2012*). *Liebherr & Will (1996)* provided a key to the North American species of *Platynus* known at the time, including all Californian species.

**Platynus brunneomarginatus (Mannerheim, 1843)**
Nomenclatural Authority: *Bousquet (2012)*
Literature Records: Santa Catalina (*Seavey, 1892*: 262; *Fall, 1897*: 236), Santa Cruz (*LeConte, 1876*: 298; *Fall, 1897*: 236; *Fall & Davis, 1934*: 143; *Liebherr & Will, 1996*: 317), Santa Rosa (*Fall, 1897*: 236)
Digitized Records: San Clemente (3 CASC; 27 LACM; 6 SBMNH), San Miguel (10 SBMNH), Santa Catalina (3 CASC; 4 LACM; 2 SBMNH), Santa Cruz (19 CASC; 10 LACM; 19 SBMNH; 6 TAMU; 1 UCSB; 11 UASM), Santa Rosa (7 SBMNH)
Range: Also known from mainland (*Liebherr & Will, 1996*; *Bousquet, 2012*).
Notes. Reported from the "Channel Islands" by *Bousquet (2012*: 1244).

**Tanystoma Motschulsky, 1845**
Nomenclatural Authority: *Bousquet (2012)*
Notes. Four species of *Tanystoma* have been recorded from California (*Bousquet, 2012*). The species were revised by *Liebherr (1985)*.

**Tanystoma cuyama Liebherr, 1985**
Nomenclatural Authority: *Bousquet (2012)*
Literature Records: none
Digitized Records: Santa Rosa (9 SBMNH)
Range: Also known from mainland (*Liebherr, 1985*).

**Tanystoma maculicolle (Dejean, 1828)**
Nomenclatural Authority: *Bousquet (2012)*
Literature Records: Anacapa (*Liebherr, 1985*: 1193 [map]; *Liebherr & Hajek, 1986*: 22), San Clemente (*Liebherr, 1984*: 538; *Liebherr, 1985*: 1193 [map]; *Liebherr & Hajek, 1986*: 22), San Miguel (*Liebherr, 1985*: 1193 [map]; *Liebherr & Hajek, 1986*: 22), San Nicolas (*Liebherr, 1985*: 1193 [map]; *Liebherr & Hajek, 1986*: 22), Santa Catalina (*Baker, 1905*: 59; *Liebherr, 1984*: 538; *Liebherr, 1985*: 1193 [map]; *Liebherr & Hajek, 1986*: 22), Santa Cruz (*Fall & Davis, 1934*: 143; *Liebherr, 1984*: 538; *Liebherr, 1985*: 1193 [map]; *Liebherr & Hajek, 1986*: 22), Santa Rosa (*Fall, 1897*: 236; *Liebherr, 1985*: 1193 [map]; *Liebherr & Hajek, 1986*: 22)
Digitized Records: San Clemente (8 CASC; 2 LACM; 4 SBMNH), San Miguel (10 CASC; 12 SBMNH), San Nicolas (14 CASC; 1 SBMNH), Santa Catalina (18 CASC; 3 LACM; 13 SBMNH), Santa Cruz (10 CASC; 14 SBMNH; 1 iNat), Santa Rosa (22 CASC; 29 SBMNH; 1 iNat)
Range: Also known from mainland (*Liebherr, 1984*, *1985*; *Bousquet, 2012*).
Notes. This species was recorded as *Platynus maculicollis* by *Baker (1905)*, *Fall (1897)*, and *Fall & Davis (1934)*. Populations of this species from the Channel Islands are predominantly brachypterous (*Liebherr & Hajek, 1986*). Reported from the "Channel Islands" by *Bousquet (2012*: 1197).

**Pogonini**

Notes. Two genera and three species of Pogonini have been recorded from California (*Bousquet, 2012*). The tribe was revised for the Western Hemisphere by *Bousquet & Laplante (1997)*.

**Thalassotrechus Van Dyke, 1918**

Nomenclatural Authority: *Bousquet (2012)*

Notes. One species of *Thalassotrechus* occurs in North America (*Bousquet, 2012*).

**Thalassotrechus barbarae (Horn, 1892)**

Nomenclatural Authority: *Bousquet (2012)*

Literature Records: none

Digitized Records: San Clemente (2 SBMNH), Santa Catalina (14 SBMNH)

Range: Also known from mainland (*Bousquet, 2012*).

Notes. *Bousquet & Laplante (1997)* provided a map of known records but did not include any Channel Islands specimens.

**Pterostichini**

Notes. Two genera and 78 species of Pterostichini are known to occur in California (*Bousquet, 2012*).

**Poecilus Bonelli, 1810**

Nomenclatural Authority: *Bousquet (2012)*

Digitized Records (genus-only): Santa Cruz (1 iNat)

Notes. Five species of *Poecilus* have been recorded from California (*Bousquet, 2012*). The genus needs revision (*Bousquet, 2012*), but *Lindroth's (1966)* key includes all but one of the California species. The iNaturalist observation cited above appears to represent *P. laetulus*.

**Poecilus (Poecilus) laetulus (LeConte, 1863)**

Nomenclatural Authority: *Bousquet (2012)*

Literature Records: Santa Catalina (*Cockerell, 1940*: 285), Santa Cruz (*LeConte, 1876*: 298; *Fall, 1897*: 236; *Fall & Davis, 1934*: 143), Santa Rosa (*Fall, 1897*: 236)

Digitized Records: San Clemente (1 SBMNH), Santa Catalina (1 SBMNH)

Range: Also known from mainland (*Fall, 1901*; *Bousquet, 2012*).

Notes. *Fall (1901*: 45) collectively reported this and other species from "the islands off the coast". This species was reported as *Pterostichus laetulus* LeConte by *LeConte (1876)*, *Fall (1897, 1901)*, and *Fall & Davis (1934)*. Reported from the "Channel Islands" by *Bousquet (2012*: 775).

**Pterostichus Bonelli, 1810**

Nomenclatural Authority: *Bousquet (2012)*

Literature Records (genus-only): Santa Cruz (*Naughton et al., 2014*: 303), Santa Rosa (*Fall, 1897*: 236)

Digitized Records (genus-only): San Clemente (24 LACM; 11 SBMNH), San Miguel (1 LACM; 5 SBMNH), Santa Catalina (25 LACM; 5 SBMNH), Santa Cruz (11 LACM; 26 SBMNH; 1 UASM), Santa Rosa (5 LACM; 18 SBMNH)

Notes. Five subgenera and 73 species of *Pterostichus* have been recorded from California (*Bousquet, 2012*). *Fall (1901*: 45) collectively reported an undetermined species and other species of *Pterostichus* from "the islands off the coast". *Hypherpes* Chaudoir, 1838 is by far the largest subgenus of *Pterostichus* in California, with 48 species recorded (*Bousquet, 2012*). It is badly in need of revision (*Bousquet, 2012*). Most or all genus-only digitized records cited above belong to this subgenus. A record of the northern Californian *Pterostichus* (*Hypherpes*) *congestus* (Ménétriés, 1843) from Santa Catalina Island provided to the California Beetle Database probably originated from a specimen determined by H.C. Fall in the Museum of Comparative Zoology, Harvard University (S. Miller, 2022, personal communication); this species record is the result of an erroneous synonymy (see *Bousquet, 2012*: 846) with *P. illustris* (K. Will, 2022, personal communication). A record of *Pterostichus* (*Hypherpes*) *lama* (Ménétriés, 1843) from Santa Rosa Island exists in the SBMNH database, but no specimen was found. This species record is geographically suspect (K. Will, 2021, personal communication).

### *Pterostichus* (*Bothriopterus*) *lustrans* LeConte, 1851
Nomenclatural Authority: *Bousquet (2012)*
Literature Records: Santa Cruz (*Fall & Davis, 1934*: 143)
Digitized Records: Santa Cruz (10 SBMNH)
Range: Also known from mainland (*Bousquet, 2012*).
Notes. Reported from the "Channel Islands" by *Bousquet (2012*: 795). Some of the digital records were erroneously identified as *Pterostichus adstrictus* Eschscholtz, 1823 previously.

### *Pterostichus* (*Hypherpes*) *gliscans* Casey, 1913
Nomenclatural Authority: *Bousquet (2012)*
Literature Records: San Clemente (*Casey, 1913a*: 119; *Miller, 1985a*: 19; *Bousquet, 2012*: 847)
Digitized Records: San Miguel (28 LACM)
Range: Endemic (*Casey, 1913a*; *Miller, 1985a*; *Bousquet, 2012*).

### *Pterostichus* (*Hypherpes*) *illustris* LeConte, 1851
Nomenclatural Authority: *Bousquet (2012)*
Literature Records: none
Digitized Records: Santa Catalina (2 CASC; 1 EMEC)
Range: Also known from mainland (*Bousquet, 2012*).

### *Pterostichus* (*Hypherpes*) *inermis* Fall, 1901
Nomenclatural Authority: *Bousquet (2012)*
Literature Records: none
Digitized Records: Santa Cruz (28 EMEC)
Range: Also known from mainland (*Bousquet, 2012*).

*Pterostichus* (*Hypherpes*) *isabellae* LeConte, 1851
Nomenclatural Authority: *Bousquet (2012)*
Literature Records: San Clemente (*Fall, 1897*: 236), Santa Catalina (*Fall, 1897*: 236)
Digitized Records: none
Range: Also known from mainland (*Fall, 1901*; *Bousquet, 2012*).
Notes. *Fall (1901*: 45) collectively reported this and other species from "the islands off the coast". Reported from the "Channel Islands" by *Bousquet (2012*: 847).

*Pterostichus* (*Hypherpes*) *jacobinus* Casey, 1913
Nomenclatural Authority: *Bousquet (2012)*
Literature Records: none
Digitized Records: Santa Catalina (1 EMEC)
Range: Also known from mainland (*Bousquet, 2012*).

*Pterostichus* (*Hypherpes*) *menetriesii* LeConte, 1873
Nomenclatural Authority: *Bousquet (2012)*
Literature Records: Santa Rosa (*Fall, 1897*: 236; *Fall, 1901*: 44; *Bousquet, 2012*: 849)
Digitized Records: San Miguel (84 CASC), Santa Cruz (4 CASC)
Range: Also known from mainland (*Bousquet, 2012*).
Notes. This species was thought to be endemic to the Channel Islands by *Fall (1901)*.

**Sphodrini**
Notes. Two genera and four species of Sphodrini are known to occur in California (*Bousquet, 2012*).

*Calathus* Bonelli, 1810
Nomenclatural Authority: *Bousquet (2012)*
Digitized Records (genus-only): Santa Cruz (18 UASM)
Notes. Three species of *Calathus* have been recorded from California (*Bousquet, 2012*; M. L. Gimmel, 2022, unpublished data). The Western Hemisphere species were revised by *Ball & Negre (1972)*.

*Calathus* (*Neocalathus*) *ruficollis* Dejean, 1828
Nomenclatural Authority: *Bousquet (2012)*
Literature Records: Anacapa (*Cockerell, 1940*: 285; *Ball & Negre, 1972*: 481 [map]), San Clemente (*Ball & Negre, 1972*: 481 [map]), San Miguel (*Cockerell, 1940*: 285; *Ball & Negre, 1972*: 481 [map]), Santa Catalina (*Fall, 1897*: 236; *Baker, 1905*: 59; *Cockerell, 1940*: 285; *Ball & Negre, 1972*: 481 [map]), Santa Cruz (*LeConte, 1876*: 298; *Fall, 1897*: 236; *Fall & Davis, 1934*: 143; *Cockerell, 1940*: 285; *Ball & Negre, 1972*: 481 [map]; *Naughton et al., 2014*: 303), Santa Rosa (*Fall, 1897*: 236; *Ball & Negre, 1972*: 481 [map])
Digitized Records: Anacapa (6 LACM; 1 SBMNH), San Miguel (183 LACM; 16 SBMNH), Santa Catalina (1 CSUC; 9 LACM; 21 SBMNH), Santa Cruz (18 LACM; 31 SBMNH; 4 UCSB), Santa Rosa (55 LACM; 18 SBMNH)
Range: Also known from mainland (*Ball & Negre, 1972*; *Bousquet, 2012*).

Notes. This species was cited as *C. ruficollis*, without subspecies, by authors prior to *Ball & Negre (1972)*, who determined that the only subspecies occurring in southern California is the nominate subspecies, *C. r. ruficollis* Dejean, 1828. Prior records of *Calathus obscurus* LeConte (*Baker, 1905*: 59; *Fall, 1897*: 236; *Cockerell, 1940*: 285) also refer to this species, which was later synonymized with *C. ruficollis* (see *Ball & Negre, 1972*). *Cockerell's (1940*: 285) records from Anacapa, San Miguel, and Santa Cruz were cited as *Calathus insularis* Casey, which was also later synonymized with *C. ruficollis* (see *Ball & Negre, 1972*). Reported from the "Channel Islands" by *Bousquet (2012*: 1169).

*Laemostenus* **Bonelli, 1810**
Nomenclatural Authority: *Bousquet (2012)*
Notes. One adventive species of *Laemostenus* occurs in California (*Bousquet, 2012*).

*Laemostenus* (*Laemostenus*) *complanatus* (Dejean, 1828)
Nomenclatural Authority: *Bousquet (2012)*
Literature Records: none
Digitized Records: San Nicolas (9 SBMNH), Santa Catalina (1 CASC; 2 LACM)
Range: Also known from mainland (*Bousquet, 2012*).
Notes. This species was introduced from the Palearctic region to North America (*Bousquet, 2012*: 1173).

**Zabrini**
Notes. One genus and 36 species of Zabrini are known to occur in California (*Bousquet, 2012*).

*Amara* **Bonelli, 1810**
Nomenclatural Authority: *Bousquet (2012)*
Digitized Records (genus-only): Anacapa (6 ASUHIC), San Clemente (7 LACM), San Nicolas (3 LACM; 1 SBMNH), Santa Barbara (17 LACM; 4 SBMNH), Santa Cruz (7 UCSB; 1 iNat), Santa Rosa (1 LACM; 9 SBMNH)
Notes. Nine subgenera and 36 species of *Amara* have been recorded from California (*Bousquet, 2012*). *Miller & Miller (1985)* reported many specimens from Santa Barbara Island not determined to species, noting the need for revision of the genus.

*Amara* (*Amara*) *aurata* Dejean, 1828
Nomenclatural Authority: *Bousquet (2012)*
Literature Records: San Clemente (*Casey, 1918*: 274; *Miller, 1985a*: 19; *Hieke, 1993*: 121; *Bousquet, 2012*: 929)
Digitized Records: San Clemente (3 CASC), Santa Cruz (2 CASC)
Range: Also known from mainland (*Hieke, 1993*; *Bousquet, 2012*).
Notes. This species was recorded as the purported endemic *Celia clementina* Casey, 1918 by *Casey (1918)*, and by *Miller (1985a)* as *Amara clementina*. It was subsequently synonymized with *A. aurata* by *Hieke (1993)*.

### Amara (Amara) conflata LeConte, 1855

Nomenclatural Authority: *Bousquet (2012)*

Literature Records: none

Digitized Records: Santa Cruz (4 CASC)

Range: Also known from mainland (*Bousquet, 2012*).

Notes. Reported from the "Channel Islands" by *Bousquet (2012*: 939).

### Amara (Amara) pomona Casey, 1918

Nomenclatural Authority: *Bousquet (2012)*

Literature Records: Santa Cruz (*Hieke, 1993*: 114), Santa Rosa (*Hieke, 1993*: 114)

Digitized Records: none

Range: Also known from mainland (*Hieke, 1993*).

Notes. This species was recorded as *Amara brunnipes* Motschulsky, 1859 by *Hieke (1993)*, which is a junior primary homonym with *A. pomona* as the next available name.

### Amara (Bradytus) insignis Dejean, 1831

Nomenclatural Authority: *Bousquet (2012)*

Literature Records: Santa Catalina (*Fall, 1897*: 236; *Baker, 1905*: 59), Santa Rosa (*Fall, 1897*: 236)

Digitized Records: Santa Catalina (5 CASC), Santa Cruz (4 CASC)

Range: Also known from mainland (*Fall, 1901*; *Bousquet, 2012*).

Notes. *Fall (1901*: 45) reported this species from "islands". Reported from the "Channel Islands" by *Bousquet (2012*: 905). SBMNH specimens of the "*Amara insignis* group", represented by the two species *A. insignis* and *A. insularis*, were determined based on the key in *Lindroth (1968*: 659). The characters cited by Lindroth in both the key and species accounts (*Lindroth, 1968*: 659) involve the punctation of the pronotum and development of the hind wings. However, while mainland specimens of *A. insignis* appear to be consistently punctate basally, this character appears to break down, at least among northern island populations (M. L. Gimmel, 2021, personal observation). The shape of the aedeagal apices do not seem to covary with this character and, in fact, impunctate "*A. insularis*" forms do not appear to possess a distinctive aedeagus when compared to mainland *A. insignis*. Additionally, hind wings appear to be developed in all specimens where this character is visible (M. L. Gimmel, 2021, personal observation). In the end, all island members of the group housed in SBMNH were determined as *A. insularis*.

### Amara (Bradytus) insularis Horn, 1875

Nomenclatural Authority: *Bousquet (2012)*

Literature Records: San Clemente (*Horn, 1875*: 128; *Fall, 1897*: 236; *Casey, 1918*: 295; *Cockerell, 1940*: 285; *Lindroth, 1968*: 692; *Miller, 1985a*: 19; *Miller & Miller, 1985*: 123), San Nicolas (*Fall, 1897*: 236; *Hayward, 1908*: 51; *Cockerell, 1940*: 285), Santa Barbara (*Fall, 1897*: 236; *Hayward, 1908*: 51; *Cockerell, 1940*: 285), Santa Rosa (*Hayward, 1908*: 51)

Digitized Records: Anacapa (4 CASC), San Clemente (31 CASC; 1 LACM; 10 SBMNH), San Miguel (2 CASC; 7 LACM; 2 SBMNH), San Nicolas (18 CASC; 201 LACM; 2 SBMNH), Santa Barbara (3 CASC; 89 LACM; 36 SBMNH), Santa Catalina (6 CASC), Santa Cruz (3 CASC; 8 SBMNH), Santa Rosa (3 CASC; 2 LACM; 18 SBMNH)

Range: Endemic (*Horn, 1875*; *Fall, 1897*; *Hayward, 1908*; *Casey, 1918*; *Cockerell, 1940*; *Lindroth, 1968*; *Miller, 1985a*; *Bousquet, 2012*).

Notes. *Fall (1901)* reported this species from "all the islands". *Lindroth (1968*: 692) incorrectly stated that this species was only known from the type locality (San Clemente Island), though *Miller (1985a)* noted that "published records from other islands (than San Clemente) cannot be trusted." This species, however, is questionably distinct from *A. insignis* (see Notes in that species account above).

### *Amara* (*Celia*) *californica* Dejean, 1828

Nomenclatural Authority: *Bousquet (2012)*

Literature Records: San Clemente (*Casey, 1918*: 294; *Lindroth, 1968*: 693; *Lindroth, 1975*: 131; *Bousquet, 2012*: 915), Santa Cruz (*LeConte, 1876*: 298; *Fall, 1897*: 236; *Fall & Davis, 1934*: 143), Santa Rosa (*Fall, 1897*: 236)

Digitized Records: Anacapa (40 CASC), San Clemente (5 CASC), San Miguel (21 CASC), San Nicolas (17 CASC), Santa Cruz (3 CASC), Santa Rosa (1 CASC)

Range: Also known from mainland (*Lindroth, 1968*; *Bousquet, 2012*).

Notes. Recorded by *Fall & Davis (1934)* as *Celia californica*. *Amara perspecta* Casey, 1918, described as endemic from San Clemente Island (*Casey, 1918*: 294), was synonymized with *A. californica* by *Lindroth (1968*: 693). The only subspecies of *A. californica* occurring in California is the nominate subspecies, *A. c. californica* Dejean, 1828 (*Bousquet, 2012*).

### *Amara* (*Zezea*) *scitula* Zimmermann, 1832

Nomenclatural Authority: *Bousquet (2012)*

Literature Records: none

Digitized Records: San Miguel (1 CASC)

Range: Also known from mainland (*Bousquet, 2012*).

### Cicindelidae

Notes. The North American fauna of tiger beetles (often treated as a subfamily of Carabidae) was the subject of a thorough field guide by *Pearson et al. (2015)*. *Nagano (1982, 1985)* assessed the Channel Islands fauna in detail. Three tribes, eight genera, and 34 species have been recorded from California (*Bousquet, 2012*; *Pearson et al., 2015*).

### Cicindelini

Notes. The California fauna of Cicindelini consists of five genera and 29 species (*Bousquet, 2012*; *Pearson et al., 2015*). We follow the genus-level arrangement of *Pearson et al. (2015)* below.

### *Cicindela* Linnaeus, 1758

Nomenclatural Authority: *Bousquet (2012)*

Digitized Records: Santa Cruz (6 LACM)

Notes. Seventeen species of *Cicindela* are known to occur in California (*Bousquet, 2012*; *Pearson et al., 2015*).

*Cicindela hirticollis* Say, 1817

Nomenclatural Authority: *Bousquet (2012)*, *Pearson et al. (2015)*

Literature Records: Santa Catalina (*Nagano, 1982*: 36; *Nagano, 1985*: 106; *Graves, Krejci & Graves, 1988*: 653 [map]), Santa Rosa (*Nagano, 1982*: 36; *Nagano, 1985*: 106; *Graves, Krejci & Graves, 1988*: 653 [map])

Digitized Records: none

Range: Also known from mainland (*Nagano, 1982*, *1985*; *Graves, Krejci & Graves, 1988*; *Bousquet, 2012*).

Notes. All coastal Southern California *C. hirticollis* belong to the subspecies *C. h. gravida* LeConte, 1851 (*Nagano, 1982*, *1985*; *Graves, Krejci & Graves, 1988*). Recorded from the "Channel Islands" by *Bousquet (2012*: 362).

*Cicindela oregona* LeConte, 1856

Nomenclatural Authority: *Bousquet (2012)*, *Pearson et al. (2015)*

Literature Records: Anacapa (*Nagano, 1985*: 106), San Miguel (*Nagano, 1982*: 36; *Nagano, 1985*: 105), San Nicolas (*Nagano, 1982*: 35; *Nagano, 1985*: 106), Santa Catalina (*Nagano, 1982*: 36; *Nagano, 1985*: 106), Santa Cruz (*Freitag, 1965*: 136; *Nagano, 1982*: 35; *Nagano, 1985*: 105), Santa Rosa (*Fall, 1897*: 236; *Nagano, 1982*: 36; *Nagano, 1985*: 105)

Digitized Records: Anacapa (1 LACM), San Miguel (1 SBMNH; 14 LACM), San Nicolas (26 LACM), Santa Cruz (20 SBMNH; 1 UCSB; 36 LACM; 12 YPMC), Santa Rosa (2 SBMNH; 17 LACM)

Range: Also known from mainland (*Freitag, 1965*; *Nagano, 1982*, *1985*; *Bousquet, 2012*).

Notes. All coastal Southern California *C. oregona* belong to the nominate subspecies, *C. o. oregona* LeConte, 1856 (*Nagano, 1982*, *1985*). Recorded from the "Channel Islands" by *Bousquet (2012*: 367).

*Cicindela senilis* Horn, 1867

Nomenclatural Authority: *Bousquet (2012)*, *Pearson et al. (2015)*

Literature Records: San Clemente (*Nagano, 1982*: 37; *Nagano, 1985*: 107)

Digitized Records: San Clemente (60 LACM; 3 SBMNH)

Range: Also known from mainland (*Nagano, 1982*, *1985*; *Bousquet, 2012*).

Notes. The Channel Islands form of this species was referred to as the subspecies *C. s. frosti* Varas Arangua, 1928 by *Nagano (1982*, *1985)*, but subspecies were not recognized for this species by *Bousquet (2012)*. *Pearson et al. (2015)* noted that "some authors" recognize these subspecies. This species was recorded from the "Channel Islands" by *Bousquet (2012*: 325).

*Cicindelidia* Rivalier, 1954

Nomenclatural Authority: *Bousquet (2012)*, *Pearson et al. (2015)*

Notes. Four species of *Cicindelidia* are known to occur in California (*Bousquet, 2012*; *Pearson et al., 2015*). *Bousquet (2012)* treated this taxon as a subgenus of *Cicindela*.

*Cicindelidia hemorrhagica* (LeConte, 1851)

Nomenclatural Authority: *Bousquet (2012)*, *Pearson et al. (2015)*

Literature Records: San Nicolas (*Nagano, 1982*: 39; *Nagano, 1985*: 109), Santa Cruz (*Nagano, 1982*: 39; *Nagano, 1985*: 109)

Digitized Records: San Nicolas (19 SBMNH), Santa Cruz (2 SBMNH; 2 UCSB; 62 YPMC)

Range: Also known from mainland (*Nagano, 1982*, *1985*; *Bousquet, 2012*).

Notes. All Channel Islands *C. hemorrhagica* belong to the nominate subspecies, *C. h. hemorrhagica* (LeConte, 1851) (*Nagano, 1982*, *1985*). Recorded from the "Channel Islands" by *Bousquet (2012*: 314); *Nagano (1982*, *1985*) listed this species as *Cicindela haemorrhagica*, and *Bousquet (2012)* as *Cicindela hemorrhagica*.

### *Cicindelidia trifasciata* (Fabricius, 1781)
Nomenclatural Authority: *Bousquet (2012)*, *Pearson et al. (2015)*

Literature Records: Santa Catalina (*Nagano, 1982*: 38; *Nagano, 1985*: 109)

Digitized Records: Santa Catalina (3 SBMNH; 132 LACM; 1 SDNHM)

Range: Also known from mainland (*Nagano, 1982*, *1985*; *Bousquet, 2012*).

Notes. All Southern California *C. trifasciata* belong to the subspecies *C. t. sigmoidea* (LeConte, 1851) (*Nagano, 1982*, *1985*). Recorded from the "Channel Islands" by *Bousquet (2012*: 326). *Nagano (1982*, *1985)* and *Bousquet (2012)* listed this species as *Cicindela trifasciata sigmoidea*.

### Dytiscidae
Notes. Seven subfamilies, 36 genera, and 156 species of this family are known to occur in California (*Challet & Brett, 1998*; M. L. Gimmel, 2022, unpublished data). *Challet & Brett (1998)* provided an excellent county-by-county summary of their distribution in California, though the nomenclature of some of the genera is now out of date. Likewise, *Larson, Alarie & Roughley (2000)* provided excellent illustrated keys for most North American species of the family, but the generic classification has changed in many groups. *Nilsson & Hájek (2018)* provided an updated world catalog for the family.

### Agabinae: Agabini
Notes. Two tribes, six genera, and 38 species of Agabinae are known from California, of which just one species belongs to Hydrotrupini and the remainder belong to Agabini (*Challet & Brett, 1998*; M. L. Gimmel, 2022, unpublished data).

### *Agabinus* Crotch, 1873
Nomenclatural Authority: *Nilsson & Hájek (2018)*

Digitized Records (genus-only): Santa Cruz (1 UCSB)

Notes. Two species of *Agabinus* are known to occur in California (*Challet & Brett, 1998*).

### *Agabinus glabrellus* (Motschulsky, 1859)
Nomenclatural Authority: *Nilsson & Hájek (2018)*

Literature Records: Santa Catalina (*Fall, 1897*: 236; *Fall, 1901*: 53; *Challet, 1987*: 13), Santa Cruz (*Furlong & Wenner, 2002*: 250)

Digitized Records: Santa Catalina (1 LACM; 8 SBMNH), Santa Cruz (3 SBMNH), Santa Rosa (4 SBMNH)

Range: Also known from mainland (*Challet & Brett, 1998*).

**Agabinus sculpturellus Zimmermann, 1919**
Nomenclatural Authority: *Nilsson & Hájek (2018)*
Literature Records: Santa Cruz (*Furlong & Wenner, 2002*: 250)
Digitized Records: none
Range: Also known from mainland (*Challet & Brett, 1998*).

**Agabus Leach, 1817**
Nomenclatural Authority: *Nilsson & Hájek (2018)*
Notes. Sixteen species of *Agabus* are known to occur in California (*Challet & Brett, 1998*; M. L. Gimmel, 2022, unpublished data). These species are best separated by examination of the male genitalia (*Larson, Alarie & Roughley, 2000*).

**Agabus obsoletus LeConte, 1858**
Nomenclatural Authority: *Nilsson & Hájek (2018)*
Literature Records: none
Digitized Records: San Miguel (5 SBMNH)
Range: Also known from mainland (*Challet & Brett, 1998*).

**Ilybiosoma Crotch, 1873**
Nomenclatural Authority: *Nilsson & Hájek (2018)*
Notes. Eight species of *Ilybiosoma* are known to occur in California (*Challet & Brett, 1998*; M. L. Gimmel, 2022, unpublished data).

**Ilybiosoma lugens (LeConte, 1852)**
Nomenclatural Authority: *Nilsson & Hájek (2018)*
Literature Records: Santa Catalina (*Challet, 1987*: 13), Santa Rosa (*Fall, 1897*: 236)
Digitized Records: San Nicolas (2 SBMNH), Santa Catalina (58 SBMNH), Santa Cruz (9 SBMNH), Santa Rosa (6 LACM; 10 SBMNH)
Range: Also known from mainland (*Challet & Brett, 1998*).
Notes. Reported as *Agabus lugens* by *Fall (1897)* and *Challet (1987)*. Specimens of this species are indistinguishable from another California species, *Ilybiosoma perplexum* (Sharp, 1882), except by male genitalia (*Larson, Alarie & Roughley, 2000*). The San Nicolas Island specimens were both female; consequently, once discovered, male specimens matching the description of *I. lugens* from that island should be dissected to confirm their identity.

**Ilybiosoma regulare (LeConte, 1852)**
Nomenclatural Authority: *Nilsson & Hájek (2018)*
Literature Records: none
Digitized Records: Santa Cruz (5 SBMNH), Santa Rosa (3 SBMNH)
Range: Also known from mainland (*Challet & Brett, 1998*).

**Ilybiosoma seriatum (Say, 1823)**
Nomenclatural Authority: *Nilsson & Hájek (2018)*
Literature Records: none

Digitized Records: Santa Cruz (2 SBMNH), Santa Rosa (4 SBMNH)
Range: Also known from mainland (*Challet & Brett, 1998*).

### *Ilybius* Erichson, 1832
Nomenclatural Authority: *Nilsson & Hájek (2018)*
Notes. Ten species of *Ilybius* are known to occur in California (*Challet & Brett, 1998*; M. L. Gimmel, 2022, unpublished data). These species are best separated by examination of the male genitalia (*Larson, Alarie & Roughley, 2000*).

### *Ilybius discors* (LeConte, 1861)
Nomenclatural Authority: *Nilsson & Hájek (2018)*
Literature Records: Santa Cruz (*Furlong & Wenner, 2002*: 250)
Digitized Records: none
Range: Also known from mainland (*Challet & Brett, 1998*).
Notes. Reported by *Furlong & Wenner (2002)* as *Agabus discors*.

### *Ilybius lineellus* (LeConte, 1861)
Nomenclatural Authority: *Nilsson & Hájek (2018)*
Literature Records: none
Digitized Records: Santa Cruz (2 SBMNH)
Range: Also known from mainland (*Challet & Brett, 1998*).

### *Ilybius walsinghami* (Crotch, 1873)
Nomenclatural Authority: *Nilsson & Hájek (2018)*
Literature Records: San Clemente (*Larson, 1996*: 665 [map only])
Digitized Records: Santa Rosa (17 SBMNH)
Range: Also known from mainland (*Larson, 1996*; *Challet & Brett, 1998*).
Notes. Reported by *Larson (1996)* as *Agabus walsinghami*.

### Colymbetinae
Notes. Two genera, and 10 species of Colymbetinae have been recorded from California (*Challet & Brett, 1998*; M. L. Gimmel, 2022, unpublished data), all belonging to the tribe Colymbetini.

### *Rhantus* Dejean, 1833
Nomenclatural Authority: *Nilsson & Hájek (2018)*
Notes. Seven species of *Rhantus* are known to occur in California (*Challet & Brett, 1998*; M. L. Gimmel, 2022, unpublished data).

### *Rhantus gutticollis* (Say, 1830)
Nomenclatural Authority: *Nilsson & Hájek (2018)*
Literature Records: San Clemente (*Zimmerman & Smith, 1975*: 52), Santa Catalina (*Challet, 1987*: 13), Santa Cruz (*Furlong & Wenner, 2002*: 250)
Digitized Records: San Clemente (28 SBMNH), San Nicolas (1 SBMNH), Santa Catalina (57 LACM; 6 SBMNH), Santa Cruz (5 SBMNH), Santa Rosa (3 SBMNH)
Range: Also known from mainland (*Zimmerman & Smith, 1975*; *Challet & Brett, 1998*).

### Dytiscinae

Notes. Five tribes, seven genera, and 14 species of Dytiscinae have been recorded from California (M. L. Gimmel, 2022, unpublished data).

### Dytiscini

Notes. One genus and four species of Dytiscini are known to occur in California (*Challet & Brett, 1998*).

### *Dytiscus* Linnaeus, 1758

Nomenclatural Authority: *Nilsson & Hájek (2018)*

Notes. Four species of *Dytiscus* are known to occur in California (*Challet & Brett, 1998*).

### *Dytiscus marginicollis* LeConte, 1845

Nomenclatural Authority: *Nilsson & Hájek (2018)*

Literature Records: Santa Catalina (*Challet, 1987*: 13)

Digitized Records: San Clemente (6 SBMNH), Santa Catalina (4 LACM)

Range: Also known from mainland (*Challet & Brett, 1998*).

### Eretini

Notes. One genus and species of Eretini is known to occur in California (*Challet & Brett, 1998*).

### *Eretes* Laporte, 1833

Nomenclatural Authority: *Nilsson & Hájek (2018)*

Notes. One species of *Eretes* has been reported from California (*Challet & Brett, 1998*). This genus was revised by *Miller (2002)*.

### *Eretes sticticus* (Linnaeus, 1767)

Nomenclatural Authority: *Nilsson & Hájek (2018)*

Literature Records: none

Digitized Records: Santa Catalina (2 LACM)

Range: Also known from mainland (*Challet & Brett, 1998*; *Miller, 2002*).

### Hydroporinae

Notes. Six tribes, 19 genera, and 85 species of Hydroporinae are known to occur in California (*Challet & Brett, 1998*; M. L. Gimmel, 2022, unpublished data).

### Bidessini

Notes. Three genera and 10 species of Bidessini are known to occur in California (M. L. Gimmel, 2022, unpublished data).

### *Liodessus* Guignot, 1939

Nomenclatural Authority: *Nilsson & Hájek (2018)*

Notes. Two species of *Liodessus* have been reported from California (*Miller, 1998*). The *L. affinis* group, containing both California species, was revised by *Miller (1998)*.

***Liodessus obscurellus*** (LeConte, 1852)

Nomenclatural Authority: *Nilsson & Hájek (2018)*

Literature Records: none

Digitized Records: San Miguel (3 SBMNH), Santa Catalina (1 SBMNH), Santa Cruz (11 SBMNH), Santa Rosa (8 SBMNH)

Range: Also known from mainland (*Miller, 1998*).

Notes. This species was treated as a junior synonym of *Liodessus affinis* (Say, 1823) until recently (*Miller, 1998*).

***Neoclypeodytes*** Young, 1967

Nomenclatural Authority: *Nilsson & Hájek (2018)*

Notes. Seven species of *Neoclypeodytes* have been reported from California (*Miller, 2001*). The genus was revised by *Miller (2001)*.

***Neoclypeodytes pictodes*** (Sharp, 1882)

Nomenclatural Authority: *Nilsson & Hájek (2018)*

Literature Records: none

Digitized Records: Santa Rosa (16 SBMNH)

Range: Also known from mainland (*Challet & Brett, 1998*; *Miller, 2001*).

***Uvarus*** Guignot, 1939

Nomenclatural Authority: *Nilsson & Hájek (2018)*

Notes. One species of *Uvarus* is known to occur in California (*Challet & Brett, 1998*).

***Uvarus subtilis*** (LeConte, 1852)

Nomenclatural Authority: *Nilsson & Hájek (2018)*

Literature Records: none

Digitized Records: Santa Cruz (6 SBMNH), Santa Rosa (2 SBMNH)

Range: Also known from mainland (*Challet & Brett, 1998*).

**Hydroporini**

Notes. Eleven genera and 49 species of Hydroporini are known to occur in California (*Challet & Brett, 1998*; M. L. Gimmel, 2022, unpublished data).

***Leconectes*** Fery & Ribera, 2018

Nomenclatural Authority: *Fery & Ribera (2018)*

Notes. One species of *Leconectes* is known to occur in California (M. L. Gimmel, 2022, unpublished data).

***Leconectes striatellus*** (LeConte, 1852)

Nomenclatural Authority: *Fery & Ribera (2018)*

Literature Records: Santa Catalina (*Challet, 1987*: 13), Santa Cruz (*Short & Caterino, 2009*: 405), Santa Rosa (*Fall, 1897*: 236; *Short & Caterino, 2009*: 405)

Digitized Records: San Clemente (1 SBMNH), San Miguel (2 SBMNH), Santa Catalina (54 LACM; 2 SBMNH), Santa Cruz (14 SBMNH; 1 UCSB), Santa Rosa (11 SBMNH)

Range: Also known from mainland (*Challet & Brett, 1998*; *Short & Caterino, 2009*; *Fery & Ribera, 2018*).

Notes. *Fall (1897)* and *Challet (1987)* reported this species as *Deronectes striatellus*, while *Short & Caterino (2009)* reported it as *Stictotarsus striatellus*; it has also been known as *Boreonectes striatellus* until recently (*Fery & Ribera, 2018*).

**Sanfilippodytes Franciscolo, 1979**
Nomenclatural Authority: *Nilsson & Hájek (2018)*
Digitized Records (genus-only): Santa Cruz (8 EMEC)
Notes. Thirteen species of *Sanfilippodytes* are known to occur in California (*Challet & Brett, 1998*).

**Sanfilippodytes barbarensis (Wallis, 1933)**
Nomenclatural Authority: *Nilsson & Hájek (2018)*
Literature Records: none
Digitized Records: Santa Catalina (7 SBMNH), Santa Cruz (9 SBMNH), Santa Rosa (34 SBMNH)
Range: Also known from mainland (*Nilsson & Hájek, 2018*).

**Sanfilippodytes latebrosus (LeConte, 1852)**
Nomenclatural Authority: *Nilsson & Hájek (2018)*
Literature Records: none
Digitized Records: Santa Catalina (1 SBMNH), Santa Cruz (3 SBMNH), Santa Rosa (12 SBMNH)
Range: Also known from mainland (*Challet & Brett, 1998*).

**Sanfilippodytes vilis (LeConte, 1852)**
Nomenclatural Authority: *Nilsson & Hájek (2018)*
Literature Records: Santa Catalina (*Challet, 1987*: 13), Santa Cruz (*Furlong & Wenner, 2002*: 250)
Digitized Records: San Miguel (7 SBMNH), San Nicolas (8 SBMNH)
Range: Also known from mainland (*Challet & Brett, 1998*).
Notes. According to *Challet (1987*: 13), *Fall (1897)* originally reported this species as *Hydroporus vilis* from Santa Catalina and Santa Rosa, but later (*Fall, 1923*) indicated that these records referred to *H. belfragei* (see *S. williami* below). All of the foregoing species were subsequently transferred to *Sanfilippodytes*.

**Sanfilippodytes williami (Rochette, 1986)**
Nomenclatural Authority: *Nilsson & Hájek (2018)*
Literature Records: Santa Catalina (*Fall, 1897*: 236; *Fall, 1923*: 59; *Rochette, 1986*: 341; *Challet, 1987*: 13), Santa Rosa (*Fall, 1897*: 236)
Digitized Records: Santa Cruz (3 SBMNH)
Range: Also known from mainland (*Rochette, 1986*; *Challet & Brett, 1998*).
Notes. *Fall (1897*: 236) misidentified this species as *Hydroporus vilis* LeConte. *Fall (1923)* and *Challet (1987)* corrected the identification to *Hydroporus belfragei* Sharp, 1882. Finally,

*Rochette (1986)* described the new species *H. williami* that included the island material. All of the foregoing species were subsequently transferred to *Sanfilippodytes*.

### Hydrovatini
Notes. One genus and two species of Hydrovatini are known to occur in California (*Challet & Brett, 1998*).

### *Hydrovatus* Motschulsky, 1853
Nomenclatural Authority: *Nilsson & Hájek (2018)*
Notes. Two species of *Hydrovatus* are known to occur in California (*Challet & Brett, 1998*).

### *Hydrovatus brevipes* Sharp, 1882
Nomenclatural Authority: *Nilsson & Hájek (2018)*
Literature Records: Santa Cruz (*Furlong & Wenner, 2002*: 250)
Digitized Records: none
Range: Also known from mainland (*Challet & Brett, 1998*).

### Hygrotini
Notes. Two genera and 21 species of Hygrotini are known to occur in California (*Challet & Brett, 1998*).

### *Hygrotus* Stephens, 1828
Nomenclatural Authority: *Nilsson & Hájek (2018)*
Notes. Twenty species of *Hygrotus* are known to occur in California (*Challet & Brett, 1998*).

### *Hygrotus* (*Leptolambus*) *lutescens* (LeConte, 1852)
Nomenclatural Authority: *Nilsson & Hájek (2018)*
Literature Records: Santa Catalina (*Challet, 1987*: 13)
Digitized Records: San Clemente (14 SBMNH), San Nicolas (7 SBMNH), Santa Catalina (63 LACM; 10 SBMNH), Santa Cruz (12 SBMNH)
Range: Also known from mainland (*Challet & Brett, 1998*).

### Laccophilinae
Notes. One genus and six species of Laccophilinae are known to occur in California (*Challet & Brett, 1998*), all belonging to the tribe Laccophilini.

### *Laccophilus* Leach, 1815
Nomenclatural Authority: *Nilsson & Hájek (2018)*
Notes. Six species of *Laccophilus* are known to occur in California (*Challet & Brett, 1998*).

### *Laccophilus fasciatus* Aubé, 1838
Nomenclatural Authority: *Nilsson & Hájek (2018)*
Literature Records: Santa Catalina (*Challet, 1987*: 13)
Digitized Records: San Clemente (16 SBMNH), Santa Catalina (34 LACM; 1 SBMNH)
Range: Also known from mainland (*Challet & Brett, 1998*).

Notes. *Challet (1987)* indicated that the subspecies occurring on the islands is *L. f. terminalis* Sharp, 1882.

### *Laccophilus maculosus* (Germar, 1823)
Nomenclatural Authority: *Nilsson & Hájek (2018)*
Literature Records: Santa Catalina (*Challet, 1987*: 13)
Digitized Records: Santa Catalina (24 LACM; 6 SBMNH)
Range: Also known from mainland (*Challet & Brett, 1998*).
Notes. *Challet (1987)* indicated the subspecies occurring on the islands is *L. m. decipiens* LeConte, 1852.

### Gyrinidae
Notes. Two genera and 11 species of this family are known to occur in California, all of which belong to the subfamily Gyrininae (M. L. Gimmel, 2022, unpublished data).

### *Gyrinus* Müller, 1764
Nomenclatural Authority: *Oygur & Wolfe (1991)*
Notes. Nine species of *Gyrinus* are known to occur in California (*Oygur & Wolfe, 1991*). Additional species of *Gyrinus* are likely to occur on the Channel Islands. *Oygur & Wolfe (1991)* revised the genus for North America and provided an identification key to species.

### *Gyrinus plicifer* LeConte, 1852
Nomenclatural Authority: *Oygur & Wolfe (1991)*
Literature Records: Santa Cruz (*Furlong & Wenner, 2002*: 250)
Digitized Records: Santa Catalina (6 SBMNH), Santa Cruz (1 CASC; 20 SBMNH; 3 UCSB), Santa Rosa (8 SBMNH)
Range: Also known from mainland (*Oygur & Wolfe, 1991*).

### Haliplidae
Notes. Three genera and 20 species of this family are known to occur in California (*van Vondel, 2021*). *van Vondel (2021)* provided a recent revision of the North American species. *Leech & Chandler (1956)* provided keys to the California species known at the time.

### *Haliplus* Latreille, 1802
Nomenclatural Authority: *van Vondel (2021)*
Notes. Thirteen species of *Haliplus* have been reported to occur in California (*van Vondel, 2021*).

### *Haliplus* undetermined species
Literature Records: none
Digitized Records: Santa Catalina (4 LACM)

### *Peltodytes* Régimbart, 1879
Nomenclatural Authority: *van Vondel (2021)*
Notes. Five species of *Peltodytes* have been reported to occur in California (*van Vondel, 2021*).

*Peltodytes* (*Neopeltodytes*) *simplex* (LeConte, 1852)
Nomenclatural Authority: *van Vondel (2021)*
Literature Records: Santa Catalina (*van Vondel, 2021*: 258), Santa Cruz (*Furlong & Wenner, 2002*: 250; *van Vondel, 2021*: 258)
Digitized Records: Santa Catalina (1 SBMNH), Santa Cruz (17 SBMNH; 1 UCSB), Santa Rosa (9 SBMNH)
Range: Also known from mainland (*van Vondel, 2021*).

## MYXOPHAGA

### Hydroscaphidae
Notes. Only one species of this family is known to occur in California (*Reichardt, 1973*). The New World Hydroscaphidae were treated by *Reichardt & Hinton (1976)*.

*Hydroscapha* LeConte, 1874
Nomenclatural Authority: *Reichardt & Hinton (1976)*
Notes. One species of *Hydroscapha* is known from California (*Reichardt, 1973*).

*Hydroscapha natans* LeConte, 1874
Nomenclatural Authority: *Reichardt & Hinton (1976)*
Literature Records: Santa Cruz (*Furlong & Wenner, 2002*: 250)
Digitized Records: Santa Cruz (9 SBMNH), Santa Rosa (9 SBMNH)
Range: Also known from mainland (*Reichardt & Hinton, 1976*).

### Sphaeriusidae, NEW FAMILY RECORD
Notes. One species of this family is known to occur in California (*Reichardt, 1973*). This family has also been known as Sphaeridae, Sphaeriidae, and Microsporidae in the literature.

*Sphaerius* Waltl, 1838
Nomenclatural Authority: *Reichardt (1973)*
Notes. One species of *Sphaerius* has been recorded from California (*Reichardt, 1973*).

*Sphaerius politus* Horn, 1868
Nomenclatural Authority: *Reichardt (1973)*
Literature Records: none
Digitized Records: Santa Catalina (4 SBMNH), Santa Cruz (11 SBMNH)
Range: Also known from mainland (*Horn, 1868*).
Notes. This species was described from Visalia, Tulare County, California (*Horn, 1868*).

## POLYPHAGA

## SCIRTOIDEA

### Scirtidae
Notes. Four genera and 14 species of Scirtidae are known from California (M. L. Gimmel, 2022, unpublished data).

Scirtidae undetermined genus and species
Literature Records: Santa Cruz (*Furlong & Wenner, 2002*: 250)
Digitized Records: none
Notes. *Furlong & Wenner (2002)* did not indicate number of specimens, adult or larva, or specimen deposition of their record of "Scirtidae" from Santa Cruz Island.

## CLAMBOIDEA

### Clambidae
Notes. There are two genera and five species of Clambidae recorded from California (*Endrődy-Younga, 1981*). This family was treated for North America by *Endrődy-Younga (1981)*, who provided keys to all known species.

### *Clambus* Fischer von Waldheim, 1820
Nomenclatural Authority: *Endrődy-Younga (1981)*
Notes. Four species of *Clambus* have been recorded from California (*Endrődy-Younga, 1981*).

### *Clambus* undetermined species
Literature Records: none
Digitized Records: Santa Cruz (1 SBMNH)
Notes. The Santa Cruz Island specimen housed in SBMNH is a female, and therefore cannot presently be determined to species (M. L. Gimmel, 2021, personal observation).

### *Loricaster* Mulsant & Rey, 1861
Nomenclatural Authority: *Endrődy-Younga (1981)*
Notes. One species of *Loricaster* has been recorded from California (*Endrődy-Younga, 1981*).

### *Loricaster rotundus* Grigarick & Schuster, 1961
Nomenclatural Authority: *Endrődy-Younga (1981)*
Literature Records: none
Digitized Records: San Clemente (2 SBMNH), Santa Catalina (10 SBMNH)
Range: Also known from mainland (*Endrődy-Younga, 1981*).
Notes. The genus *Loricaster* was recorded from Santa Catalina Island by *Caterino & Chandler (2010)*: 191); that record certainly refers to this species.

## DASCILLOIDEA

### Dascillidae, NEW FAMILY RECORD
Notes. Two subfamilies, two genera, and three species of Dascillidae are known to occur in California (*Johnston & Gimmel, 2020*). *Johnston & Gimmel (2020)* reviewed the family for North America.

### Karumiinae
Notes. One genus and two species of Karumiinae have been recorded from California (*Johnston & Gimmel, 2020*).

*Anorus* **LeConte, 1859**
Nomenclatural Authority: *Johnston & Gimmel (2020)*
Notes. Two species of *Anorus* have been recorded from California (*Johnston & Gimmel, 2020*).

*Anorus piceus* **LeConte, 1859**
Nomenclatural Authority: *Johnston & Gimmel (2020)*
Literature Records: none
Digitized Records: San Clemente (1 LACM), Santa Catalina (3 LACM), Santa Cruz (4 LACM), Santa Rosa (23 LACM)
Range: Also known from mainland (*Johnston & Gimmel, 2020*).
Notes. *Johnston & Gimmel (2020)* saw no specimens from the Channel Islands during the course of their study. The specimens listed above were examined by us and are certainly within the *A. piceus* concept of the recent revision and are reliably labeled from the islands from multiple collecting events. The flightless morphology and life history of females of this species (*Johnston & Gimmel, 2020*) presents a fascinating question of how these island populations became established.

**Rhipiceridae, NEW FAMILY RECORD**
Notes. One genus and two species of Rhipiceridae are known to occur in California (*Schnepp & Powell, 2018*).

*Sandalus* **Knoch, 1801**
Nomenclatural Authority: *Schnepp & Powell (2018)*
Notes. Two species of *Sandalus* have been recorded from California (*Schnepp & Powell, 2018*).

*Sandalus cribricollis* **Van Dyke, 1923**
Nomenclatural Authority: *Schnepp & Powell (2018)*
Literature Records: none
Digitized Records: Santa Catalina (1 UCRC)
Range: Also known from mainland (*Van Dyke, 1923*).

**BUPRESTOIDEA**

**Buprestidae**
Notes. Four subfamilies, 35 genera, and 319 species of Buprestidae are known to occur in California (*Nelson et al., 2008*; M. L. Gimmel, 2022, unpublished data). This family was the subject of an extensive distributional catalog and bibliography by *Nelson et al. (2008)*.

**Agrilinae: Agrilini**
Notes. Two tribes, three genera, and 33 species of Agrilinae are known to occur in California, of which just one species belongs to Tracheini and the remainder to Agrilini (*Nelson et al., 2008*; M. L. Gimmel, 2022, unpublished data).

*Agrilus* **Curtis, 1825**
Nomenclatural Authority: *Nelson et al. (2008)*

Notes. Thirty-one species of *Agrilus* have been reported from California (*Nelson et al., 2008*; M. L. Gimmel, 2022, unpublished data).

### *Agrilus quadriguttatus* Gory, 1841
Nomenclatural Authority: *Nelson et al. (2008)*
Literature Records: none
Digitized Records: Santa Cruz (14 SBMNH)
Range: Also known from mainland (*Nelson et al., 2008*).
Notes. The subspecies of *A. quadriguttatus* occurring on the Channel Islands is *A. q. niveiventris* Horn, 1891 (N. Woodley, 2021, personal communication).

### Buprestinae
Notes. Seven tribes, 15 genera, and 132 species of Buprestinae are known to occur in California (*Nelson et al., 2008*; M. L. Gimmel, 2022, unpublished data).

### Anthaxiini
Notes. One genus and 24 species of Anthaxiini are known to occur in California (*Nelson et al., 2008*).

### *Anthaxia* Eschscholtz, 1829
Nomenclatural Authority: *Nelson et al. (2008)*
Digitized Records (genus-only): Santa Cruz (9 SBMNH; 6 UCRC; 1 UCSB)
Notes. Twenty-four species of *Anthaxia* have been reported from California (*Nelson et al., 2008*). All but one of the undetermined specimens reported above belong to the *A. aeneogaster* species group, a taxonomic complex that needs revision (N. Woodley, 2021, personal communication).

### *Anthaxia* (*Melanthaxia*) *aeneogaster* Gory & Laporte, 1839
Nomenclatural Authority: *Nelson et al. (2008)*
Literature Records: none
Digitized Records: Santa Cruz (4 SBMNH)
Range: Also known from mainland (*Nelson et al., 2008*).

### Buprestini
Notes. Three genera and 19 species of Buprestini are known to occur in California (*Nelson et al., 2008*).

### *Buprestis* Linnaeus, 1758
Nomenclatural Authority: *Nelson et al. (2008)*
Notes. Fourteen species of *Buprestis* have been reported from California (*Nelson et al., 2008*). The genus was revised for North America by *Helfer (1941)*.

### *Buprestis* (*Cypriacis*) *aurulenta* Linnaeus, 1767
Nomenclatural Authority: *Nelson et al. (2008)*
Literature Records: Santa Cruz (*Cockerell, 1940*: 286)
Digitized Records: Santa Cruz (6 CASC; 6 SBMNH)

Range: Also known from mainland (*Nelson et al., 2008*).

**Chrysobothrini**
Notes. Three genera and 68 species of Chrysobothrini are known to occur in California (*Nelson et al., 2008*; M. L. Gimmel, 2022, unpublished data).

*Chrysobothris* **Eschscholtz, 1829**
Nomenclatural Authority: *Nelson et al. (2008)*
Notes. Sixty-five species of *Chrysobothris* have been recorded from California (M. L. Gimmel, 2022, unpublished data).

*Chrysobothris mali* **Horn, 1886**
Nomenclatural Authority: *Nelson et al. (2008)*
Literature Records: none
Digitized Records: Santa Catalina (1 CASC), Santa Cruz (1 SBMNH)
Range: Also known from mainland (*Nelson et al., 2008*).

**Melanophilini**
Notes. Four genera and 13 species of Melanophilini have been recorded from California (*Nelson et al., 2008*).

*Melanophila* **Eschscholtz, 1829**
Nomenclatural Authority: *Nelson et al. (2008)*
Notes. Four species of *Melanophila* have been recorded from California (*Nelson et al., 2008*). The genus was revised for North America by *Sloop (1937)*.

*Melanophila consputa* **LeConte, 1857**
Nomenclatural Authority: *Nelson et al. (2008)*
Literature Records: none
Digitized Records: Santa Catalina (4 iNat)
Range: Also known from mainland (*Nelson et al., 2008*).

**Polycestinae: Acmaeoderini**
Notes. Five tribes, eight genera, and 119 species of Polycestinae are known to occur in California, of which four genera and 102 species belong to Acmaeoderini (*Nelson et al., 2008*).

*Acmaeodera* **Eschscholtz, 1829**
Nomenclatural Authority: *Nelson et al. (2008)*
Digitized Records (genus-only): Santa Cruz (4 UCSB)
Notes. *Cockerell (1940*: 286) relayed information from H.C. Fall that *Acmaeodera connexa* LeConte, 1859 (which he misspelled as "*A. convexa*") was erroneously recorded from Santa Rosa (*Fall, 1897*: 237). A total of 78 species of *Acmaeodera* has been recorded from California (*Nelson et al., 2008*).

***Acmaeodera* (*Acmaeodera*) *hepburnii* LeConte, 1860**
Nomenclatural Authority: *Nelson et al. (2008)*
Literature Records: Santa Catalina (*Bellamy, 1982*: 359), Santa Cruz (*Nelson, 1962*: 56; *Bellamy, 1982*: 359), Santa Rosa (*Fall, 1897*: 237; *Cockerell, 1940*: 286)
Digitized Records: Santa Catalina (1 SBMNH), Santa Cruz (41 CASC; 10 LACM; 29 SBMNH; 6 UCSB; 1 iNat), Santa Rosa (2 SBMNH)
Range: Also known from mainland (*Nelson, 1962*; *Nelson et al., 2008*).
Notes. Often misspelled *Acmaeodera hepburni*.

***Acmaeodera* (*Acmaeodera*) *mariposa* Horn, 1878**
Nomenclatural Authority: *Nelson et al. (2008)*
Literature Records: none
Digitized Records: Santa Cruz (3 SBMNH)
Range: Also known from mainland (*Nelson et al., 2008*).
Notes. The subspecies occurring on the islands is *A. m. dohrni* Horn, 1878 (N. Woodley, 2021, personal communication).

***Acmaeodera* (*Acmaeodera*) *prorsa* Fall, 1899**
Nomenclatural Authority: *Nelson et al. (2008)*
Literature Records: Santa Cruz (*Bellamy, 1982*: 359)
Digitized Records: Santa Catalina (1 CASC), Santa Cruz (1 CASC; 1 SBMNH; 1 UCSB)
Range: Also known from mainland (*Bellamy, 1982*; *Nelson et al., 2008*).

**DRYOPOIDEA**

**Dryopidae**
Notes. Three genera and five species of Dryopidae are known to occur in California (*Shepard, 1993*). *Shepard (1993)* provided the most recent published checklist of California Dryopidae. *Brown (1972)*, although outdated, remains the best identification guide for North America.

***Postelichus* Nelson, 1989**
Nomenclatural Authority: *Shepard (1993)*
Notes. Two species of *Postelichus* have been recorded from California (*Shepard, 1993*; *Barr & Shepard, 2022*). A key to the species of *Postelichus* was provided by *Barr & Shepard (2022)*.

***Postelichus productus* (LeConte, 1852)**
Nomenclatural Authority: *Barr & Shepard (2022)*
Literature Records: Santa Catalina (*Fall, 1897*: 237)
Digitized Records: none
Range: Also known from mainland (*Brown, 1972*; *Barr & Shepard, 2022*).
Notes. *Fall (1897)* reported this species as *Dryops productus*.

**Elmidae**

Notes. Fourteen genera and 24 species of Elmidae are known to occur in California (M. L. Gimmel, 2022, unpublished data). *Shepard (1993)* provided the most recent published checklist of California Elmidae. *Brown (1972)*, although outdated, remains the best identification guide for North America.

***Ordobrevia* Sanderson, 1953**

Nomenclatural Authority: *Shepard (1993)*

Notes. One species of *Ordobrevia* has been recorded from California (*Shepard, 1993*).

***Ordobrevia nubifera* (Fall, 1901)**

Nomenclatural Authority: *Shepard (1993)*

Literature Records: Santa Cruz (*Furlong & Wenner, 2002*: 250)

Digitized Records: none

Range: Also known from mainland (*Brown, 1972*).

**Heteroceridae**

Notes. Three genera and 11 species of Heteroceridae are known to occur in California (M. L. Gimmel, 2022, unpublished data). *Shepard (1993)* provided the most recent published checklist of California Heteroceridae. *Pacheco (1964)* monographed the species for the New World, though most of his genera were not recognized by *King, Starr & Lago (2011)*.

***Heterocerus* Fabricius, 1792**

Nomenclatural Authority: *Shepard (1993)*

Digitized Records (genus-only): San Clemente (11 SBMNH), San Nicolas (5 SBMNH), Santa Catalina (7 SBMNH), Santa Cruz (7 SBMNH), Santa Rosa (16 SBMNH)

Notes. Eight species of *Heterocerus* have been recorded from California (*Shepard, 1993*).

***Heterocerus mexicanus* Sharp, 1882**

Nomenclatural Authority: *Shepard (1993)*, *King, Starr & Lago (2011)*

Literature Records: Santa Cruz (*Furlong & Wenner, 2002*: 250)

Digitized Records: Santa Cruz (3 SBMNH)

Range: Also known from mainland (*Brown, 1972*).

Notes. This species has been known in the literature (*Pacheco, 1964*; *Shepard, 1993*) as *Dampfius mexicanus*. However, the genus *Dampfius* Pacheco, 1964 was synonymized with *Heterocerus* by *King, Starr & Lago (2011)*.

**Limnichidae, NEW FAMILY RECORD**

Notes. Six genera and 13 species of Limnichidae are known to occur in California (*Shepard, 1993*). *Shepard (1993)* provided the most recent published checklist of California Limnichidae.

***Limnichites* Casey, 1889**

Nomenclatural Authority: *Wooldridge (1977)*

Notes. Three species of *Limnichites* have been recorded from California (*Shepard, 1993*). *Wooldridge (1977)* provided the most recent revision of this genus.

### *Limnichites nebulosus* (LeConte, 1879)
Nomenclatural Authority: *Wooldridge (1977)*
Literature Records: none
Digitized Records: Santa Cruz (5 SBMNH)
Range: Also known from mainland (*Wooldridge, 1977*).

## ELATEROIDEA

### Cantharidae
Notes. Four subfamilies, 13 genera, and 157 species of Cantharidae have been recorded from California (M. L. Gimmel, 2022, unpublished data).

### Cantharinae
Notes. Two tribes, seven genera, and 51 species of Cantharinae have been recorded from California (M. L. Gimmel, 2022, unpublished data).

### Cantharini
Notes. Five genera and 16 species of Cantharini have been recorded from California (M. L. Gimmel, 2022, unpublished data).

### *Cultellunguis* McKey-Fender, 1950
Nomenclatural Authority: *Ramsdale (2002)*
Digitized Records (genus-only): Santa Catalina (3 SBMNH), Santa Cruz (8 SBMNH)
Notes. This genus contains nine species restricted to the Pacific coast of North America (*Ramsdale, 2002*), all of which occur in California (*McKey-Fender, 1950*).

### *Cultellunguis americanus* (Pic, 1906)
Nomenclatural Authority: *McKey-Fender (1950)*, *Ramsdale (2002)*
Literature Records: Santa Catalina (*Fall, 1897*: 237)
Digitized Records: none
Range: Also known from mainland (*McKey-Fender, 1950*).
Notes. Listed as "*Telephorus notatus* Mann. var." by *Fall (1897)*. *Telephorus notatus* Mannerheim, 1843 is an unavailable homonym which was replaced by *C. americanus* (see *McKey-Fender, 1950*).

### *Cultellunguis hatchi* (McKey-Fender, 1950)
Nomenclatural Authority: *McKey-Fender (1950)*, *Fender (1968)*, *Ramsdale (2002)*
Literature Records: Santa Catalina (*McKey-Fender, 1950*: 65; *Fender, 1968*: 301; *Miller, 1985a*: 21), Santa Cruz (*McKey-Fender, 1950*: 65; *Naughton et al., 2014*: 303)
Digitized Records: Santa Cruz (13 SBMNH)
Range: Also known from mainland (*McKey-Fender, 1950*; *Fender, 1968*).
Notes. This species, as *Cantharis* (*Cultellunguis*) *hatchi* McKey-Fender, was originally recorded from both Santa Catalina and Santa Cruz by *McKey-Fender (1950)*; however, a

Santa Catalina subspecies, as *Cantharis* (*Cultellunguis*) *hatchi dorothyae Fender, 1968*, was later split off by *Fender (1968)*. This subspecies, considered to be endemic to Santa Catalina, is now known as *Cultellunguis h. dorothyae* (*Fender, 1968*), while the nominate subspecies, *Cultellunguis h. hatchi* (*McKey-Fender, 1950*), occurs on Santa Cruz Island (*Fender, 1968*; *Miller, 1985a*). The latter subspecies also occurs on the mainland (*Fender, 1968*).

### *Pacificanthia* Kazantsev, 2002

Nomenclatural Authority: *Kazantsev (2002)*

Notes. One species of the genus *Pacificanthia* occurs in California (*Kazantsev, 2002*). *Kazantsev (2002)* provided a generic description and key to species for the genus.

### *Pacificanthia consors* (LeConte, 1851)

Nomenclatural Authority: *Kazantsev (2002)*

Literature Records: Santa Cruz (*Fall & Davis, 1934*: 144)

Digitized Records: Santa Catalina (32 LACM; 3 SBMNH; 1 iNat), Santa Cruz (11 LACM; 15 SBMNH; 8 UCSB), Santa Rosa (1 LACM)

Range: Also known from mainland (*Kazantsev, 2002*).

Notes. *Fall & Davis (1934)* recorded this species as *Cantharis consors*.

### Podabrini

Notes. The two California genera of tribe Podabrini, *Dichelotarsus* Motschulsky, 1860 and *Podabrus* Westwood, 1838, while both valid, have not had their species properly assigned yet (*Ramsdale, 2002*). A collective total of 35 species belonging to both genera is known to occur in California (M. L. Gimmel, 2022, unpublished data).

### *Podabrus pruinosus* LeConte, 1851

Nomenclatural Authority: *Fall (1927)*, *Fender (1949)*

Literature Records: none

Digitized Records: Santa Cruz (13 SBMNH)

Range: Also known from mainland (*Fall, 1927*).

Notes. This species was included in *Fender's (1949)* "Group I", and therefore will probably stay in the genus *Podabrus* (see *Ramsdale, 2002*). The subspecies occurring on the islands is *P. p. pruinosus* LeConte, 1851.

### *Podabrus* undetermined species

Nomenclatural Authority: *Fender (1948)*, *Fender (1949)*

Literature Records: none

Digitized Records: Santa Cruz (1 SBMNH)

Notes. This species keyed to *Fender's (1949)* "Group VIII", and therefore may end up in the genus *Dichelotarsus* (see *Ramsdale, 2002*). However, it did not fit any of the species concepts in *Fender's (1948)* revision of that group (M. L. Gimmel, 2021, personal observation).

### Malthininae

Notes. Two genera and 60 species of Malthininae are known to occur in California (M. L. Gimmel, 2022, unpublished data).

### *Frostia* **Fender, 1951**

Nomenclatural Authority: *Fender (1951)*

Notes. Four species of *Frostia* have been recorded from California (*Fender, 1951*). This genus was described and revised by *Fender (1951)*.

### *Frostia laticollis* (LeConte, 1866)

Nomenclatural Authority: *Fender (1951)*

Literature Records: Santa Cruz (*LeConte, 1861*: 351; *Fall, 1897*: 237; *Fall, 1919*: 35; *Fall & Davis, 1934*: 144; *Naughton et al., 2014*: 303)

Digitized Records: Santa Cruz (20 SBMNH)

Range: Also known from mainland (*Fall, 1919*; *Fender, 1951*).

Notes. *LeConte (1861)* recorded this species as *Malthodes transversus* LeConte, 1861, but the name was subsequently corrected to *Malthodes laticollis* by *LeConte (1866b*: 53).
At that time the species was considered to be endemic to Santa Cruz Island. *Fall (1897, 1919)* and *Fall & Davis (1934)* recorded this species as *Malthodes laticollis*. *Fender (1951*: 524) transferred this species to *Frostia*. *Naughton et al. (2014)* reported two specimens of the genus *Frostia* from Santa Cruz Island; one voucher (in SBMNH) was identified by MLG as *F. laticollis*.

### Silinae

Notes. Three genera and 43 species of Silinae are known to occur in California (M. L. Gimmel, 2022, unpublished data).

### *Silis* **Charpentier, 1825**

Nomenclatural Authority: *Ramsdale (2002)*

Digitized Records (genus-only): Santa Cruz (1 SBMNH)

Notes. Forty species of *Silis* are known to occur in California (M. L. Gimmel, 2022, unpublished data). *Green (1966)* revised the species of *Silis* known from North America at the time. The specimen the above record is based on is female, and therefore not identifiable to species (*Green, 1966*). However, it is likely that it belongs to *S. carmelita*, cited below.

### *Silis carmelita* **Green, 1966**

Nomenclatural Authority: *Green (1966)*

Literature Records: none

Digitized Records: Santa Rosa (1 SBMNH)

Range: Also known from mainland (*Green, 1966*).

Notes. The single specimen is a male, which was dissected for examination of the genitalia for identification (M. L. Gimmel, 2021, personal observation).

**Elateridae**

Notes. Eight subfamilies, 67 genera, and 362 species of Elateridae are known to occur in California (M. L. Gimmel, 2022, unpublished data).

**Agrypninae: Oophorini**

Notes. Three tribes, nine genera, and 23 species of Agrypninae are known to occur in California, of which four genera and 11 species belong to Oophorini (M. L. Gimmel, 2022, unpublished data).

***Heteroderes* Latreille, 1834**

Nomenclatural Authority: *Kundrata et al. (2019)*

Notes. Two species of *Heteroderes* are known from California (M. L. Gimmel, 2022, unpublished data).

***Heteroderes amplicollis* (Gyllenhal, 1808)**

Nomenclatural Authority: *Stone (1975)*

Literature Records: none

Digitized Records: Santa Rosa (1 LACM)

Range: Also known from mainland (*Stone, 1975*).

Notes. This species is a destructive pest introduced from South America (*Stone, 1975*). It has often been included in *Conoderus* Eschscholtz, 1829 in the literature (including by *Stone (1975)*), but is now commonly treated as a *Heteroderes* and will soon be moved to a new genus (P. Johnson, 2021, personal communication).

**Cardiophorinae**

Notes. Five genera and 52 species of Cardiophorinae are known to occur in California (M. L. Gimmel, 2022, unpublished data).

***Cardiophorus* Eschscholtz, 1829**

Nomenclatural Authority: *Douglas (2003)*

Digitized Records (genus-only): Santa Cruz (1 SBMNH), Santa Rosa (5 SBMNH)

Notes. One of the Santa Rosa Island specimens above was identified by Hume Douglas during 2006 as "*Cardiophorus tenebrosus* group". The other specimens from both islands are similar in appearance and likely represent the same species, which may be conspecific with the specimens identified as *C. tenebrosus* below. However, this is a highly diverse genus with 28 described species of *Cardiophorus* recorded from California in the literature (M. L. Gimmel, 2022, unpublished data), even after the recent splitting off of *Paracardiophorus* Schwarz, 1895 by *Douglas (2017)* (13 species recorded from California; M. L. Gimmel, 2022, unpublished data).

***Cardiophorus tenebrosus* LeConte, 1853**

Nomenclatural Authority: *Douglas (2003)*

Literature Records: none

Digitized Records: Santa Cruz (2 SBMNH)

Range: Also known from mainland (M. L. Gimmel, 2022, unpublished data).

Notes. A specimen in the H.C. Fall collection in the Museum of Comparative Zoology, Harvard University from San Clemente Island was questionably identified as this species (S. Miller, 2022, personal communication). This record needs verification.

### *Horistonotus* Candèze, 1860
Nomenclatural Authority: *Wells (2000)*
Notes. Five species of this genus have been recorded from California (M. L. Gimmel, 2022, unpublished data). *Wells (2000)* provided a key to all North American species.

### *Horistonotus inanus* (LeConte, 1853)
Nomenclatural Authority: *Wells (2000)*
Literature Records: none
Digitized Records: Santa Catalina (10 SBMNH), Santa Cruz (1 SBMNH)
Range: Also known from mainland (*Wells, 2000*).
Notes. This is a dimorphic species with regard to dorsal color pattern, with some specimens showing distinct light coloration at the base of the elytra. However, all examined specimens from the Channel Islands are not or very weakly bicolored.

### Dendrometrinae; Dendrometrini
Notes. Two tribes, 23 genera, and 138 species of Dendrometrinae have been recorded from California, of which eight genera and 62 species belong to Dendrometrini (M. L. Gimmel, 2022, unpublished data).

### *Athous* Eschscholtz, 1829
Nomenclatural Authority: *Etzler (2020b)*
Notes. Seventeen species of *Athous* have been recorded from California (*Becker, 1979*).

### *Athous axillaris* Horn, 1871
Nomenclatural Authority: *Becker (1979)*
Literature Records: none
Digitized Records: Santa Cruz (17 SBMNH), Santa Rosa (7 SBMNH)
Range: Also known from mainland (*Becker, 1979*).

### *Athous nigropilis* Motschulsky, 1859
Nomenclatural Authority: *Becker (1979)*
Literature Records: none
Digitized Records: Santa Catalina (3 LACM)
Range: Also known from mainland (*Becker, 1979*).

### *Athous rufiventris* (Eschscholtz, 1822)
Nomenclatural Authority: *Becker (1979)*
Literature Records: none
Digitized Records: Santa Catalina (3 SBMNH), Santa Cruz (3 SBMNH)
Range: Also known from mainland (*Becker, 1979*).

*Hemicrepidius* Germar, 1839
Nomenclatural Authority: *Etzler (2020b)*
Notes. Twelve species of *Hemicrepidius* have been recorded from California (*Etzler, 2020b*).

*Hemicrepidius californicus* Becker, 1979
Nomenclatural Authority: *Etzler (2020b)*
Literature Records: none
Digitized Records: San Miguel (2 SBMNH), San Nicolas (4 LACM; 2 SBMNH)
Range: Also known from mainland (*Etzler, 2020b*).

*Hemicrepidius tumescens* (LeConte, 1861)
Nomenclatural Authority: *Etzler (2020b)*
Literature Records: Santa Cruz (*LeConte, 1861*: 348; *Fall, 1897*: 237; *Fall, 1901*: 114; *Van Dyke, 1932*: 444; *Etzler, 2020b*: 86, 87)
Range: Also known from mainland (*Van Dyke, 1932*; *Etzler, 2020b*).
Notes. Recorded by *LeConte (1861)* and *Fall (1897, 1901)* as *Asaphes tumescens*. At the time of *LeConte (1861)*, this species was considered to be endemic to Santa Cruz Island.

*Limonius* Eschscholtz, 1829
Nomenclatural Authority: *Etzler (2019)*
Notes. Eleven species of *Limonius* are known to occur in California (M. L. Gimmel, 2022, unpublished data). While *Al Dhafer (2009)* revised the North American species of *Limonius*, *Etzler (2019)* reclassified the world species to various genera.

*Limonius canus* LeConte, 1853
Nomenclatural Authority: *Al Dhafer (2009)*, *Etzler (2019)*
Literature Records: none
Digitized Records: San Clemente (1 LACM), Santa Cruz (1 SBMNH)
Range: Also known from mainland (*Al Dhafer, 2009*).

**Elaterinae**
Notes. Five tribes, 17 genera, and 127 species of Elaterinae are known to occur in California (M. L. Gimmel, 2022, unpublished data).

**Agriotini**
Notes. Four genera and 41 species of Agriotini are known to occur in California (M. L. Gimmel, 2022, unpublished data).

*Dalopius* Eschscholtz, 1829
Nomenclatural Authority: *Brown (1934)*
Digitized Records: Santa Catalina (1 LACM), Santa Cruz (1 LACM)
Notes. Twenty-four species of *Dalopius* have been recorded from California (M. L. Gimmel, 2022, unpublished data). This genus was revised for North America in a multi-part paper by *Brown (1934)*.

***Dalopius luteolus* Brown, 1934**
Nomenclatural Authority: *Brown (1934)*
Literature Records: none
Digitized Records: Santa Cruz (2 LACM)
Range: Also known from mainland (*Brown, 1934*).

***Dalopius* undetermined species**
Literature Records: none
Digitized Records: Santa Cruz (17 SBMNH), Santa Rosa (1 SBMNH)
Notes. The rather uniform specimens above (SBMNH) were examined and they do not match the description of *D. luteolus*. One male was dissected and the median lobe does not match any of the illustrated species in *Brown (1934)*, though it is fairly close to that of *Dalopius partitus* Brown, 1934. The species may be undescribed.

**Ampedini**
Notes. Four genera and 35 species of Ampedini are known to occur in California (M. L. Gimmel, 2022, unpublished data).

***Ampedus* Dejean, 1833**
Nomenclatural Authority: *Ramberg (1979)*
Notes. Twenty-four species of *Ampedus* have been recorded from California (*Ramberg, 1979*). *Ramberg (1979)* revised the species for North America. Unfortunately, most of the taxonomic acts in this thesis work have not been validly published.

***Ampedus longicornis* (LeConte, 1884)**
Nomenclatural Authority: *Ramberg (1979)*
Literature Records: Santa Catalina (*Ramberg, 1979*: 318)
Digitized Records: Santa Catalina (2 LACM; 2 SBMNH)
Range: Also known from mainland (*Ramberg, 1979*).

***Ampedus rhodopus* (LeConte, 1884)**
Nomenclatural Authority: *Ramberg (1979)*
Literature Records: none
Digitized Records: Santa Rosa (3 SBMNH)
Range: Also known from mainland (*Ramberg, 1979*).

***Anchastus* LeConte, 1853**
Nomenclatural Authority: *Johnson (2002)*
Notes. Six species of *Anchastus* are known to occur in California (M. L. Gimmel, 2022, unpublished data). *Van Dyke (1932)* provided a key to the North American species of *Anchastus*.

***Anchastus cinereipennis* (Eschscholtz, 1829)**
Nomenclatural Authority: *Van Dyke (1932)*
Literature Records: San Nicolas (*Miller & Miller, 1985*: 126), Santa Barbara (*Miller & Miller, 1985*: 126)

Digitized Records: San Clemente (3 SBMNH), San Nicolas (6 SBMNH), Santa Catalina (1 SBMNH), Santa Rosa (1 iNat)

Range: Also known from mainland (M. L. Gimmel, 2022, unpublished data).

### *Melanotus* Eschscholtz, 1829

Nomenclatural Authority: *Johnson (2002)*

Notes. Four species of *Melanotus* have been recorded from California (*Quate & Thompson, 1967*). The genus was revised for North America by *Quate & Thompson (1967)*.

### *Melanotus longulus* (LeConte, 1853)

Nomenclatural Authority: *Quate & Thompson (1967)*

Literature Records: Santa Catalina (*Fall, 1897*: 237; *Fall, 1901*: 111; *Quate & Thompson, 1967*: 61)

Digitized Records: Santa Catalina (1 LACM; 12 SBMNH; 6 TAMU), Santa Cruz (10 SBMNH), Santa Rosa (2 SBMNH)

Range: Also known from mainland (*Quate & Thompson, 1967*).

Notes. The nominate subspecies, *M. l. longulus* (LeConte, 1853), is the only subspecies of *M. longulus* known from south of the Tehachapi Mountains in southern California (*Quate & Thompson, 1967*). This species was recorded as *Melanotus variolatus* LeConte, 1861 by *Fall (1897, 1901)*, which was synonymized with *M. longulus* by *Quate & Thompson (1967)*.

### Aplastini

Notes. Three genera and 28 species of Aplastini are known to occur in California (M. L. Gimmel, 2022, unpublished data).

### *Euthysanius* LeConte, 1853

Nomenclatural Authority: *Johnson (2002)*

Notes. Seven species of *Euthysanius* are known to occur in California (*Johnson, 2002*). *Van Dyke (1932)* provided a key to the North American species of *Euthysanius*.

### *Euthysanius lautus* LeConte, 1853

Nomenclatural Authority: *Van Dyke (1932)*

Literature Records: none

Digitized Records: Santa Cruz (9 LACM), Santa Rosa (1 LACM; 1 SBMNH)

Range: Also known from mainland (*Van Dyke, 1932*).

### *Octinodes* Candèze, 1863

Nomenclatural Authority: *Johnson (2002)*

Notes. Nine species of *Octinodes* are known to occur in California (*Johnson, 2002*). *Van Dyke (1932)* provided a key to separate some North American species of *Octinodes* (as *Plastocerus* LeConte, 1853).

### *Octinodes frater* (LeConte, 1859)

Nomenclatural Authority: *Van Dyke (1932)*

Literature Records: none

Digitized Records: Santa Cruz (1 LACM; 1 SBMNH)

Range: Also known from mainland (*Van Dyke, 1932*).

### Elaterini
Notes. Five genera and 12 species of Elaterini are known to occur in California (M. L. Gimmel, 2022, unpublished data).

### *Elater* Linnaeus, 1758
Nomenclatural Authority: *Johnson (2002)*
Notes. Four species of *Elater* are known from California (M. L. Gimmel, 2022, unpublished data). The genus was revised for North America by *Roache (1961)*.

### *Elater lecontei* (Horn, 1871)
Nomenclatural Authority: *Roache (1961)*
Literature Records: none
Digitized Records: Santa Cruz (3 SBMNH)
Range: Also known from mainland (*Roache, 1961*).

### Negastriinae
Notes. Five genera and 11 species of Negastriinae are known to occur in California (M. L. Gimmel, 2022, unpublished data).

### *Paradonus* Stibick, 1971
Nomenclatural Authority: *Etzler (2020a)*
Notes. Three species of *Paradonus* have been recorded from California (*Etzler, 2020a*).

### *Paradonus inops* (LeConte, 1853)
Nomenclatural Authority: *Etzler (2020a)*
Literature Records: none
Digitized Records: Santa Cruz (20 SBMNH)
Range: Also known from mainland (*Etzler, 2020a*).

### Oxynopterinae
Notes. Only two genera and two species of Oxynopterinae are known to occur in California (M. L. Gimmel, 2022, unpublished data).

### *Melanactes* LeConte, 1853
Nomenclatural Authority: *Mathieu (1961)*
Notes. Only one species of the widespread genus *Melanactes* has been recorded from California (*Mathieu, 1961*). The genus was revised by *Mathieu (1961)*.

### *Melanactes densus* LeConte, 1853
Nomenclatural Authority: *Mathieu (1961)*
Literature Records: none
Digitized Records: Santa Catalina (2 USNM)
Range: Also known from mainland (*Mathieu, 1961*).

## Eucnemidae

Notes. Six subfamilies, 14 genera, and 22 species of Eucnemidae have been recorded from California (*Muona, 2000*; M. L. Gimmel, 2022, unpublished data). The North American fauna was revised by *Muona (2000)*.

### Macraulacinae

Notes. Three genera and seven species of Macraulacinae have been recorded from California (*Muona, 2000*).

### *Asiocnemis* Mamaev, 1976

Nomenclatural Authority: *Muona (2000)*.
Notes. Five species of *Asiocnemis* have been reported from California (*Muona, 2000*).

### *Asiocnemis hospitalis* (Blanchard, 1904)

Nomenclatural Authority: *Muona (2000)*
Literature Records: Santa Rosa (*Muona, 2000*: 81)
Digitized Records: Santa Rosa (1 LACM)
Range: Also known from mainland (*Muona, 2000*).

## Lampyridae, NEW FAMILY RECORD

Notes. Three subfamilies, nine genera, and 22 species of Lampyridae are known to occur in California (M. L. Gimmel, 2022, unpublished data).

### Lampyrinae

Notes. Six genera and 16 species of Lampyrinae are known to occur in California (M. L. Gimmel, 2022, unpublished data).

### *Pyropyga* Motschulsky, 1852

Nomenclatural Authority: *Green (1961)*
Notes. One species of *Pyropyga* has been recorded from California (*Green, 1961*).

### *Pyropyga nigricans* (Say, 1823)

Nomenclatural Authority: *Green (1961)*
Literature Records: none
Digitized Records: Santa Cruz (1 UCSB), Santa Rosa (2 LACM)
Range: Also known from mainland (*Green, 1961*).

### Pterotinae

Notes. One genus and two species of Pterotinae have been recorded from California (*Chemsak, 1978*).

### *Pterotus* LeConte, 1859

Nomenclatural Authority: *Chemsak (1978)*
Notes. Two species of *Pterotus* are known, both of them occurring in California (*Chemsak, 1978*).

*Pterotus obscuripennis* LeConte, 1859
Nomenclatural Authority: *Chemsak (1978)*
Literature Records: none
Digitized Records: Santa Catalina (1 LACM; 3 SBMNH)
Range: Also known from mainland (*Chemsak, 1978*).

**Phengodidae, NEW FAMILY RECORD**
Notes. Two subfamilies, four genera, and seven species of Phengodidae are known to occur in California (M. L. Gimmel, 2022, unpublished data).

**Phengodinae**
Notes. Two genera and four species of Phengodinae have been recorded from California (M. L. Gimmel, 2022, unpublished data)

*Zarhipis* LeConte, 1880
Nomenclatural Authority: *Linsdale (1964)*
Notes. Three species of *Zarhipis* have been recorded from California (*Linsdale, 1964*). The genus was revised by *Linsdale (1964)*.

*Zarhipis integripennis* (LeConte, 1874)
Nomenclatural Authority: *Linsdale (1964)*
Literature Records: none
Digitized Records: Santa Catalina (1 LACM)
Range: Also known from mainland (*Linsdale, 1964*).

**Throscidae, NEW FAMILY RECORD**
Notes. Three genera and five species of Throscidae are known to occur in California (M. L. Gimmel, 2022, unpublished data).

*Trixagus* Kugelann, 1794
Nomenclatural Authority: *Yensen (1975)*
Notes. Three species of *Trixagus* have been recorded from California (*Yensen, 1975*).

*Trixagus sericeus* (LeConte, 1868)
Nomenclatural Authority: *Yensen (1975)*
Literature Records: none
Digitized Records: Santa Cruz (2 SBMNH)
Range: Also known from mainland (*Yensen, 1975*).

**HISTEROIDEA**

**Histeridae**
Notes. Seven subfamilies, 39 genera, and 141 species of Histeridae are known to occur in California (M. L. Gimmel, 2022, unpublished data). Although no histerid taxa below are recorded for Anacapa Island, LACM has undetermined material from that island.

**Abraeinae**

Notes. Three tribes, six genera, and 13 species of Abraeinae are known to occur in California (M. L. Gimmel, 2022, unpublished data).

**Abraeini**

Notes. Two genera and five species of Abraeini are known to occur in California (M. L. Gimmel, 2022, unpublished data).

***Plegaderus* Erichson, 1834**

Nomenclatural Authority: *Mazur (2011)*

Notes. Four species of *Plegaderus* are known to occur in California (M. L. Gimmel, 2022, unpublished data).

***Plegaderus* undetermined species**

Literature Records: none
Digitized Records: Santa Cruz (2 SBMNH)

**Acritini**

Notes. Three genera and four species of Acritini are known to occur in California (M. L. Gimmel, 2022, unpublished data).

***Halacritus* Schmidt, 1893**

Nomenclatural Authority: *Mazur (2011)*

Notes. One species of *Halacritus* has been recorded from California (*Mazur, 2011*).

***Halacritus maritimus* (LeConte, 1851)**

Nomenclatural Authority: *Mazur (2011)*
Literature Records: none
Digitized Records: San Clemente (1 SBMNH), San Nicolas (2 SBMNH)
Range: Also known from mainland.

**Dendrophilinae**

Notes. Four tribes, five genera, and 20 species of Dendrophilinae are known to occur in California (M. L. Gimmel, 2022, unpublished data).

**Bacaniini**

Notes. One genus and two species of Bacaniini have been recorded from California (*Mazur, 2011*).

***Bacanius* LeConte, 1853**

Nomenclatural Authority: *Mazur (2011)*

Notes. Two species of *Bacanius* have been recorded from California (*Mazur, 2011*).

***Bacanius* undetermined species**

Literature Records: Santa Catalina (*Caterino & Chandler, 2010*: 191)
Digitized Records: Santa Catalina (1 SBMNH)

Notes. The male specimen (SBMNH) from Santa Catalina Island does not appear to match either of the two species previously recorded for California, *Bacanius* (*Gomyister*) *acuminatus* Casey, 1893 or *Bacanius* (*s.str.*) *globulinus* Casey, 1893. It may represent an undescribed species.

**Paromalini**
Notes. Two genera and 12 species of Paromalini are known to occur in California (M. L. Gimmel, 2022, unpublished data).

*Carcinops* **Marseul, 1855**
Nomenclatural Authority: *Mazur (2011)*; *Reese & Swanson (2017)*
Notes. Ten species of *Carcinops* are known to occur in California (M. L. Gimmel, 2022, unpublished data). This genus was reported from Santa Barbara Island by *Miller & Miller (1985*: 123) from stems of *Coreopsis gigantea* (Kellogg) H.M. Hall (Asteraceae).

*Carcinops opuntiae* **(LeConte, 1851)**
Nomenclatural Authority: *Reese & Swanson (2017)*
Literature Records: none
Digitized Records: Santa Catalina (1 SBMNH)
Range: Also known from mainland (*Reese & Swanson, 2017*).

**Histerinae**
Notes. Four tribes, nine genera, and 30 species of Histerinae are known to occur in California (M. L. Gimmel, 2022, unpublished data).

**Histerini**
Notes. Five genera and 20 species of Histerini are known to occur in California (M. L. Gimmel, 2022, unpublished data).

*Margarinotus* **Marseul, 1854**
Nomenclatural Authority: *Mazur (2011)*
Notes. Eleven species of *Margarinotus* have been recorded from California (*Caterino, 2010*), belonging to two subgenera, *Paralister* Bickhardt, 1917 and *Ptomister* Houlbert & Monnot, 1922. The California species were revised by *Caterino (2010)*.

*Margarinotus* (*Ptomister*) *sexstriatus* **(LeConte, 1851)**
Nomenclatural Authority: *Caterino (2010)*
Literature Records: Santa Cruz (*Caterino, 2010*: 10), Santa Rosa (*Caterino, 2010*: 10)
Digitized Records: Santa Cruz (1 SBMNH)
Range: Also known from mainland (*Caterino, 2010*).

**Hololeptini**
Notes. Two genera and six species of Hololeptini have been recorded from California (*Mazur, 2011*).

### *Hololepta* Paykull, 1811
Nomenclatural Authority: *Mazur (2011)*

Notes. Five species of *Hololepta* have been recorded from California (*Mazur, 2011*) in two subgenera, *Hololepta* (*s.str.*) and *Leionota* Marseul, 1853.

### *Hololepta* (*Leionota*) *vicina* LeConte, 1851
Nomenclatural Authority: *Mazur (2011)*

Literature Records: Santa Catalina (*Fall, 1897*: 237)

Digitized Records: none

Range: Also known from mainland (*Fall, 1901*).

### *Iliotona* Carnochan, 1917
Nomenclatural Authority: *Mazur (2011)*

Notes. One species of *Iliotona* has been recorded from California (*Mazur, 2011*).

### *Iliotona cacti* (LeConte, 1851)
Nomenclatural Authority: *Mazur (2011)*

Literature Records: none

Digitized Records: San Miguel (1 SBMNH)

Range: Also known from mainland (*Fall, 1901*).

### Saprininae
Notes. Eleven genera and 53 species of Saprininae are known to occur in California (M. L. Gimmel, 2022, unpublished data).

### *Aphelosternus* Wenzel, 1962
Nomenclatural Authority: *Mazur (2011)*

Notes. Only one species is contained in the genus *Aphelosternus* (*Mazur, 2011*).

### *Aphelosternus interstitialis* (LeConte, 1851)
Nomenclatural Authority: *Mazur (2011)*

Literature Records: Santa Catalina (*Fall, 1897*: 237; *Fall, 1901*: 96)

Digitized Records: none

Range: Also known from mainland (*Fall, 1901*).

Notes. This species was recorded as *Saprinus interstitialis* by *Fall (1897, 1901)*.

### *Euspilotus* Lewis, 1907
Nomenclatural Authority: *Mazur (2011)*

Digitized Records (genus-only): Santa Catalina (2 SBMNH), Santa Rosa (1 SBMNH)

Notes. Eleven species of *Euspilotus* are known to occur in California (M. L. Gimmel, 2022, unpublished data), distributed among three subgenera, *Hesperosaprinus* Wenzel, 1962, *Neosaprinus* Bickhardt, 1909, and *Platysaprinus* Bickhardt, 1916.

### *Euspilotus* (*Hesperosaprinus*) *scissus* (LeConte, 1851)
Nomenclatural Authority: *Mazur (2011)*

Literature Records: none

Digitized Records: San Miguel (9 SBMNH), San Nicolas (8 SBMNH), Santa Cruz (5 SBMNH), Santa Rosa (4 SBMNH)

Range: Also known from mainland (*Mazur, 2011*).

### *Euspilotus* (*Hesperosaprinus*) species near *laridus* (LeConte, 1851)

Nomenclatural Authority: *Mazur (2011)*

Literature Records: Santa Catalina (*Fall, 1897*: 237)

Digitized Records: none

Range: Unknown.

Notes. This species was recorded as "*Saprinus* sp. near *laridus*" by *Fall (1897)*.

### *Geomysaprinus* Ross, 1940

Nomenclatural Authority: *Mazur (2011)*

Notes. Six species of *Geomysaprinus* have been recorded from California (*Mazur, 2011*). All of these belong to the subgenus *Priscosaprinus* Wenzel, 1962 (*Mazur, 2011*).

### *Geomysaprinus* undetermined species

Literature Records: none

Digitized Records: Santa Catalina (1 SBMNH), Santa Rosa (1 SBMNH)

### *Hypocaccus* Thomson, 1867

Nomenclatural Authority: *Mazur (2011)*

Notes. Seven species of *Hypocaccus* are known to occur in California (M. L. Gimmel, 2022, unpublished data), distributed among two subgenera, *Baeckmanniolus* Reichardt, 1926 and *Hypocaccus* (*s.str.*).

### *Hypocaccus* (*Baeckmanniolus*) *gaudens* (LeConte, 1851)

Nomenclatural Authority: *Mazur (2011)*

Literature Records: none

Digitized Records: San Miguel (4 SBMNH), San Nicolas (1 LACM; 1 SBMNH), Santa Catalina (2 SBMNH), Santa Cruz (3 SBMNH), Santa Rosa (10 SBMNH)

Range: Also known from mainland (*Mazur, 2011*).

### *Hypocaccus* (*Baeckmanniolus*) *serrulatus* (LeConte, 1851)

Nomenclatural Authority: [none]

Literature Records: none

Digitized Records: Santa Catalina (1 SBMNH)

Range: Also known from mainland.

Notes. This name is missing from the catalog of *Mazur (2011)*.

### *Hypocaccus* (*Hypocaccus*) *bigemmeus* (LeConte, 1851)

Nomenclatural Authority: *Mazur (2011)*

Literature Records: none

Digitized Records: San Clemente (1 SBMNH), San Miguel (10 SBMNH), San Nicolas (7 SBMNH), Santa Cruz (8 SBMNH), Santa Rosa (4 SBMNH)

Range: Also known from mainland (*Mazur, 2011*).

### *Hypocaccus* (*Hypocaccus*) *lucidulus* (LeConte, 1851)

Nomenclatural Authority: *Mazur (2011)*

Literature Records: San Clemente (*Caterino, Chatzimanolis & Richmond, 2015*: 278), San Miguel (*Caterino, Chatzimanolis & Richmond, 2015*: 278), San Nicolas (*Cockerell, 1940*: 285; *Caterino, Chatzimanolis & Richmond, 2015*: 278), Santa Cruz (*Caterino, Chatzimanolis & Richmond, 2015*: 278), Santa Rosa (*Caterino, Chatzimanolis & Richmond, 2015*: 278)

Digitized Records: San Clemente (11 SBMNH), San Miguel (19 SBMNH), San Nicolas (12 SBMNH), Santa Cruz (21 SBMNH), Santa Rosa (13 SBMNH)

Range: Also known from mainland (*Cockerell, 1940*; *Mazur, 2011*).

Notes. *Cockerell (1940)* reported this species as *Saprinus lucidulus*.

### *Neopachylopus* Reichardt, 1926

Nomenclatural Authority: *Mazur (2011)*

Notes. Two species of *Neopachylopus* have been recorded from California (*Mazur, 2011*).

### *Neopachylopus sulcifrons* (Mannerheim, 1843)

Nomenclatural Authority: *Mazur (2011)*

Literature Records: San Nicolas (*Cockerell, 1940*: 285)

Digitized Records: San Clemente (1 SBMNH), San Miguel (5 SBMNH), San Nicolas (1 SBMNH), Santa Catalina (1 SBMNH), Santa Cruz (14 SBMNH), Santa Rosa (2 SBMNH)

Range: Also known from mainland (*Cockerell, 1940*; *Mazur, 2011*).

Notes. *Cockerell (1940)* reported this species as *Saprinus sulcifrons*.

### *Saprinus* Erichson, 1834

Nomenclatural Authority: *Mazur (2011)*

Digitized Records (genus-only): Santa Rosa (2 LACM)

Notes. Four species of *Saprinus* have been recorded from California, all of them belonging to the nominate subgenus (M. L. Gimmel, 2022, unpublished data).

### *Saprinus* (*Saprinus*) *lugens* Erichson, 1834

Nomenclatural Authority: *Mazur (2011)*

Literature Records: San Clemente (*Fall, 1897*: 237; *Miller & Miller, 1985*: 123), San Nicolas (*Fall, 1897*: 237; *Miller & Miller, 1985*: 123), Santa Barbara (*Fall, 1897*: 237; *Miller & Miller, 1985*: 123), Santa Cruz (*Fall & Davis, 1934*: 144; *Miller & Miller, 1985*: 123), Santa Rosa (*Fall, 1897*: 237; *Miller & Miller, 1985*: 123)

Digitized Records: San Miguel (4 SBMNH), San Nicolas (1 SBMNH), Santa Barbara (1 SBMNH), Santa Catalina (3 SBMNH; 1 TAMU), Santa Cruz (7 SBMNH), Santa Rosa (26 SBMNH)

Range: Also known from mainland (*Mazur, 2011*).

### *Saprinus* (*Saprinus*) *oregonensis* LeConte, 1844

Nomenclatural Authority: *Mazur (2011)*

Literature Records: Santa Cruz (*Fall & Davis, 1934*: 144)

Digitized Records: none

Range: Also known from mainland (*Mazur, 2011*).

### *Xerosaprinus* Wenzel, 1962

Nomenclatural Authority: *Mazur (2011)*

Digitized Records (genus-only): Santa Rosa (2 LACM)

Notes. Thirteen species of *Xerosaprinus* are known from California (M. L. Gimmel, 2022, unpublished data), belonging to two subgenera, *Vastosaprinus* Wenzel, 1962 and *Xerosaprinus* (s.str.).

### *Xerosaprinus* (*Xerosaprinus*) *fimbriatus* (LeConte, 1851)

Nomenclatural Authority: *Mazur (2011)*

Literature Records: Santa Catalina (*Fall, 1897*: 237)

Digitized Records: none

Range: Also known from mainland.

Notes. This species was recorded as *Saprinus fimbriatus* by *Fall (1897)*.

### *Xerosaprinus* (*Xerosaprinus*) *lubricus* (LeConte, 1851)

Nomenclatural Authority: *Mazur (2011)*

Literature Records: San Clemente (*Fall, 1897*: 237), Santa Catalina (*Fall, 1897*: 237), Santa Cruz (*Fall & Davis, 1934*: 144), Santa Rosa (*Fall, 1897*: 237)

Digitized Records: Santa Catalina (3 SBMNH), Santa Cruz (5 SBMNH; 3 UCSB), Santa Rosa (13 SBMNH)

Range: Also known from mainland (*Mazur, 2011*).

Notes. This species was recorded as *Saprinus lubricus* by *Fall (1897)* and *Fall & Davis (1934)*.

### *Xerosaprinus* (*Xerosaprinus*) *vitiosus* (LeConte, 1851)

Nomenclatural Authority: *Mazur (2011)*

Literature Records: Santa Catalina (*Seavey, 1892*: 262; *Fall, 1897*: 237)

Digitized Records: none

Range: Also known from mainland (*Mazur, 2011*).

Notes. This species was recorded as *Saprinus vitiosus* by *Seavey (1892)* and *Fall (1897)*.

## HYDROPHILOIDEA

### Helophoridae, NEW FAMILY RECORD

Notes. The family Helophoridae contains a single genus, *Helophorus*. The species (as Hydrophilidae subfamily Helophorinae) were revised for North America by *Smetana (1985)*.

### *Helophorus* Fabricius, 1775

Nomenclatural Authority: *Hansen (1999)*

Notes. Nineteen species of *Helophorus* have been recorded from California, all of them belonging to the subgenus *Rhopalohelophorus* Kuwert, 1886 (*Hansen, 1999*).

*Helophorus* (*Rhopalohelophorus*) *linearis* LeConte, 1855
Nomenclatural Authority: *Smetana (1985)*
Literature Records: none
Digitized Records: San Clemente (15 SBMNH)
Range: Also known from mainland (*Smetana, 1985*).

**Hydrophilidae**
Notes. Five subfamilies, 21 genera, and 117 species of Hydrophilidae are known to occur in California (*Hansen, 1999*; M. L. Gimmel, 2022, unpublished data).

**Acidocerinae**
Notes. One species of Acidocerinae has been recorded from California (*Hansen, 1999*).

*Helochares* Mulsant, 1844
Nomenclatural Authority: *Hansen (1999)*
Notes. One species of *Helochares* has been recorded from California (*Hansen, 1999*), belonging to the subgenus *Hydrobaticus* MacLeay, 1871.

*Helochares* (*Hydrobaticus*) *normatus* (LeConte, 1861)
Nomenclatural Authority: *Hansen (1999)*; *Short & Girón (2018)*
Literature Records: Santa Cruz (*Furlong & Wenner, 2002*: 250)
Digitized Records: Santa Cruz (16 SBMNH), Santa Rosa (12 SBMNH)
Range: Also known from mainland (*Hansen, 1999*; *Short & Girón, 2018*).

**Chaetarthriinae**
Notes. Two tribes, three genera, and 20 species of Chaetarthriinae are known to occur in California (*Hansen, 1999*; M. L. Gimmel, 2022, unpublished data).

**Anacaenini**
Notes. Two genera and nine species of Anacaenini are known to occur in California (*Hansen, 1999*; M. L. Gimmel, 2022, unpublished data).

*Anacaena* Thomson, 1859
Nomenclatural Authority: *Hansen (1999)*
Notes. Three species of *Anacaena* are known from California (M. L. Gimmel, 2022, unpublished data).

*Anacaena signaticollis* (Fall, 1924)
Nomenclatural Authority: *Hansen (1999)*
Literature Records: Santa Cruz (*Furlong & Wenner, 2002*: 250; *Short & Caterino, 2009*: 405)
Digitized Records: San Miguel (3 SBMNH), Santa Cruz (50 SBMNH), Santa Rosa (3 SBMNH)
Range: Also known from mainland (*Hansen, 1999*; *Short & Caterino, 2009*).

**Chaetarthriini**
Notes. One genus and 11 species of Chaetarthriini have been recorded from California (*Hansen, 1999*).

***Chaetarthria* Stephens, 1835**
Nomenclatural Authority: *Hansen (1999)*
Digitized Records (genus-only): Santa Cruz (37 SBMNH), Santa Rosa (1 SBMNH)
Notes. Eleven species of *Chaetarthria* have been recorded from California (*Hansen, 1999*). The species of this genus were revised for the New World by *Miller (1974)*.

***Chaetarthria hespera* Miller, 1974**
Nomenclatural Authority: *Miller (1974)*, *Hansen (1999)*
Literature Records: Santa Catalina (*Fall, 1897*: 236; *Miller, 1974*: 43)
Digitized Records: Santa Catalina (17 SBMNH), Santa Cruz (2 SBMNH)
Range: Also known from mainland (*Miller, 1974*; *Hansen, 1999*).
Notes. Based on material examined in *Miller (1974)* (who accidentally indicated Santa Catalina as being in Orange Co.), *Fall's (1897)* record of *C. nigrella* apparently refers to this species.

***Chaetarthria nigrella* (LeConte, 1861)**
Nomenclatural Authority: *Miller (1974)*, *Hansen (1999)*
Literature Records: none
Digitized Records: Santa Cruz (4 SBMNH), Santa Rosa (5 SBMNH)
Range: Also known from mainland (*Miller, 1974*; *Hansen, 1999*).
Notes. *Fall's (1897)* island record of *C. nigrella* refers to *C. hespera* (see that species).

***Chaetarthria punctulata* Sharp, 1882**
Nomenclatural Authority: *Miller (1974)*, *Hansen (1999)*
Literature Records: none
Digitized Records: Santa Cruz (2 SBMNH)
Range: Also known from mainland (*Miller, 1974*; *Hansen, 1999*).

***Chaetarthria pusilla* Sharp, 1882**
Nomenclatural Authority: *Miller (1974)*, *Hansen (1999)*
Literature Records: none
Digitized Records: Santa Cruz (12 SBMNH)
Range: Also known from mainland (*Miller, 1974*; *Hansen, 1999*).

**Enochrinae**
Notes. Two genera and 21 species of Enochrinae have been recorded from California (*Hansen, 1999*; M. L. Gimmel, 2022, unpublished data).

***Cymbiodyta* Bedel, 1881**
Nomenclatural Authority: *Hansen (1999)*
Notes. Nine species of *Cymbiodyta* have been recorded from California (*Hansen, 1999*). The genus was revised by *Smetana (1974)*.

### *Cymbiodyta columbiana* Leech, 1948

Nomenclatural Authority: *Smetana (1974)*, *Hansen (1999)*

Literature Records: none

Digitized Records: Santa Cruz (5 SBMNH)

Range: Also known from mainland (*Smetana, 1974*; *Hansen, 1999*).

Notes. This species is morphologically extremely similar to *C. dorsalis* (see *Smetana, 1974*: 44). Consequently, Channel Island records of these two species should be interpreted with caution.

### *Cymbiodyta dorsalis* (Motschulsky, 1859)

Nomenclatural Authority: *Smetana (1974)*, *Hansen (1999)*

Literature Records: San Miguel (*Smetana, 1974*: 38 [map]), Santa Catalina (*Fall, 1897*: 236; *Smetana, 1974*: 38 [map]), Santa Cruz (*Winters, 1927*: 27; *Leech, 1948*: 449; *Smetana, 1974*: 38 [map]), Santa Rosa (*Fall, 1897*: 236)

Digitized Records: San Miguel (22 SBMNH), San Nicolas (8 SBMNH), Santa Catalina (8 SBMNH), Santa Cruz (13 SBMNH), Santa Rosa (37 SBMNH)

Range: Also known from mainland (*Winters, 1927*; *Leech, 1948*; *Smetana, 1974*; *Hansen, 1999*).

Notes. See note under *C. columbiana* above.

### *Cymbiodyta punctatostriata* (Horn, 1873)

Nomenclatural Authority: *Smetana (1974)*, *Hansen (1999)*

Literature Records: Santa Cruz (*Smetana, 1974*: 24)

Digitized Records: Santa Cruz (10 SBMNH)

Range: Also known from mainland (*Smetana, 1974*; *Hansen, 1999*).

### *Enochrus* Thomson, 1859

Nomenclatural Authority: *Hansen (1999)*

Notes. Twelve species of *Enochrus* are known from California (M. L. Gimmel, 2022, unpublished data); these belong to three subgenera, *Enochrus* (*s.str.*), *Lumetus* Zaitzev, 1908, and *Methydrus* Rey, 1885. The genus was revised for North America by *Gundersen (1978)*.

### *Enochrus* (*Enochrus*) *carinatus* (LeConte, 1855)

Nomenclatural Authority: *Gundersen (1978)*, *Hansen (1999)*

Literature Records: San Miguel (*Cockerell, 1940*: 285)

Digitized Records: Santa Cruz (1 SBMNH)

Range: Also known from mainland (*Gundersen, 1978*).

Notes. The nominate subspecies, *E. c. carinatus* (LeConte, 1855), is the only subspecies occurring in California (*Gundersen, 1978*).

### *Enochrus* (*Enochrus*) *piceus* Miller, 1964

Nomenclatural Authority: *Gundersen (1978)*, *Hansen (1999)*

Literature Records: none

Digitized Records: San Nicolas (1 SBMNH), Santa Catalina (5 SBMNH), Santa Cruz (21 SBMNH), Santa Rosa (3 SBMNH)

Range: Also known from mainland (*Gundersen, 1978*; *Hansen, 1999*).

Notes. The nominate subspecies, *E. p. piceus* Miller, 1964, is the only subspecies occurring in California (*Gundersen, 1978*).

### *Enochrus* (*Lumetus*) *hamiltoni* (Horn, 1890)
Nomenclatural Authority: *Gundersen (1978)*, *Hansen (1999)*

Literature Records: none

Digitized Records: San Miguel (5 SBMNH)

Range: Also known from mainland (*Gundersen, 1978*; *Hansen, 1999*).

### *Enochrus* (*Methydrus*) *cristatus* (LeConte, 1855)
Nomenclatural Authority: *Gundersen (1978)*, *Hansen (1999)*

Literature Records: none

Digitized Records: Santa Cruz (5 SBMNH)

Range: Also known from mainland (*Gundersen, 1978*; *Hansen, 1999*).

### *Enochrus* (*Methydrus*) *pygmaeus* (Fabricius, 1792)
Nomenclatural Authority: *Gundersen (1978)*, *Hansen (1999)*

Literature Records: none

Digitized Records: Santa Catalina (8 SBMNH)

Range: Also known from mainland (*Gundersen, 1978*; *Hansen, 1999*).

Notes. *Enochrus p. pectoralis* (LeConte, 1855) is the subspecies occurring in coastal California (*Gundersen, 1978*).

### Hydrophilinae
Notes. Four tribes, eight genera, and 47 species of Hydrophilinae are known to occur in California (*Hansen, 1999*; M. L. Gimmel, 2022, unpublished data).

### Berosini
Notes. One genus and 11 species of Berosini are known to occur in California (*Van Tassell, 1966*; *Hansen, 1999*).

### *Berosus* Leach, 1817
Nomenclatural Authority: *Hansen (1999)*

Notes. Eleven species of *Berosus* in two subgenera, *Berosus* (*s.str.*) and *Enoplurus* Hope, 1838, are known to occur in California (M. L. Gimmel, 2022, unpublished data). The species were revised for North America by *Van Tassell (1966)*.

### *Berosus* (*Berosus*) *fraternus* LeConte, 1855
Nomenclatural Authority: *Van Tassell (1966)*, *Hansen (1999)*

Literature Records: Santa Catalina (*Van Tassell, 1966*: 223 [map])

Digitized Records: none

Range: Also known from mainland (*Van Tassell, 1966*; *Hansen, 1999*).

**Berosus (Berosus) hatchi Miller, 1965**
Nomenclatural Authority: *Van Tassell (1966)*, *Hansen (1999)*
Literature Records: Santa Catalina (*Van Tassell, 1966*: 214)
Digitized Records: none
Range: Also known from mainland (*Van Tassell, 1966*; *Hansen, 1999*).

**Berosus (Berosus) infuscatus LeConte, 1855**
Nomenclatural Authority: *Van Tassell (1966)*, *Hansen (1999)*
Literature Records: none
Digitized Records: San Clemente (1 SBMNH)
Range: Also known from mainland (*Van Tassell, 1966*; *Hansen, 1999*).

**Berosus (Enoplurus) punctatissimus LeConte, 1852**
Nomenclatural Authority: *Van Tassell (1966)*, *Hansen (1999)*
Literature Records: Santa Cruz (*Furlong & Wenner, 2002*: 250)
Digitized Records: Santa Catalina (10 SBMNH), Santa Cruz (11 SBMNH), Santa Rosa (3 SBMNH)
Range: Also known from mainland (*Van Tassell, 1966*; *Hansen, 1999*).

**Hydrobiusini**
Notes. Two genera and three species of Hydrobiusini have been recorded from California (*Hansen, 1999*).

**Hydrobius Leach, 1815**
Nomenclatural Authority: *Hansen (1999)*
Digitized Records (genus-only): Santa Cruz (1 UCSB)
Notes. Only one species of *Hydrobius* is known to occur in California (*Hansen, 1999*).

**Hydrobius fuscipes (Linnaeus, 1758)**
Nomenclatural Authority: *Hansen (1999)*
Literature Records: Santa Cruz (*Furlong & Wenner, 2002*: 250)
Digitized Records: none
Range: Also known from mainland (*Hansen, 1999*).

**Hydrophilini**
Notes. Three genera and 12 species of Hydrophilini are known to occur in California (M. L. Gimmel, 2022, unpublished data).

**Hydrochara Berthold, 1827**
Nomenclatural Authority: *Hansen (1999)*
Notes. Two species of *Hydrochara* have been recorded from California (*Hansen, 1999*). The genus was revised by *Smetana (1980)*.

**Hydrochara lineata (LeConte, 1855)**
Nomenclatural Authority: *Smetana (1980)*, *Hansen (1999)*

Literature Records: Santa Cruz (*LeConte, 1876*: 298; *Fall, 1897*: 236; *Fall, 1901*: 56; *Fall & Davis, 1934*: 144; *Smetana, 1980*: 67), Santa Rosa (*Fall, 1897*: 236; *Fall, 1901*: 56; *Smetana, 1980*: 67)

Digitized Records: Santa Cruz (16 LACM; 9 SBMNH; 2 UCSB)

Range: Also known from mainland (*Fall, 1901*; *Smetana, 1980*; *Hansen, 1999*).

Notes. This species was recorded as *Hydrocharis glaucus* LeConte by *LeConte (1876)* and *Fall (1897, 1901)*; this name is now a junior synonym of *H. lineata* (see *Smetana, 1980*). *Fall & Davis (1934)* recorded this species as *Hydrophilus lineatus* (LeConte). The dot on *Smetana's (1980*: 23) "Map 3" in the Pacific Ocean south of San Clemente Island is in error (*Miller & Menke, 1981*: 68).

***Hydrophilus* Geoffroy, 1762**
Nomenclatural Authority: *Hansen (1999)*

Notes. Two species of *Hydrophilus*, both belonging to the nominate subgenus, have been recorded from California (*Short & McIntosh, 2014*). The North American species were reviewed by *Short & McIntosh (2014)*.

***Hydrophilus* (*Hydrophilus*) *triangularis* Say, 1823**
Nomenclatural Authority: *Hansen (1999)*, *Short & McIntosh (2014)*

Literature Records: Santa Cruz (*Furlong & Wenner, 2002*: 250)

Digitized Records: San Clemente (1 iNat), Santa Cruz (1 SBMNH)

Range: Also known from mainland (*Short & McIntosh, 2014*).

***Tropisternus* Solier, 1834**
Nomenclatural Authority: *Hansen (1999)*

Digitized Records (genus-only): Santa Cruz (3 UCSB)

Notes. Eight species of *Tropisternus* are known to occur in California, all belonging to the nominate subgenus (M. L. Gimmel, 2022, unpublished data).

***Tropisternus* (*Tropisternus*) *affinis* Motschulsky, 1859**
Nomenclatural Authority: *Hansen (1999)*

Literature Records: Santa Catalina (*Fall, 1897*: 236)

Digitized Records: Santa Catalina (8 SBMNH; 66 LACM), Santa Cruz (40 SBMNH; 12 LACM; 6 UCSB), Santa Rosa (18 SBMNH; 42 LACM)

Range: Also known from mainland (*Hansen, 1999*).

Notes. This species has been reported as *Tropisternus ellipticus* (LeConte, 1855) by most prior workers, including *Fall (1897)*. However, that name is a junior primary homonym and is permanently invalid (*Hansen, 1999*).

***Tropisternus* (*Tropisternus*) *californicus* (LeConte, 1855)**
Nomenclatural Authority: *Hansen (1999)*

Literature Records: Santa Catalina (*Seavey, 1892*: 262; *Fall, 1897*: 236), Santa Cruz (*LeConte, 1876*: 298; *Fall, 1897*: 236; *Fall & Davis, 1934*: 144)

Digitized Records: none

Range: Also known from mainland (*Hansen, 1999*).

**Laccobiini**

Notes. Two genera and 21 species of Laccobiini are known to occur in California (*Hansen, 1999*; M. L. Gimmel, 2022, unpublished data).

*Laccobius* **Erichson, 1837**

Nomenclatural Authority: *Hansen (1999)*

Notes. The species of *Laccobius* were revised for the United States in two publications by *Gentili (1986a*, *1986b)*. Fifteen species of the genus have been recorded from California (M. L. Gimmel, 2022, unpublished data).

*Laccobius* (*Hydroxenus*) *californicus* **d'Orchymont, 1942**

Nomenclatural Authority: *Hansen (1999)*

Literature Records: Santa Cruz (*Gentili, 1986b*: 47)

Digitized Records: Santa Cruz (4 SBMNH)

Range: Also known from mainland (*Gentili, 1986b*).

*Laccobius* (*Hydroxenus*) *ellipticus* **LeConte, 1855**

Nomenclatural Authority: *Hansen (1999)*

Literature Records: Santa Catalina (*Fall, 1897*: 236), Santa Cruz (*Winters, 1926*: 50; *d'Orchymont, 1942*: 4; *Gentili, 1986b*: 49), Santa Rosa (*Fall, 1897*: 236)

Digitized Records: Santa Catalina (4 SBMNH), Santa Cruz (14 SBMNH), Santa Rosa (4 SBMNH)

Range: Also known from mainland (*Gentili, 1986b*).

Notes. *Fall (1897)* mistakenly recorded this species as "*Laccophilus ellipticus*".

*Laccobius* (*Microlaccobius*) *insolitus* **d'Orchymont, 1942**

Nomenclatural Authority: *Hansen (1999)*

Literature Records: none

Digitized Records: San Nicolas (4 SBMNH), Santa Rosa (7 SBMNH)

Range: Also known from mainland (*Gentili, 1986b*).

**Sphaeridiinae**

Notes. Three tribes, seven genera, and 28 species of Sphaeridiinae are known to occur in California (M. L. Gimmel, 2022, unpublished data).

**Megasternini**

Notes. Five genera and 23 species of Megasternini are known to occur in California (M. L. Gimmel, 2022, unpublished data).

*Agna* **Smetana, 1978**

Nomenclatural Authority: *Hansen (1999)*

Notes. One species of *Agna* has been recorded from California (*Hansen, 1999*). The species were reviewed by *Smetana (1978)* and *Arriaga-Varela, Cortés-Aguilar & Fikáček (2019)*.

*Agna capillata* (**LeConte, 1855**)

Nomenclatural Authority: *Smetana (1978)*, *Hansen (1999)*

Literature Records: none

Digitized Records: Santa Catalina (1 SBMNH)

Range: Also known from mainland (*Smetana, 1978*; *Arriaga-Varela, Cortés-Aguilar & Fikáček, 2019*).

Notes. This species apparently develops exclusively in rotting cacti and other succulents (*Arriaga-Varela, Cortés-Aguilar & Fikáček, 2019*).

### *Cercyon* Leach, 1817

Nomenclatural Authority: *Smetana (1978)*, *Hansen (1999)*

Digitized Records (genus-only): San Miguel (223 LACM), San Nicolas (53 LACM)

Notes. Eighteen species of *Cercyon* are known to occur in California, belonging to three subgenera, *Cercyon* (*s.str.*), *Paracercyon* Seidlitz, 1888, and *Prostercyon* Smetana, 1978 (M. L. Gimmel, 2022, unpublished data). *Smetana (1978)* revised the North American species, and *Suzumura, Kobayashi & Ôhara (2019)* provided an update to certain western coastal species. "*Cercyon* sp. larvae" were reported from Santa Catalina Island by *Straughan & Hadley (1980)*: 392).

### *Cercyon* (*Cercyon*) *fimbriatus* Mannerheim, 1852

Nomenclatural Authority: *Smetana (1978)*, *Hansen (1999)*, *Suzumura, Kobayashi & Ôhara (2019)*

Literature Records: San Miguel (*Smetana, 1978*: 145 [map]), Santa Catalina (*Smetana, 1978*: 145 [map])

Digitized Records: San Clemente (5 SBMNH), San Miguel (19 SBMNH), San Nicolas (2 SBMNH), Santa Catalina (7 SBMNH), Santa Cruz (33 SBMNH), Santa Rosa (8 SBMNH)

Range: Also known from mainland (*Smetana, 1978*; *Hansen, 1999*; *Suzumura, Kobayashi & Ôhara, 2019*).

### *Cercyon* (*Cercyon*) *haemorrhoidalis* (Fabricius, 1775)

Nomenclatural Authority: *Smetana (1978)*, *Hansen (1999)*

Literature Records: Santa Barbara (*Miller & Miller, 1985*: 123)

Digitized Records: Santa Rosa (4 SBMNH)

Range: Also known from mainland (*Smetana, 1978*; *Hansen, 1999*).

Notes. Introduced to North America from the Palearctic realm (*Smetana, 1978*).

Misspelled by *Miller & Miller (1985)* as "*Cercyon haemorrhoides*".

### *Cercyon* (*Cercyon*) *luniger* Mannerheim, 1853

Nomenclatural Authority: *Suzumura, Kobayashi & Ôhara (2019)*

Literature Records: San Miguel (*Straughan & Hadley, 1980*: 392), Santa Catalina (*Fall, 1897*: 236; *Fall, 1901*: 58; *Blackwelder, 1931*: 24; *Leech, 1948*: 458; *Smetana, 1978*: 149), Santa Cruz (*Smetana, 1978*: 149)

Digitized Records: San Clemente (3 SBMNH), Santa Cruz (8 SBMNH)

Range: Also known from mainland (*Suzumura, Kobayashi & Ôhara, 2019*).

Notes. This species was originally referred to as *C. luniger* by *Fall (1897*, *1901)*, *Blackwelder (1931)*, *Leech (1948)*, and *Straughan & Hadley (1980)*. *Smetana (1978)* did not report his

concept of *C. luniger* from the islands, but included Fall and CASC island material among the paratypes of his *Cercyon spathifer* Smetana, 1978. However, *Suzumura, Kobayashi & Ôhara (2019)* synonymized *C. spathifer* with *C. luniger*; what was previously considered *C. luniger* was given a new name. The distribution map in *Suzumura, Kobayashi & Ôhara (2019*: 480) also shows a Channel Island record for *C. luniger*, presumably from an earlier instance of the California Beetle Project pages, but the resolution does not allow identification of the island.

### *Cercyon (Cercyon) quisquilius* (Linnaeus, 1761)
Nomenclatural Authority: *Smetana (1978)*, *Hansen (1999)*
Literature Records: none
Digitized Records: Santa Catalina (1 SBMNH), Santa Cruz (1 SBMNH)
Range: Also known from mainland (*Smetana, 1978*; *Hansen, 1999*).
Notes. Introduced to North America from the Palearctic realm (*Smetana, 1978*).

### Sphaeridiini
Notes. One genus and three species of Sphaeridiini are known to occur in California (M. L. Gimmel, 2022, unpublished data).

### *Sphaeridium* Fabricius, 1775
Nomenclatural Authority: *Hansen (1999)*
Notes. Three introduced species of *Sphaeridium* are known to occur in California (M. L. Gimmel, 2022, unpublished data). These species were treated by *Smetana (1978)*.

### *Sphaeridium scarabaeoides* (Linnaeus, 1758)
Nomenclatural Authority: *Smetana (1978)*, *Hansen (1999)*
Literature Records: none
Digitized Records: Santa Catalina (2 LACM), Santa Cruz (4 SBMNH), Santa Rosa (44 LACM; 3 SBMNH)
Range: Also known from mainland (*Smetana, 1978*; *Hansen, 1999*).
Notes. Introduced to North America from the Palearctic realm (*Smetana, 1978*)

### SCARABAEOIDEA

### Geotrupidae
Notes. Two subfamilies, five genera, and seven species of Geotrupidae have been recorded from California (*Howden, 1984*; M. L. Gimmel, 2022, unpublished data).

### Bolboceratinae
Notes. Four genera and six species of Bolboceratinae have been recorded from California (*Howden, 1984*; M. L. Gimmel, 2022, unpublished data).

### *Bolbocerastes* Cartwright, 1953
Nomenclatural Authority: *Smith (2009)*
Notes. Two species of *Bolbocerastes* are known from California (*Howden, 1984*).
The species of the genus were treated by *Cartwright (1953)*.

### *Bolbocerastes regalis* Cartwright, 1953

Nomenclatural Authority: *Cartwright (1953)*, *Howden (1984)*

Literature Records: San Clemente (*von Bloeker, 1939b*: 153; *Cartwright, 1953*: 108)

Digitized Records: none

Range: Also known from mainland (*Cartwright, 1953*; *Howden, 1984*).

Notes. This species was misidentified as *Bolboceras serratus* LeConte by *von Bloeker (1939b*: 153).

### *Odonteus* Samouelle, 1819

Nomenclatural Authority: *Smith (2009)*

Notes. One species of *Odonteus* is known from California (*Howden, 1984*). This genus has been known in recent literature as *Bolboceras* Kirby, 1818.

### *Odonteus obesus* LeConte, 1859

Nomenclatural Authority: *Smith (2009)*

Literature Records: none

Digitized Records: Santa Rosa (1 SBMNH)

Range: Also known from mainland (*Howden, 1984*).

Notes. This species was formerly known as *Bolboceras obesus* (*e.g.*, in *Howden (1984)*).

### Scarabaeidae

Notes. Seven subfamilies, 76 genera, and 321 species of Scarabaeidae are known to occur in California (M. L. Gimmel, 2022, unpublished data). The subfamily Rutelinae has yet to be recorded from the Channel Islands.

### Aphodiinae

Notes. Five tribes, 36 genera, and 102 species of Aphodiinae are known to occur in California (M. L. Gimmel, 2022, unpublished data).

### Aegialiini

Notes. Two genera and 14 species of Aegialiini have been recorded from California (*Gordon & Cartwright, 1988*).

### *Aegialia* Latreille, 1807

Nomenclatural Authority: *Smith (2009)*

Notes. Thirteen species of *Aegialia* are known to occur in California (*Gordon & Cartwright, 1988*); these are distributed among two subgenera, *Aegialia* (*s.str.*) and *Psammoporus* Thomson, 1863. The North American fauna was treated by *Gordon & Cartwright (1988)*.

### *Aegialia* (*Aegialia*) *convexa* Fall, 1932

Nomenclatural Authority: *Gordon & Cartwright (1988)*

Literature Records: San Clemente (*Gordon & Cartwright, 1988*: 22)

Digitized Records: none

Range: Also known from mainland (*Gordon & Cartwright, 1988*).

***Aegialia* (*Aegialia*) *crassa* LeConte, 1860**
Nomenclatural Authority: *Gordon & Cartwright (1988)*
Literature Records: San Clemente (*von Bloeker, 1939b*: 153)
Digitized Records: none
Range: Also known from mainland (*Gordon & Cartwright, 1988*).

***Aegialia* (*Aegialia*) *nigrella* Brown, 1931**
Nomenclatural Authority: *Gordon & Cartwright (1988)*
Literature Records: San Nicolas (*Gordon & Cartwright, 1988*: 24)
Digitized Records: none
Range: Also known from mainland (*Gordon & Cartwright, 1988*).

***Aegialia* (*Aegialia*) *punctata* Brown, 1931**
Nomenclatural Authority: *Gordon & Cartwright (1988)*
Literature Records: San Nicolas (*Gordon & Cartwright, 1988*: 25)
Digitized Records: none
Range: Also known from mainland (*Gordon & Cartwright, 1988*).

**Aphodiini**
Notes. Twenty-eight genera and 66 species of Aphodiini are known to occur in California (*Gordon & Skelley, 2007*; M. L. Gimmel, 2022, unpublished data). The tribe was monographed for North America by *Gordon & Skelley (2007)*.

***Aphodius* Illiger, 1798**
Nomenclatural Authority: *Gordon & Skelley (2007)*
Digitized Records (genus-only): Santa Catalina (1 LACM)
Notes. Prior to *Gordon & Skelley (2007)*, this genus was considered to encompass most of the North American fauna of Aphodiini. Only a single, adventive species occurs in North America (*Gordon & Skelley, 2007*).

***Aphodius fimetarius* (Linnaeus, 1758)**
Nomenclatural Authority: *Gordon & Skelley (2007)*
Literature Records: none
Digitized Records: Anacapa (4 LACM; 1 SBMNH), Santa Catalina (9 LACM; 1 SBMNH; 1 iNat), Santa Cruz (2 SBMNH; 3 UCSB)
Range: Also known from mainland (*Gordon & Skelley, 2007*).
Notes. Adventive in North America, originating in Europe (*Gordon & Skelley, 2007*).

***Calamosternus* Motschulsky, 1859**
Nomenclatural Authority: *Gordon & Skelley (2007)*
Notes. Only a single, adventive species occurs in North America (*Gordon & Skelley, 2007*).

***Calamosternus granarius* (Linnaeus, 1767)**
Nomenclatural Authority: *Gordon & Skelley (2007)*
Literature Records: Santa Rosa (*von Bloeker, 1939b*: 153)

Digitized Records: Anacapa (1 SBMNH), San Nicolas (3 LACM; 1 SBMNH), Santa Catalina (1 LACM), Santa Rosa (2 LACM)

Range: Also known from mainland (*Gordon & Skelley, 2007*).

Notes. *von Bloeker (1939b)* recorded this species as *Aphodius granarius* and incorrectly attributed the species to LeConte. This species was introduced to North America from Europe (*Gordon & Skelley, 2007*).

### *Cinacanthus* Schmidt, 1913

Nomenclatural Authority: *Gordon & Skelley (2007)*

Notes. Three species of this genus are known to occur in California (*Gordon & Skelley, 2007*).

### *Cinacanthus militaris* (LeConte, 1858)

Nomenclatural Authority: *Gordon & Skelley (2007)*

Literature Records: San Nicolas (*von Bloeker, 1939b*: 153)

Digitized Records: none

Range: Also known from mainland (*Gordon & Skelley, 2007*).

Notes. This species was recorded by *von Bloeker (1939b)* as *Aphodius militaris*.

### *Labarrus* Mulsant & Rey, 1869

Nomenclatural Authority: *Gordon & Skelley (2007)*

Notes. Two species of this genus occur in North America (*Gordon & Skelley, 2007*). At least one occurs in California, while the status of the other in the state remains unknown (see below).

### *Labarrus pseudolividus* (Balthasar, 1941)

Nomenclatural Authority: *Gordon & Skelley (2007)*

Literature Records: San Clemente (*von Bloeker, 1939b*: 153), San Miguel (*von Bloeker, 1939b*: 153), Santa Catalina (*Cockerell, 1940*: 286), Santa Cruz (*von Bloeker, 1939b*: 153), Santa Rosa (*von Bloeker, 1939b*: 153)

Digitized Records: Anacapa (1 SBMNH; 24 LACM), San Clemente (7 LACM), San Miguel (22 LACM), San Nicolas (32 LACM), Santa Cruz (5 SBMNH; 198 LACM), Santa Rosa (5 LACM)

Range: Also known from mainland (*Gordon & Skelley, 2007*).

Notes. *von Bloeker (1939b)* and *Cockerell (1940)* recorded this species as *Aphodius lividus* [now *Labarrus lividus* (Olivier, 1789)]. *Gordon & Skelley (2007*: 263) noted that most North American specimens identified as *L. lividus* probably represent *L. pseudolividus*. After examination of the *von Bloeker (1939b)* vouchers housed in SBMNH, we have determined that these indeed represent *L. pseudolividus*. There are no confirmed published vouchers of true *L. lividus* reported from California. *Labarrus lividus* is probably adventive in North America, while *L. pseudolividus* is probably native (*Gordon & Skelley, 2007*).

### *Otophorus* Mulsant, 1842

Nomenclatural Authority: *Gordon & Skelley (2007)*

Notes. Only a single, adventive species of this genus is known from North America (*Gordon & Skelley, 2007*).

### *Otophorus haemorrhoidalis* (Linnaeus, 1758)

Nomenclatural Authority: *Gordon & Skelley (2007)*

Literature Records: none

Digitized Records: Santa Cruz (1 SBMNH)

Range: Also known from mainland (*Gordon & Skelley, 2007*).

Notes. Introduced to North America from the Palearctic realm (*Gordon & Skelley, 2007*).

### *Planolinellus* Dellacasa & Dellacasa, 2005

Nomenclatural Authority: *Gordon & Skelley (2007)*

Notes. Only one species of this genus occurs in North America (*Gordon & Skelley, 2007*).

### *Planolinellus vittatus* (Say, 1825)

Nomenclatural Authority: *Gordon & Skelley (2007)*

Literature Records: San Nicolas (*von Bloeker, 1939b*: 153), Santa Rosa (*von Bloeker, 1939b*: 153)

Digitized Records: San Nicolas (41 LACM; 1 SBMNH), Santa Rosa (43 LACM; 1 SBMNH)

Range: Also known from mainland (*Gordon & Skelley, 2007*).

Notes. Recorded by *von Bloeker (1939b)* as *Aphodius vittatus*.

### *Rugaphodius* Gordon & Skelley, 2007

Nomenclatural Authority: *Gordon & Skelley (2007)*

Notes. Only one species of this genus occurs in North America (*Gordon & Skelley, 2007*).

### *Rugaphodius rugatus* (Schmidt, 1907)

Nomenclatural Authority: *Gordon & Skelley (2007)*

Literature Records: San Nicolas (*von Bloeker, 1939b*: 153), Santa Rosa (*von Bloeker, 1939b*: 153)

Digitized Records: San Nicolas (1 LACM), Santa Rosa (1 LACM)

Range: Also known from mainland (*Gordon & Skelley, 2007*).

Notes. This species was recorded by *von Bloeker (1939b)* as *Aphodius rugatus*.

### Psammodiini

Notes. Six genera and 11 species of Psammodiini are known to occur in California (M. L. Gimmel, 2022, unpublished data).

### *Tesarius* Rakovič, 1981

Nomenclatural Authority: *Rakovič (1984)*

Notes. Four species of *Tesarius* have been recorded from California (*Rakovič, 1984*). The species of the genus were treated by *Rakovič (1984)*.

### *Tesarius mcclayi* (Cartwright, 1955)

Nomenclatural Authority: *Rakovič (1984)*

Literature Records: none

Digitized Records: San Nicolas (1 SBMNH), Santa Rosa (4 SBMNH)

Range: Also known from mainland (*Cartwright, 1955*).

### Cetoniinae

Notes. Four tribes, five genera, and 14 species of Cetoniinae have been recorded from California (M. L. Gimmel, 2022, unpublished data).

### Cremastocheilini

Notes. Two genera and 10 species of Cremastocheilini have been recorded from California (M. L. Gimmel, 2022, unpublished data).

### *Cremastocheilus* Knoch, 1801

Nomenclatural Authority: *Smith (2009)*

Digitized Records (genus-only): Santa Catalina (2 SBMNH)

Notes. Nine species of *Cremastocheilus* are known from California (M. L. Gimmel, 2022, unpublished data).

### *Cremastocheilus schaumii* LeConte, 1853

Nomenclatural Authority: *Smith (2009)*

Literature Records: Santa Catalina (*von Bloeker, 1939b*: 156; *Cockerell, 1940*: 286)

Digitized Records: Santa Catalina (1 SBMNH)

Range: Also known from mainland.

Notes. *von Bloeker (1939b)* reported "one adult collected at edge of ant-hill". Reported as *Cremastocheilus schaumi* by *von Bloeker (1939b)* and *Cockerell (1940)*.

### Gymnetini

Notes. One species of Gymnetini has been recorded from California (*Goodrich, 1966*).

### *Cotinis* Burmeister, 1842

Nomenclatural Authority: *Smith (2009)*

Notes. One species of *Cotinis* is known from California (*Goodrich, 1966*). The genus was revised by *Goodrich (1966)*.

### *Cotinis mutabilis* (Gory & Percheron, 1833)

Nomenclatural Authority: *Goodrich (1966)*; *Smith (2009)*

Literature Records: none

Digitized Records: Santa Catalina (1 iNat)

Range: Also known from mainland (*Goodrich, 1966*).

### Dynastinae

Notes. Five tribes, seven genera, and 12 species of Dynastinae have been recorded from California (*Ratcliffe & Cave, 2017*). The North American fauna of the subfamily was treated by *Ratcliffe & Cave (2017)*.

**Cyclocephalini**

Notes. One genus and five or six species of Cyclocephalini have been recorded from California (*Ratcliffe & Cave, 2017*).

***Cyclocephala* Dejean, 1821**

Nomenclatural Authority: *Ratcliffe & Cave (2017)*

Notes. Five or six species of *Cyclocephala* have been recorded from California (*Ratcliffe & Cave, 2017*).

***Cyclocephala borealis* Arrow, 1911**

Nomenclatural Authority: *Ratcliffe & Cave (2017)*

Literature Records: San Clemente (*von Bloeker, 1939b*: 156; *Ratcliffe & Cave (2017*: 61), Santa Catalina (*Fall, 1897*: 238; *von Bloeker, 1939b*: 156)

Digitized Records: none

Range: Also known from mainland (*Ratcliffe & Cave, 2017*).

Notes. *von Bloeker (1939b)* recorded this species as *Ochrosidia villosa* (Burmeister, 1855); this name is preoccupied in *Cyclocephala* and *C. borealis* is the currently valid name for this taxon. This species is otherwise known to occur only in the eastern USA; the Channel Islands records are suspect. *Ratcliffe & Cave (2017*: 61) wrote: "We have a strange record of one male specimen collected in April, 1939 from San Clemente Island in the Channel Islands…".

***Cyclocephala hirta* LeConte, 1861**

Nomenclatural Authority: *Ratcliffe & Cave (2017)*

Literature Records: none

Digitized Records: Santa Catalina (18 LACM)

Range: Also known from mainland (*Ratcliffe & Cave, 2017*).

Notes. According to *Ratcliffe & Cave (2017)*, the subspecies present in southern California is the nominate subspecies, *C. h. hirta* LeConte, 1861.

***Cyclocephala longula* LeConte, 1863**

Nomenclatural Authority: *Ratcliffe & Cave (2017)*

Literature Records: San Clemente (*von Bloeker, 1939b*: 156), San Nicolas (*von Bloeker, 1939b*: 156), Santa Cruz (*von Bloeker, 1939b*: 156), Santa Rosa (*von Bloeker, 1939b*: 156)

Digitized Records: none

Range: Also known from mainland (*Ratcliffe & Cave, 2017*).

Notes. *von Bloeker (1939b)* recorded this species as *Ochrosidia longula* from Santa Cruz and Santa Rosa; he also recorded it as *Ochrosidia obesula* Casey, 1915 from San Clemente and San Nicolas. The latter name is now considered a synonym of *C. longula* (see *Ratcliffe & Cave, 2017*).

***Cyclocephala melanocephala* (Fabricius, 1775)**

Nomenclatural Authority: *Ratcliffe & Cave (2017)*

Literature Records: San Miguel (*von Bloeker, 1939b*: 156), Santa Cruz (*von Bloeker, 1939b*: 156), Santa Rosa (*von Bloeker, 1939b*: 156)

Digitized Records: none

Range: Also known from mainland (*Ratcliffe & Cave, 2017*).

Notes. *von Bloeker (1939b)* recorded this species as *Dichromina dimidiata* (Burmeister, 1847) (now considered a junior synonym of *C. melanocephala*, see *Ratcliffe & Cave, 2017*) and noted that it was "fairly common in flowers of *Datura metalloides*, August".

### *Cyclocephala pasadenae* (Casey, 1915)

Nomenclatural Authority: *Ratcliffe & Cave (2017)*

Literature Records: San Miguel (*von Bloeker, 1939b*: 156), Santa Cruz (*von Bloeker, 1939b*: 156), Santa Rosa (*von Bloeker, 1939b*: 156)

Digitized Records: none

Range: Also known from mainland (*Ratcliffe & Cave, 2017*).

Notes. *von Bloeker (1939b)* recorded this species as *Ochrosidia pasadenae*.

### Pentodontini

Notes. Two genera and three species of Pentodontini have been recorded from California (*Ratcliffe & Cave, 2017*).

### *Ligyrus* Casey, 1915

Nomenclatural Authority: *López-García & Deloya (2022)*

Notes. One species of *Ligyrus* has been recorded from California (*Ratcliffe & Cave, 2017*; *López-García & Deloya, 2022*).

### *Ligyrus gibbosus* (DeGeer, 1774)

Nomenclatural Authority: *López-García & Deloya (2022)*

Literature Records: San Clemente (*von Bloeker, 1939b*: 156), San Miguel (*von Bloeker, 1939b*: 156), San Nicolas (*von Bloeker, 1939b*: 156), Santa Rosa (*von Bloeker, 1939b*: 156)

Digitized Records: San Clemente (40 LACM; 2 SBMNH), San Miguel (18 LACM; 1 SBMNH), San Nicolas (26 LACM; 1 SBMNH), Santa Cruz (13 LACM; 1 SBMNH), Santa Rosa (13 LACM; 1 SBMNH)

Range: Also known from mainland (*Ratcliffe & Cave, 2017*).

Notes. *von Bloeker (1939b)* recorded this species as two different taxa, *Ligyrus californicus* Casey, 1909 and *L. scitulus* Casey, 1915. The former was reported from San Clemente, San Miguel, San Nicolas, and Santa Rosa and was noted to be "common in loose sand under *Abronia maritima*, *A. alba*, *Franseria bipinnatifida*, *Astragalus nevinii* and *A. miguelensis*, and occasional at lights" (*von Bloeker, 1939b*: 156); the latter was reported from San Clemente, San Miguel, and San Nicolas and was noted to be "occasional at lights, much less common than the preceding species [*L. scitulus*]" (*von Bloeker, 1939b*: 156). Both taxa are now considered junior synonyms of *L. gibbosus* (see *Ratcliffe & Cave, 2017*), and the species has been recently placed in the genus *Tomarus* Erichson, 1847 but subsequently transferred back to *Ligyrus* (*López-García & Deloya, 2022*).

### Melolonthinae

Notes. Nine tribes, 15 genera, and 172 species of Melolonthinae are known to occur in California (M. L. Gimmel, 2022, unpublished data).

**Dichelonychini**

Notes. Three genera and 46 species of Dichelonychini are known to occur in California (M. L. Gimmel, 2022, unpublished data).

**Coenonycha Horn, 1876**

Nomenclatural Authority: *Evans & Smith (2009)*

Notes. Twenty-six species of *Coenonycha* have been recorded from California (*Evans & Smith, 1986*). The species of the genus were keyed by *Evans & Smith (1986)*.

**Coenonycha clementina Casey, 1909**

Nomenclatural Authority: *Evans & Smith (2009)*

Literature Records: San Clemente (*Casey, 1909*: 281; *von Bloeker, 1939b*: 155; *Cazier & McClay, 1943*: 17; *Evans, 1985*: 86; *Miller, 1985a*: 19; *Evans & Smith, 1986*: 86; *Evans & d'Hotman, 1988*: 207)

Digitized Records: San Clemente (2 ASUHIC; 5 LACM; 6 SBMNH; 6 USNM)

Range: Endemic (*Casey, 1909*; *Cazier & McClay, 1943*; *Evans, 1985*; *Miller, 1985a*; *Evans & Smith, 1986*; *Evans & d'Hotman, 1988*).

Notes. A flightless species (*Cazier & McClay, 1943*: 7; *Evans, 1985*: 86). Adults were reported from among the roots of a perennial *Lupinus* (Fabaceae) species during March (*Evans, 1985*: 86). This species was assessed by the US Fish and Wildlife Service and available data did not suggest Endangered or Threatened status at the time of assessment (*Greenwalt, 1977*).

**Coenonycha clypeata McClay, 1943**

Nomenclatural Authority: *Evans & Smith (2009)*

Literature Records: Santa Catalina (*Cazier & McClay, 1943*: 23; *Miller, 1985a*: 19; *Caterino & Chandler, 2010*: 187)

Digitized Records: Santa Catalina (3 LACM; 1 SBMNH)

Range: Endemic (*Cazier & McClay, 1943*; *Miller, 1985a*; *Caterino & Chandler, 2010*).

Notes. Early Santa Catalina records of *Coenonycha rotundata* LeConte, 1856 (*Fall, 1897*: 238; *von Bloeker, 1939b*: 155) probably referred to this species or to *C. fulva*.

**Coenonycha fulva McClay, 1943**

Nomenclatural Authority: *Evans & Smith (2009)*

Literature Records: Santa Catalina (*Cazier & McClay, 1943*: 23; *Evans, 1985*: 86; *Miller, 1985a*; *Evans & Smith, 1986*: 90; *Caterino & Chandler, 2010*: 187)

Digitized Records: Anacapa (1 LACM), Santa Catalina (2 ASUHIC; 43 LACM; 12 SBMNH)

Range: Endemic (*Cazier & McClay, 1943*; *Miller, 1985a*; *Evans & Smith, 1986*; *Caterino & Chandler, 2010*).

Notes. Early Santa Catalina records of *C. rotundata* (*Fall, 1897*: 238; *von Bloeker, 1939b*: 155) probably referred to this species or to *C. clypeata*. *Evans (1985*: 86) reported a large number of individuals from *Adenostoma fasciculatum* Hook. & Arn. (Rosaceae).

*Coenonycha santacruzae* Evans, 1986

Nomenclatural Authority: *Evans & Smith (2009)*

Literature Records: Santa Cruz (*Evans & Smith, 1986*: 82)

Digitized Records: Santa Cruz (2 ASUHIC; 1 SBMNH)

Range: Endemic (*Evans & Smith, 1986*).

Notes. *Evans & Smith (1986)* reported adults from *Adenostoma fasciculatum*, *Cercocarpus traskae* Eastw. (Rosaceae), and *Artemisia californica* Less. (Asteraceae).

*Dichelonyx* Harris, 1827

Nomenclatural Authority: *Evans & Smith (2009)*

Notes. Seventeen species of *Dichelonyx* are known from California (M. L. Gimmel, 2022, unpublished data).

*Dichelonyx backii* Kirby, 1837

Nomenclatural Authority: *Evans & Smith (2009)*

Literature Records: none

Digitized Records: Santa Cruz (5 LACM)

Range: Also known from mainland (*Evans & Smith, 2009*).

*Dichelonyx fulgida* LeConte, 1856

Nomenclatural Authority: *Evans & Smith (2009)*

Literature Records: Santa Cruz (*von Bloeker, 1939b*: 155)

Digitized Records: none

Range: Also known from mainland (*Evans & Smith, 2009*).

Notes. The subspecies occurring on the islands is *D. f. crotchii* Horn, 1876; it was recorded as *Dichelonyx crotchi* by *von Bloeker (1939b)*.

*Dichelonyx pusilla* LeConte, 1856

Nomenclatural Authority: *Evans & Smith (2009)*

Literature Records: San Miguel (*von Bloeker, 1939b*: 155), Santa Cruz (*von Bloeker, 1939b*: 155), Santa Rosa (*von Bloeker, 1939b*: 155)

Digitized Records: San Miguel (2 LACM), Santa Cruz (3 LACM), Santa Rosa (2 LACM)

Range: Also known from mainland (*Evans & Smith, 2009*).

Diplotaxini

Notes. One genus and 22 species of Diplotaxini are known to occur in California (M. L. Gimmel, 2022, unpublished data).

*Diplotaxis* Kirby, 1837

Nomenclatural Authority: *Evans & Smith (2009)*

Notes. Twenty-two species of *Diplotaxis* are known to occur in California (M. L. Gimmel, 2022, unpublished data). The genus was revised in two papers by *Vaurie (1958*, *1960)*.

### *Diplotaxis fimbriata* Fall, 1909

Nomenclatural Authority: *Vaurie (1960)*, *Evans & Smith (2009)*
Literature Records: San Clemente (*von Bloeker, 1939b*: 154), Santa Cruz (*von Bloeker, 1939b*: 154)
Digitized Records: none
Range: Also known from mainland (*Vaurie, 1960*).

### *Diplotaxis subangulata* LeConte, 1856

Nomenclatural Authority: *Vaurie (1960)*, *Evans & Smith (2009)*
Literature Records: San Clemente (*von Bloeker, 1939b*: 154), Santa Cruz (*von Bloeker, 1939b*: 154), Santa Rosa (*von Bloeker, 1939b*: 154)
Digitized Records: San Clemente (1 LACM; 1 SBMNH), Santa Cruz (3 LACM), Santa Rosa (1 LACM; 1 SBMNH)
Range: Also known from mainland (*Vaurie, 1960*).

### Hopliini

Notes. One genus and four species of Hopliini have been recorded from California (*Hardy, 1977*).

### *Hoplia* Illiger, 1803

Nomenclatural Authority: *Evans & Smith (2009)*
Notes. Four species of *Hoplia* have been recorded from California (*Hardy, 1977*). The species were revised by *Hardy (1977)*.

### *Hoplia callipyge* LeConte, 1856

Nomenclatural Authority: *Evans & Smith (2009)*
Literature Records: Santa Cruz (*Hardy, 1977*: 14)
Digitized Records: Santa Cruz (2 SBMNH)
Range: Also known from mainland (*Hardy, 1977*).

### Melolonthini

Notes. Five genera and 28 species of Melolonthini have been recorded from California (M. L. Gimmel, 2022, unpublished data).

### *Amblonoxia* Reitter, 1902

Nomenclatural Authority: *Evans & Smith (2009)*
Notes. Six species of *Amblonoxia* have been recorded from California (*Hardy, 1974*). The species were revised by *Hardy (1974)*. Members of this genus were until recently placed in the genus *Parathyce* Hardy, 1974, which is now a junior synonym of *Amblonoxia* (*Evans & Smith, 2009*).

### *Amblonoxia palpalis* (Horn, 1880)

Nomenclatural Authority: *Evans & Smith (2009)*
Literature Records: San Clemente (*von Bloeker, 1939b*: 155; *Hardy, 1974*: 20; *Miller & Miller, 1985*: 124), San Nicolas (*von Bloeker, 1939b*: 155; *Hardy, 1974*: 20; *Miller & Miller, 1985*: 124), Santa Barbara (*von Bloeker, 1939b*: 155; *Miller & Miller, 1985*: 124)

Digitized Records: Anacapa (2 SBMNH), San Nicolas (1 SBMNH)

Range: Also known from mainland (*Hardy, 1974*).

Notes. This species was recorded as *Thyce blaisdelli* Casey by *von Bloeker (1939b)*, which is now considered a junior synonym of *A. palpalis* (see *Evans & Smith, 2009*). It was recorded as *Parathyce palpalis* by *Hardy (1974)* and *Miller & Miller (1985)*.

### *Polyphylla* Harris, 1841

Nomenclatural Authority: *Evans & Smith (2009)*

Literature Records (genus-only): San Miguel (*von Bloeker, 1939b*: 155), Santa Cruz (*von Bloeker, 1939b*: 155)

Digitized Records (genus-only): Santa Cruz (1 UCSB)

Notes. Seventeen species of *Polyphylla* have been recorded from California (M. L. Gimmel, 2022, unpublished data). The genus was revised by *Young (1988)*. The above literature records were deemed unidentifiable to species (because of specimen damage) by *von Bloeker (1939b)*.

### *Polyphylla crinita* LeConte, 1856

Nomenclatural Authority: *Young (1988)*, *Evans & Smith (2009)*

Literature Records: Santa Cruz (*von Bloeker, 1939a*: 148; *von Bloeker, 1939b*: 154; *Young, 1967*: 307; *Young, 1988*: 55), Santa Rosa (*von Bloeker, 1939a*: 149; *von Bloeker, 1939b*: 154; *Young, 1967*: 307; *Young, 1988*: 55)

Digitized Records: Santa Catalina (16 LACM), Santa Cruz (47 LACM; 1 SBMNH)

Range: Also known from mainland (*Young, 1967*, *1988*).

Notes. This species was first recorded from the Channel Islands as the Santa Cruz-endemic *Polyphylla ona* von Bloeker, 1939 and the Santa Rosa-endemic *Polyphylla santarosae* von Bloeker, 1939 by *von Bloeker (1939a*, *1939b)*. These were synonymized with *P. crinita* by *Cazier (1940*: 137), who also synonymized *Polyphylla nigra* Casey, 1914, *Polyphylla martini* von Bloeker, 1939, and *Polyphylla santacruzae* von Bloeker, 1939 under the same name; *Young (1967*: 305) followed this arrangement. Later, however, *Young (1988)* removed the latter three taxa from synonymy with *P. crinita* (see entry for *P. nigra* below).

### *Polyphylla nigra* Casey, 1914

Nomenclatural Authority: *Young (1988)*, *Evans & Smith (2009)*

Literature Records: Santa Cruz (*von Bloeker, 1939a*: 149; *von Bloeker, 1939b*: 154; *Young, 1988*: 68), Santa Rosa (*von Bloeker, 1939a*: 149; *von Bloeker, 1939b*: 155; *Young, 1988*: 68)

Digitized Records: Santa Catalina (1 SBMNH), Santa Cruz (11 SBMNH)

Range: Also known from mainland (*Young, 1988*).

Notes. This species was first recorded from the Channel Islands as the Santa Rosa-endemic *P. martini* and the Santa Cruz-endemic *P. santacruzae* by *von Bloeker (1939a*, *1939b)*. These were synonymized with *P. crinita* by *Cazier (1940*: 137), which was followed by *Young (1967*: 305). Later, *Young (1988*: 67) synonymized these two names with the newly resurrected *P. nigra*. The larva of "*P. santacruzae*" was reported from "beneath roots of *Eucalyptus*, August" (*von Bloeker, 1939b*).

**Phobetusini**
Notes. One genus and nine species of Phobetusini have been recorded from California (*Hardy, 1978*).

***Phobetus* LeConte, 1856**
Nomenclatural Authority: *Evans & Smith (2009)*
Literature Records (genus-only): San Clemente (*Doyen, 1974*: 87), Santa Barbara (*Miller & Miller, 1985*: 125)
Digitized Records (genus-only): San Clemente (4 SBMNH), San Nicolas (11 SBMNH), Santa Rosa (26 SBMNH)
Notes. Nine species of *Phobetus* have been recorded from California (*Hardy, 1978*). *Barrett (1935)* and *Hardy (1978)* provided keys to species. The San Clemente, San Miguel, and Santa Barbara records of *Phobetus comatus* LeConte, 1856 in *von Bloeker (1939b*: 155) and San Clemente, San Nicolas, and Santa Rosa specimens in the SBMNH collection apparently represent one or more new species in the *Phobetus testaceus* group; the southern island specimens appear to be brachypterous (M. L. Gimmel, 2021, personal observation). The genus needs revision (A. Evans, 2021, personal communication).

***Phobetus ciliatus* Barrett, 1935**
Nomenclatural Authority: *Evans & Smith (2009)*
Literature Records: Santa Catalina (*Barrett, 1935*: 51; *Cazier, 1937*: 84; *Cockerell, 1940*: 286; *Miller, 1985a*: 19; *Caterino & Chandler, 2010*: 187)
Digitized Records: Santa Catalina (17 LACM; 4 SBMNH)
Range: Endemic (*Barrett, 1935*; *Cazier, 1937*; *Hardy, 1978*; *Miller, 1985a*; *Caterino & Chandler, 2010*).
Notes. Recorded simply from "Channel Islands, California" by *Hardy (1978)*. However, this distinctive species appears to be restricted to Santa Catalina Island (A. Evans, 2021, personal communication). *Fall's (1897*: 238; 1901: 141) and *von Bloeker's (1939b*: 155) records of *P. comatus* from that island apparently represent *P. ciliatus* (*Cockerell, 1940*: 286).

***Phobetus testaceus* LeConte, 1862**
Nomenclatural Authority: *Evans & Smith (2009)*
Literature Records: Santa Cruz (*LeConte, 1861*: 346; *Casey, 1909*: 282; *Fall & Davis, 1934*: 144; *Cazier, 1937*: 85; *Cockerell, 1940*: 286; *Evans, 1985*: 87; *Miller, 1985a*: 19)
Digitized Records: Santa Cruz (22 SBMNH; 3 UCSB)
Range: Endemic (*LeConte, 1861*; *Cazier, 1937*; *Hardy, 1978*; *Evans, 1985*; *Miller, 1985a*).
Notes. *Casey (1909*: 282) opined that this was "probably… a well-marked subspecies of *comatus*." *von Bloeker (1939b*: 155) noted that *Fall (1897*: 238) recorded this species from Santa Cruz Island under *P. comatus*, and did not provide any new records or comments on its validity. *Evans (1985*: 87) reported adults copulating and feeding on *Cercocarpus betuloides blancheae* (C. K. Schneid.) Little. Recorded simply from "Channel Islands, California" by *Hardy (1978)*. Apparently endemic to Santa Cruz Island (A. Evans, 2021, personal communication).

**Rhizotrogini**

Notes. One genus and 11 species of Rhizotrogini are known to occur in California (M. L. Gimmel, 2022, unpublished data).

***Phyllophaga* Harris, 1827**

Nomenclatural Authority: *Evans & Smith (2009)*

Notes. Eleven species of *Phyllophaga* are known from California, belonging to two subgenera, *Listrochelus* Blanchard, 1851 and *Phyllophaga* (*s.str.*) (M. L. Gimmel, 2022, unpublished data).

***Phyllophaga* (*Listrochelus*) *mucorea* (LeConte, 1856)**

Nomenclatural Authority: *Evans & Smith (2009)*

Literature Records: San Clemente (*von Bloeker, 1939b*: 154)

Digitized Records: none

Range: Also known from mainland (*Evans & Smith, 2009*).

Notes. Reported by *von Bloeker (1939b)* as *Listrochelus mucoreus*.

**Sericini**

Notes. One genus and 49 species of Sericini have been recorded from California (M. L. Gimmel, 2022, unpublished data).

***Serica* MacLeay, 1819**

Nomenclatural Authority: *Evans & Smith (2009)*

Literature Records (genus-only): San Clemente (*Doyen, 1974*: 87), Santa Barbara (*Miller & Miller, 1985*: 125)

Digitized Records (genus-only): Santa Cruz (1 UCSB)

Notes. Forty-nine species of *Serica* have been recorded from California (M. L. Gimmel, 2022, unpublished data). The genus is under revision by P. Lago (2022, personal communication).

***Serica alternata* LeConte, 1856**

Nomenclatural Authority: *Evans & Smith (2009)*

Literature Records: San Clemente (*von Bloeker, 1939b*: 154), San Nicolas (*von Bloeker, 1939b*: 154), Santa Barbara (*von Bloeker, 1939b*: 154)

Digitized Records: none

Range: Also known from mainland (*Dawson, 1933*).

Notes. *Miller & Miller (1985)* rejected *von Bloeker's (1939b)* island records of this species, since *Dawson (1933)* had revised the species previously included under *S. alternata*, and *von Bloeker (1939b)* apparently ignored this.

***Serica catalina* Dawson, 1947**

Nomenclatural Authority: *Evans & Smith (2009)*

Literature Records: Santa Catalina (*Dawson, 1947*: 234; *Miller, 1985a*: 19; *Caterino & Chandler, 2010*: 187)

Digitized Records: none
Range: Endemic (*Dawson, 1947*; *Miller, 1985a*; *Caterino & Chandler, 2010*).

### *Serica cruzi* Saylor, 1939
Nomenclatural Authority: *Evans & Smith (2009)*
Literature Records: Santa Cruz (*Saylor, 1939*: 55; *Miller, 1985a*: 19)
Digitized Records: Santa Cruz (5 LACM; 1 SBMNH; 1 UCSB; 1 USNM)
Range: Endemic (*Saylor, 1939*; *Miller, 1985a*).

### *Serica mixta* LeConte, 1856
Nomenclatural Authority: *Evans & Smith (2009)*
Literature Records: San Clemente (*von Bloeker, 1939b*: 154), San Miguel (*von Bloeker, 1939b*: 154), San Nicolas (*von Bloeker, 1939b*: 154), Santa Cruz (*von Bloeker, 1939b*: 154), Santa Rosa (*von Bloeker, 1939b*: 154)
Digitized Records: none
Range: Also known from mainland (*Dawson, 1947*).

### Scarabaeinae
Notes. Five tribes, seven genera, and nine species of Scarabaeinae are known to occur in California (M. L. Gimmel, 2022, unpublished data).

### *Canthon* Hoffmannsegg, 1817
Nomenclatural Authority: *Smith (2009)*
Notes. One species of *Canthon* has been recorded from California (*Robinson, 1948*), belonging to the subgenus *Boreocanthon* Halffter, 1958.

### *Canthon* (*Boreocanthon*) *simplex* LeConte, 1857
Nomenclatural Authority: *Robinson (1948)*
Literature Records: San Clemente (*von Bloeker, 1939b*: 153)
Digitized Records: none
Range: Also known from mainland (*Robinson, 1948*).

### Trogidae
Notes. This family is represented by two genera and eight species in California (*Vaurie, 1955*). The family was treated for North America by *Vaurie (1955)*.

### *Trox* Fabricius, 1775
Nomenclatural Authority: *Smith (2009)*
Notes. Six species of *Trox* have been recorded from California (*Vaurie, 1955*).

### *Trox atrox* LeConte, 1854
Nomenclatural Authority: *Vaurie (1955)*
Literature Records: San Clemente (*von Bloeker, 1939b*: 154)
Digitized Records: San Clemente (2 LACM)
Range: Also known from mainland (*Vaurie, 1955*).

*Trox gemmulatus* **Horn, 1874**
Nomenclatural Authority: *Vaurie (1955)*
Literature Records: San Clemente (*von Bloeker, 1939b*: 154)
Digitized Records: San Clemente (5 LACM; 1 SBMNH)
Range: Also known from mainland (*Vaurie, 1955*).

**STAPHYLINOIDEA**

**Colonidae, NEW FAMILY RECORD**
Notes. One genus and 14 species of Colonidae have been recorded from California (*Peck & Stephan, 1996*; *Peck & Newton, 2017*). Until recently, this family was treated as a subfamily of Leiodidae (see *Cai et al., 2022*).

*Colon* **Herbst, 1797**
Nomenclatural Authority: *Peck & Newton (2017)*
Notes. *Peck & Stephan (1996)* reported 14 species of *Colon* from California belonging to three subgenera (*Colon* (*s.str.*), *Eurycolon* Ganglbauer, 1899, and *Myloechus* Latreille, 1806) in their revision of the North American species of the genus.

*Colon* (*Myloechus*) *forceps* **Hatch, 1957**
Nomenclatural Authority: *Peck & Newton (2017)*
Literature Records: none
Digitized Records: Santa Cruz (1 SBMNH)
Range: Also known from mainland (*Peck & Stephan, 1996*).

**Hydraenidae**
Notes. Two subfamilies, five genera, and 49 species of Hydraenidae have been recorded from California (*Perkins, 1980*). This family was treated for the New World by *Perkins (1980)*.

**Hydraeninae**
Notes. Two genera and 16 species of Hydraeninae have been recorded from California (*Perkins, 1980*).

*Hydraena* **Kugelann, 1794**
Nomenclatural Authority: *Perkins (1980)*
Digitized Records (genus-only): San Clemente (6 SBMNH), Santa Cruz (23 SBMNH), Santa Rosa (10 SBMNH)
Notes. Twelve species of *Hydraena* have been recorded from California (*Perkins, 1980*).

*Hydraena arenicola* **Perkins, 1980**
Nomenclatural Authority: *Perkins (1980)*
Literature Records: Santa Cruz (*Perkins, 1980*: 485)
Digitized Records: none
Range: Also known from mainland (*Perkins, 1980*).

*Hydraena circulata* **Perkins, 1980**
Nomenclatural Authority: *Perkins (1980)*
Literature Records: Santa Cruz (*Perkins, 1980*: 483)
Digitized Records: none
Range: Also known from mainland (*Perkins, 1980*).

*Hydraena vandykei* **d'Orchymont, 1923**
Nomenclatural Authority: *Perkins (1980)*
Literature Records: Santa Cruz (*Perkins, 1980*: 493)
Digitized Records: none
Range: Also known from mainland (*Perkins, 1980*).

**Ochthebiinae**
Notes. Three genera and 33 species of Ochthebiinae have been recorded from California (*Perkins, 1980*).

*Ochthebius* **Leach, 1815**
Nomenclatural Authority: *Perkins (1980)*
Digitized Records (genus-only): San Clemente (52 SBMNH), San Miguel (50 SBMNH), San Nicolas (22 SBMNH), Santa Catalina (14 SBMNH), Santa Cruz (30 SBMNH), Santa Rosa (103 SBMNH)
Notes. Thirty species of *Ochthebius* have been recorded from California (*Perkins, 1980*).

*Ochthebius discretus* **LeConte, 1878**
Nomenclatural Authority: *Perkins (1980)*
Literature Records: Santa Catalina (*Fall, 1897*: 236)
Digitized Records: none
Range: Also known from mainland (*Perkins, 1980*).

*Ochthebius interruptus* **LeConte, 1852**
Nomenclatural Authority: *Perkins (1980)*
Literature Records: Santa Cruz (*Perkins, 1980*: 507; *Furlong & Wenner, 2002*: 250)
Digitized Records: none
Range: Also known from mainland (*Perkins, 1980*).

*Ochthebius puncticollis* **LeConte, 1852**
Nomenclatural Authority: *Perkins (1980)*
Literature Records: Santa Cruz (*Perkins, 1980*: 524)
Digitized Records: Santa Cruz (3 SBMNH)
Range: Also known from mainland (*Perkins, 1980*).

**Leiodidae**
Notes. There are four subfamilies, 19 genera, and 101 species of Leiodidae known from California (*Peck & Newton, 2017*; M. L. Gimmel, 2022, unpublished data). A distributional catalog of the North American fauna of the family was provided by *Peck & Newton (2017)*.

### Catopocerinae

Notes. One genus and 23 species of Catopocerinae have been recorded from California (*Peck & Cook, 2011*).

### *Pinodytes* Horn, 1880

Nomenclatural Authority: *Peck & Newton (2017)*

Notes. A total of 23 species of this genus is known from California (*Peck & Cook, 2011*), and it is likely that additional species will be discovered in the Channel Islands. The genus was revised by *Peck & Cook (2011)*.

### *Pinodytes gibbosus* Peck & Cook, 2011

Nomenclatural Authority: *Peck & Newton (2017)*

Literature Records: Santa Catalina (*Peck & Cook, 2011*: 26), Santa Cruz (*Peck & Cook, 2011*: 26), Santa Rosa (*Peck & Cook, 2011*: 26)

Digitized Records: Santa Catalina (7 SBMNH), Santa Cruz (4 SBMNH), Santa Rosa (36 SBMNH)

Range: Also known from mainland (*Peck & Cook, 2011*).

Notes. *Caterino & Chandler (2010*: 191) reported the genus *Catopocerus* Motschulsky, 1870 from Santa Catalina Island; this record presumably refers to this species.

### Leiodinae

Notes. Three tribes, 13 genera, and 63 species of Leiodinae have been recorded from California (*Peck & Newton, 2017*; M. L. Gimmel, 2022, unpublished data).

### *Agathidium* Panzer, 1797

Nomenclatural Authority: *Peck & Newton (2017)*

Notes. Twenty-three species of *Agathidium* have been recorded from California (*Miller & Wheeler, 2005*; *Wheeler & Miller, 2005*). This genus was monographed for North America in two papers by *Miller & Wheeler (2005)* and *Wheeler & Miller (2005)*.

### *Agathidium pulchrum* LeConte, 1853

Nomenclatural Authority: *Peck & Newton (2017)*

Literature Records: none

Digitized Records: Santa Cruz (1 SBMNH)

Range: Also known from mainland (*Miller & Wheeler, 2005*).

### *Agathidium virile* Fall, 1901

Nomenclatural Authority: *Peck & Newton (2017)*

Literature Records: none

Digitized Records: San Clemente (29 SBMNH), Santa Catalina (2 SBMNH)

Range: Also known from mainland (*Wheeler & Miller, 2005*).

### *Leiodes* Latreille, 1797

Nomenclatural Authority: *Peck & Newton (2017)*

Literature Records (genus-only): Santa Catalina (*Caterino & Chandler, 2010*: 191)

Notes. Sixteen species of *Leiodes* have been recorded from California (*Baranowski, 1993*). This genus was revised for North America by *Baranowski (1993)*. *Caterino & Chandler (2010*: 191) reported this genus as a new record from Santa Catalina Island.

**Leiodes antennata (Fall, 1910)**
Nomenclatural Authority: *Peck & Newton (2017)*
Literature Records: none
Digitized Records: Santa Catalina (3 SBMNH)
Range: Also known from mainland (*Baranowski, 1993*).

**Leiodes paludicola (Crotch, 1874)**
Nomenclatural Authority: *Peck & Newton (2017)*
Literature Records: none
Digitized Records: Santa Catalina (3 SBMNH)
Range: Also known from mainland (*Baranowski, 1993*).

**Ptiliidae, NEW FAMILY RECORD**
Notes. Two subfamilies, 11 genera, and 29 species of Ptiliidae are known to occur in California (M. L. Gimmel, 2022, unpublished data).

**Nossidiinae**
Notes. Two genera and two species of Nossidiinae are known to occur in California (M. L. Gimmel, 2022, unpublished data).

**Motschulskium Matthews, 1872**
Nomenclatural Authority: *Hall (2000)*
Notes. One species of *Motschulskium* has been recorded from the west coast of North America, including California (*Hall, 2000*).

**Motschulskium sinuatocolle Matthews, 1872**
Nomenclatural Authority: *Hall (2000)*
Literature Records: none
Digitized Records: San Clemente (6 SBMNH), San Nicolas (6 SBMNH), Santa Catalina (1 SBMNH)
Range: Also known from mainland (*Hall, 2000*).

**Ptiliinae**
Notes. Five tribes, nine genera, and 27 species of Ptiliinae are known to occur in California (M. L. Gimmel, 2022, unpublished data).

**Acrotrichini**
Notes. Two genera and 11 species of Acrotrichini are known to occur in California (M. L. Gimmel, 2022, unpublished data).

**Acrotrichis Motschulsky, 1848**
Nomenclatural Authority: *Hall (2000)*

Notes. Ten species of *Acrotrichis* have been recorded from California (M. L. Gimmel, 2022, unpublished data).

### *Acrotrichis* undetermined species
Literature Records: none
Digitized Records: Santa Cruz (3 SBMNH), Santa Rosa (1 SBMNH)

### Ptenidiini
Notes. One genus and four species of Ptenidiini have been recorded from California (M. L. Gimmel, 2022, unpublished data).

### *Ptenidium* Erichson, 1845
Nomenclatural Authority: *Hall (2000)*
Notes. Four species of *Ptenidium* have been recorded from California (M. L. Gimmel, 2022, unpublished data).

### *Ptenidium* undetermined species
Literature Records: none
Digitized Records: Santa Rosa (33 SBMNH)

### Ptiliini
Notes. Three genera and nine species of Ptiliini are known to occur in California (M. L. Gimmel, 2022, unpublished data).

### *Actidium* Matthews, 1869
Nomenclatural Authority: *Hall (2000)*
Notes. Four species of *Actidium* have been recorded from California (M. L. Gimmel, 2022, unpublished data).

### *Actidium* undetermined species
Literature Records: none
Digitized Records: Santa Cruz (1 SBMNH)

### *Ptiliolum* Flach, 1888
Nomenclatural Authority: *Hall (2000)*
Notes. No species of *Ptiliolum* have been recorded from California in the literature (M. L. Gimmel, 2022, unpublished data); this represents a **new state record** for the genus.

### *Ptiliolum* undetermined species
Literature Records: none
Digitized Records: San Clemente (9 SBMNH), Santa Catalina (38 SBMNH), Santa Rosa (30 SBMNH)

### Ptinellini
Notes. Two genera and two species of Ptinellini are known to occur in California (M. L. Gimmel, 2022, unpublished data).

### *Pteryx* Matthews, 1859
Nomenclatural Authority: *Hall (2000)*
Notes. No species of *Pteryx* have been recorded from California in the literature (M. L. Gimmel, 2022, unpublished data); this represents a **new state record** for the genus.

### *Pteryx* undetermined species
Literature Records: none
Digitized Records: Santa Cruz (27 SBMNH), Santa Rosa (24 SBMNH)

### Staphylinidae
Notes. Twenty-five subfamilies, 306 genera, and, 1,349 species of Staphylinidae are known to occur in California, making it the largest family of beetles in the state (M. L. Gimmel, 2022, unpublished data). The staphylinid subfamilies Dasycerinae, Euaesthetinae, Micropeplinae, Osoriinae, Proteininae, Scaphidiinae, Steninae, Trichophyinae, and Trigonurinae occur on nearby mainland but no Channel Islands records are known.

### Aleocharinae
Notes. Nineteen tribes, 103 genera, and 360 species of this enormously diverse and poorly understood subfamily are known from California (M. L. Gimmel, 2022, unpublished data). "Aleocharinae, genus near *Oxypoda*" was reported from Santa Barbara Island by *Miller & Miller (1985*: 124); the specimen is housed in SBMNH but was on loan during this study.

### Aleocharini
Notes. Three genera and 36 species of Aleocharini are known to occur in California (M. L. Gimmel, 2022, unpublished data).

### *Aleochara* Gravenhorst, 1802
Nomenclatural Authority: *Newton et al. (2000)*
Notes. Thirty-three species of *Aleochara* have been recorded from California, belonging to seven subgenera, *Aleochara* (*s.str.*), *Calochara* Casey, 1906, *Coprochara* Mulsant & Rey, 1874, *Echocara* Casey, 1906, *Emplenota* Casey, 1884, *Maseochara* Sharp, 1883, and *Xenochara* Mulsant & Rey, 1874 (M. L. Gimmel, 2022, unpublished data). One specimen from Santa Catalina Island in the H.C. Fall collection in the Museum of Comparative Zoology, Harvard University was identified more recently as *Aleochara* (*Emplenota*) *pacifica* (Casey, 1893) by J. Klimaszewski in 1982 (S. Miller, 2022, personal communication). However, this specimen did not appear in *Klimaszewski's (1984)* revision and needs to be verified.

### *Aleochara* (*Coprochara*) *bimaculata* Gravenhorst, 1802
Nomenclatural Authority: *Klimaszewski (1984)*
Literature Records: San Clemente (*Fall, 1897*: 236), Santa Catalina (*Fall, 1897*: 236)
Digitized Records: none
Range: Also known from mainland (*Klimaszewski, 1984*).

***Aleochara* (*Coprochara*) *densissima* Bernhauer, 1906**

Nomenclatural Authority: *Klimaszewski (1984)*

Literature Records: Santa Catalina (*Cockerell, 1940*: 285; *Klimaszewski, 1984*: 27)

Digitized Records: none

Range: Also known from mainland (*Klimaszewski, 1984*).

***Aleochara* (*Coprochara*) *sulcicollis* Mannerheim, 1843**

Nomenclatural Authority: *Klimaszewski (1984)*

Literature Records: San Miguel (*Cockerell, 1940*: 285), San Nicolas (*Cockerell, 1940*: 285), Santa Cruz (*Klimaszewski, 1984*: 33), Santa Rosa (*Fall, 1897*: 236)

Digitized Records: San Clemente (5 SBMNH), San Miguel (17 SBMNH), San Nicolas (5 SBMNH), Santa Catalina (4 SBMNH), Santa Cruz (40 SBMNH), Santa Rosa (2 SBMNH)

Range: Also known from mainland (*Klimaszewski, 1984*).

Notes. This species was recorded as *Baryodma sulcicollis* by *Cockerell (1940)*.

***Aleochara* (*Emplenota*) *curtidens* Klimaszewski, 1984**

Nomenclatural Authority: *Klimaszewski (1984)*

Literature Records: Santa Barbara (*Klimaszewski, 1984*: 102; *Miller & Miller, 1985*: 124)

Digitized Records: none

Range: Also known from mainland (*Klimaszewski, 1984*).

Notes. Misspelled by *Miller & Miller (1985)* as *Aleochara curtedens*.

***Aleochara* (*Emplenota*) *littoralis* (Mäklin, 1853)**

Nomenclatural Authority: *Klimaszewski (1984)*

Literature Records: Santa Barbara (*Fall, 1897*: 236), Santa Catalina (*Fall, 1897*: 236; *Klimaszewski, 1984*: 99)

Digitized Records: none

Range: Also known from mainland (*Fall, 1901*; *Klimaszewski, 1984*).

Notes. *Fall (1897)* recorded this species as *Polistoma arenaria* Casey and *Fall (1901)* recorded it as *Polystoma arenaria* from "the islands off the coast"; *P. arenaria* was synonymized with *A. littoralis* by *Klimaszewski (1984*: 98).

***Aleochara* (*Maseochara*) *valida* LeConte, 1858**

Nomenclatural Authority: *Klimaszewski (1984)*

Literature Records: Santa Catalina (*Klimaszewski, 1984*: 107)

Digitized Records: San Clemente (5 SBMNH), San Miguel (12 SBMNH), Santa Rosa (5 SBMNH)

Range: Also known from mainland (*Klimaszewski, 1984*).

***Aleochara* (*Xenochara*) *fumata* Gravenhorst, 1802**

Nomenclatural Authority: *Klimaszewski (1984)*

Literature Records: Santa Catalina (*Klimaszewski, 1984*: 53)

Digitized Records: none

Range: Also known from mainland (*Klimaszewski, 1984*).

Notes. This species was introduced from the Palearctic region (*Klimaszewski, 1984*).

### *Aleochara* (*Xenochara*) *lanuginosa* Gravenhorst, 1802

Nomenclatural Authority: *Klimaszewski (1984)*

Literature Records: Santa Rosa (*Klimaszewski, 1984*: 50)

Digitized Records: Santa Rosa (4 SBMNH)

Range: Also known from mainland (*Klimaszewski, 1984*).

Notes. This species was introduced from the Palearctic region (*Klimaszewski, 1984*).

### Athetini

Notes. Twenty-eight genera and 124 species of Athetini are known to occur in California (M. L. Gimmel, 2022, unpublished data).

### *Acrotona* Thomson, 1859

Nomenclatural Authority: *Newton et al. (2000)*

Digitized Records (genus-only): San Nicolas (1 SBMNH), Santa Cruz (4 SBMNH)

Notes. Six species of *Acrotona* have been reported from California (M. L. Gimmel, 2022, unpublished data).

### *Acrotona recondita* (Erichson, 1839)

Nomenclatural Authority: *Klimaszewski et al. (2015)*

Literature Records: Santa Catalina (*Casey, 1910*: 136)

Digitized Records: none

Range: Also known from mainland (*Casey, 1910*; *Klimaszewski et al., 2015*).

Notes. Reported by *Casey (1910)* as *Arisota umbrina* Casey, 1910, currently a synonym of *Ac. recondita* (see *Klimaszewski et al., 2015*).

### *Acrotona sonomana* (Casey, 1910)

Nomenclatural Authority: *Gusarov (2003b)*

Literature Records: Santa Catalina (*Casey, 1911*: 166; *Miller, 1985a*: 19; *Gusarov, 2003b*: 106)

Digitized Records: none

Range: Also known from mainland (*Gusarov, 2003b*).

Notes. Reported as the "endemic" *Strigota* (*Eustrigota*) *seclusa* Casey, 1911 by *Casey (1911)* and *Miller (1985a)*. This species was synonymized with *A. sonomana* by *Gusarov (2003b)*.

### *Adota* Casey, 1910

Nomenclatural Authority: *Newton et al. (2000)*, *Gusarov (2003a)*

Notes. Of the three *Adota* species in North America, two have been reported from California (*Gusarov, 2003a*), both from seashore environments.

### *Adota maritima* (Mannerheim, 1843)

Nomenclatural Authority: *Gusarov (2003a)*

Literature Records: Santa Catalina (*Gusarov, 2003a*: 11)

Digitized Records: none

Range: Occurs along much of the west coast of North America, from southern California to Alaska (*Gusarov, 2003a*).

### *Atheta* Thomson, 1858
Nomenclatural Authority: *Newton et al. (2000)*
Notes. Sixteen species of *Atheta* have been recorded from California (M. L. Gimmel, 2022, unpublished data).

### *Atheta hampshirensis* Bernhauer, 1909
Nomenclatural Authority: *Gusarov (2003b)*
Literature Records: none
Digitized Records: San Nicolas (1 SBMNH)
Range: Also known from mainland (*Gusarov, 2003b*).

### *Hydrosmecta* Thomson, 1858
Nomenclatural Authority: *Newton et al. (2000)*
Notes. As many as nine species of *Hydrosmecta* may occur in California (M. L. Gimmel & M. S. Caterino, 2022, personal data), but the genus has not been recently revised. Several may not be congeneric, and some may be synonyms (*Seevers, 1978*).

### *Hydrosmecta* undetermined species
Literature Records: none
Digitized Records: Santa Cruz (20 SBMNH)

### *Pontomalota* Casey, 1885
Nomenclatural Authority: *Newton et al. (2000)*
Notes. Two species of *Pontomalota* have been recorded from California (*Ahn & Ashe, 1992*). The species of this genus were revised by *Ahn & Ashe (1992)*.

### *Pontomalota opaca* (LeConte, 1863)
Nomenclatural Authority: *Ahn & Ashe (1992)*
Literature Records: San Miguel (*Ahn & Ashe, 1992*: 352)
Digitized Records: San Miguel (30 SBMNH), San Nicolas (6 SBMNH), Santa Rosa (7 SBMNH)
Range: Also known from mainland (*Ahn & Ashe, 1992*).

### "*Sonomota*" Casey, 1911
Nomenclatural Authority: *Newton et al. (2000)*
Notes. Currently this genus-group name is a synonym of *Atheta* (*Microdota*) Mulsant & Rey, 1873, not of *Geostiba* Thomson, 1858 as listed by *Newton et al. (2000)* (see *Gusarov, 2002*). However, it is being used here to designate a distinctive group of West Coast athetines (V. Gusarov, 2022, personal communication).

### *Sonomota* undetermined species
Literature Records: none
Digitized Records: San Clemente (7 SBMNH), Santa Cruz (7 SBMNH)

***Tarphiota* Casey, 1893**
Nomenclatural Authority: *Newton et al. (2000)*
Notes. Two species of *Tarphiota* have been recorded from California (*Ahn, 1996b*; *Klimaszewski, Majka & Langor, 2006*). The North American species were revised by *Ahn (1996b)*; *Klimaszewski, Majka & Langor (2006)* provided an update.

***Tarphiota fucicola* (Mäklin, 1852)**
Nomenclatural Authority: *Ahn (1996b)*
Literature Records: none
Digitized Records: San Miguel (24 SBMNH), San Nicolas (5 SBMNH), Santa Cruz (29 SBMNH), Santa Rosa (16 SBMNH)
Range: Also known from mainland (*Ahn, 1996b*).

***Tarphiota geniculata* (Mäklin, 1852)**
Nomenclatural Authority: *Ahn (1996b)*
Literature Records: none
Digitized Records: San Clemente (9 SBMNH), San Miguel (17 SBMNH), San Nicolas (5 SBMNH), Santa Catalina (10 SBMNH), Santa Cruz (30 SBMNH), Santa Rosa (13 SBMNH)
Range: Also known from mainland (*Ahn, 1996b*).

***Thinusa* Casey, 1893**
Nomenclatural Authority: *Newton et al. (2000)*
Digitized Records: Santa Cruz (1 UCRC)
Notes. Two species of *Thinusa* have been recorded from California (*Ahn, 1997*). The species of this genus were revised by *Ahn (1997)*.

***Thinusa fletcheri* Casey, 1906**
Nomenclatural Authority: *Ahn (1997)*
Literature Records: none
Digitized Records: San Clemente (2 SBMNH), San Nicolas (1 SBMNH), Santa Catalina (1 SBMNH), Santa Rosa (1 SBMNH)
Range: Also known from mainland (*Ahn, 1997*).

***Thinusa maritima* (Casey, 1885)**
Nomenclatural Authority: *Ahn (1997)*
Literature Records: Santa Cruz (*Ahn, 1997*: 80)
Digitized Records: Santa Cruz (2 SBMNH)
Range: Also known from mainland (*Ahn, 1997*).

**Falagriini**
Notes. Six genera and seven species of Falagriini have been recorded from California (M. L. Gimmel, 2022, unpublished data). The species of Falagriini were revised for North America by *Hoebeke (1985)*.

*Falagriota* **Casey, 1906**

Nomenclatural Authority: *Newton et al. (2000)*

Notes. One species of *Falagriota* has been recorded from California (*Hoebeke, 1985*).

*Falagriota occidua* **(Casey, 1885)**

Nomenclatural Authority: *Hoebeke (1985)*

Literature Records: Santa Cruz (*Naughton et al., 2014*: 304)

Digitized Records: Santa Cruz (11 SBMNH), Santa Rosa (2 SBMNH)

Range: Also known from mainland (*Hoebeke, 1985*).

**Homalotini**

Notes. Nine genera and 32 species of Homalotini have been recorded from California (M. L. Gimmel, 2022, unpublished data).

*Diestota* **Mulsant & Rey, 1871**

Nomenclatural Authority: *Newton et al. (2000)*

Notes. Two species, *Diestota angustula* (Casey, 1906) and *Diestota spissula* (Casey, 1911), have been recorded from California (*Seevers, 1978*, as *Apheloglossa* Casey, 1893).

*Diestota* **undetermined species**

Literature Records: none

Digitized Records: San Clemente (3 SBMNH), San Miguel (14 SBMNH), San Nicolas (4 SBMNH), Santa Catalina (2 SBMNH), Santa Cruz (2 SBMNH), Santa Rosa (1 SBMNH)

*Stictalia* **Casey, 1906**

Nomenclatural Authority: *Newton et al. (2000)*

Notes: The sixteen North American species of *Stictalia* are all western (*Newton et al., 2000*). Of these, 12 have been recorded from California, but essentially only from their original type localities (*Seevers, 1978*).

*Stictalia* **undetermined species**

Literature Records: none

Digitized Records: San Clemente (36 SBMNH), Santa Cruz (5 SBMNH), Santa Rosa (15 SBMNH)

**Hypocyphtini**

Notes. Three genera and five species of Hypocyphtini are known to occur in California (M. L. Gimmel, 2022, unpublished data).

*Holobus* **Solier, 1849**

Nomenclatural Authority: *Newton et al. (2000)*

Notes: Only one species, *Holobus oviformis* Casey, 1893, has been recorded from California, from Los Angeles and San Diego counties. It is likely that the undetermined Santa Catalina Island specimen below is referable to this species.

***Holobus* undetermined species**
Literature Records: none
Digitized Records: Santa Catalina (1 SBMNH)

***Oligota* Mannerheim, 1830**
Nomenclatural Authority: *Newton et al. (2000)*
Notes: Three of the 10 North American species of *Oligota* have been recorded from California (M. L. Gimmel, 2022, unpublished data).

***Oligota* undetermined species**
Literature Records: none
Digitized Records: San Nicolas (1 SBMNH), Santa Cruz (4 SBMNH), Santa Rosa (15 SBMNH)

**Liparocephalini**
Notes. Four genera and seven species of Liparocephalini are known to occur in California (M. L. Gimmel, 2022, unpublished data).

***Diaulota* Casey, 1893**
Nomenclatural Authority: *Newton et al. (2000)*
Notes. Four species of *Diaulota* have been recorded from California (*Ahn, 1996a*). The species of this genus were reviewed by *Ahn (1996a).*

***Diaulota fulviventris* Moore, 1956**
Nomenclatural Authority: *Ahn (1996a)*
Literature Records: none
Digitized Records: Santa Cruz (2 SBMNH), Santa Rosa (1 SBMNH)
Range: Also known from mainland (*Ahn, 1996a*).

**Myllaenini**
Notes. Two genera and eight species of Myllaenini have been recorded from California (M. L. Gimmel, 2022, unpublished data).

***Bryothinusa* Casey, 1904**
Nomenclatural Authority: *Newton et al. (2000)*
Notes. One species of *Bryothinusa* has been recorded from California (*Moore & Orth, 1979*).

***Bryothinusa catalinae* Casey, 1904**
Nomenclatural Authority: *Moore & Orth (1979)*
Literature Records: Santa Catalina (*Casey, 1904*: 313; *Baker, 1905*: 57; *Moore, 1956*: 132)
Digitized Records: Santa Catalina (8 USNM)
Range: Also known from mainland (*Moore, 1956*; *Moore & Orth, 1979*).
Notes. *Moore & Orth (1979)* described the larva of this odd seashore species. The species was considered endemic to Santa Catalina Island until *Moore (1956)* reported on mainland specimens.

*Myllaena* Erichson, 1837

Nomenclatural Authority: *Newton et al. (2000)*

Notes. Seven species of *Myllaena* have been reported from California (*Klimaszewski, 1982*). At least four of these occur in coastal southern California, potentially conspecific with *Myllaena* from the Channel Islands. The species were revised for North America by *Klimaszewski (1982)*.

*Myllaena* **undetermined species**

Literature Records: none

Digitized Records: San Miguel (3 SBMNH), Santa Catalina (1 SBMNH), Santa Cruz (11 SBMNH), Santa Rosa (1 SBMNH)

**Oxypodini**

Notes. Twenty genera and 78 species of Oxypodini are known to occur in California (M. L. Gimmel, 2022, unpublished data).

**Meoticina undetermined genus and species**

Literature Records: none

Digitized Records: Santa Cruz (8 SBMNH)

Notes: Four genera of this oxypodine subtribe are known from California, *Gyronycha* Casey, 1893, *Alisalia* Casey, 1911, *Apimela* Mulsant & Rey, 1874, and *Bamona* Sharp, 1883 (*Newton et al., 2000*), though none, yet, from the Channel Islands. This record is certainly referable to one of them.

*Blepharhymenus* Solier, 1849

Nomenclatural Authority: *Newton et al. (2000)*

Notes: Nineteen species of *Blepharhymenus* have been described from California (*Seevers, 1978*), all in need of revision.

*Blepharhymenus* **undetermined species**

Literature Records: none

Digitized Records: San Clemente (2 SBMNH), Santa Cruz (2 SBMNH), Santa Rosa (10 SBMNH)

*Oxypoda* Mannerheim, 1830

Nomenclatural Authority: *Newton et al. (2000)*

Notes. Twenty-eight species of *Oxypoda* have been recorded from California (M. L. Gimmel, 2022, unpublished data).

*Oxypoda* **undetermined species**

Literature Records: none

Digitized Records: Santa Catalina (4 SBMNH), Santa Cruz (10 SBMNH), Santa Rosa (15 SBMNH)

*Phloeopora* Erichson, 1837

Nomenclatural Authority: *Newton et al. (2000)*

Notes: Two of the eight North American species of the cosmopolitan genus *Phloeopora* have been reported from California (*Seevers, 1978*).

### *Phloeopora* undetermined species

Literature Records: none
Digitized Records: Santa Cruz (3 SBMNH)

### Tachyusini

Notes. Seven genera and 27 species of Tachyusini are known to occur in California (M. L. Gimmel, 2022, unpublished data).

### *Gnypeta* Thomson, 1858

Nomenclatural Authority: *Newton et al. (2000)*
Notes: Eighteen species of this large, widespread genus have been reported from California (*Seevers, 1978*), but they are in need of revision.

### *Gnypeta* undetermined species

Literature Records: none
Digitized Records: Santa Cruz (4 SBMNH), Santa Rosa (2 SBMNH)

### Habrocerinae

Notes. The subfamily Habrocerinae, which contains one genus and species in California, was revised by *Assing & Wunderle (1995)*.

### *Habrocerus* Erichson, 1839

Nomenclatural Authority: *Herman (2001)*
Notes. One adventive species of *Habrocerus* occurs in California (*Herman, 2001*).

### *Habrocerus capillaricornis* (Gravenhorst, 1806)

Nomenclatural Authority: *Herman (2001)*
Literature Records: none
Digitized Records: Santa Cruz (1 SBMNH)
Range: Also known from mainland (*Assing & Wunderle, 1995*).
Notes. This species was introduced to North America from the Western Palearctic (*Assing & Wunderle, 1995*).

### Leptotyphlinae

Notes. The subfamily Leptotyphlinae contains seven genera and 12 described species in California (M. L. Gimmel, 2022, unpublished data).

### Leptotyphlinae undetermined genus and species

Literature Records: none
Digitized Records: Santa Cruz (2 SBMNH).
Notes. The two specimen lots recorded above from SBMNH are currently housed in ethanol tubes and collectively represent >30 specimens. A large number of undescribed

species of Leptotyphlinae are known to occur in California (V. Gusarov, 2022, personal communication); the Channel Island specimens are almost certainly undescribed.

**Mycetoporinae**
Notes. This subfamily was recently split from Tachyporinae by *Yamamoto (2021)*.
It contains nine genera and 29 species in California (M. L. Gimmel, 2022, unpublished data).

***Bryoporus* Kraatz, 1857**
Nomenclatural Authority: *Herman (2001)*
Notes. One species of *Bryoporus* has been recorded from California (*Campbell, 1993b*). The genus was revised for North America by *Campbell (1993b)*.

***Bryoporus rufescens* LeConte, 1863**
Nomenclatural Authority: *Herman (2001)*
Literature Records: Santa Cruz (*Naughton et al., 2014*: 304)
Digitized Records: San Clemente (32 SBMNH), Santa Cruz (2 SBMNH), Santa Rosa (1 SBMNH)
Range: Also known from mainland (*Campbell, 1993b*).

***Lordithon* Thomson, 1859**
Nomenclatural Authority: *Herman (2001)*
Notes. Four species of *Lordithon* have been recorded from California (*Campbell, 1982*). This genus was revised for North America by *Campbell (1982)*. Additional species are likely to occur on the Channel Islands.

***Lordithon thoracicus* (Fabricius, 1777)**
Nomenclatural Authority: *Herman (2001)*
Literature Records: none
Digitized Records: Santa Catalina (2 SBMNH), Santa Cruz (7 SBMNH)
Range: Also known from mainland (*Campbell, 1982*).
Notes. The nominate subspecies, *L. t. thoracicus* (Fabricius, 1777), is the only subspecies occurring in California (*Campbell, 1982*). The species has a Holarctic distribution (*Campbell, 1982*).

***Mycetoporus* Mannerheim, 1830**
Nomenclatural Authority: *Herman (2001)*
Notes. Six species of *Mycetoporus* have been recorded from California (*Campbell, 1991*). The genus was revised for North America by *Campbell (1991)*. Additional species of *Mycetoporus* possibly occur on the Channel Islands.

***Mycetoporus neotomae* Fall, 1910**
Nomenclatural Authority: *Herman (2001)*
Literature Records: none
Digitized Records: Santa Catalina (1 SBMNH), Santa Cruz (5 SBMNH)
Range: Also known from mainland (*Campbell, 1991*).

**Omaliinae: Omaliini**
Notes. Five tribes, 33 genera, and 95 species of Omaliinae are known to occur in California, with 10 genera and 22 species of Omaliini known to occur in the state (*Herman, 2001*; M. L. Gimmel, 2022, unpublished data).

*Omalium* Gravenhorst, 1802
Nomenclatural Authority: *Herman (2001)*
Notes. Eight species of *Omalium* have been recorded from California (*Herman, 2001*).

*Omalium algarum* Casey, 1885
Nomenclatural Authority: *Herman (2001)*
Literature Records: none
Digitized Records: San Nicolas (1 LACM)
Range: Also known from mainland (*Frank & Ahn, 2011*).
Notes. One of the few seashore-inhabiting species of the genus, *O. algarum* occurs along much of the West Coast of North America, from southern California to British Columbia (*Frank & Ahn, 2011*).

**Oxytelinae**
Notes. Four tribes, 13 genera, and 79 species of Oxytelinae are known to occur in California (*Herman, 2001*; M. L. Gimmel, 2022, unpublished data). "Oxytelinae" was reported from Santa Cruz Island by *Straughan & Hadley (1980*: 392).

**Blediini**
Notes. One genus and 34 species of Blediini have been recorded from California (*Herman, 2001*).

*Bledius* Leach, 1819
Nomenclatural Authority: *Herman (2001)*
Literature Records (genus-only): Santa Catalina (*Straughan & Hadley, 1980*: 392)
Digitized Records (genus-only): Santa Cruz (1 SBMNH)
Notes. Thirty-four species of *Bledius* have been recorded from California (*Herman, 2001*). The species were revised in a series of papers by *Herman (1972*, *1976*, *1983*, *1986)*.

*Bledius albonotatus* Mäklin, 1853
Nomenclatural Authority: *Herman (2001)*
Literature Records: San Miguel (*Herman, 1983*: 123), San Nicolas (*Herman, 1983*: 123)
Digitized Records: San Miguel (16 SBMNH), San Nicolas (3 SBMNH), Santa Catalina (1 SBMNH), Santa Cruz (3 SBMNH), Santa Rosa (11 SBMNH)
Range: Also known from mainland (*Herman, 1983*).

*Bledius fenyesi* Bernhauer & Schubert, 1911
Nomenclatural Authority: *Herman (2001)*
Literature Records: San Miguel (*Herman, 1976*: 164), San Nicolas (*Herman, 1976*: 164), Santa Cruz (*Herman, 1976*: 164)

Digitized Records: San Clemente (7 SBMNH), San Miguel (38 SBMNH), San Nicolas (3 SBMNH), Santa Catalina (7 SBMNH), Santa Cruz (19 SBMNH), Santa Rosa (10 SBMNH)

Range: Also known from mainland (*Herman, 1976*).

### *Bledius opacifrons* LeConte, 1877

Nomenclatural Authority: *Herman (2001)*

Literature Records: none

Digitized Records: Santa Cruz (6 SBMNH), Santa Rosa (16 SBMNH)

Range: Also known from mainland (*Herman, 1976*).

### *Bledius ruficornis* LeConte, 1863

Nomenclatural Authority: *Herman (2001)*

Literature Records: San Clemente (*Herman, 1983*: 128)

Digitized Records: San Clemente (1 SBMNH), Santa Cruz (2 SBMNH), Santa Rosa (11 SBMNH)

Range: Also known from mainland (*Herman, 1983*).

### Oxytelini

Notes. Ten genera and 42 species of Oxytelini have been recorded from California (*Herman, 2001*; M. L. Gimmel, 2022, unpublished data).

### *Aploderus* Stephens, 1833

Nomenclatural Authority: *Herman (2001)*

Digitized Records (genus-only): Santa Cruz (15 SBMNH), Santa Rosa (20 SBMNH)

Notes. Seven species of *Aploderus* have been recorded from California (*Herman, 2001*).

### *Aploderus flavipennis* Casey, 1889

Nomenclatural Authority: *Herman (2001)*

Literature Records: Santa Catalina (*Fall, 1897*: 237)

Digitized Records: none

Range: Also known from mainland (*Herman, 2001*).

Notes. Reported by *Fall (1897)* as *Haploderus flavipennis*.

### *Apocellus* Erichson, 1839

Nomenclatural Authority: *Herman (2001)*

Notes. Three species of *Apocellus* have been recorded from California (*Herman, 2001*).

### *Apocellus analis* LeConte, 1877

Nomenclatural Authority: *Herman (2001)*

Literature Records: Santa Catalina (*Fall, 1897*: 237)

Digitized Records: none

Range: Also known from mainland (*Herman, 2001*).

### *Carpelimus* Leach, 1819

Nomenclatural Authority: *Herman (2001)*

Notes. Thirteen species of *Carpelimus* have been recorded from California (*Herman, 2001*).

**_Carpelimus_ undetermined species**
Literature Records: none
Digitized Records: San Clemente (1 SBMNH), San Nicolas (3 SBMNH), Santa Catalina (2 SBMNH), Santa Cruz (19 SBMNH), Santa Rosa (2 SBMNH)

**_Platystethus_ Mannerheim, 1830**
Nomenclatural Authority: *Herman (2001)*
Notes. Two species of *Platystethus* have been recorded from California (*Moore & Legner, 1971*). The North American species were reviewed by *Moore & Legner (1971)*.

**_Platystethus americanus_ Erichson, 1840**
Nomenclatural Authority: *Herman (2001)*
Literature Records: none
Digitized Records: Santa Rosa (3 SBMNH)
Range: Also known from mainland (*Moore & Legner, 1971*).

**_Thinobius_ Kiesenwetter, 1844**
Nomenclatural Authority: *Herman (2001)*
Notes. Six species of *Thinobius* have been recorded from California (*Herman, 2001*).

**_Thinobius_ undetermined species**
Literature Records: none
Digitized Records: San Nicolas (10 SBMNH), Santa Cruz (8 SBMNH), Santa Rosa (41 SBMNH)

**Paederinae: Paederini**
Notes. Two tribes, 19 genera, and 101 species of Paederinae are known to occur in California, of which 17 genera and 99 species belong to Paederini (M. L. Gimmel, 2022, unpublished data).

**_Astenus_ Dejean, 1833**
Nomenclatural Authority: *Newton et al. (2000)*
Notes. This genus contains 24 species in North America (*Newton et al., 2000*); four of these have been recorded from California (M. L. Gimmel, 2022, unpublished data).

**_Astenus_ undetermined species**
Literature Records: none
Digitized Records: San Clemente (1 SBMNH), Santa Cruz (2 SBMNH)

**_Lobrathium_ Mulsant & Rey, 1878**
Nomenclatural Authority: *Newton et al. (2000)*
Digitized Records (genus-only): Santa Cruz (35 SBMNH)

Notes. Seven species of *Lobrathium* have been recorded from California (M. L. Gimmel, 2022, unpublished data).

### *Lobrathium jacobinum* (LeConte, 1863)
Nomenclatural Authority: *Casey (1905)*
Literature Records: Santa Rosa (*Fall, 1897*: 237)
Range: Also known from mainland (*Casey, 1905*).
Notes. Reported by *Fall (1897)* as *Lathrobium jacobinum*.

### *Medon* Stephens, 1833
Nomenclatural Authority: *Newton et al. (2000)*
Notes. Twenty-eight species of *Medon* have been recorded from California (M. L. Gimmel, 2022, unpublished data). *Naughton et al. (2014)* reported two separate, unidentified species of *Medon* from Santa Cruz Island.

### *Medon* undetermined species
Literature Records: Santa Cruz (*Naughton et al., 2014*: 304)
Digitized Records: San Clemente (18 SBMNH), San Miguel (3 SBMNH), Santa Catalina (4 SBMNH), Santa Cruz (80 SBMNH), Santa Rosa (51 SBMNH)

### *Orus* Casey, 1884
Nomenclatural Authority: *Newton et al. (2000)*
Notes. Twelve species of *Orus* have been recorded from California (*Herman, 1964*, *1965*; *Moore & Legner, 1972*). The species of this genus were revised by *Herman (1964*, *1965)*, with a modification by *Moore & Legner (1972)*.

### *Orus* undetermined species
Literature Records: none
Digitized Records: Santa Cruz (1 SBMNH)
Notes. The one SBMNH specimen from Santa Cruz Island cited above appears to be a female, and therefore not determinable to species using the keys of *Herman (1964*, *1965)*.

### *Sunius* Stephens, 1829
Nomenclatural Authority: *Newton et al. (2000)*
Literature Records (genus-only): Santa Barbara (*Miller & Miller, 1985*: 124), Santa Catalina (*Fall, 1897*: 237)
Notes. Fourteen species of *Sunius* have been recorded from California (M. L. Gimmel, 2022, unpublished data). The genus-only record reported by *Fall (1897)* as "*Caloderma* sp." was presumably a species different from *S. mobilis* or *S. reductus*, which were reported in the same publication. The former record is presumably in reference to a specimen in the Museum of Comparative Zoology, Harvard University that was later identified by H.C. Fall as *Sunius exilis* (Casey, 1905) (S. Miller, 2022, personal communication). The species reported by *Miller & Miller (1985)* was said to be near *Sunius cuneicollis* (Casey, 1886); they noted that the genus needed revision.

### *Sunius mobilis* (Casey, 1886)

Nomenclatural Authority: *Casey (1905)*, *Newton et al. (2000)*

Literature Records: Santa Catalina (*Fall, 1897*: 237)

Digitized Records: none

Range: Also known from mainland (*Casey, 1905*).

Notes. Reported by *Fall (1897)* as *Caloderma mobile*.

### *Sunius reductus* (Casey, 1886)

Nomenclatural Authority: *Casey (1905)*, *Newton et al. (2000)*

Literature Records: Santa Catalina (*Fall, 1897*: 237)

Digitized Records: none

Range: Also known from mainland (*Casey, 1905*).

Notes. Reported by *Fall (1897)* as *Caloderma reductum*.

### Pselaphinae

Notes. Twelve tribes, 35 genera, and 255 species of Pselaphinae are known to occur in California (*Chandler, 1997*; M. L. Gimmel, 2022, unpublished data).

### Euplectini

Notes. Eleven genera and 58 species of Euplectini have been recorded from California (*Chandler, 1997*; M. L. Gimmel, 2022, unpublished data).

### *Actium* Casey, 1886

Nomenclatural Authority: *Chandler (1997)*

Notes. Twenty-eight species of *Actium* have been recorded from California (*Chandler, 1997*; *Caterino & Chandler, 2010*). The genus was revised by *Grigarick & Schuster (1971)*, with an addition by *Caterino & Chandler (2010)*.

### *Actium californicum* (LeConte, 1878)

Nomenclatural Authority: *Grigarick & Schuster (1971)*

Literature Records: Santa Cruz (*Grigarick & Schuster, 1971*: 26; *Caterino & Chandler, 2010*: 191)

Digitized Records: none

Range: Also known from mainland (*Grigarick & Schuster, 1971*).

Notes. *Grigarick & Schuster (1971)* reported that most specimens of this species collected on Santa Cruz Island were taken by stripping bark of fallen *Quercus agrifolia* Née (Fagaceae) lying close to small, intermittent streams during the spring.

### *Actium vestigialis* Caterino & Chandler, 2010

Nomenclatural Authority: *Caterino & Chandler (2010)*

Literature Records: Santa Catalina (*Caterino & Chandler, 2010*: 188)

Digitized Records: Santa Catalina (5 SBMNH)

Range: Endemic (*Caterino & Chandler, 2010*).

**Faronini**

Notes. Two genera and 34 species of Faronini have been recorded from California (M. L. Gimmel, 2022, unpublished data).

***Sonoma* Casey, 1886**

Nomenclatural Authority: *Chandler (1997)*

Literature Records (genus-only): San Clemente (*Caterino & Chandler, 2010*: 191), Santa Catalina (*Caterino & Chandler, 2010*: 191), Santa Cruz (*Caterino & Chandler, 2010*: 191)

Digitized Records (genus-only): San Clemente (27 SBMNH), Santa Catalina (6 SBMNH), Santa Cruz (1 SBMNH)

Notes. Twenty-eight species of *Sonoma* have been recorded from California (*Ferro, 2016*). The genus was revised for western North America by *Ferro (2016)*. *Caterino & Chandler (2010)* reported occurrence of this genus in leaf litter on San Clemente, Santa Catalina, and Santa Cruz islands. The records of the former two, and perhaps all three, islands presumably refer to *S. isabellae*.

***Sonoma isabellae* (LeConte, 1851)**

Nomenclatural Authority: *Ferro (2016)*

Literature Records: San Clemente (*Ferro, 2016*: 49), Santa Catalina (*Ferro, 2016*: 49)

Digitized Records: San Clemente (7 SBMNH), Santa Catalina (5 SBMNH)

Range: Also known from mainland (*Ferro, 2016*).

**Trogastrini**

Notes. Three genera and 37 species of Trogastrini have been recorded from California (*Chandler, 1997*; M. L. Gimmel, 2022, unpublished data).

***Oropus* Casey, 1886**

Nomenclatural Authority: *Chandler (1997)*

Notes. Twenty-seven species of *Oropus* have been recorded from California (*Chandler, 1997*). The genus was revised by *Schuster & Grigarick (1960)*.

***Oropus* undetermined species**

Literature Records: Santa Catalina (*Caterino & Chandler, 2010*: 191), Santa Cruz (*Caterino & Chandler, 2010*: 191)

Digitized Records: Santa Cruz (2 SBMNH)

Notes. *Caterino & Chandler (2010)* reported occurrence of the genus *Oropus* in leaf litter on Santa Catalina and Santa Cruz islands.

**Tychini**

Notes. Two genera and 22 species of Tychini have been recorded from California (*Chandler, 1997*; M. L. Gimmel, 2022, unpublished data).

***Hesperotychus* Schuster & Marsh, 1958**

Nomenclatural Authority: *Chandler (1997)*

Notes. Thirteen species of *Hesperotychus* have been recorded from California (M. L. Gimmel, 2022, unpublished data). The species were revised by *Schuster & Marsh (1958)*.

### *Hesperotychus* undetermined species

Literature Records: Santa Catalina (*Caterino & Chandler, 2010*: 191)

Digitized Records: Santa Catalina (1 SBMNH)

Notes. *Caterino & Chandler (2010)* reported occurrence of this genus in leaf litter on Santa Catalina Island.

### Pseudopsinae

Notes. This subfamily is represented by four genera and eight species in California (*Herman, 2001*). So far only *Pseudopsis* has been identified from the Channel Islands, though *Nanobius serricollis* (LeConte, 1875), which occurs in the southern California coast ranges (*Herman, 1977*), may also occur there.

### *Pseudopsis* Newman, 1834

Nomenclatural Authority: *Herman (2001)*

Literature Records (genus-only): Santa Catalina (*Fall, 1897*: 237)

Notes. Four species of *Pseudopsis* have been recorded from California (*Herman, 2001*). *Fall (1897*: 237) reported an undetermined species of this genus from Santa Catalina Island. This may not represent *P. minuta*, as he later (*Fall, 1901*: 227) described that species only from "the cañons of the southern Sierras". This could represent either an undescribed species or the later-described *P. montoraria* Herman, 1975, which occurs in the coastal mountain ranges of southern California and has not yet been identified from the Channel Islands. *Herman (1975)* revised the genus.

### *Pseudopsis minuta* Fall, 1901

Nomenclatural Authority: *Herman (1975*, *2001*)

Literature Records: Santa Cruz (*Naughton et al., 2014*: 304)

Digitized Records: Santa Cruz (15 SBMNH)

Range: Also known from mainland (*Herman, 1975*).

Notes. *Naughton et al. (2014*: 304) identified a single specimen only to "*Pseudopsis* sp.". The voucher in SBMNH was examined and this represents *P. minuta*.

### Scydmaeninae

Notes. Five tribes, 13 genera, and 57 species of Scydmaeninae are known to occur in California (M. L. Gimmel, 2022, unpublished data).

### Cephenniini

Notes. Two genera and nine species of Cephenniini are known to occur in California (*Hopp & Caterino, 2009*; M. L. Gimmel, 2022, unpublished data).

### *Cephennium* Müller & Kunze, 1822

Nomenclatural Authority: *O'Keefe (2000)*

Notes. Eight species of *Cephennium* have been recorded from California (*Hopp & Caterino, 2009*). The Californian species were revised by *Hopp & Caterino (2009)*.

### *Cephennium urbanum* Hopp & Caterino, 2009

Nomenclatural Authority: *Hopp & Caterino (2009)*

Literature Records: Santa Catalina (*Caterino & Chandler, 2010*: 191)
Digitized Records: Santa Catalina (9 SBMNH)
Range: Also known from mainland (*Hopp & Caterino, 2009*).
Notes. Only the genus *Cephennium* was reported from Santa Catalina Island by *Caterino & Chandler (2010)*; that report presumably referred to this species.

**Glandulariini**
Notes. Seven genera and 41 species of Glandulariini have been recorded from California (M. L. Gimmel, 2022, unpublished data).

*Brachycepsis* **Brendel, 1889**
Nomenclatural Authority: *O'Keefe (2000)*
Notes. Two species of *Brachycepsis* have been recorded from California (M. L. Gimmel, 2022, unpublished data). Species of this genus are currently unidentifiable, as there are several undescribed species and the last treatment is over 120 years old (see *O'Keefe, 2000*).

*Brachycepsis* **undetermined species**
Literature Records: Santa Catalina (*Caterino & Chandler, 2010*: 191), Santa Cruz (*Naughton et al., 2014*: 304)
Digitized Records: Santa Cruz (4 SBMNH), Santa Rosa (4 SBMNH)

*Euconnus* **Thomson, 1862**
Nomenclatural Authority: *O'Keefe (2000)*
Notes. Seven species of *Euconnus* have been recorded from California, belonging to two subgenera, *Drastophus* Casey, 1897 and *Napochus* Thomson, 1862 (M. L. Gimmel, 2022, unpublished data).

*Euconnus* **undetermined species**
Literature Records: none
Digitized Records: San Clemente (1 SBMNH), Santa Catalina (3 SBMNH), Santa Rosa (24 SBMNH)
Notes. All members of *Euconnus* observed from the Channel Islands belong to the subgenus *Drastophus*. Based on morphology, there are at least two species represented among this material, and probably more.

*Stenichnus* **Thomson, 1859**
Nomenclatural Authority: *O'Keefe (2000)*
Notes. There are several described species of this genus in North America, and the latest treatment is over 120 years old (see *O'Keefe, 2000*). Undescribed species probably exist.

*Stenichnus* **undetermined species**
Literature Records: Santa Catalina (*Caterino & Chandler, 2010*: 191)
Digitized Records: Santa Catalina (3 SBMNH)

**Silphinae**

Notes. Long known as the family Silphidae, this group was recently made a subfamily of Staphylinidae (*Cai et al., 2022*). There are two tribes, four genera, and nine species of Silphinae recorded from California (*Miller & Peck, 1979*; *Peck & Miller, 1993*). *Miller & Peck (1979)* provided a guide to the group for California, while *Peck & Miller (1993)* provided a catalog for North America. Members of this subfamily have been extensively surveyed and investigated for the Channel Islands, and it is doubtful that additional species will be discovered there.

**Nicrophorini**

Notes. One genus and four species of Nicrophorini have been recorded from California (*Peck & Miller, 1993*).

***Nicrophorus* Fabricius, 1775**

Nomenclatural Authority: *Sikes, Madge & Newton (2002)*

Digitized Records (genus-only): Santa Cruz (2 EMEC)

Notes. Four species of *Nicrophorus* have been recorded from California (*Peck & Miller, 1993*).

***Nicrophorus guttula* Motschulsky, 1845**

Nomenclatural Authority: *Sikes, Madge & Newton (2002)*

Literature Records: San Clemente (*Fall, 1897*: 236; *Fall, 1901*: 58; *Miller & Peck, 1979*: 97; *Anderson & Peck, 1986*: 296; *Peck & Kaulbars, 1988*: 72 [map])

Digitized Records: San Clemente (1 LACM), Santa Catalina (2 LACM), Santa Rosa (2 SBMNH)

Range: Also known from mainland (*Fall, 1901*; *Miller & Peck, 1979*; *Anderson & Peck, 1986*; *Peck & Kaulbars, 1988*; *Peck & Miller, 1993*; *Sikes, Madge & Newton, 2002*).

***Nicrophorus marginatus* Fabricius, 1801**

Nomenclatural Authority: *Sikes, Madge & Newton (2002)*

Literature Records: none

Digitized Records: San Miguel (1 LACM)

Range: Also known from mainland (*Fall, 1901*; *Miller & Peck, 1979*; *Anderson & Peck, 1986*; *Peck & Kaulbars, 1988*; *Peck & Miller, 1993*; *Sikes, Madge & Newton, 2002*).

Notes. While this is the most widespread species of *Nicrophorus* in North America, only one specimen has been reported from the Channel Islands, collected in 1985.

***Nicrophorus nigrita* Mannerheim, 1843**

Nomenclatural Authority: *Sikes, Madge & Newton (2002)*

Literature Records: Anacapa (*Miller & Peck, 1979*: 96; *Miller & Miller, 1985*: 124), San Clemente (*Fall, 1897*: 236; *Miller & Peck, 1979*: 96; *Miller & Miller, 1985*: 124; *Peck & Kaulbars, 1988*: 69 [map]), Santa Barbara (*Miller & Peck, 1979*: 96; *Anderson, 1982*: 262; *Miller & Miller, 1985*: 124), Santa Catalina (*Miller & Miller, 1985*: 124), Santa Cruz (*Miller & Peck, 1979*: 96; *Miller & Miller, 1985*: 124; *Peck & Kaulbars, 1988*: 69 [map]), Santa Rosa (*Fall, 1897*: 236; *Miller & Miller, 1985*: 124; *Sikes, Madge & Newton, 2002*: 139)

Digitized Records: Santa Catalina (1 DMNS; 6 LACM; 1 SBMNH), Santa Cruz (4 LACM; 15 SBMNH), Santa Rosa (2 SBMNH)

Range: Also known from mainland (*Fall, 1901*; *Miller & Peck, 1979*; *Peck & Kaulbars, 1988*; *Peck & Miller, 1993*; *Sikes, Madge & Newton, 2002*).

Notes. *Fall (1901)* recorded this species as *Necrophorus pustulatus* var. *nigritus*, and reported it from "both islands" (presumably San Clemente and Santa Rosa). *Miller & Miller (1985)* recorded it "under dead mice and at lanterns" on Santa Barbara Island.

**Silphini**

Notes. Three genera and five species of Silphini have been recorded from California (*Miller & Peck, 1979*; *Peck & Miller, 1993*).

***Heterosilpha* Portevin, 1926**

Nomenclatural Authority: *Peck & Miller (1993)*

Notes. Two species of *Heterosilpha* have been recorded from California (*Peck & Miller, 1993*).

***Heterosilpha ramosa* (Say, 1823)**

Nomenclatural Authority: *Peck & Miller (1993)*

Literature Records: San Miguel (*Miller & Peck, 1979*: 93), Santa Cruz (*Fall & Davis, 1934*: 144; *Miller & Peck, 1979*: 93; *Peck & Kaulbars, 1988*: 57 [map]), Santa Rosa (*Fall, 1897*: 236; *Miller & Peck, 1979*: 93)

Digitized Records: San Clemente (1 UCRC), Santa Cruz (2 LACM; 11 SBMNH; 23 TAMU; 2 UCRC), Santa Rosa (24 LACM; 11 SBMNH)

Range: Also known from mainland (*Miller & Peck, 1979*; *Peck & Kaulbars, 1988*; *Peck & Miller, 1993*).

Notes. *Fall (1897)* and *Fall & Davis (1934)* recorded this species as *Silpha ramosa*.

***Thanatophilus* Leach, 1815**

Nomenclatural Authority: *Peck & Miller (1993)*

Notes. Two species of *Thanatophilus* have been recorded from California (*Miller & Peck, 1979*).

***Thanatophilus lapponicus* (Herbst, 1793)**

Nomenclatural Authority: *Peck & Miller (1993)*

Literature Records: Santa Rosa (*Fall, 1897*: 236)

Digitized Records: none

Range: Also known from mainland (*Miller & Peck, 1979*; *Peck & Kaulbars, 1988*; *Peck & Miller, 1993*).

Notes. *Fall (1897)* recorded this species as *Silpha lapponica*. *Miller & Peck (1979*: 91) stated that they had not seen Channel Island specimens of this species, and speculated that Fall's specimens may have been destroyed in the 1906 San Francisco fire.

### Staphylininae

Notes. Three tribes, 35 genera, and 232 species of Staphylininae are known to occur in California (*Herman, 2001*; M. L. Gimmel, 2022, unpublished data).

### Staphylinini: Amblyopinina

Notes. Seven subtribes, 20 genera, and 171 species of Staphylinini are known to occur in California, of which one genus and 10 species belong to Amblyopinina (*Herman, 2001*; M. L. Gimmel, 2022, unpublished data).

### *Heterothops* Stephens, 1829

Nomenclatural Authority: *Herman (2001)*

Digitized Records (genus-only): Santa Catalina (1 SBMNH), Santa Cruz (1 SBMNH), Santa Rosa (19 SBMNH)

Notes. Ten species of *Heterothops* have been recorded from California (*Herman, 2001*). The species were revised for North America by *Smetana (1971)*.

### *Heterothops conformis* Smetana, 1971

Nomenclatural Authority: *Herman (2001)*

Literature Records: Santa Cruz (*Naughton et al., 2014*: 304)

Digitized Records: Santa Catalina (1 SBMNH), Santa Cruz (43 SBMNH), Santa Rosa (2 SBMNH)

Range: Also known from mainland (*Smetana, 1971*).

### *Heterothops fusculus* LeConte, 1863

Nomenclatural Authority: *Herman (2001)*

Literature Records: Santa Catalina (*Fall, 1897*: 236)

Digitized Records: none

Range: Also known from mainland (*Smetana, 1971*).

Notes. This species was recorded by *Fall (1897)* as *Heterothops californicus* LeConte, 1863, now considered a synonym of *H. californicus* (see *Smetana, 1971*: 26).

### Staphylinini: Erichsoniina

Notes. One genus and two species of Erichsoniina have been recorded from California (*Herman, 2001*).

### *Erichsonius* Fauvel, 1874

Nomenclatural Authority: *Herman (2001)*

Notes. Two species of *Erichsonius* have been recorded from California (*Herman, 2001*). The species were revised for the New World by *Frank (1975)*.

### *Erichsonius puncticeps* (Horn, 1884)

Nomenclatural Authority: *Herman (2001)*

Literature Records: Santa Catalina (*Fall, 1897*: 236)

Digitized Records: San Miguel (1 SBMNH), Santa Cruz (3 SBMNH), Santa Rosa (9 SBMNH)

Range: Also known from mainland (*Frank, 1975*).
Notes. *Fall (1897)* recorded this species as *Actobius puncticeps.*

**Staphylinini: Philonthina**
Notes. Six genera and 100 species of Philonthina are known to occur in California (*Herman, 2001*; M. L. Gimmel, 2022, unpublished data).

*Belonuchus* **Nordmann, 1837**
Nomenclatural Authority: *Herman (2001)*
Notes. Three species of *Belonuchus* have been recorded from California (*Herman, 2001*). The species were revised for North America by *Smetana (1995)*.

*Belonuchus ephippiatus* **(Say, 1830)**
Nomenclatural Authority: *Herman (2001)*
Literature Records: none
Digitized Records: San Miguel (2 SBMNH), Santa Catalina (3 SBMNH)
Range: Also known from mainland (*Smetana, 1995*).

*Bisnius* **Stephens, 1829**
Nomenclatural Authority: *Herman (2001)*
Notes. Thirteen species of *Bisnius* have been recorded from California (*Herman, 2001*). The species were revised for North America by *Smetana (1995)*.

*Bisnius albionicus* **(Mannerheim, 1843)**
Nomenclatural Authority: *Herman (2001)*
Literature Records: none
Digitized Records: San Miguel (7 SBMNH), Santa Rosa (1 SBMNH)
Range: Also known from mainland (*Smetana, 1995*).

*Bisnius sordidus* **(Gravenhorst, 1802)**
Nomenclatural Authority: *Herman (2001)*
Literature Records: none
Digitized Records: San Nicolas (1 SBMNH), Santa Rosa (1 SBMNH)
Range: Also known from mainland (*Smetana, 1995*).
Notes. The distribution map in *Smetana (1995*: 526) shows a record from either San Miguel or Santa Rosa. This species was introduced to North America from the Palearctic realm (*Smetana, 1995*).

*Cafius* **Curtis, 1829**
Nomenclatural Authority: *Herman (2001)*
Notes. Eight species of *Cafius* have been recorded from California (*Herman, 2001*). The species were revised for the west coast of North America by *Orth & Moore (1980)*.

*Cafius canescens* **(Mäklin, 1852)**
Nomenclatural Authority: *Herman (2001)*

Literature Records: San Nicolas (*Fall, 1897*: 236; *Cockerell, 1940*: 285), Santa Catalina (*Orth & Moore, 1980*: 186)

Digitized Records: San Miguel (34 LACM; 46 SBMNH), San Nicolas (76 LACM; 2 SBMNH), Santa Cruz (7 SBMNH), Santa Rosa (7 LACM; 15 SBMNH; 2 UTCI)

Range: Also known from mainland (*Orth & Moore, 1980*).

### *Cafius lithocharinus* (LeConte, 1863)

Nomenclatural Authority: *Herman (2001)*

Literature Records: San Nicolas (*Straughan & Hadley, 1980*: 392), Santa Rosa (*Fall, 1897*: 236)

Digitized Records: San Clemente (9 SBMNH), San Miguel (1 SBMNH), San Nicolas (5 SBMNH), Santa Catalina (3 SBMNH), Santa Cruz (4 SBMNH), Santa Rosa (47 SBMNH)

Range: Also known from mainland (*Orth & Moore, 1980*).

### *Cafius luteipennis* Horn, 1884

Nomenclatural Authority: *Herman (2001)*

Literature Records: Santa Catalina (*Orth & Moore, 1980*: 192), Santa Rosa (*Fall, 1897*: 236)

Digitized Records: San Clemente (7 SBMNH), San Miguel (6 SBMNH), San Nicolas (5 SBMNH), Santa Catalina (5 SBMNH), Santa Cruz (14 SBMNH), Santa Rosa (11 SBMNH)

Range: Also known from mainland (*Orth & Moore, 1980*).

### *Cafius opacus* (LeConte, 1863)

Nomenclatural Authority: *Herman (2001)*

Literature Records: Santa Catalina (*Fall, 1897*: 237)

Digitized Records: none

Range: Also known from mainland (*Orth & Moore, 1980*).

### *Cafius seminitens* Horn, 1884

Nomenclatural Authority: *Herman (2001)*

Literature Records: San Miguel (*Cockerell, 1940*: 285; *Orth & Moore, 1980*: 185), San Nicolas (*Cockerell, 1940*: 285)

Digitized Records: San Clemente (6 SBMNH), San Miguel (10 LACM; 21 SBMNH), San Nicolas (64 LACM; 9 SBMNH), Santa Cruz (1 LACM; 3 SBMNH), Santa Rosa (7 LACM; 12 SBMNH; 2 UTCI)

Range: Also known from mainland (*Orth & Moore, 1980*).

### *Cafius sulcicollis* (LeConte, 1863)

Nomenclatural Authority: *Herman (2001)*

Literature Records: Santa Cruz (*Orth & Moore, 1980*: 195), Santa Rosa (*Fall, 1897*: 237)

Digitized Records: San Clemente (2 SBMNH), Santa Rosa (6 SBMNH)

Range: Also known from mainland (*Orth & Moore, 1980*).

***Gabrius*** **Stephens, 1829**

Nomenclatural Authority: *Herman (2001)*

Digitized Records (genus-only): San Nicolas (1 SBMNH), Santa Cruz (1 SBMNH)

Notes. Sixteen species of *Gabrius* have been recorded from California (*Herman, 2001*). The genus was revised for North America by *Smetana (1995).*

***Gabrius nigritulus*** **(Gravenhorst, 1802)**

Nomenclatural Authority: *Herman (2001)*

Literature Records: Santa Catalina (*Fall, 1897*: 236)

Digitized Records: San Nicolas (4 SBMNH)

Range: Also known from mainland (*Smetana, 1995*).

Notes. This species was recorded as *Philonthus nigritulus* by *Fall (1897)*. It was introduced to North America from the western Palearctic realm (*Smetana, 1995*).

***Neobisnius*** **Ganglbauer, 1895**

Nomenclatural Authority: *Herman (2001)*

Notes. Eight species of *Neobisnius* have been recorded from California (*Herman, 2001*). The species were revised for the New World by *Frank (1981).*

***Neobisnius occidentoides*** **Frank, 1981**

Nomenclatural Authority: *Herman (2001)*

Literature Records: San Clemente (*Frank, 1981*: 49)

Digitized Records: San Clemente (7 SBMNH; 4 UTCI), Santa Catalina (2 SBMNH), Santa Cruz (1 SBMNH), Santa Rosa (1 SBMNH)

Range: Also known from mainland (*Frank, 1981*).

***Neobisnius sobrinus*** **(Erichson, 1840)**

Nomenclatural Authority: *Herman (2001)*

Literature Records: none

Digitized Records: Santa Rosa (1 SBMNH)

Range: Also known from mainland (*Frank, 1981*).

***Neobisnius terminalis*** **(LeConte, 1863)**

Nomenclatural Authority: *Herman (2001)*

Literature Records: none

Digitized Records: Santa Cruz (3 SBMNH)

Range: Also known from mainland (*Frank, 1981*).

***Philonthus*** **Stephens, 1829**

Nomenclatural Authority: *Herman (2001)*

Notes. Fifty-two species of *Philonthus* have been recorded from California (M. L. Gimmel, 2022, unpublished data). The species were revised for North America by *Smetana (1995).*

***Philonthus cruentatus*** **(Gmelin, 1790)**

Nomenclatural Authority: *Herman (2001)*

Literature Records: Santa Catalina (*Cockerell, 1940*: 285)

Digitized Records: Santa Barbara (1 LACM), Santa Catalina (1 SBMNH), Santa Cruz (1 SBMNH), Santa Rosa (26 LACM; 1 SBMNH)

Range: Also known from mainland (*Smetana, 1995*).

Notes. Species introduced from the Palearctic (*Smetana, 1995*). *Cockerell (1940)* recorded it as "*Philonthus cruentus*".

### *Philonthus davus* Smetana, 1995

Nomenclatural Authority: *Herman (2001)*

Literature Records: none

Digitized Records: Santa Cruz (3 SBMNH), Santa Rosa (3 SBMNH)

Range: Also known from mainland (*Smetana, 1995*).

### *Philonthus flavolimbatus* Erichson, 1840

Nomenclatural Authority: *Herman (2001)*

Literature Records: none

Digitized Records: Santa Catalina (1 SBMNH)

Range: Also known from mainland (*Smetana, 1995*).

### *Philonthus hepaticus* Erichson, 1840

Nomenclatural Authority: *Herman (2001)*

Literature Records: none

Digitized Records: Santa Catalina (2 SBMNH)

Range: Also known from mainland (*Smetana, 1995*).

### *Philonthus lecontei* Horn, 1884

Nomenclatural Authority: *Herman (2001)*

Literature Records: Santa Rosa (*Fall, 1897*: 236)

Digitized Records: none

Range: Also known from mainland (*Smetana, 1995*).

### *Philonthus longicornis* Stephens, 1832

Nomenclatural Authority: *Herman (2001)*

Literature Records: Santa Catalina (*Fall, 1897*: 236)

Digitized Records: none

Range: Also known from mainland (*Smetana, 1995*).

Notes. This species was introduced from the Palearctic region (*Smetana, 1995*).

### *Philonthus quadrulus* Horn, 1884

Nomenclatural Authority: *Herman (2001)*

Literature Records: none

Digitized Records: Santa Cruz (10 SBMNH)

Range: Also known from mainland (*Smetana, 1995*).

### *Philonthus triangulum* Horn, 1884

Nomenclatural Authority: *Herman (2001)*

Literature Records: Santa Catalina (*Smetana, 1995*: 340)

Digitized Records: none
Range: Also known from mainland (*Smetana, 1995*).

**Staphylinini: Quediina**
Notes. Two genera and 40 species of Quediina have been recorded from California (*Herman, 2001*; M. L. Gimmel, 2022, unpublished data).

*Quedius* **Stephens, 1829**
Nomenclatural Authority: *Herman (2001)*
Literature Records (genus-only): Santa Cruz (*Naughton et al., 2014*: 304)
Digitized Records (genus-only): Santa Cruz (1 SBMNH)
Notes. Thirty-nine species of *Quedius* have been recorded from California, belonging to five subgenera, *Distichalius* Casey, 1915, *Microsaurus* Dejean, 1833, *Paraquedius* Casey, 1915, *Quedius* (*s.str.*), and *Raphirus* Stephens, 1829 (M. L. Gimmel, 2022, unpublished data). The species were revised for North America by *Smetana (1971)*. One of the two voucher specimens (SBMNH, June specimen) of the "*Quedius* sp." from the *Naughton et al. (2014)* study was examined by MLG; this specimen represents the digitized genus-only record above. It is a female of either *Q. limbifer* or *Q.* (*Microsaurus*) *pellax* Smetana, 1971.

*Quedius* (*Microsaurus*) *limbifer* **Horn, 1878**
Nomenclatural Authority: *Smetana (1971)*, *Herman (2001)*
Literature Records: Santa Cruz (*Smetana, 1971*: 113)
Digitized Records: Santa Cruz (10 SBMNH), Santa Rosa (1 SBMNH)
Range: Also known from mainland (*Smetana, 1971*).

**Staphylinini: Staphylinina**
Notes. Eight genera and 14 species of Staphylinina are known to occur in California (*Herman, 2001*; M. L. Gimmel, 2022, unpublished data).

*Creophilus* **Leach, 1819**
Nomenclatural Authority: *Herman (2001)*
Notes. One species of *Creophilus* has been recorded from California (*Herman, 2001*). The species were revised for the world by *Clarke (2011)*.

*Creophilus maxillosus* **(Linnaeus, 1758)**
Nomenclatural Authority: *Clarke (2011)*
Literature Records: San Clemente (*Fall, 1897*: 236; *Fall, 1901*: 68), Santa Catalina (*Fall, 1897*: 236)
Digitized Records: San Clemente (1 LACM; 1 SBMNH), Santa Cruz (8 SBMNH), Santa Rosa (1 LACM; 6 SBMNH)
Range: Also known from mainland (*Fall, 1901*; *Clarke, 2011*).
Notes. This species was recorded as *C. villosus* by *Fall (1897*, *1901)*. *Creophilus m. villosus* (Gravenhorst, 1802) is the only subspecies of *C. maxillosus* (Linnaeus, 1758) over most of North America (*Clarke, 2011*); this is the subspecies occurring in the islands. The

distribution map in *Clarke (2011*: 765) shows records in the Channel Islands, but these are not listed.

### *Hadrotes* Mäklin, 1852

Nomenclatural Authority: *Herman (2001)*

Notes. One species of *Hadrotes* has been recorded from California (*Herman, 2001*).

### *Hadrotes crassus* (Mannerheim, 1846)

Nomenclatural Authority: *Herman (2001)*

Literature Records: San Clemente (*Caterino, Chatzimanolis & Richmond, 2015*: 278), San Nicolas (*Cockerell, 1940*: 285; *Caterino, Chatzimanolis & Richmond, 2015*: 278), Santa Catalina (*Caterino, Chatzimanolis & Richmond, 2015*: 278), Santa Cruz (*Caterino, Chatzimanolis & Richmond, 2015*: 278), Santa Rosa (*Fall, 1897*: 236; *Caterino, Chatzimanolis & Richmond, 2015*: 278)

Digitized Records: Anacapa (1 SBMNH), San Clemente (11 SBMNH), San Miguel (1 LACM; 8 SBMNH), San Nicolas (13 SBMNH), Santa Catalina (12 SBMNH), Santa Cruz (13 SBMNH), Santa Rosa (11 SBMNH; 2 UTCI)

Range: Also known from mainland (*Caterino, Chatzimanolis & Richmond, 2015*).

### *Tasgius* Stephens, 1829

Nomenclatural Authority: *Herman (2001)*

Notes. Two adventive species of *Tasgius* have been recorded from California (*Miller & Miller, 1985*; *Herman, 2001*).

### *Tasgius ater* (Gravenhorst, 1802)

Nomenclatural Authority: *Herman (2001)*

Literature Records: San Miguel (*Miller & Miller, 1985*: 124), Santa Barbara (*Miller & Miller, 1985*: 124)

Digitized Records: San Miguel (2 SBMNH), San Nicolas (1 SBMNH), Santa Rosa (6 SBMNH)

Range: Also known from mainland (*Miller & Miller, 1985*).

Notes. Introduced to North America from Europe (*Miller & Miller, 1985*). Recorded as *Staphylinus ater* (Gravenhorst) from *Suaeda* (Amaranthaceae) by *Miller & Miller (1985)*.

### *Thinopinus* LeConte, 1852

Nomenclatural Authority: *Herman (2001)*

Notes. Only one species of *Thinopinus* is known (*Herman, 2001*).

### *Thinopinus pictus* LeConte, 1852

Nomenclatural Authority: *Herman (2001)*

Literature Records: San Miguel (*Caterino, Chatzimanolis & Richmond, 2015*: 278), San Nicolas (*Caterino, Chatzimanolis & Richmond, 2015*: 278), Santa Catalina (*Caterino, Chatzimanolis & Richmond, 2015*: 278), Santa Cruz (*Caterino, Chatzimanolis & Richmond, 2015*: 278), Santa Rosa (*Caterino, Chatzimanolis & Richmond, 2015*: 278)

Digitized Records: San Miguel (12 SBMNH), San Nicolas (1 LACM; 13 SBMNH), Santa Catalina (1 SEMC; 11 SBMNH; 1 iNat), Santa Cruz (10 SBMNH), Santa Rosa (10 SBMNH; 2 UTCI; 1 iNat)

Range: Also known from mainland (*Caterino, Chatzimanolis & Richmond, 2015*).

**Xantholinini**

Notes. Fourteen genera and 55 species of Xantholinini are known to occur in California (*Herman, 2001*). The species of the tribe were revised for North America by *Smetana (1982)*. A specimen deposited in LACM of an additional genus and species occurring on Santa Rosa Island, *Neohypnus picipennis* (LeConte, 1880), was captured by Scott Miller during the 1980s, but no specimen was located to substantiate this record (G-A. Kung, 2022, personal communication).

***Linohesperus* Smetana, 1982**

Nomenclatural Authority: *Herman (2001)*

Digitized Records (genus-only): San Clemente (2 SBMNH), Santa Catalina (16 SBMNH), Santa Cruz (3 SBMNH)

Notes. Twenty-two species of *Linohesperus* have been recorded from California (*Herman, 2001*).

***Linohesperus borealis* (Casey, 1906)**

Nomenclatural Authority: *Herman (2001)*

Literature Records: Santa Cruz (*Naughton et al., 2014*: 304)

Digitized Records: Santa Rosa (1 SBMNH)

Range: Also known from mainland (*Smetana, 1982*).

***Linohesperus cuspifer* Smetana, 1982**

Nomenclatural Authority: *Herman (2001)*

Literature Records: Santa Cruz (*Smetana, 1988*: 545)

Digitized Records: none

Range: Also known from mainland (*Smetana, 1982*, *1988*).

***Nudobius* Thomson, 1860**

Nomenclatural Authority: *Herman (2001)*

Notes. Only one species of *Nudobius* has been recorded from California (*Herman, 2001*).

***Nudobius pugetanus* Casey, 1906**

Nomenclatural Authority: *Smetana (1982)*, *Herman (2001)*

Literature Records: none

Digitized Records: Santa Cruz (1 SBMNH)

Range: Also known from mainland (*Smetana, 1982*).

**Tachyporinae**

Notes. Three tribes, eight genera, and 32 species of Tachyporinae are known to occur in California (M. L. Gimmel, 2022, unpublished data). This subfamily was recently restricted

(*i.e.*, the previous tribe "Mycetoporini" excluded as a separate subfamily) by *Yamamoto (2021)*.

**Tachinusini**
Notes. Three genera and 15 species of Tachinusini are known to occur in California (M. L. Gimmel, 2022, unpublished data).

*Nitidotachinus* **Campbell, 1993**
Nomenclatural Authority: *Herman (2001)*
Notes. Three species of *Nitidotachinus* have been recorded from California (*Campbell, 1993a*). This genus was reviewed by *Campbell (1993a)*, who provided a key to species.

*Nitidotachinus agilis* **(Horn, 1877)**
Nomenclatural Authority: *Herman (2001)*
Literature Records: none
Digitized Records: Santa Cruz (1 SBMNH), Santa Rosa (1 SBMNH)
Range: Also known from mainland (*Campbell, 1993a*).

*Tachinus* **Gravenhorst, 1802**
Nomenclatural Authority: *Herman (2001)*
Notes. Ten species of *Tachinus* have been recorded from California (M. L. Gimmel, 2022, unpublished data). Additional species of *Tachinus* are likely to occur on the Channel Islands. The genus was revised for North America by *Campbell (1973)* and updated by *Campbell (1988)*.

*Tachinus debilis* **Horn, 1877**
Nomenclatural Authority: *Herman (2001)*
Literature Records: none
Digitized Records: Santa Cruz (9 SBMNH)
Range: Also known from mainland (*Campbell, 1973*).

**Tachyporini**
Notes. Three genera and 11 species of Tachyporini have been recorded from California (M. L. Gimmel, 2022, unpublished data).

*Palporus* **Campbell, 1979**
Nomenclatural Authority: *Yamamoto (2021)*
Notes. Only one species of *Palporus* occurs in California (*Campbell, 1979*). This genus was revised for North America by *Campbell (1979)*, as *Tachyporus* (*Palporus*); the subgenus was subsequently elevated to genus by *Yamamoto (2021)*.

*Palporus nitidulus* **(Fabricius, 1781)**
Nomenclatural Authority: *Yamamoto (2021)*
Literature Records: none
Digitized Records: San Nicolas (1 SBMNH), Santa Cruz (1 SBMNH)
Range: Also known from mainland (*Campbell, 1979*).

***Sepedophilus* Gistel, 1856**
Nomenclatural Authority: *Herman (2001)*
Notes. Three species of *Sepedophilus* have been recorded from California (*Campbell, 1976*). The genus was revised for North America by *Campbell (1976)*.

***Sepedophilus castaneus* (Horn, 1877)**
Nomenclatural Authority: *Herman (2001)*
Literature Records: none
Digitized Records: Santa Cruz (25 SBMNH), Santa Rosa (14 SBMNH)
Range: Also known from mainland (*Campbell, 1976*).

***Tachyporus* Gravenhorst, 1802**
Nomenclatural Authority: *Herman (2001)*
Literature Records (genus-only): San Miguel (*Miller & Davis, 1986*: 550)
Notes. Seven species of *Tachyporus* have been recorded from California (*Campbell, 1979*). Additional species of *Tachyporus* are likely to occur on the Channel Islands. This genus was revised for North America by *Campbell (1979)*.

***Tachyporus californicus* Horn, 1877**
Nomenclatural Authority: *Herman (2001)*
Literature Records: Santa Catalina (*Fall, 1897*: 237), Santa Rosa (*Fall, 1897*: 237)
Digitized Records: San Clemente (1 SBMNH), San Nicolas (4 SBMNH), Santa Cruz (17 SBMNH), Santa Rosa (1 SBMNH)
Range: Also known from mainland (*Campbell, 1979*).

**BOSTRICHOIDEA**

**Bostrichidae**
Notes. Five subfamilies, 19 genera, and 35 species of Bostrichidae are known to occur in California (M. L. Gimmel, 2022, unpublished data). The works of *Fisher (1950*; all subfamilies except Lyctinae) and *Gerberg (1957*; Lyctinae) adequately cover the North American fauna known at the time. *Borowski & Węgrzynowicz (2007)* provided a world catalog of this group; we follow their classification below.

**Bostrichinae**
Notes. Four tribes, 10 genera, and 15 species of Bostrichinae are known to occur in California (M. L. Gimmel, 2022, unpublished data).

**Bostrichini**
Notes. Four genera and seven species of Bostrichini are known to occur in California (M. L. Gimmel, 2022, unpublished data).

***Amphicerus* LeConte, 1861**
Nomenclatural Authority: *Borowski & Węgrzynowicz (2007)*
Notes. Three species of *Amphicerus* are known to occur in California (M. L. Gimmel, 2022, unpublished data). The species were keyed by *Fisher (1950)*.

*Amphicerus cornutus* (Pallas, 1772)

Nomenclatural Authority: *Fisher (1950)*, *Borowski & Węgrzynowicz (2007)*

Literature Records: none

Digitized Records: Santa Catalina (4 LACM), Santa Cruz (9 SBMNH; 1 UCSB)

Range: Also known from mainland (*Fisher, 1950*; *Borowski & Węgrzynowicz, 2007*).

**Xyloperthini**

Notes. Four genera and six species of Xyloperthini have been recorded from California (*Fisher, 1950*).

*Scobicia* Lesne, 1901

Nomenclatural Authority: *Borowski & Węgrzynowicz (2007)*

Notes. Two species of *Scobicia* are known to occur in California (*Fisher, 1950*). These species were keyed by *Fisher (1950)*.

*Scobicia declivis* (LeConte, 1857)

Nomenclatural Authority: *Fisher (1950)*, *Borowski & Węgrzynowicz (2007)*

Literature Records: San Nicolas (*Fall, 1897*: 238; *Fall, 1901*: 133; *Burke, Hartman & Snyder, 1922*: 12; *Fisher, 1950*: 111)

Digitized Records: Santa Cruz (4 SBMNH)

Range: Also known from mainland (*Fall, 1901*; *Fisher, 1950*; *Borowski & Węgrzynowicz, 2007*).

Notes. *Fall (1897*, *1901)* recorded this species as *Sinoxylon declive*. *Borowski & Węgrzynowicz (2007)* falsely stated the year of publication of the species as 1859.

*Scobicia suturalis* (Horn, 1878)

Nomenclatural Authority: *Fisher (1950)*, *Borowski & Węgrzynowicz (2007)*

Literature Records: none

Digitized Records: Anacapa (1 SBMNH), Santa Catalina (1 SBMNH), Santa Cruz (1 SBMNH)

Range: Also known from mainland (*Fisher, 1950*; *Borowski & Węgrzynowicz, 2007*).

**Dinoderinae**

Notes. Four genera and eight species of Dinoderinae are known to occur in California (M. L. Gimmel, 2022, unpublished data).

*Stephanopachys* Waterhouse, 1888

Nomenclatural Authority: *Borowski & Węgrzynowicz (2007)*

Notes. Four species of *Stephanopachys* have been reported from California (M. L. Gimmel, 2022, unpublished data). The species were keyed out by *Fisher (1950)*.

*Stephanopachys substriatus* (Paykull, 1800)

Nomenclatural Authority: *Fisher (1950)*, *Borowski & Węgrzynowicz (2007)*

Literature Records: none

Digitized Records: Santa Catalina (11 LACM)

Range: Also known from mainland (*Fisher, 1950*; *Borowski & Węgrzynowicz, 2007*).

**Lyctinae**

Notes. Two genera and six species of Lyctinae have been recorded from California (*Gerberg, 1957*).

***Lyctus* Fabricius, 1792**

Nomenclatural Authority: *Borowski & Węgrzynowicz (2007)*

Notes. Five species have been reported from California (*Gerberg, 1957*). These were keyed out by *Gerberg (1957)*.

***Lyctus cavicollis* LeConte, 1866**

Nomenclatural Authority: *Gerberg (1957)*, *Borowski & Węgrzynowicz (2007)*

Literature Records: none

Digitized Records: Santa Cruz (7 SBMNH)

Range: Also known from mainland (*Gerberg, 1957*; *Borowski & Węgrzynowicz, 2007*).

***Lyctus linearis* (Goeze, 1777)**

Nomenclatural Authority: *Gerberg (1957)*, *Borowski & Węgrzynowicz (2007)*

Literature Records: none

Digitized Records: Santa Cruz (1 SBMNH)

Range: Also known from mainland (*Gerberg, 1957*; *Borowski & Węgrzynowicz, 2007*).

Notes. This species is cosmopolitan (*Gerberg, 1957*).

***Lyctus planicollis* LeConte, 1858**

Nomenclatural Authority: *Gerberg (1957)*, *Borowski & Węgrzynowicz (2007)*

Literature Records: none

Digitized Records: Santa Cruz (1 SBMNH)

Range: Also known from mainland (*Gerberg, 1957*; *Borowski & Węgrzynowicz, 2007*).

Notes. This species was indicated as a synonym of the older *Lyctus carbonarius* Waltl, 1832 by *Lesne (1916)*; however, the synonymy was seen as tentative by *Gerberg (1957*: 26). *Borowski & Węgrzynowicz (2007)* treated *L. carbonarius* as valid with priority over *L. planicollis*.

**Polycaoninae**

Notes. Two genera and four species of Polycaoninae have been recorded from California (*Fisher, 1950*).

***Melalgus* Dejean, 1833**

Nomenclatural Authority: *Borowski & Węgrzynowicz (2007)*

Notes. Two species of *Melalgus* are known from California (*Fisher, 1950*). The species were keyed out by *Fisher (1950)*.

***Melalgus confertus* (LeConte, 1866)**

Nomenclatural Authority: *Fisher (1950)*, *Borowski & Węgrzynowicz (2007)*

Literature Records: none

Digitized Records: Santa Catalina (22 LACM)

Range: Also known from mainland (*Fisher, 1950*; *Borowski & Węgrzynowicz, 2007*).

*Polycaon* Castelnau, 1836

Nomenclatural Authority: *Borowski & Węgrzynowicz (2007)*

Notes. Two species of *Polycaon* are known from California (*Fisher, 1950*). The species were keyed out by *Fisher (1950)*.

*Polycaon stoutii* (LeConte, 1853)

Nomenclatural Authority: *Fisher (1950)*, *Borowski & Węgrzynowicz (2007)*

Literature Records: Santa Cruz (*LeConte, 1876*: 299; *Fall, 1897*: 238; *Fall & Davis, 1934*: 144)

Digitized Records: Santa Catalina (6 LACM; 2 SBMNH; 1 iNat)

Range: Also known from mainland (*Fisher, 1950*; *Borowski & Węgrzynowicz, 2007*).

Notes. In addition to *P. stoutii*, *LeConte (1876)* also reported *Polycaon ovicollis* (LeConte, 1857) from Santa Cruz Island. The latter is now considered a junior synonym of the former (see *Fisher, 1950*).

Psoinae

Notes. One genus and two species of Psoinae have been recorded from California (*Fisher, 1950*), belonging to the tribe Psoini.

*Psoa* Herbst, 1797

Nomenclatural Authority: *Borowski & Węgrzynowicz (2007)*

Notes. Two species of *Psoa* are known from California (*Fisher, 1950*). The species were keyed out by *Fisher (1950)*.

*Psoa maculata* (LeConte, 1852)

Nomenclatural Authority: *Fisher (1950)*, *Borowski & Węgrzynowicz (2007)*

Literature Records: none

Digitized Records: Santa Catalina (4 LACM)

Range: Also known from mainland (*Fisher, 1950*; *Borowski & Węgrzynowicz, 2007*).

*Psoa quadrisignata* (Horn, 1868)

Nomenclatural Authority: *Fisher (1950)*, *Borowski & Węgrzynowicz (2007)*

Literature Records: Santa Catalina (*Cockerell, 1940*: 286)

Digitized Records: none

Range: Also known from mainland (*Fisher, 1950*; *Borowski & Węgrzynowicz, 2007*).

Dermestidae

Notes. Six subfamilies, 18 genera, and 77 species are known to occur in California (*Háva & Herrmann, 2021*; M. L. Gimmel, 2022, unpublished data). *Beal (2003)* provided a distributional checklist of Dermestidae from North America, which was updated by *Háva & Herrmann (2021)*. We use the classification of the latter publication below.

Dermestinae

Notes. One genus and 14 species of Dermestinae are known to occur in California, belonging to the tribe Dermestini (*Háva & Herrmann, 2021*).

### Dermestes Linnaeus, 1758

Nomenclatural Authority: *Beal (2003)*

Digitized Records (genus-only): Anacapa (9 LACM), San Clemente (1 LACM; 22 YPMC), Santa Barbara (3 LACM), Santa Catalina (21 LACM; 2 YPMC), Santa Cruz (5 YPMC)

Notes. Fourteen species of *Dermestes* are known to occur in California, belonging to two subgenera, *Dermestes* (*s.str.*) and *Dermestinus* Zantiev, 1967 (*Beal, 2003*; *Háva & Herrmann, 2021*). These were all keyed out in the work of *Lepesme (1949)*.

### Dermestes (Dermestinus) caninus Germar, 1824

Nomenclatural Authority: *Beal (2003)*, *Háva & Herrmann (2021)*

Literature Records: San Clemente (*Fall, 1897*: 237), San Nicolas (*Fall, 1897*: 237), Santa Barbara (*Fall, 1897*: 237; *Miller & Miller, 1985*: 125), Santa Rosa (*Fall, 1897*: 237)

Digitized Records: Santa Barbara (2 SBMNH)

Range: Also known from mainland (*Beal, 2003*; *Háva & Herrmann, 2021*).

Notes. *Fall (1897)* recorded this species as *Dermestes mannerheimii* LeConte, which is a junior synonym of *D. caninus* (see *Miller & Miller, 1985*). These taxa represent two valid subspecies according to *Háva & Herrmann (2021)*.

### Dermestes (Dermestinus) frischi Kugelann, 1792

Nomenclatural Authority: *Beal (2003)*

Literature Records: San Clemente (*Miller & Miller, 1985*: 125), San Miguel (*Miller & Miller, 1985*: 125), Santa Barbara (*Miller & Miller, 1985*: 125), Santa Cruz (*Fall & Davis, 1934*: 143; *Miller & Miller, 1985*: 125)

Digitized Records: San Clemente (5 LACM; 3 SBMNH), San Miguel (1 SBMNH), San Nicolas (8 SBMNH), Santa Barbara (3 SBMNH), Santa Cruz (5 SBMNH)

Range: Also known from mainland (*Beal, 2003*; *Háva & Herrmann, 2021*).

Notes. The species epithet was spelled *frischii* by *Háva & Herrmann (2021)*, who did not include California in the list of states in which the species was known to occur. This species is cosmopolitan (*Háva & Herrmann, 2021*).

### Dermestes (Dermestinus) marmoratus Say, 1823

Nomenclatural Authority: *Beal (2003)*, *Háva & Herrmann (2021)*

Literature Records: San Clemente (*Fall, 1897*: 237), San Nicolas (*Fall, 1897*: 237), Santa Catalina (*Fall, 1897*: 237), Santa Rosa (*Fall, 1897*: 237)

Digitized Records: San Clemente (1 YPMC)

Range: Also known from mainland (*Beal, 2003*; *Háva & Herrmann, 2021*).

### Dermestes (Dermestinus) rattus LeConte, 1854

Nomenclatural Authority: *Beal (2003)*, *Háva & Herrmann (2021)*

Literature Records: Santa Rosa (*Fall, 1897*: 237)

Digitized Records: Santa Cruz (1 SBMNH)

Range: Also known from mainland (*Beal, 2003*; *Beal & Seeno, 1977*; *Háva & Herrmann, 2021*).

Notes. This species was recorded as *Dermestes tristis* by *Fall (1897)*; the subspecies of *D. rattus* occurring on the islands is *D. r. tristis* Fall, 1897.

### *Dermestes* (*Dermestinus*) *talpinus* Mannerheim, 1843

Nomenclatural Authority: *Beal (2003)*, *Háva & Herrmann (2021)*
Literature Records: Santa Cruz (*LeConte, 1876*: 298; *Fall, 1897*: 237; *Fall & Davis, 1934*: 143)
Digitized Records: Santa Rosa (1 SBMNH)
Range: Also known from mainland (*Beal, 2003*; *Háva & Herrmann, 2021*).

### Megatominae

Notes. Two tribes, seven genera, and 42 species of Dermestidae are known to occur in California (*Háva & Herrmann, 2021*; M. L. Gimmel, 2022, unpublished data).

### Anthrenini

Notes. One genus and 10 species of Anthrenini have been recorded from California (*Háva & Herrmann, 2021*).

### *Anthrenus* Geoffroy, 1762

Nomenclatural Authority: *Kadej (2011)*, *Háva & Herrmann (2021)*
Notes. Ten species of *Anthrenus* have been reported from California, belonging to four subgenera, *Anthrenops* Reitter, 1881, *Anthrenus* (*s.str.*), *Florilinus* Mulsant & Rey, 1868, and *Nathrenus* Casey, 1900 (*Beal, 2003*; *Háva & Herrmann, 2021*). The North American species were revised by *Beal (1998)*; *Kadej (2011)* described an additional species from California and provided an updated key to North American species.

### *Anthrenus* (*Anthrenus*) *lepidus* LeConte, 1854

Nomenclatural Authority: *Beal (1998)*, *Beal (2003)*, *Háva & Herrmann (2021)*
Literature Records: none
Digitized Records: Santa Cruz (2 SBMNH), Santa Rosa (3 SBMNH)
Range: Also known from mainland (*Beal, 1998*, *2003*; *Háva & Herrmann, 2021*).

### *Anthrenus* (*Nathrenus*) *verbasci* (Linnaeus, 1767)

Nomenclatural Authority: *Beal (1998)*, *Beal (2003)*, *Háva & Herrmann (2021)*
Literature Records: Santa Catalina (*Cockerell, 1940*: 286)
Digitized Records: Santa Catalina (1 iNat)
Range: Also known from mainland (*Beal, 1998*, *2003*; *Háva & Herrmann, 2021*).
Notes. This is a cosmopolitan species (*Beal, 2003*).

### Megatomini

Notes. Six genera and 32 species of Megatomini are known to occur in California (*Háva & Herrmann, 2021*; M. L. Gimmel, 2022, unpublished data).

### *Cryptorhopalum* Guérin-Méneville, 1838

Nomenclatural Authority: *Háva & Herrmann (2021)*

Notes. Nine species of *Cryptorhopalum* have been recorded from California (*Beal, 2003*; *Háva & Herrmann, 2021*). These were mostly revised and keyed by *Beal (1979)*, who later published a revision of the entire Nearctic fauna (*Beal, 1985*).

### *Cryptorhopalum apicale* (Mannerheim, 1843)
Nomenclatural Authority: *Beal (1985)*, *Beal (2003)*, *Háva & Herrmann (2021)*
Literature Records: Santa Cruz (*Beal, 1979*: 16 [map]; *Beal, 1985*: 191 [map])
Digitized Records: Santa Cruz (18 SBMNH)
Range: Also known from mainland (*Beal, 1979*, *1985*, *2003*; *Háva & Herrmann, 2021*).

### *Cryptorhopalum triste* LeConte, 1854
Nomenclatural Authority: *Beal (1985)*, *Beal (2003)*, *Háva & Herrmann (2021)*
Literature Records: none
Digitized Records: Santa Rosa (1 SBMNH)
Range: Also known from mainland (*Beal, 1979*, *1985*, *2003*; *Háva & Herrmann, 2021*).

### *Megatoma* Herbst, 1791
Nomenclatural Authority: *Háva & Herrmann (2021)*
Digitized Records (genus-only): Santa Rosa (6 SBMNH)
Notes. Eleven species of *Megatoma* have been reported from California, distributed among two subgenera, *Megatoma* (*s.str.*) and *Pseudohadrotoma* Kalík, 1957 (*Háva & Herrmann, 2021*). The species in North America were revised by *Beal (1967)*.

### *Megatoma* (*Megatoma*) *variegata* (Horn, 1875)
Nomenclatural Authority: *Beal (2003)*, *Háva & Herrmann (2021)*
Literature Records: none
Digitized Records: Santa Cruz (10 SBMNH), Santa Rosa (11 SBMNH)
Range: Also known from mainland (*Beal, 2003*; *Háva & Herrmann, 2021*).

### *Trogoderma* Dejean, 1821
Nomenclatural Authority: *Beal (2003)*, *Háva & Herrmann (2021)*
Notes. Nine species of *Trogoderma* have been reported from California (*Beal, 2003*; *Háva & Herrmann, 2021*). The North American species were revised by *Beal (1954)*.

### *Trogoderma sternale* Jayne, 1882
Nomenclatural Authority: *Beal (1954)*, *Beal (2003)*, *Háva & Herrmann (2021)*
Literature Records: Santa Barbara (*Miller & Miller, 1985*: 125), Santa Catalina (*Fall, 1897*: 237; *Fall, 1901*: 93; *Beal, 1954*: 72 [map]; *Miller & Miller, 1985*: 125)
Digitized Records: Santa Barbara (1 SBMNH), Santa Catalina (4 SBMNH), Santa Cruz (75 SBMNH; 1 UCSB), Santa Rosa (3 SBMNH)
Range: Also known from mainland (*Beal, 1954*, *2003*; *Háva & Herrmann, 2021*).
Notes. The subspecies of *T. sternale* present on the islands is the nominate subspecies, *T. s. sternale* Jayne, 1882 (*Beal, 1954*).

**Ptinidae**

Notes. Nine subfamilies, 41 genera, and 145 species of Ptinidae are known to occur in California (M. L. Gimmel, 2022, unpublished data). The family Ptinidae in the modern sense corresponds with the combined former concepts of Anobiidae and Ptinidae. The species (excluding Ptininae) were cataloged for North America by *White (1982)*. The classification we employ below follows *Philips (2002)*, with the exception of the use of Ptinidae over Anobiidae.

**Anobiinae**

Notes. Seven tribes, 13 genera, and 31 species of Anobiinae are known to occur in California (M. L. Gimmel, 2022, unpublished data).

**Anobiini**

Notes. Three genera and seven species of Anobiini are known to occur in California (M. L. Gimmel, 2022, unpublished data).

*Hemicoelus* **LeConte, 1861**

Nomenclatural Authority: *White (1976a)*, *White (1982)*

Notes. Four species of *Hemicoelus* are known to occur in California, including three previously reported (*White, 1982*) and the new state record below. *White (1976a)* provided a key to North American species.

*Hemicoelus nelsoni* **(Hatch, 1961)**

Nomenclatural Authority: *White (1976a)*, *White (1982)*

Literature Records: none

Digitized Records: Santa Cruz (1 SBMNH), Santa Rosa (1 SBMNH)

Range: Also known from mainland (*White, 1982*).

Notes. This species represents a **new state record** for California.

**Colposternini**

Notes. One species of Colposternini has been recorded from California (*White, 1982*).

*Colposternus* **Fall, 1905**

Nomenclatural Authority: *White (1982)*

Notes. One species of *Colposternus* has been recorded from California (*White, 1982*).

*Colposternus tenuilineatus* **(Horn, 1894)**

Nomenclatural Authority: *White (1982)*

Literature Records: Santa Catalina (*Fall, 1897*: 238; *Fall, 1901*: 132; *Fall, 1905*: 191)

Digitized Records: Santa Cruz (1 SBMNH)

Range: Also known from mainland (*White, 1982*).

Notes. *Fall (1897, 1901)* recorded this species as *Trypopitys tenuilineata*; *Fall (1905)* subsequently transferred it to *Colposternus*.

**Euceratocerini**

Notes. Two or three genera and 10 or 11 species of Euceratocerini have been recorded from California (*White, 1982*; M. L. Gimmel, 2022, unpublished data).

***Actenobius* Fall, 1905**

Nomenclatural Authority: *White (1982)*

Notes. One species of *Actenobius* has been recorded from California (*White, 1982*).

***Actenobius pleuralis* (Casey, 1898)**

Nomenclatural Authority: *White (1982)*

Literature Records: none

Digitized Records: Santa Cruz (1 SBMNH)

Range: Also known from mainland (*White, 1982*).

***Euceratocerus* LeConte, 1874**

Nomenclatural Authority: *White (1982)*

Notes. No species of *Euceratocerus* have been reported from California (*White, 1982*). The species were treated for North America by *Hinson (2021)*.

***Euceratocerus hornii* LeConte, 1874**

Nomenclatural Authority: *White (1982)*

Literature Records: Santa Catalina (*Fall, 1897*: 238)

Digitized Records: none

Range: Also known from mainland (*White, 1982*; *Hinson, 2021*).

Notes. This species is otherwise reported only from Texas, with all other species of the genus being from points farther east (*White, 1982*), casting extreme doubt on *Fall's (1897)* record from Santa Catalina. Fall's record may have referred to Santa Catalina Island specimens of another ptinid species with pectinate antennae, *Ptilinus basalis* LeConte, 1858, in the H.C. Fall collection at the Museum of Comparative Zoology, Harvard University (S. Miller, 2022, personal communication). This species was not mentioned in *Fall (1897)*, so these specimens were probably (re)determined by H.C. Fall after that publication.

**Hadrobregmini**

Notes. Two genera and three species of Hadrobregmini have been recorded from California (*White, 1982*).

***Priobium* Motschulsky, 1845**

Nomenclatural Authority: *White (1982)*

Notes. One species of *Priobium* has been recorded from California (*White, 1982*).

***Priobium punctatum* (LeConte, 1859)**

Nomenclatural Authority: *White (1982)*

Literature Records: Santa Cruz (*Cockerell, 1940*: 286)

Digitized Records: Santa Catalina (1 iNat), Santa Cruz (7 SBMNH)

Range: Also known from mainland (*White, 1982*).

Notes. *Cockerell (1940)* reported this species as *Trypopitys punctatus*.

**Stegobiini**
Notes. Two genera and five species of Stegobiini are known to occur in California (M. L. Gimmel, 2022, unpublished data).

***Oligomerus* Redtenbacher, 1849**
Nomenclatural Authority: *White (1982)*
Notes. Four species of *Oligomerus* are known to occur in California, including one putatively undescribed species (M. L. Gimmel, 2022, unpublished data). *White (1976a)* provided a key to species of *Oligomerus*.

***Oligomerus delicatulus* (Fall, 1920)**
Nomenclatural Authority: *White (1976a)*, *White (1982)*
Literature Records: none
Digitized Records: Anacapa (1 SBMNH)
Range: Also known from mainland (*White, 1982*).

***Stegobium* Motschulsky, 1860**
Nomenclatural Authority: *White (1982)*
Notes. One species of *Stegobium* is known to occur in California (M. L. Gimmel, 2022, unpublished data).

***Stegobium paniceum* (Linnaeus, 1758)**
Nomenclatural Authority: *White (1982)*
Literature Records: none
Digitized Records: Santa Rosa (4 SBMNH)
Range: Also known from mainland (*White, 1982*).
Notes. This pest species, known as the drugstore beetle, has been introduced around the world.

**Dorcatominae**
Notes. Three tribes, five genera, and seven species of Dorcatominae are known to occur in California (M. L. Gimmel, 2022, unpublished data).

**Dorcatomini**
Notes. Three genera and five species of Dorcatomini are known to occur in California (M. L. Gimmel, 2022, unpublished data).

***Byrrhodes* LeConte, 1878**
Nomenclatural Authority: *White (1982)*
Notes. Two species of *Byrrhodes* have been recorded from California, including one putatively undescribed species (M. L. Gimmel, 2022, unpublished data). *White (1973a)* provided a key to North American species.

### Byrrhodes ulkei (Fall, 1905)

Nomenclatural Authority: *White (1973a)*, *White (1982)*
Literature Records: none
Digitized Records: Santa Catalina (1 SBMNH)
Range: Also known from mainland (*White, 1982*).
Notes. In the SBMNH collection, T.K. Philips identified a possible new species of *Byrrhodes* from specimens collected on the coastal side of the Santa Ynez Mountains in Santa Barbara County; the above record of *B. ulkei* possibly represents this species.

### Petaliini

Notes. One species of Petaliini has been recorded from California (*White, 1982*).

### Petalium LeConte, 1861

Nomenclatural Authority: *Ford (1973)*, *White (1982)*
Notes. One species of *Petalium* has been recorded from California (*White, 1982*). *Ford (1973)* provided a key to North American species of this genus.

### Petalium californicum Fall, 1905

Nomenclatural Authority: *Ford (1973)*, *White (1982)*
Literature Records: none
Digitized Records: Santa Cruz (1 SBMNH)
Range: Also known from mainland (*Ford, 1973*; *White, 1982*).

### Ernobiinae

Notes. Three tribes, seven genera, and 27 species of Ernobiinae are known to occur in California (*White, 1982*; M. L. Gimmel, 2022, unpublished data).

### Ernobiini

Notes. Three genera and 19 species of Ernobiini are known to occur in California (*White, 1982*; M. L. Gimmel, 2022, unpublished data).

### Ernobius Thomson, 1859

Nomenclatural Authority: *White (1982)*
Notes. Seventeen species of *Ernobius* are known to occur in California, including one putatively undescribed species (M. L. Gimmel, 2022, unpublished data). The species of this genus were keyed for California by *Ruckes (1957)*.

### Ernobius debilis LeConte, 1865

Nomenclatural Authority: *Ruckes (1957)*, *White (1982)*
Literature Records: Santa Cruz (*LeConte, 1865*: 225; *Fall, 1897*: 238; *Fall, 1901*: 131)
Digitized Records: Santa Cruz (10 SBMNH)
Range: Also known from mainland (*Fall, 1901*).
Notes. This species was considered endemic to Santa Cruz Island by *LeConte (1865)* and *Fall (1897)*, but subsequently discovered on the mainland (*Fall, 1901*).

*Ernobius punctulatus* (LeConte, 1859)
Nomenclatural Authority: *Ruckes (1957)*, *White (1982)*
Literature Records: Santa Cruz (*Fall & Davis, 1934*: 144)
Digitized Records: none
Range: Also known from mainland (*White, 1982*).

**Ozognathini**
Notes. Two genera and three species of Ozognathini have been recorded from California (*White, 1982*).

*Ozognathus* LeConte, 1861
Nomenclatural Authority: *White (1982)*
Notes. One species of *Ozognathus* has been reported from California (*White, 1982*).

*Ozognathus cornutus* (LeConte, 1859)
Nomenclatural Authority: *White (1982)*
Literature Records: none
Digitized Records: Santa Catalina (1 iNat), Santa Cruz (2 SBMNH), Santa Rosa (1 SBMNH)
Range: Also known from mainland (*White, 1982*).

*Xarifa* Fall, 1905
Nomenclatural Authority: *White (1982)*
Notes. Originally described as a genus endemic to the Channel Islands, one species, *Xarifa lobata* Fall, 1929, was subsequently described from mainland California (Carmel; *Fall, 1929a*). These remain the only two species known; they were keyed by *White (1974)*.

*Xarifa insularis* Fall, 1905
Nomenclatural Authority: *White (1982)*
Literature Records: San Clemente (*Fall, 1905*: 138; *Miller, 1985a*: 20), Santa Catalina (*Fall, 1905*: 138; *Miller, 1985a*: 20; *White, 1982*: 3)
Digitized Records: Santa Catalina (3 SBMNH), Santa Cruz (1 LACM; 6 SBMNH), Santa Rosa (12 SBMNH)
Range: Endemic (*Fall, 1905*; *Miller, 1985a*).

**Xestobiini**
Notes. Two genera and five species of Xestobiini have been recorded from California (*White, 1982*).

*Xestobium* Motschulsky, 1845
Nomenclatural Authority: *White (1982)*
Notes. Three species of *Xestobium* have been reported from California (*White, 1982*). These species were partially keyed by *White (1975)*, with another species added by *White (1976b)*.

***Xestobium marginicolle*** **(LeConte, 1859)**
Nomenclatural Authority: *White (1982)*
Literature Records: Santa Barbara (*Miller & Miller, 1985*: 123)
Digitized Records: Santa Cruz (2 SBMNH)
Range: Also known from mainland (*White, 1982*).
Notes. Reported from *Hemizonia* (Asteraceae) on Santa Barbara Island by *Miller & Miller (1985)*.

**Eucradinae**
Notes. Only one genus, belonging to the tribe Hedobiini, and three species of Eucradinae have been recorded from California (*White, 1982*).

***Ptinomorphus*** **Mulsant & Rey, 1868**
Nomenclatural Authority: *Zahradník & Háva (2014)*
Notes. Three species of *Ptinomorphus* have been reported from California (*White, 1982*, as *Hedobia* Dejean, 1821).

***Ptinomorphus granosus*** **(LeConte, 1874)**
Nomenclatural Authority: *White (1982)*, *Zahradník & Háva (2014)*
Literature Records: none
Digitized Records: Santa Rosa (1 SBMNH)
Range: Also known from mainland (*White, 1982*).
Notes. This species was previously known as *Hedobia granosa*.

**Mesocoelopodinae**
Notes. Only one genus, belonging to the tribe Tricorynini, and 20 species of Mesocoelopodinae have been recorded from California (M. L. Gimmel, 2022, unpublished data).

***Tricorynus*** **Waterhouse, 1849**
Nomenclatural Authority: *White (1982)*
Literature Records (genus-only): Santa Cruz (*Naughton et al., 2014*: 303)
Digitized Records (genus-only): Anacapa (2 SBMNH), San Clemente (5 SBMNH), San Nicolas (2 SBMNH), Santa Barbara (1 SBMNH), Santa Catalina (3 SBMNH), Santa Cruz (7 SBMNH)
Notes. Twenty species of *Tricorynus* have been recorded from California (M. L. Gimmel, 2022, unpublished data). This genus was revised for North America by *White (1965)*.

***Tricorynus nubilus*** **(Fall, 1905)**
Nomenclatural Authority: *White (1965)*, *White (1982)*
Literature Records: Santa Catalina (*White, 1965*: 333)
Digitized Records: none
Range: Also known from mainland (*White, 1965*, *1982*).

***Tricorynus obsoletus*** **(LeConte, 1865)**
Nomenclatural Authority: *White (1965)*, *White (1982)*

Literature Records: Santa Catalina (*Fall, 1897*: 238)

Digitized Records: none

Range: Also known from mainland (*White, 1965*).

Notes. *Fall (1897)* listed this species as *Hemiptychus obsoletus* with a question mark.

### Ptilininae

Notes. Only one genus, belonging to the tribe Ptilinini, and four species of Ptilininae have been recorded from California (*White, 1982*).

### *Ptilinus* Müller, 1764

Nomenclatural Authority: *White (1982)*

Notes. Four species of *Ptilinum* have been recorded from California (*White, 1982*). No modern key exists to separate these species.

### *Ptilinus basalis* LeConte, 1858

Nomenclatural Authority: *White (1982)*

Literature Records: none

Digitized Records: Santa Cruz (3 SBMNH)

Range: Also known from mainland (*White, 1982*).

Notes. See Notes for *Euceratocerus hornii* above.

### Ptininae: Ptinini

Notes. Four tribes, six genera, and 23 species of Ptininae, including three genera and 19 species of Ptinini, are known to occur in California (M. L. Gimmel, 2022, unpublished data).

### *Ptinus* Linnaeus, 1766

Nomenclatural Authority: *Papp & Okumura (1959)*

Notes. Fifteen species of *Ptinus* have been recorded from California, distributed among three subgenera, *Gynopterus* Mulsant & Rey, 1868, *Ptinus* (*s.str.*), and *Tectoptinus* Iablokoff-Khnzorian & Karapetyan, 1986 (*Papp & Okumura, 1959*; *Zahradník & Háva, 2014*). The key of *Papp & Okumura (1959)* can be used to separate these. *Naughton et al. (2014*: 303) recorded the genus only from Santa Cruz Island, but this has since been identified as *P. agnatus* (see below).

### *Ptinus* (*Gynopterus*) *fallax* Fall, 1905

Nomenclatural Authority: *Papp & Okumura (1959)*

Literature Records: none

Digitized Records: Santa Catalina (6 SBMNH; 1 iNat)

Range: Also known from mainland (*Papp & Okumura, 1959*).

### *Ptinus* (*Ptinus*) *agnatus* Fall, 1905

Nomenclatural Authority: *Papp & Okumura (1959)*

Literature Records: none

Digitized Records: Santa Cruz (3 SBMNH), Santa Rosa (13 SBMNH)

Range: Also known from mainland (*Papp & Okumura, 1959*).

Notes. The Santa Cruz Island specimen of "*Ptinus*" from *Naughton et al. (2014)* exists as a voucher (in SBMNH) and was identified to this species by MLG.

### Xyletininae
Notes. Two tribes, six genera, and 29 species of Xyletininae have been recorded from California (*White, 1982*; M. L. Gimmel, 2022, unpublished data).

### Lasiodermini
Notes. Two genera and five species of Lasiodermini have been recorded from California (M. L. Gimmel, 2022, unpublished data).

### *Lasioderma* Stephens, 1835
Nomenclatural Authority: *White (1982)*
Notes. Two species of *Lasioderma* are known to occur in California (M. L. Gimmel, 2022, unpublished data).

### *Lasioderma serricorne* (Fabricius, 1792)
Nomenclatural Authority: *White (1982)*
Literature Records: none
Digitized Records: Santa Cruz (15 LACM)
Range: Also known from mainland (*White, 1982*).
Notes. This cosmopolitan pest is known as the cigarette beetle.

### Xyletinini
Notes. Four genera and 24 species of Xyletinini have been recorded from California (*White, 1982*).

### *Euvrilletta* Fall, 1905
Nomenclatural Authority: *White (1982)*
Notes. Seven species of *Euvrilletta* have been recorded from California (*White, 1982*). The key in *White (1985)* can be used to identify them.

### *Euvrilletta catalinae* (Fall, 1905)
Nomenclatural Authority: *White (1982)*
Literature Records: Santa Catalina (*Fall, 1897*: 238; *Fall, 1901*: 131; *Fall, 1905*: 162; *Miller, 1985a*: 20; *Caterino & Chandler, 2010*: 187)
Digitized Records: none
Range: Endemic (*Fall, 1905*; *Miller, 1985a*; *Caterino & Chandler, 2010*).
Notes. *Fall (1905)* recorded this species as *Oligomerodes catalinae* Fall; the genus *Oligomerodes* Fall was subsequently synonymized with *Euvrilletta* Fall by *White (1976a*: 164), and the species was included in a key by *White (1985*: 191). Specimens referred to by *Fall (1897*: 238, *1901*: 131) as "*Oligomerus*? new species" represent this species. Adults were collected from foliage of *Heteromeles arbutifolia* (Lindl.) M.Roem. (Rosaceae) (*Fall, 1901*; *White, 1982*).

### *Euvrilletta occidentalis* (Fall, 1905)

Nomenclatural Authority: *White (1982)*

Literature Records: none

Digitized Records: Santa Cruz (1 SBMNH)

Range: Also known from mainland (*White, 1982*).

### *Vrilletta* LeConte, 1874

Nomenclatural Authority: *White (1982)*

Notes. Ten species of *Vrilletta* have been recorded from California (*White, 1982*). *White (1980)* reviewed the genus and provided a key to species.

### *Vrilletta blaisdelli* Fall, 1905

Nomenclatural Authority: *White (1982)*

Literature Records: Santa Catalina (*Cockerell, 1940*: 286)

Digitized Records: Santa Cruz (1 SBMNH), Santa Rosa (11 SBMNH)

Range: Also known from mainland (*White, 1982*).

### *Xyletinus* Latreille, 1809

Nomenclatural Authority: *White (1977)*, *White (1982)*

Notes. Six species of *Xyletinus* have been recorded from California (*White, 1982*). *White (1977)* provided a partial, updated key to these species, supplementing that of *White (1973b)*.

### *Xyletinus* undetermined species

Literature Records: none

Digitized Records: Santa Rosa (1 SBMNH)

Notes. The single SBMNH specimen roughly keys to *Xyletinus rotundicollis* White, 1977 in *White (1977)*. However, without adequate comparative material we hesitate to make a species-level identification.

### CLEROIDEA

### Byturidae

Notes. Two genera and two species are known from California (*Springer & Goodrich, 1983*). *Springer & Goodrich (1983)* provided a revision of the family for North America.

### *Xerasia* Lewis, 1895

Nomenclatural Authority: *Springer & Goodrich (1991)*

Notes. Only one species of *Xerasia* is known from North America (*Springer & Goodrich, 1991*).

### *Xerasia grisescens* (Jayne, 1882)

Nomenclatural Authority: *Springer & Goodrich (1991)*

Literature Records: Santa Catalina (*Cockerell, 1940*: 286; *Springer & Goodrich, 1983*: 190 [map only]), Santa Cruz (*Springer & Goodrich, 1983*: 190 [map only]; *Naughton et al., 2014*: 303)

Digitized Records: San Miguel (1 SBMNH), Santa Catalina (2 LACM; 11 SBMNH), Santa Cruz (2 SBMNH), Santa Rosa (4 SBMNH)

Range: Also known from mainland (*Springer & Goodrich, 1983*).

Notes. This species was recorded by *Cockerell (1940)* as *Byturus grisescens*, and by *Springer & Goodrich (1983)* as *Byturellus grisescens*.

## Cleridae

Notes. Four subfamilies, 18 genera, and about 88 species of Cleridae are known from California (M. L. Gimmel, 2022, unpublished data).

## Clerinae

Notes. Five tribes, six genera and 43 species of Clerinae are known to occur in California (M. L. Gimmel, 2022, unpublished data). This subfamily was recently redefined and reclassified by *Bartlett (2021)*.

## Dieropsini

Notes. One genus and four species of Dieropsini are known to occur in California (*Foster, 1976*).

### *Trichodes* Herbst, 1792

Nomenclatural Authority: *Wolcott (1947)*

Notes. Four species of *Trichodes* are known from California (*Foster, 1976*). These were keyed by *Foster (1976)*.

### *Trichodes ornatus* Say, 1823

Nomenclatural Authority: *Foster (1976)*

Literature Records: none

Digitized Records: Santa Cruz (7 SBMNH; 6 UCSB)

Range: Also known from mainland (*Foster, 1976*).

Notes. The subspecies of *T. ornatus* occurring in coastal California is *T. o. douglasianus* White, 1849 (*Foster, 1976*).

## Hydnocerini

Notes. One genus and 15 species of Hydnocerini are known to occur in California (M. L. Gimmel, 2022, unpublished data).

### *Phyllobaenus* Dejean, 1837

Nomenclatural Authority: *Wolcott (1947)*

Literature Records (genus-only): Santa Cruz (*Naughton et al., 2014*: 303)

Digitized Records (genus-only): Santa Catalina (1 SBMNH), Santa Cruz (2 SBMNH; 2 UCSB)

Notes. Fifteen species of *Phyllobaenus* are known to occur in California (M. L. Gimmel, 2022, unpublished data). The two SBMNH specimens from Santa Cruz are vouchers for the *Naughton et al. (2014)* study.

***Phyllobaenus funebris*** (Chevrolat, 1874)
Nomenclatural Authority: *Wolcott (1947)*
Literature Records: none
Digitized Records: San Miguel (2 SBMNH), Santa Cruz (14 SBMNH)
Range: Also known from mainland (*Wolcott, 1947*).

***Phyllobaenus scaber*** (LeConte, 1852)
Nomenclatural Authority: *Wolcott (1947)*
Literature Records: none
Digitized Records: Santa Cruz (1 SBMNH), Santa Rosa (1 SBMNH)
Range: Also known from mainland (*Wolcott, 1947*).

**Korynetinae**
Notes. Six genera and 13 species of Korynetinae are known to occur in California (M. L. Gimmel, 2022, unpublished data).

***Loedelia*** Lucas, 1920
Nomenclatural Authority: *Wolcott (1947)*
Notes. One species of *Loedelia* is known from California (*Wolcott, 1947*).

***Loedelia maculicollis*** (LeConte, 1874)
Nomenclatural Authority: *Wolcott (1947)*
Literature Records: none
Digitized Records: Santa Cruz (2 SBMNH; 1 UCSB)
Range: Also known from mainland (*Wolcott, 1947*).

***Necrobia*** Olivier, 1795
Nomenclatural Authority: *Wolcott (1947)*
Notes. Three species of *Necrobia* are known from California (M. L. Gimmel, 2022, unpublished data). Although these have long been thought to be introductions to North America from other parts of the world, evidence presented from tar pit material by *Holden, Barclay & Angus (2018)* casts doubt on this for the third known Californian species, *Necrobia violacea* (Linnaeus, 1758).

***Necrobia ruficollis*** (Fabricius, 1775)
Nomenclatural Authority: *Papp (1959)*
Literature Records: San Clemente (*Fall, 1897*: 238; *Fall, 1901*: 130)
Digitized Records: Anacapa (1 SBMNH), Santa Cruz (5 LACM; 2 SBMNH)
Range: Also known from mainland (*Wolcott, 1947*).
Notes. *Fall (1901)* recorded this species as *Corynetes ruficollis*. This is a cosmopolitan species (*Papp, 1959*).

***Necrobia rufipes*** (DeGeer, 1775)
Nomenclatural Authority: *Papp (1959)*
Literature Records: San Clemente (*Fall, 1897*: 238), Santa Catalina (*Fall, 1897*: 238), Santa Rosa (*Fall, 1897*: 238)

Digitized Records: Anacapa (9 LACM), San Clemente (22 LACM; 1 SBMNH), San Miguel (4 LACM; 3 SBMNH), San Nicolas (28 LACM; 7 SBMNH), Santa Catalina (1 LACM; 1 SBMNH), Santa Cruz (98 LACM; 8 SBMNH; 2 UCSB), Santa Rosa (3 SBMNH)

Range: Also known from mainland (*Fall, 1901*; *Wolcott, 1947*). This is a cosmopolitan species (*Papp, 1959*).

**Tillinae**

Notes. Six genera and 32 species of Tillinae are known to occur in California (*Burke, Leavengood & Zolnerowich, 2015*; M. L. Gimmel, 2022, unpublished data).

*Cymatodera* **Gray, 1832**

Nomenclatural Authority: *Burke, Leavengood & Zolnerowich (2015)*

Digitized Records (genus-only): Santa Cruz (1 UCRC)

Notes. Twenty-five species of the genus *Cymatodera* have been recorded from California (M. L. Gimmel, 2022, unpublished data).

*Cymatodera angustata* **Spinola, 1844**

Nomenclatural Authority: *Burke, Leavengood & Zolnerowich (2015)*

Literature Records: Santa Rosa (*Fall, 1897*: 238)

Digitized Records: none

Range: Also known from mainland (*Rifkind, 2019*).

Notes. This flightless species was not reported from the islands by *Rifkind (2019)*, and *Fall's (1897)* record may actually refer to *C. caterinoi*.

*Cymatodera caterinoi* **Rifkind & Burke, 2019**

Nomenclatural Authority: *Rifkind (2019)*

Literature Records: Anacapa (*Rifkind, 2019*: 556), Santa Cruz (*Rifkind, 2019*: 556), Santa Rosa (*Rifkind, 2019*: 556)

Digitized Records: Santa Cruz (5 SBMNH; 1 UCSB), Santa Rosa (1 CASC; 1 CSCA; 4 JNRC; 3 SBMNH)

Range: Endemic (*Rifkind, 2019*).

Notes. Flightless (*Rifkind, 2019*).

*Cymatodera insularis* **Rifkind, 2019**

Nomenclatural Authority: *Rifkind (2019)*

Literature Records: San Clemente (*Rifkind, 2019*: 553), Santa Catalina (*Rifkind, 2019*: 553)

Digitized Records: San Nicolas (1 SBMNH), Santa Catalina (2 SBMNH)

Range: Endemic (*Rifkind, 2019*).

Notes. Flightless (*Rifkind, 2019*).

*Cymatodera ovipennis* **LeConte, 1859**

Nomenclatural Authority: *Burke, Leavengood & Zolnerowich (2015)*

Literature Records: Santa Catalina (*Fall, 1897*: 238; *Fall, 1901*: 129)

Digitized Records: none

Range: Also known from mainland (*Fall, 1901*; *Burke, Leavengood & Zolnerowich, 2015*).

**Melyridae**

Notes. Two subfamilies, 29 genera, and 302 species of Melyridae are known to occur in California (M. L. Gimmel, 2022, unpublished data).

**Dasytinae**

Notes. Two tribes, 19 genera, and 211 species of Dasytinae are known to occur in California (M. L. Gimmel, 2022, unpublished data).

**Dasytini**

Notes. Eight genera and 40 species of Dasytini are known to occur in California (M. L. Gimmel, 2022, unpublished data).

***Dasytastes* Casey, 1895**

Nomenclatural Authority: M. L. Gimmel & A. Mayor, 2023, unpublished data

Digitized Records (genus-only): Anacapa (3 SBMNH), San Clemente (3 SBMNH), Santa Barbara (35 SBMNH), Santa Cruz (24 SBMNH), Santa Rosa (2 SBMNH)

Notes. Like most genera of Dasytinae, *Dasytastes* is in desperate need of revision. Until this is completed, distributional status and endemicity of the species cannot be determined. Forms from San Clemente Island appear externally divergent.

***Dasytastes catalinae* (LeConte, 1866)**

Nomenclatural Authority: M. L. Gimmel & A. Mayor, 2023, unpublished data

Literature Records: Santa Catalina (*LeConte, 1866c*: 361; *Casey, 1895*: 583; *Fall, 1901*: 128; *Miller, 1985a*: 20; *Caterino & Chandler, 2010*: 187)

Digitized Records: Santa Catalina (12 SBMNH)

Range: Endemic (*LeConte, 1866c*; *Casey, 1895*; *Fall, 1901*; *Miller, 1985a*; *Caterino & Chandler, 2010*).

Notes. *LeConte (1866c)* recorded this species as *Dasytes catalinae*; it was transferred to the genus *Dasytastes* by *Casey (1895)*. *Fall (1901*: 128) reported it to be "quite common on various flowers in July".

***Dasytastes insularis* Fall, 1901**

Nomenclatural Authority: M. L. Gimmel & A. Mayor, 2023, unpublished data

Literature Records: Santa Catalina (*Fall, 1897*: 238; *Fall, 1901*: 251; *Miller, 1985a*: 20)

Digitized Records: none

Range: Endemic (*Fall, 1897, 1901*; *Miller, 1985a*).

Notes. Recorded as "*Dasytes*, sp. nov." by *Fall (1897)*.

**"*Dasytes*" Paykull, 1799**

Nomenclatural Authority: M. L. Gimmel & A. Mayor, 2023, unpublished data

Digitized Records (genus-only): Santa Cruz (6 SBMNH)

Notes. The native North American species currently placed in *Dasytes* will soon receive a generic reassignment; true *Dasytes* only occurs in the Palearctic region (M. L. Gimmel, 2022, unpublished data). The North American fauna is in dire need of revision.

### *Dasytes clementae* Fall, 1901

Nomenclatural Authority: M. L. Gimmel & A. Mayor, 2023, unpublished data

Literature Records: San Clemente (*Fall, 1897*: 238; *Fall, 1901*: 251; *Miller, 1985a*: 20)

Digitized Records: none

Range: Endemic (*Fall, 1897*, *1901*; *Miller, 1985a*).

Notes. Recorded as "*Dasytes*, sp. nov." by *Fall (1897)*.

### *Eschatocrepis* LeConte, 1862

Nomenclatural Authority: M. L. Gimmel & A. Mayor, 2023, unpublished data

Notes. This genus contains only one species in North America (M. L. Gimmel, 2022, unpublished data).

### *Eschatocrepis constrictus* (LeConte, 1852)

Nomenclatural Authority: M. L. Gimmel & A. Mayor, 2023, unpublished data

Literature Records: Santa Catalina (*Fall, 1897*: 238), Santa Cruz (*Fall & Davis, 1934*: 144)

Digitized Records: Anacapa (1 SBMNH), San Miguel (3 SBMNH), Santa Cruz (3 LACM; 35 SBMNH), Santa Rosa (5 SBMNH)

Range: Also known from mainland (*Howell, 1985*).

### Listrini

Notes. Eleven genera and 171 species of Listrini are known to occur in California (M. L. Gimmel, 2022, unpublished data).

### *Listrus* Motschulsky, 1860

Nomenclatural Authority: M. L. Gimmel & A. Mayor, 2023, unpublished data

Literature Records (genus-only): San Clemente (*Fall, 1897*: 238), Santa Barbara (*Miller & Miller, 1985*: 126), Santa Rosa (*Fall, 1897*: 238)

Digitized Records (genus-only): Anacapa (1 SBMNH), San Clemente (74 LACM; 23 SBMNH), San Miguel (26 SBMNH), Santa Catalina (11 LACM; 14 SBMNH), Santa Cruz (6 LACM; 60 SBMNH), Santa Rosa (8 SBMNH)

Notes. Reported from *Coreopsis* (Asteraceae) on Santa Barbara Island by *Miller & Miller (1985)*. This genus currently contains 56 species known to occur in California (M. L. Gimmel, 2022, unpublished data). It is in desperate need of revision.

### *Listrus anacapensis* Blaisdell, 1924

Nomenclatural Authority: M. L. Gimmel & A. Mayor, 2023, unpublished data

Literature Records: Anacapa (*Blaisdell, 1924a*: 21; *Miller, 1985a*: 20)

Digitized Records: none

Range: Endemic (*Blaisdell, 1924a*; *Miller, 1985a*).

Notes. This species, along with all other *Listrus*, was inexplicably moved to the genus *Amecocerus* by *Pic (1937*: 98), and was reported as *Amecocerus anacapensis* by *Miller (1985a)*.

### *Listrus interruptus* LeConte, 1866

Nomenclatural Authority: M. L. Gimmel & A. Mayor, 2023, unpublished data

Literature Records: Santa Cruz (*LeConte, 1866c*: 357)

Digitized Records: none

Range: Also known from mainland (*Casey, 1895*).

Notes. *Casey (1895*: 547): "This species extends westward to the crests of the Sierras in California, but does not descend the western slope of the mountains." The Santa Cruz Island record was thought to be incorrect by *Fall (1901*: 127).

### *Microasydates* Gimmel & Mayor, 2022

Nomenclatural Authority: *Gimmel & Mayor (2022)*

Notes. This genus contains four species, all of which occur in California (*Gimmel & Mayor, 2022*). It was revised by *Gimmel & Mayor (2022)*.

### *Microasydates punctipennis* (LeConte, 1866)

Nomenclatural Authority: *Gimmel & Mayor (2022)*

Literature Records: Santa Catalina (*LeConte, 1866c*: 355; *Casey, 1895*: 532; *Fall, 1897*: 237; *Fall, 1901*: 126; *Gimmel & Mayor, 2022*: 555)

Digitized Records: Santa Catalina (1 SBMNH)

Range: Endemic (*LeConte, 1866c*; *Casey, 1895*; *Fall, 1901*; *Gimmel & Mayor, 2022*); reported from mainland (*Blaisdell, 1930*: 19), but based on a misidentification of *Asydates kumeyaay* Mayor & Gimmel, 2019 (*Gimmel & Mayor, 2022*: 555).

Notes. *LeConte (1866c)* and *Fall (1897)* recorded this species as *Pristoscelis punctipennis*; *Casey (1895)* and *Fall (1901)* recorded it as *Trichochrous punctipennis*.

### *Microasydates sanclemente* Gimmel & Mayor, 2022

Nomenclatural Authority: *Gimmel & Mayor (2022)*

Literature Records: San Clemente (*Gimmel & Mayor, 2022*: 556)

Digitized Records: San Clemente (7 SBMNH)

Range: Endemic (*Gimmel & Mayor, 2022*).

### *Microasydates santabarbara* Gimmel & Mayor, 2022

Nomenclatural Authority: *Gimmel & Mayor (2022)*

Literature Records: Anacapa (*Gimmel & Mayor, 2022*: 558), Santa Cruz (*Gimmel & Mayor, 2022*: 558), Santa Rosa (*Gimmel & Mayor, 2022*: 558)

Digitized Records: Anacapa (9 SBMNH), Santa Cruz (9 SBMNH; 15 LACM), Santa Rosa (3 LACM)

Range: Also known from mainland (*Gimmel & Mayor, 2022*).

### *Pseudasydates* Blaisdell, 1938

Nomenclatural Authority: M. L. Gimmel & A. Mayor, 2023, unpublished data

Notes. The genus *Pseudasydates* contains two currently described species in California, although there are additional species waiting to be described (M. L. Gimmel & A. Mayor, 2022, unpublished data).

### *Pseudasydates explanatus* (Casey, 1895)

Nomenclatural Authority: M. L. Gimmel & A. Mayor, 2023, unpublished data

Literature Records: none

Digitized Records: Santa Catalina (11 LACM)

Range: Also known from mainland (M. L. Gimmel & A. Mayor, 2022, unpublished data).

Notes. The specimens from Santa Catalina Island, collected by George P. Kanakoff (LACM) in 1941, are an unexpected outlier for this otherwise Mojave Desert-Central Valley genus. Effort should be made to recollect it on the island during early spring.

### *Trichochrous* Motschulsky, 1860

Nomenclatural Authority: M. L. Gimmel & A. Mayor, 2023, unpublished data

Literature Records (genus-only): San Miguel (*Miller & Davis, 1986*: 550), Santa Barbara (*Miller & Miller, 1985*: 126)

Notes. *Miller & Miller (1985)* reported 1–2 species of *Trichochrous* from Santa Barbara Island, including from *Avena* (Poaceae) and *Frankenia* (Frankeniaceae). *Miller & Davis (1986)* reported two undetermined species of this genus from San Miguel Island occurring on *Malacothrix* (Malvaceae).

### *Trichochrous brevicornis* (LeConte, 1852)

Nomenclatural Authority: M. L. Gimmel & A. Mayor, 2023, unpublished data

Literature Records: none

Digitized Records: Santa Cruz (16 SBMNH), Santa Rosa (1 SBMNH)

Range: Also known from mainland (M. L. Gimmel & A. Mayor, 2022, unpublished data).

### *Trichochrous calcaratus* Fall, 1934

Nomenclatural Authority: M. L. Gimmel & A. Mayor, 2023, unpublished data

Literature Records: Anacapa (*Cockerell, 1940*: 285; *Miller, 1985a*: 20), San Miguel (*Cockerell, 1940*: 285), Santa Cruz (*Fall & Davis, 1934*: 144; *Cockerell, 1940*: 285; *Miller, 1985a*: 20), Santa Rosa (*Fall, 1897*: 237; *Fall, 1934*: 143; *Cockerell, 1940*: 285; *Miller, 1985a*: 20)

Digitized Records: Anacapa (12 SBMNH), San Miguel (37 SBMNH), Santa Cruz (11 LACM; 57 SBMNH), Santa Rosa (1 LACM; 35 SBMNH)

Range: Endemic (*Fall, 1934*; *Cockerell, 1940*; *Miller, 1985a*).

Notes. *Falls' (1897)* record from Santa Rosa Island was reported as *Pristoscelis aenescens* [= *Trichochrous aenescens* (LeConte, 1852)] and almost certainly represents *T. calcaratus*. *LeConte (1866c*: 355) recorded *P. aenescens* from "San Diego and the Islands off Santa Barbara". *Miller & Miller (1985*: 132) reported that these records are in error.

The populations on the four northern islands may each represent different endemic species (M. L. Gimmel, 2022, unpublished data).

### *Trichochrous pedalis* (LeConte, 1866)

Nomenclatural Authority: M. L. Gimmel & A. Mayor, 2023, unpublished data

Literature Records: Santa Catalina (*LeConte, 1866c*: 355; *Casey, 1895*: 529; *Fall, 1897*: 237; *Falls' 1901*: 126; *Cockerell, 1940*: 285; *Miller, 1985a*: 20)

Digitized Records: Santa Catalina (9 LACM; 3 SBMNH)

Range: Also known from mainland (M. L. Gimmel & A. Mayor, 2022, unpublished data).

Notes. *LeConte (1866c)* and *Fall (1897)* recorded this species as *Pristoscelis pedalis*; it was transferred to *Trichochrous* by *Casey (1895)*. Records of this species from San Clemente by *Fall (1897*: 237), *Cockerell (1940*: 285), and *Miller (1985a*: 20) refer to the "undescribed species near *pedalis*" below. *Seavey's (1892*: 262) record of *Pristoscelis quadricollis* [= *Trichochrous quadricollis* (LeConte, 1859)] from Santa Catalina is in error and almost certainly represents this species; he reported that it was collected from *Heteromeles arbutifolia*. *Fall (1897*: 235) also doubted the validity of this identification. All prior authors have considered *T. pedalis* to be endemic to Santa Catalina Island; however, individuals from a population occurring in the Palos Verdes Hills in mainland Los Angeles County have been collected which are not diagnosable from the island populations, neither externally nor using male genitalia, which are otherwise diagnostic within the genus (M. L. Gimmel & A. Mayor, 2022, unpublished data).

### *Trichochrous* undescribed species 1 near *brevicornis*
Literature Records: none
Digitized Records: Anacapa (1 SBMNH), San Miguel (1 LACM) Santa Cruz (4 SBMNH), Santa Rosa (12 SBMNH; 9 LACM)
Range: Endemic (M. L. Gimmel & A. Mayor, 2022, unpublished data).
Notes. In our record set, the identifications of this new species are listed merely as "*Trichochrous*".

### *Trichochrous* undescribed species 2 near *brevicornis*
Literature Records: none
Digitized Records: San Nicolas (19 SBMNH; 1 LACM)
Range: Endemic (M. L. Gimmel & A. Mayor, 2022, unpublished data).
Notes. In our record set, the identifications of this new species are listed merely as "*Trichochrous*".

### *Trichochrous* undescribed species near *pedalis*
Literature Records: San Clemente (*Fall, 1897*: 237; *Cockerell, 1940*: 285; *Miller, 1985a*: 20), Santa Barbara (*Fall, 1897*: 237)
Digitized Records: San Clemente (35 SBMNH), San Nicolas (40 SBMNH; 58 LACM), Santa Barbara (18 LACM)
Range: Endemic (M. L. Gimmel & A. Mayor, 2022, unpublished data).
Notes. The San Clemente Island records were recorded as *T. pedalis* by *Fall (1897)*, *Cockerell (1940)*, and *Miller (1985a)*. *Fall's (1897)* record from Santa Barbara Island was of *Pristoscelis aenescens* [= *Trichochrous aenescens* (LeConte, 1852)] but almost certainly represents this species. In our record set, the identifications of this new species are listed merely as "*Trichochrous*".

### Malachiinae
Notes. Five tribes, 10 genera, and 91 species of Malachiinae are known to occur in California (M. L. Gimmel, 2022, unpublished data).

**Apalochrini**

Notes. One genus and 21 species of Apalochrini are known to occur in California (M. L. Gimmel, 2022, unpublished data).

***Collops* Erichson, 1840**

Nomenclatural Authority: A. Mayor (2022, unpublished catalog)

Digitized Records (genus-only): San Nicolas (1 SBMNH)

Notes. Twenty-one species of *Collops* have been recorded from California (A. Mayor, 2022, unpublished catalog). The genus-only record from San Nicolas Island above refers to a larval specimen.

***Collops cribrosus* LeConte, 1852**

Nomenclatural Authority: A. Mayor (2022, unpublished catalog)

Literature Records: Santa Rosa (*Fall, 1897*: 237)

Digitized Records: San Miguel (1 SBMNH), Santa Cruz (32 SBMNH; 1 UCSB), Santa Rosa (11 SBMNH)

Range: Also known from mainland (A. Mayor, 2022, unpublished catalog).

Notes. *Fall (1912)* reported that this species lacks hind wings, but observation of specimens in SBMNH reveal that hind wings are present in at least most specimens of both sexes from both island and mainland populations. The wings extend about two-thirds the length of the elytra (M. L. Gimmel, 2021, personal observation).

***Collops crusoe* Fall, 1910 (Fig. 5)**

Nomenclatural Authority: A. Mayor (2022, unpublished catalog)

Literature Records: San Nicolas (*Fall, 1910*: 140; *Fall, 1912*: 269; *Cockerell, 1940*: 285; *Miller, 1985a*: 20), Santa Cruz (*Miller, 1985a*: 20), Santa Rosa (*Miller, 1985a*: 20)

Digitized Records: San Miguel (2 LACM), San Nicolas (60 LACM; 15 SBMNH; 3 iNat), Santa Cruz (2 LACM), Santa Rosa (1 LACM)

Range: Endemic (*Fall, 1910*, *1912*; *Cockerell, 1940*; *Miller, 1985a*).

Notes. *Fall (1912)* reported that this species lacks hind wings. However, specimens in SBMNH appear to have poorly developed hind wings in both sexes, extending about half the length of the elytra (M. L. Gimmel, 2021, personal observation). Almost certainly they are flightless.

***Collops vittatus* (Say, 1823)**

Nomenclatural Authority: A. Mayor (2022, unpublished catalog)

Literature Records: none

Digitized Records: Santa Catalina (3 SBMNH)

Range: Also known from mainland (A. Mayor, 2022, unpublished catalog).

Notes. Island specimens of both sexes of this species present in SBMNH are fully winged (M. L. Gimmel, 2021, personal observation).

**Attalini**

Notes. Three genera and 38 species of Attalini are known to occur in California (M. L. Gimmel, 2022, unpublished data).

*Attalus* **Erichson, 1840**
Nomenclatural Authority: A. Mayor (2022, unpublished catalog)
Notes. Seventeen described species of *Attalus* are known from California (A. Mayor, 2022, unpublished catalog). The two species below may prove to belong to the genus *Attalusinus* Leng, 1918 (tribe Troglopini) upon further study. The latter genus contains one described species in California (A. Mayor, 2022, unpublished catalog).

*Attalus transmarinus* **Fall, 1898**
Nomenclatural Authority: A. Mayor (2022, unpublished catalog)
Literature Records: San Clemente (*Fall, 1897*: 237; *Fall, 1901*: 125; *Miller, 1985a*: 20)
Digitized Records: none
Range: Endemic (*Fall, 1897, 1901*; *Miller, 1985a*).
Notes. *Fall (1897)* originally recorded this species as *Attalus subfasciatus* Fall, 1897; this was, however, discovered to be a homonym and replaced with *A. transmarinus* by *Fall (1898b*: 267). *Champion (1914*: 65) subsequently and unnecessarily proposed *Attalus falli* Champion, 1914 as a replacement name.

*Attalus* **undescribed species**
Nomenclatural Authority: A. Mayor (2021, personal communication)
Literature Records: none
Digitized Records: San Clemente (1 iNat)
Range: Endemic (A. Mayor, 2021, personal communication).
Notes. This record is based on the specimen of *Attalus* on iNaturalist here (www.inaturalist.org/observations/71409011), which has been confirmed to belong to an undescribed species (A. Mayor, 2021, personal communication).

*Endeodes* **LeConte, 1859**
Nomenclatural Authority: A. Mayor (2022, unpublished catalog)
Notes. The genus *Endeodes* contains three species recorded from California (A. Mayor, 2022, unpublished catalog), all of which are flightless and beach-dwelling, and all of which are represented on the Channel Islands.

*Endeodes basalis* **(LeConte, 1852)**
Nomenclatural Authority: *Moore & Legner (1975)*
Literature Records: Santa Catalina (*Fall, 1897*: 237)
Digitized Records: San Clemente (6 SBMNH), San Miguel (15 SBMNH), San Nicolas (14 SBMNH), Santa Cruz (14 SBMNH), Santa Rosa (9 SBMNH)
Range: Also known from mainland (*Moore & Legner, 1975*).
Notes. This species was reported by *Fall (1897)* as *Endeodes abdominalis* (LeConte, 1852), which is now considered a junior synonym of *E. basalis* (see *Moore & Legner, 1975*). Some SBMNH specimens from San Miguel and Santa Rosa are nearly all-black (M. L. Gimmel, 2021, personal observation).

*Endeodes collaris* **(LeConte, 1852)**
Nomenclatural Authority: *Moore & Legner (1975)*

Literature Records: Santa Catalina (*Fall, 1901*: 124), Santa Rosa (*Fall, 1897*: 237)

Digitized Records: Santa Rosa (1 SBMNH)

Range: Also known from mainland (*Fall, 1901*; *Moore & Legner, 1975*).

Notes. *Fall (1897*: 240) noted an undetermined *Endeodes* from Santa Rosa that was based on one almost entirely black specimen, which may be a color variety of *E. collaris*.

### *Endeodes insularis* Blackwelder, 1932

Nomenclatural Authority: *Moore & Legner (1975)*

Literature Records: San Miguel (*Blackwelder, 1932*: 134; *Moore, 1954*: 198; *Moore & Legner, 1975*: 80), Santa Catalina (*Fall, 1897*: 237)

Digitized Records: San Nicolas (1 SBMNH), Santa Rosa (5 SBMNH)

Range: Also known from mainland (*Moore, 1954*).

Notes. The Santa Catalina record from *Fall (1897)* is based on a specimen cited as an undetermined *Endeodes* taken during July and possessing very minute elytra; this almost certainly refers to *E. insularis*. This species was originally thought to be endemic to the islands until it was collected on the mainland in Gaviota State Park, Santa Barbara County by *Moore (1954)*.

### Ebaeini

Notes. Three genera and five species of Ebaeini are known to occur in California (M. L. Gimmel, 2022, unpublished data).

### *Charopus* Erichson, 1840

Nomenclatural Authority: A. Mayor (2022, unpublished catalog)

Notes. The genus *Charopus* contains two described species known from California (A. Mayor, 2022, unpublished catalog), plus the undescribed species below.

### *Charopus* undescribed species

Nomenclatural Authority: A. Mayor (2021, personal communication)

Literature Records: Santa Catalina (*Fall, 1917*: 78)

Digitized Records: Santa Cruz (3 SBMNH)

Range: Also known from mainland (A. Mayor, 2021, personal communication).

Notes. Recorded by *Fall (1917)* as *Microlipus longicollis* (now *Charopus longicollis* Motschulsky, 1860). However, according to A. Mayor (2021, personal communication), the island populations represent an undescribed species of *Charopus* that also occurs on the mainland.

### Malachiini

Notes. Two genera and 26 species of Malachiini have been recorded from California (M. L. Gimmel, 2022, unpublished data).

### "*Malachius*" Fabricius, 1775

Nomenclatural Authority: A. Mayor (2022, unpublished catalog)

Notes. Twenty-one species of *Malachius* have been reported from California (A. Mayor, 2022, unpublished catalog). All native North American species currently placed in this genus will soon be reassigned to *Hapalorhinus* LeConte, 1859.

**"*Malachius*" undetermined species**
Literature Records: Santa Rosa (*Fall, 1897*: 237)
Digitized Records: none
Notes. The genus *Malachius* has not been recorded from the Channel Islands since *Fall's (1897)* record of "*Malachius*, sp. nov.?" from Santa Rosa (A. Mayor, 2021, personal communication). Fall's original specimens, if still extant, should be reexamined.

**Microlipus LeConte, 1852**
Nomenclatural Authority: A. Mayor (2022, unpublished catalog)
Notes. Five species of the genus *Microlipus* have been reported from California (A. Mayor, 2022, unpublished catalog).

**Microlipus laticeps LeConte, 1852**
Nomenclatural Authority: A. Mayor (2022, unpublished catalog)
Literature Records: none
Digitized Records: San Miguel (1 SBMNH), Santa Catalina (1 SBMNH), Santa Cruz (6 SBMNH; 1 UCSB), Santa Rosa (13 SBMNH)
Range: Also known from mainland (A. Mayor, 2022, unpublished catalog).

**Trogossitidae**
Notes. The family Trogossitidae, as delimited by *Gimmel et al. (2019)*, contains two subfamilies, seven genera, and 26 species in California (*Barron, 1971*; M. L. Gimmel, 2022, unpublished data). These species were keyed by *Barron (1971)*.

**Trogossitinae**
Notes. Six genera and 24 species of Trogossitinae are known to occur in California (*Barron, 1971*; M. L. Gimmel, 2022, unpublished data).

**Temnoscheila Westwood, 1830**
Nomenclatural Authority: *Kolibáč (2013)*
Notes. The genus *Temnoscheila*, often misspelled *Temnochila*, contains six species in California (*Barron, 1971*).

**Temnoscheila chlorodia (Mannerheim, 1843)**
Nomenclatural Authority: *Barron (1971)*, *Kolibáč (2013)*
Literature Records: Santa Cruz (*Barron, 1971*: 84)
Digitized Records: Santa Cruz (1 LACM; 1 SBMNH; 6 UCSB)
Range: Also known from mainland (*Barron, 1971*).
Notes. *Barron (1971)* recorded this species as *Temnochila chlorodia*.

**Tenebroides Piller & Mitterpacher, 1783**
Nomenclatural Authority: *Kolibáč (2013)*

Notes. The genus *Tenebroides* contains six species in California (M. L. Gimmel, 2022, unpublished data).

### *Tenebroides crassicornis* (Horn, 1862)
Nomenclatural Authority: *Barron (1971)*, *Kolibáč (2013)*
Literature Records: Santa Cruz (*Barron, 1971*: 97)
Digitized Records: none
Range: Also known from mainland (*Barron, 1971*).

### *Tenebroides occidentalis* Fall, 1910
Nomenclatural Authority: *Barron (1971)*, *Kolibáč (2013)*
Literature Records: none
Digitized Records: Santa Cruz (9 SBMNH; 2 UCSB)
Range: Also known from mainland (*Barron, 1971*).

## TENEBRIONOIDEA

### Anthicidae
Notes. Three subfamilies, 20 genera, and 99 species of Anthicidae are known to occur in California (M. L. Gimmel, 2022, unpublished data).

### Anthicinae
Notes. Twelve genera and 53 species of Anthicinae are known to occur in California (M. L. Gimmel, 2022, unpublished data).

### *Amblyderus* LaFerté-Sénectère, 1849
Nomenclatural Authority: *Chandler (2002)*
Notes. Two species of *Amblyderus* have been recorded from California (*Chandler, 1999*). The species of this genus were treated for North America by *Chandler (1999)*.

### *Amblyderus obesus* Casey, 1895
Nomenclatural Authority: *Chandler (1999)*
Literature Records: San Nicolas (*Chandler, 1999*: 282)
Digitized Records: San Miguel (1 SBMNH), San Nicolas (2 SBMNH)
Range: Also known from mainland (*Chandler, 1999*).

### *Amblyderus parviceps* Casey, 1895
Nomenclatural Authority: *Chandler (1999)*
Literature Records: none
Digitized Records: Santa Cruz (5 SBMNH), Santa Rosa (2 SBMNH)
Range: Also known from mainland (*Chandler, 1999*).

### *Anthicus* Paykull, 1798
Nomenclatural Authority: *Chandler (2002)*
Notes. Twenty-one species of *Anthicus* are known to occur in California (M. L. Gimmel, 2022, unpublished data). The species were treated for North America by *Werner (1964)*.

***Anthicus cribratus*** **LeConte, 1851**
Nomenclatural Authority: *Werner (1964)*
Literature Records: Santa Cruz (*Werner, 1964*: 215)
Digitized Records: Santa Cruz (11 SBMNH)
Range: Also known from mainland (*Werner, 1964*).

***Anthicus maritimus*** **LeConte, 1851**
Nomenclatural Authority: *Werner (1964)*
Literature Records: none
Digitized Records: San Nicolas (9 SBMNH)
Range: Also known from mainland (*Werner, 1964*).

***Anthicus nanus*** **LeConte, 1851**
Nomenclatural Authority: *Werner (1964)*
Literature Records: none
Digitized Records: Santa Catalina (3 SBMNH), Santa Cruz (10 SBMNH)
Range: Also known from mainland (*Werner, 1964*).

***Anthicus punctulatus*** **LeConte, 1851**
Nomenclatural Authority: *Werner (1964)*
Literature Records: none
Digitized Records: San Clemente (3 SBMNH), Santa Catalina (9 SBMNH), Santa Cruz (2 SBMNH)
Range: Also known from mainland (*Werner, 1964*).

***Anthicus rufulus*** **LeConte, 1851**
Nomenclatural Authority: *Werner (1964)*
Literature Records: none
Digitized Records: Santa Catalina (5 SBMNH)
Range: Also known from mainland (*Werner, 1964*).

***Cyclodinus*** **Mulsant & Rey, 1866**
Nomenclatural Authority: *Chandler (2002)*
Notes. Five species of *Cyclodinus* have been recorded from California (*Chandler, 2005*). The genus was revised for the New World by *Chandler (2005)*.

***Cyclodinus annectens*** **(LeConte, 1851)**
Nomenclatural Authority: *Chandler (2005)*
Literature Records: San Clemente (*Fall, 1897*: 239; *Fall, 1901*: 181), Santa Catalina (*Fall, 1897*: 239; *Fall, 1901*: 181; *Chandler, 2005*: 7)
Digitized Records: Santa Catalina (1 SBMNH)
Range: Also known from mainland (*Fall, 1901*; *Chandler, 2005*).
Notes. *Fall (1897*, *1901)* recorded this species as *Anthicus californicus* LaFerté-Senéctère, 1849. However, this name actually applies to an eastern North American species now

placed in the genus *Cyclodinus* Mulsant & Rey, 1866 (see *Chandler, 2005*). The Channel Island records of this species almost certainly pertain to *C. annectens*.

### *Ischyropalpus* LaFerté-Sénectère, 1849
Nomenclatural Authority: *Chandler (2002)*

Notes. Five species of *Ischyropalpus* are known to occur in California (M. L. Gimmel, 2022, unpublished data). The species of this genus were revised for North America by *Werner (1973)*.

### *Ischyropalpus nitidulus* (LeConte, 1851)
Nomenclatural Authority: *Werner (1973)*

Literature Records: San Clemente (*Fall, 1897*: 239), Santa Catalina (*Fall, 1897*: 239; *Werner, 1973*: 1060), Santa Cruz (*Werner, 1973*: 1060)

Digitized Records: Anacapa (1 LACM), Santa Catalina (4 LACM), Santa Cruz (20 SBMNH)

Range: Also known from mainland (*Werner, 1973*).

Notes. *Fall (1897)* originally reported this species as "*Anthicus* sp.", but later redetermined the same specimens in the Museum of Comparative Zoology, Harvard University to *Ischyropalpus sturmi* (LaFerté-Sénectère, 1849) (S. Miller, 2022, personal communication), California specimens of which are now considered to be *I. nitidulus*.

### *Omonadus* Mulsant & Rey, 1866
Nomenclatural Authority: *Chandler (2002)*

Notes. Two adventive species of *Omonadus* have been recorded from California (*Werner, 1964*). The species were revised for North America by *Werner (1964*; as part of *Anthicus*).

### *Omonadus floralis* (Linnaeus, 1758)
Nomenclatural Authority: *Werner (1964)*, *Chandler (2002)*

Literature Records: Santa Catalina (*Cockerell, 1940*: 285)

Digitized Records: none

Range: Also known from mainland (*Cockerell, 1940*; *Werner, 1964*).

Notes. This species is introduced from the Old World (*Chandler, 2002*). *Cockerell (1940)* recorded it as *Hemantus floralis*, and *Werner (1964)* treated it as *Anthicus floralis*.

### Notoxinae
Notes. Two genera and 24 species of Notoxinae have been recorded from California (*Chandler, 1983*; M. L. Gimmel, 2022, unpublished data).

### *Notoxus* Geoffroy, 1762
Nomenclatural Authority: *Chandler (2002)*

Digitized Records (genus-only): Santa Catalina (8 LACM), Santa Rosa (1 LACM; 5 SBMNH)

Notes. Twenty-three species of *Notoxus* have been recorded from California (*Chandler, 1983*). The North American species were revised by *Chandler (1983)*.

*Notoxus desertus* **Casey, 1895**
Nomenclatural Authority: *Chandler (1983)*
Literature Records: Santa Catalina (*Fall, 1897*: 238)
Digitized Records: Santa Catalina (8 SBMNH), Santa Cruz (11 SBMNH), Santa Rosa (1 SBMNH)
Range: Also known from mainland (*Chandler, 1983*).
Notes. This species was recorded as *Notoxus constrictus* Casey, 1895 by *Fall (1897)*, which is now treated as a junior synonym of *N. desertus* (see *Chandler, 1983*: 354).

*Notoxus sparsus* **LeConte, 1859**
Nomenclatural Authority: *Chandler (1983)*
Literature Records: Santa Cruz (*Chandler, 1983*: 395)
Digitized Records: none
Range: Also known from mainland (*Chandler, 1983*).

**Ciidae**
Notes. Two subfamilies, nine genera, and 27 species of Ciidae have been recorded from California (*Lawrence, 1982*). The North American species were monographed by *Lawrence (1971)*, treated for California by *Lawrence (1974)*, and cataloged for North America by *Lawrence (1982)*. *Lopes-Andrade et al. (2016)* made additional taxonomic changes affecting the California species.

**Ciinae: Ciini**
Notes. Two tribes, eight genera, and 26 species of Ciinae, seven genera and 25 species belonging to Ciini, have been recorded from California (*Lawrence, 1982*).

*Ceracis* **Mellié, 1848**
Nomenclatural Authority: *Lawrence (1982)*
Notes. Two species of *Ceracis* have been recorded from California (*Lawrence, 1982*).

*Ceracis californicus* **(Casey, 1884)**
Nomenclatural Authority: *Lawrence (1971)*, *Lawrence (1974)*
Literature Records: none
Digitized Records: Santa Rosa (1 SBMNH)
Range: Also known from mainland (*Lawrence, 1971*, *1974*).
Notes. This widespread species has not been reported from the islands in the literature, but is well-known from coastal California and the western half of North America (*Lawrence, 1971*, *1974*).

*Cis* **Latreille, 1796**
Nomenclatural Authority: *Lawrence (1982)*
Notes. Seventeen species of *Cis* have been recorded from California (*Lawrence, 1982*).

*Cis* **undetermined species**
Literature Records: Santa Catalina (*Fall, 1897*: 238)
Digitized Records: none

Notes. *Fall (1897)* recorded "*Cis* sp." from Santa Catalina. There are 17 species of this genus reported from California (M. L. Gimmel, 2022, unpublished data), but we know of no additional specimens or literature records from the Channel Islands.

### *Hadreule* Thomson, 1859
Nomenclatural Authority: *Lopes-Andrade et al. (2016)*
Notes. One species of *Hadreule* has been recorded from California (*Lawrence, 1982*). In many publications, including those of *Lawrence (1971*, *1974*, *1982)*, the genus has been misspelled and misattributed as *Hadraule* Thomson, 1863, but the proper spelling and attribution are *Hadreule* Thomson, 1859 (see *Lopes-Andrade et al., 2016*: 359).

### *Hadreule blaisdelli* (Casey, 1900)
Nomenclatural Authority: *Lawrence (1971)*, *Lawrence (1974)*
Literature Records: none
Digitized Records: Santa Cruz (1 SBMNH)
Range: Also known from mainland (*Lawrence, 1971*, *1974*).
Notes. This widespread species and occasional herbarium pest has not been reported from the islands in the literature, but it is well documented from California and the rest of North America (*Lawrence, 1971*, *1974*).

### *Orthocis* Casey, 1898
Nomenclatural Authority: *Lawrence (1982)*
Notes. One species of *Orthocis* has been recorded from California (*Lawrence, 1982*).

### *Orthocis punctatus* (Mellié, 1848)
Nomenclatural Authority: *Lawrence (1971)*, *Lawrence (1974)*
Literature Records: none
Digitized Records: Santa Catalina (2 SBMNH), Santa Rosa (7 SBMNH)
Range: Also known from mainland (*Lawrence, 1971*, *1974*).
Notes. This widespread species has not been reported from the islands in the literature, but the town of Santa Barbara was cited as a known locality by *Lawrence (1971*: 486, *1974*: 19).

### *Sulcacis* Dury, 1917
Nomenclatural Authority: *Lawrence (1982)*
Notes. One species of *Sulcacis* has been recorded from California (*Lawrence, 1982*).

### *Sulcacis curtulus* (Casey, 1898)
Nomenclatural Authority: *Lawrence (1971)*, *Lawrence (1974)*
Literature Records: none
Digitized Records: Santa Cruz (1 SBMNH)
Range: Also known from mainland (*Lawrence, 1971*, *1974*).
Notes. This widespread species has not been reported from the islands in the literature, but it is well documented from coastal California and across North America (*Lawrence, 1971*, *1974*).

**Meloidae**

Notes. Two subfamilies, 18 genera, and 122 species of Meloidae are known to occur in California (M. L. Gimmel, 2022, unpublished data).

**Meloinae**

Notes. Twelve genera and 85 species of Meloinae are known to occur in California (M. L. Gimmel, 2022, unpublished data).

*Cordylospasta* **Horn, 1875**

Nomenclatural Authority: *Pinto (1972)*

Notes. Two species of *Cordylospasta* have been recorded from California (*Pinto, 1972*). The genus was revised by *Pinto (1972)*.

*Cordylospasta opaca* **(Horn, 1868)**

Nomenclatural Authority: *Pinto (1972)*

Literature Records: Santa Cruz (*Pinto, 1972*: 1170)

Digitized Records: none

Range: Also known from mainland (*Pinto, 1972*).

*Epicauta* **Dejean, 1834**

Nomenclatural Authority: *Pinto (1991)*

Notes. *Pinto (1991)* provided keys to all North and Central American species of this large genus, of which 25 species have been recorded from California in two subgenera, *Epicauta* (*s.str.*) and *Macrobasis* LeConte, 1862 (M. L. Gimmel, 2022, unpublished data).

*Epicauta* (*Epicauta*) *puncticollis* **Mannerheim, 1843**

Nomenclatural Authority: *Pinto (1991)*

Literature Records: Santa Rosa (*Ballmer, 1980*: 79 [map])

Digitized Records: Santa Rosa (21 LACM; 2 SBMNH)

Range: Also known from mainland (*Ballmer, 1980*; *Pinto, 1991*).

*Lytta* **Fabricius, 1775**

Nomenclatural Authority: *Selander (1960)*

Notes. Thirty species of *Lytta* have been recorded from California, belonging to the subgenera *Adicolytta* Selander, 1960, *Paralytta* Selander, 1960, and *Poreospasta* Horn, 1868 (*Selander, 1960*). The species of this genus were monographed by *Selander (1960)*.

*Lytta* (*Poreospasta*) *stygica* **(LeConte, 1851)**

Nomenclatural Authority: *Selander (1960)*

Literature Records: none

Digitized Records: San Clemente (1 SBMNH)

Range: Also known from mainland (*Selander, 1960*).

*Meloe* **Linnaeus, 1758**

Nomenclatural Authority: *Pinto & Selander (1970)*

Literature Records (genus-only): Santa Catalina (*Fall, 1897*: 239)

Notes. Eleven species of *Meloe* have been recorded from California, belonging to the subgenera *Meloe* (*s.str.*) and *Treiodous* Dugès, 1869 (M. L. Gimmel, 2022, unpublished data). The species of this genus were monographed for the New World by *Pinto & Selander (1970)*. Although listed separately as "*Meloe* sp.", *Fall's (1897)* record probably refers to *M. barbarus*.

### *Meloe* (*Meloe*) *strigulosus* Mannerheim, 1852
Nomenclatural Authority: *Pinto & Selander (1970)*
Literature Records: San Miguel (*Cockerell, 1940*: 285; *Pinto & Selander, 1970*: 159)
Digitized Records: San Miguel (2 SBMNH)
Range: Also known from mainland (*Pinto & Selander, 1970*).
Notes. Triungulin larvae of this species have been reported as phoretic both on flower-visiting bee hosts and other taxa such as flies to travel from flower to flower (*Pinto et al., 2020*). It is worth noting that the single known island that this species inhabits is the only island where *M. barbarus* is not known to occur.

### *Meloe* (*Treiodous*) *barbarus* LeConte, 1861
Nomenclatural Authority: *Pinto & Selander (1970)*
Literature Records: San Clemente (*Van Dyke, 1928*: 445; *Pinto & Selander, 1970*: 120; *Huether & Huether, 2015*: 162; *Miller & Miller, 1985*: 128), San Nicolas (*Miller & Miller, 1985*: 128), Santa Barbara (*LeConte, 1861*: 354; *Van Dyke, 1928*: 445; *Fall, 1897*: 239; *Fall, 1901*: 183; *Cockerell, 1940*: 285; *Pinto & Selander, 1970*: 120; *Miller & Miller, 1985*: 128), Santa Catalina (*Pinto & Selander, 1970*: 120; *Miller & Miller, 1985*: 128), Santa Cruz (*Miller & Miller, 1985*: 128)
Digitized Records: San Clemente (3 LACM), San Nicolas (1 SBMNH), Santa Catalina (2 LACM), Santa Cruz (1 UCRC), Santa Rosa (1 UCRC)
Range: Also known from mainland (*Cockerell, 1940*; *Pinto & Selander, 1970*).
Notes. This species, originally described from Santa Barbara Island and thought to be endemic, was shown to be widespread in the Pacific coast states by *Pinto & Selander (1970)*.

### Mordellidae
Notes. Two tribes, four genera, and twenty-six species of Mordellidae are known to occur in California (*Bright, 1986*; M. L. Gimmel, 2022, unpublished data). The family was revised for North America by *Liljeblad (1945)* and cataloged for North America by *Bright (1986)*. However, generic concepts around *Mordellistena* have changed significantly since those publications.

### Mordellini
Notes. One genus and three species of Mordellini have been recorded from California (*Bright, 1986*).

### *Mordella* Linnaeus, 1758
Nomenclatural Authority: *Bright (1986)*
Notes. Three species of *Mordella* have been recorded from California (*Bright, 1986*).

### *Mordella albosuturalis* Liljeblad, 1922
Nomenclatural Authority: *Liljeblad (1945)*
Literature Records: none
Digitized Records: Santa Cruz (1 SBMNH)
Range: Also known from mainland (*Liljeblad, 1945*).

### *Mordella hubbsi* Liljeblad, 1922
Nomenclatural Authority: *Liljeblad (1945)*
Literature Records: Santa Cruz (*Naughton et al., 2014*: 304)
Digitized Records: Santa Cruz (2 LACM; 4 SBMNH)
Range: Also known from mainland (*Liljeblad, 1945*).

### Mordellistenini
Notes. Three genera and 23 species of Mordellistenini are known to occur in California (*Bright, 1986*).

### *Mordellina* Schilsky, 1908
Nomenclatural Authority: *Lisberg (2003)*
Notes. Six species now placed in *Mordellina* have been recorded from California (*Bright, 1986*; *Lisberg, 2003*).

### *Mordellina* undetermined species
Literature Records: none
Digitized Records: San Clemente (10 SBMNH), San Miguel (6 LACM), San Nicolas (9 SBMNH), Santa Barbara (3 SBMNH), Santa Catalina (1 SBMNH), Santa Cruz (4 SBMNH), Santa Rosa (5 SBMNH)
Notes. Characters used to separate putative species of *Mordellina* from *Mordellistena* in the SBMNH collection were those outlined by *Lisberg (2003)*, who noted that additional species require transfer from *Mordellistena* to *Mordellina*. Because of this issue and the lack of a modern revision, no attempt was made to determine these specimens to species.

### *Mordellistena* Costa, 1854
Nomenclatural Authority: *Bright (1986)*
Notes. Sixteen species currently placed in *Mordellistena* are known to occur in California (M. L. Gimmel, 2022, unpublished data).

### *Mordellistena* undetermined species
Literature Records: Santa Barbara (*Miller & Miller, 1985*: 128), Santa Catalina (*Fall, 1897*: 238), Santa Cruz (*Naughton et al., 2014*: 304)
Digitized Records: Anacapa (6 SBMNH), San Clemente (1 SBMNH), San Nicolas (1 SBMNH), Santa Barbara (2 SBMNH), Santa Catalina (7 SBMNH), Santa Cruz (3 SBMNH), Santa Rosa (2 SBMNH)
Notes. *Fall (1897)* listed two separate undetermined species of *Mordellistena* from Santa Catalina; *Miller & Miller (1985)* indicated that the Santa Barbara Island species belongs to a

group including *Mordellistena nubila* (LeConte, 1858), *Mordellistena ruficeps* LeConte, 1862, and *Mordellistena subfucus* Liljeblad, 1945. MLG notes that there are at least four morphospecies of *Mordellistena* represented among island material in SBMNH, including an undescribed apterous species based on four specimens from San Clemente (1), San Nicolas (1), and Santa Barbara (2) islands.

**Mycetophagidae, NEW FAMILY RECORD**
Notes. Two subfamilies, four genera, and seven species of Mycetophagidae have been recorded from California (*Parsons, 1975*). The North American fauna was revised by *Parsons (1975)*.

**Mycetophaginae**
Notes. Three genera and six species of Mycetophaginae have been recorded from California (*Parsons, 1975*).

***Litargus* Erichson, 1846**
Nomenclatural Authority: *Young (2002)*
Notes. One species of *Litargus* has been recorded from California (*Parsons, 1975*).

***Litargus balteatus* LeConte, 1856**
Nomenclatural Authority: *Parsons (1975)*
Literature Records: none
Digitized Records: Santa Cruz (4 SBMNH), Santa Rosa (6 SBMNH)
Range: Also known from mainland (*Parsons, 1975*).
Notes. This is a cosmopolitan species (*Parsons, 1975*), but is presumably native to North America.

***Mycetophagus* Hellwig, 1792**
Nomenclatural Authority: *Young (2002)*
Notes. Four species of *Mycetophagus* have been recorded from California, belonging to three subgenera, *Gratusus* Casey, 1900, *Mycetophagus* (*s.str.*), and *Parilendus* Casey, 1900 (*Parsons, 1975*).

***Mycetophagus* (*Gratusus*) *pluriguttatus* LeConte, 1856**
Nomenclatural Authority: *Parsons (1975)*
Literature Records: none
Digitized Records: Santa Cruz (4 SBMNH), Santa Rosa (3 SBMNH)
Range: Also known from mainland (*Parsons, 1975*).

***Typhaea* Curtis, 1830**
Nomenclatural Authority: *Young (2002)*
Notes. One species of *Typhaea* has been recorded from California (*Parsons, 1975*).

***Typhaea stercorea* (Linnaeus, 1758)**
Nomenclatural Authority: *Parsons (1975)*
Literature Records: none

Digitized Records: Santa Catalina (1 iNat)
Range: Also known from mainland (*Parsons, 1975*).
Notes. This is a cosmopolitan species (*Parsons, 1975*), presumably Palearctic in origin.

## Mycteridae
Notes. Three subfamilies, three genera, and six species of Mycteridae have been recorded from California (M. L. Gimmel, 2022, unpublished data).

## Eurypinae
Notes. One genus and species of Eurypinae has been recorded from California (*Pollock & Majka, 2012*).

### *Lacconotus* LeConte, 1862
Nomenclatural Authority: *Pollock & Majka (2012)*
Notes. One species of *Lacconotus* has been recorded from California, belonging to the subgenus *Alcconotus* Pollock & Majka, 2012 (*Pollock & Majka, 2012*). The North American species were reviewed by *Pollock & Majka (2012)*.

### *Lacconotus* (*Alcconotus*) *pinicola* Horn, 1879
Nomenclatural Authority: *Pollock & Majka (2012)*
Literature Records: Santa Catalina (*Pollock & Majka, 2012*: 21), Santa Cruz (*Pollock & Majka, 2012*: 22; *Naughton et al., 2014*: 304)
Digitized Records: Santa Catalina (2 SBMNH), Santa Cruz (5 SBMNH)
Range: Also known from mainland (*Pollock & Majka, 2012*).

## Oedemeridae, NEW FAMILY RECORD
Notes. Two subfamilies, 13 genera, and 33 species of Oedemeridae are known to occur in California (M. L. Gimmel, 2022, unpublished data). Most of the North American fauna of Oedemeridae was treated by *Arnett (1951)*.

## Oedemerinae
Notes. Eleven genera and 31 species of Oedemerinae are known to occur in California (M. L. Gimmel, 2022, unpublished data).

## Asclerini
Notes. Eight genera and 24 species of Asclerini are known to occur in California (M. L. Gimmel, 2022, unpublished data).

### *Copidita* LeConte, 1866
Nomenclatural Authority: *Arnett (1951)*
Notes. One species of *Copidita* has been recorded from California (*Arnett, 1951*).

### *Copidita quadrimaculata* (Motschulsky, 1853)
Nomenclatural Authority: *Arnett (1951)*
Literature Records: none

Digitized Records: San Clemente (2 SBMNH), San Miguel (2 LACM; 16 SBMNH), San Nicolas (3 LACM; 1 SBMNH), Santa Catalina (4 LACM; 1 SBMNH), Santa Cruz (33 LACM; 3 SBMNH), Santa Rosa (1 LACM; 8 SBMNH)

Range: Also known from mainland (*Arnett, 1951*).

Notes. This is a widespread beach-dwelling species on the Pacific coast of North America (*Arnett, 1951*).

### Nacerdini

Notes. Two genera and five species of Nacerdini have been recorded from California (*Arnett, 1951*).

### *Nacerdes* Dejean, 1834

Nomenclatural Authority: *Arnett (1951)*

Notes. One species of *Nacerdes* has been recorded from California (*Arnett, 1951*).

### *Nacerdes melanura* (Linnaeus, 1758)

Nomenclatural Authority: *Arnett (1951)*

Literature Records: none

Digitized Records: Santa Catalina (1 LACM)

Range: Also known from mainland (*Arnett, 1951*).

Notes. Known as the wharf borer, this species is adventive in North America, and probably occurs on coasts worldwide (*Arnett, 1951*).

### *Xanthochroa* Schmidt, 1846

Nomenclatural Authority: *Arnett (1951)*

Notes. Four species of *Xanthochroa* have been recorded from California (*Arnett, 1951*).

### *Xanthochroa marina* Horn, 1896

Nomenclatural Authority: *Arnett (1951)*

Literature Records: none

Digitized Records: Santa Cruz (8 SBMNH)

Range: Also known from mainland (*Arnett, 1951*).

Notes. This species, described from Marin County, is known from California and Oregon (*Arnett, 1951*).

### Pyrochroidae, NEW FAMILY RECORD

Notes. Four subfamilies, five genera, and 35 species of Pyrochroidae are known to occur in California (M. L. Gimmel, 2022, unpublished data).

### Pedilinae

Notes. One genus and 25 species of Pedilinae are known to occur in California (M. L. Gimmel, 2022, unpublished data).

### *Pedilus* Fischer von Waldheim, 1820

Nomenclatural Authority: *Bouchard et al. (2011)*

Digitized Records (genus-only): Santa Cruz (1 SBMNH; 1 UCSB)

Notes. Twenty-five species of *Pedilus* are known to occur in California (M. L. Gimmel, 2022, unpublished data). Most of the California species of this genus were treated by *Abdullah (1964, 1966, 1969)*.

### *Pedilus bardii* (Horn, 1874)
Nomenclatural Authority: *Abdullah (1966)*
Literature Records: none
Digitized Records: Santa Cruz (1 SBMNH)
Range: Also known from mainland (*Abdullah, 1966*).

### Salpingidae, NEW FAMILY RECORD
Notes. This family is currently divided into seven subfamilies worldwide and its composition has changed dramatically and often over the last century. Five subfamilies, six genera, and 14 species are known to occur in California (M. L. Gimmel, 2022, unpublished data).

### Salpinginae
Notes. Two genera and two or three species of Salpinginae are known to occur in California (M. L. Gimmel, 2022, unpublished data).

### *Rhinosimus* Latreille, 1802
Nomenclatural Authority: *Bouchard et al. (2011)*
Notes. One or two species of this genus have been reported from California (M. L. Gimmel, 2022, unpublished data). The North American species were treated by *Blair (1932)*, but the genus still needs revision. *Rhinosimus* has a Holarctic distribution.

### *Rhinosimus* undetermined species
Literature Records: none
Digitized Records: Santa Cruz (4 SBMNH)
Notes. *Rhinosimus* is known from coastal habitats from Alaska south, with the Santa Cruz records representing the southernmost extent of its range known to us.

### Scraptiidae
Notes. This family is divided into two subfamilies, Anaspidinae and Scraptiinae, both occurring in California; the former was historically included in Mordellidae (*Liljeblad, 1945*), while the latter was historically included within Melandryidae (*Pollock, 2002*). Five genera and 19 species of Scraptiidae are known to occur in California (M. L. Gimmel, 2022, unpublished data).

### Anaspidinae
Notes. Four genera and 17 species of Anaspidinae have been recorded from California (M. L. Gimmel, 2022, unpublished data). These were treated for North America by *Liljeblad (1945)*.

### *Anaspis* Geoffroy, 1762
Nomenclatural Authority: *Pollock (2002)*

Digitized Records (genus-only): Santa Cruz (14 SBMNH)

Notes. Seven species of *Anaspis* have been recorded from California (*Liljeblad, 1945*).

### *Anaspis atrata* Champion, 1891

Nomenclatural Authority: *Liljeblad (1945)*

Literature Records: none

Digitized Records: Santa Cruz (2 LACM; 6 SBMNH)

Range: Also known from mainland (*Liljeblad, 1945*).

### *Anaspis collaris* LeConte, 1851

Nomenclatural Authority: *Liljeblad (1945)*

Literature Records: Santa Catalina (*Fall, 1897*: 238; *Liljeblad, 1945*: 216)

Digitized Records: Santa Catalina (1 SBMNH)

Range: Also known from mainland (*Liljeblad, 1945*).

### *Pentaria* Mulsant, 1856

Nomenclatural Authority: *Pollock (2002)*

Notes. Five species of *Pentaria* have been recorded from California (*Liljeblad, 1945*).

### *Pentaria trifasciata* (Melsheimer, 1845)

Nomenclatural Authority: *Liljeblad (1945)*

Literature Records: Santa Catalina (*Fall, 1897*: 238)

Digitized Records: San Nicolas (1 SBMNH), Santa Catalina (7 SBMNH), Santa Cruz (5 LACM; 6 SBMNH)

Range: Also known from mainland (*Liljeblad, 1945*).

Notes. *Liljeblad (1945)* treated this as an extremely variable and widespread species, including *Pentaria trifasciata nubila* (LeConte, 1859) as a variety representing a color morph. *Fall (1897)* reported this species as *Pentaria nubila*. We have seen specimens from Santa Cruz Island corresponding to both color morphs in the sense of *Liljeblad (1945)*.

### Tenebrionidae

Notes. This family has a recent and reliable catalog for North America by *Bousquet et al. (2018)* which serves as a good starting point for taxonomy and finding identification references. The constituent groups that comprise the Channel Island diversity vary in accessibility of authoritative treatments and identification resources. Many of the tribes and genera require a thorough revision before the species reported from the islands can be truly verified. Eight subfamilies, 37 tribes, 110 genera, 525 species of Tenebrionidae are known to occur in California (M. L. Gimmel, 2022, unpublished data).

### Alleculinae

Notes. Six genera and 33 species of Alleculinae, all belonging to the tribe Alleculini, have been recorded from California (*Bousquet et al., 2018*). "Alleculinae larvae" were reported from San Clemente and Santa Catalina by *Straughan & Hadley (1980*: 392).

### *Hymenorus* Mulsant, 1852

Nomenclatural Authority: *Bousquet et al. (2018)*

Literature Records (genus-only): Santa Cruz (*Naughton et al., 2014*: 304)

Notes. This genus was last revised by *Fall (1931)* and requires significant work, at least for species in western North America. There are 19 species reported from California (*Bousquet et al., 2018*).

### *Hymenorus infuscatus* Casey, 1891

Nomenclatural Authority: *Bousquet et al. (2018)*

Literature Records: Santa Catalina (*Fall, 1897*: 238; *Fall, 1901*: 176; *Fall, 1931*: 186)

Digitized Records: none

Range: Also known from mainland (*Fall, 1931*).

Notes. The southern California species are difficult to identify. We have not seen any *Hymenorus* specimens from the Channel Islands, but trust the records given by Fall who completed the last revision of this genus.

### *Isomira* Mulsant, 1856

Nomenclatural Authority: *Bousquet et al. (2018)*

Digitized Records (genus-only): San Clemente (16 SBMNH), San Nicolas (2 SBMNH), Santa Catalina (9 SBMNH), Santa Cruz (21 SBMNH), Santa Rosa (11 SBMNH)

Notes. This genus was revised in the dissertation of *Marshall (1964)* which was subsequently published in numerous parts except for a final key and treatments of the western species. Species are difficult to identify without comparative material and dissected males (M. A. Johnston, 2022, personal observation). There are five species recorded from California (*Bousquet et al., 2018*).

### *Isomira comstocki* Papp, 1956

Nomenclatural Authority: *Bousquet et al. (2018)*

Literature Records: Santa Cruz (*Marshall, 1964*:145)

Digitized Records: Santa Barbara (1 LACM), Santa Cruz (4 SBMNH)

Range: Also known from mainland (*Marshall, 1964*).

Notes. This widespread and variable species is primarily identified by dissected male genitalia.

### *Isomira damnata* Marshall, 1970

Nomenclatural Authority: *Bousquet et al. (2018)*

Literature Records: none

Digitized Records: Santa Catalina (1 LACM)

Range: Also known from mainland (*Marshall, 1964*).

Notes. The single Channel Island specimen of this species is a male with extruded genitalia that was reliably determined by J.M. Campbell.

### *Isomira luscitiosa* Casey, 1891

Nomenclatural Authority: *Bousquet et al. (2018)*

Literature Records: none

Digitized Records: Santa Cruz (5 LACM), Santa Rosa (2 LACM)

Range: Also known from mainland (*Marshall, 1964*).

Notes. The specimens representing this species were reliably determined by J.M. Campbell.

### *Isomira variabilis* (Horn, 1875)

Nomenclatural Authority: *Bousquet et al. (2018)*

Literature Records: San Clemente (*Fall, 1897*: 238; *Fall, 1901*: 176)

Digitized Records: none

Range: Also known from mainland (*Marshall, 1964*).

Notes. *Marshall (1964)* recognized a species group that most workers before him had lumped into a single concept of *I. variabilis*, in which he included *I. damnata* and *I. luscitiosa*. It is possible that *Fall (1897, 1901)* was actually referring to one of the latter two species. However, *I. variabilis* is widespread and known from the coastal habitats of southern California and may well be the correct identification.

### *Mycetochara* Guérin-Méneville, 1827

Nomenclatural Authority: *Bousquet et al. (2018)*

Digitized Records (genus-only): Santa Catalina (4 SBMNH), Santa Cruz (7 SBMNH)

Notes. *Campbell (1978)* revised this genus and synonymized many of the previously described species along with providing a reliable key. Three species are recorded from California (*Bousquet et al., 2018*).

### *Mycetochara pubipennis* LeConte, 1878

Nomenclatural Authority: *Bousquet et al. (2018)*

Literature Records: Santa Catalina (*Campbell, 1978*: 936)

Digitized Records: none

Range: Also known from mainland (*Campbell, 1978*).

Notes. *Campbell (1978)* only saw male specimens from the island and noted that the eyes are slightly smaller than those on the mainland, a character which is traditionally used as part of species diagnoses.

### Blaptinae

Notes. Three tribes, 18 genera, and 122 species of Blaptinae are known to occur in California (*Bousquet et al., 2018*; M. L. Gimmel, 2022, unpublished data).

### Amphidorini

Notes. Six genera and 89 species of Amphidorini are known to occur in California (*Bousquet et al., 2018*; M. L. Gimmel, 2022, unpublished data). This tribe has seen recent treatments for many constituent species groups but has lacked a thorough revision of the genera and subgenera. The nomenclaturally unavailable dissertation of *Johnston (2018)* provides keys, diagnoses, and new concepts of genera and subgenera. Since this study has not been published in compliance with the ICZN, the taxonomy follows *Bousquet et al. (2018)* where most species are included in the large genus *Eleodes*.

### *Eleodes* Eschscholtz, 1829

Nomenclatural Authority: *Bousquet et al., 2018*

Digitized Records (genus-only): Anacapa (1 LACM), San Clemente (19 LACM; 1 UCMC), San Miguel (38 LACM; 3 SDNHM), San Nicolas (27 LACM), Santa Catalina (1 OSUC; 54 LACM), Santa Cruz (25 LACM; 1 UCSB; 1 YPMC; 1 iNat), Santa Rosa (31 LACM)

Notes. The nomenclaturally unavailable dissertation by *Johnston (2018)* breaks this genus into multiple genera and it is expected that the nomenclature reported here will soon be outdated. Digitized records identified as *Amphidora* Eschscholtz, 1829, a current subgenus of *Eleodes*, are included in the genus-only records above. This large genus of flightless arid-adapted species is restricted to western North America and has its center of diversity in California with 67 species recorded from the state. A possible fossil of *Eleodes* was reported by *Lipps (1964)* from a deposit on West Anacapa Island, though it is likely to be a modern contaminant that made its way into the sandy deposit.

### *Eleodes* (*Amphidora*) *littoralis* (Eschscholtz, 1829)

Nomenclatural Authority: *Bousquet et al. (2018)*

Literature Records: Santa Catalina (*Fall, 1897*: 238), Santa Cruz (*Fall & Davis, 1934*: 144), Santa Rosa (*Fall, 1897*: 238)

Digitized Records: Anacapa (14 LACM; 1 SBMNH), San Clemente (5 LACM), San Miguel (3 SBMNH), Santa Catalina (2 OSUC; 24 LACM; 26 SBMNH), Santa Cruz (5 SBMNH), Santa Rosa (9 SBMNH)

Range: Also known from mainland (*Blaisdell, 1909*).

Notes. This small, hirsute species is common in leaf litter along the coastal mountain ranges of California.

### *Eleodes* (*Amphidora*) *nigropilosa* (LeConte, 1851)

Nomenclatural Authority: *Bousquet et al. (2018)*

Literature Records: Santa Catalina (*Straughan & Hadley, 1980*: 392)

Digitized Records: Santa Catalina (2 LACM; 1 MAJC; 2 OSUC; 6 SBMNH), Santa Cruz (3 SBMNH), Santa Rosa (1 SBMNH)

Range: Also known from mainland (*Blaisdell, 1909*; *Triplehorn, 1996*).

Notes. This species is common in coastal habitats in California and Baja California.

### *Eleodes* (*Blapylis*) *clavicornis* Eschscholtz, 1829

Nomenclatural Authority: *Bousquet et al. (2018)*

Literature Records: none

Digitized Records: Anacapa (3 LACM)

Range: Also known from mainland (*Somerby, 1972*).

Notes. This relatively small species is known from coastal sand dunes.

### *Eleodes* (*Blapylis*) *inculta* LeConte, 1861

Nomenclatural Authority: *Bousquet et al. (2018)*

Literature Records: Anacapa (*Somerby, 1972*: 179; *Miller, 1985a*: 21), San Miguel (*Blaisdell, 1918*: 384; *Spilman, 1962*: 57; *Tanner, 1961*: 73; *Somerby, 1972*: 179; *Miller, 1985a*: 21), Santa Barbara (*LeConte, 1861*: 352; *Miller, 1985a*: 21; *Miller & Miller, 1985*: 128), Santa Cruz (*Blaisdell, 1918*: 384; *Fall & Davis, 1934*: *Tanner, 1961*: 73; 144; *Somerby, 1972*: 179;

*Miller, 1985a*: 21; *Naughton et al., 2014*: 304), Santa Rosa (*Blaisdell, 1909*: 331; *Blaisdell, 1918*: 384; *Blaisdell, 1939*: 52; *Spilman, 1962*: 57; *Somerby, 1972*: 179; *Miller, 1985a*: 21)

Digitized Records: Anacapa (1 BYUC; 10 LACM; 1 SBMNH), San Miguel (1 BYUC; 3 OSUC; 131 LACM; 2 MAJC; 76 SBMNH), Santa Catalina (2 OSUC; 1 LACM), Santa Cruz (1 BYUC; 2 OSUC; 19 LACM; 10 MAJC; 1 SBMNH; 1 iNat), Santa Rosa (20 LACM; 6 MAJC; 16 SBMNH)

Range: Endemic (*LeConte, 1861*; *Blaisdell, 1909*; *Spilman, 1962*; *Somerby, 1972*).

Notes. *Blaisdell (1909)* incorrectly documented the type locality as Santa Rosa Island, while *LeConte (1861)* stated that it was Santa Barbara Island. *Somerby (1972)* and *Miller & Miller (1985)* suggested that the type may have been mislabeled and was actually from Santa Cruz Island. The subspecies *E. inculta affinis* Blaisdell, 1918 was reported from Santa Cruz and San Miguel islands by *Blaisdell (1918)* and later synonymized by *Miller (1985a)* following the nomenclaturally unavailable thesis of *Somerby (1972)*. A single record of the species *Eleodes cordata* Eschscholtz, 1829 from Santa Cruz Island in the BYUC is here considered to almost certainly represent *E. inculta. Eleodes cordata* is otherwise not known from southern California.

### *Eleodes* (*Blapylis*) *scabripennis* LeConte, 1859

Nomenclatural Authority: *Bousquet et al. (2018)*

Literature Records: Santa Barbara (*Fall, 1897*: 238; *Fall, 1901*: 168), Santa Rosa (*Fall, 1897*: 238; *Fall, 1901*: 168)

Digitized Records: none

Range: Also known from mainland (*Somerby, 1972*).

Notes. This species was described from Fort Tejon, Kern County, California. The female holotype has a comparatively small pronotum but is quite similar to *Eleodes* (*Blapylis*) *consobrina* LeConte, 1851 (*LeConte, 1861*; *Blaisdell, 1909*). It may be that *E. scabripennis* is simply a synonym of *E. consobrina*, but it does superficially resemble *E. inculta* quite closely. The above records are almost certainly from Fall examining LeConte's material, and applying this name to *E. inculta*. However, until the genus is revised, these records cannot be fully discounted.

### *Eleodes* (*Blapylis*) *subvestita* (Blaisdell, 1939)

Nomenclatural Authority: *Bousquet et al. (2018)*

Literature Records: San Nicolas (*Blaisdell, 1939*: 55; *Cockerell, 1939*: 317; *Cockerell, 1940*: 284; *Steele, 1979*: 30; *Spilman, 1962*: 57; *Somerby, 1972*: 180; *Miller, 1985a*: 21)

Digitized Records: San Nicolas (26 LACM; 1 SBMNH)

Range: Endemic (*Blaisdell, 1939*; *Spilman, 1962*; *Somerby, 1972*).

Notes. The original description was apparently made using a combination of this actual Channel Island endemic *Eleodes* and a genitalic dissection belonging to a specimen of another subfamily, and thus was placed into a new monotypic genus and subfamily, before the error was discovered and the species placed correctly into the genus *Eleodes* subgenus *Blapylis* Horn, 1870 (*Spilman, 1962*). Accordingly, *Blaisdell (1939)*, *Cockerell (1939, 1940)*, and *Steele (1979)* reported the original combination of *Eleodopsis subvestita* Blaisdell, 1939.

### *Eleodes* (*Cratidus*) *osculans* (LeConte, 1851)

Nomenclatural Authority: *Bousquet et al. (2018)*

Literature Records: Santa Cruz (*LeConte, 1876*: 299; *Fall, 1897*: 238; *Fall & Davis, 1934*: 144), Santa Rosa (*Fall, 1897*: 238)

Digitized Records: San Miguel (2 SBMNH), Santa Catalina (5 SBMNH; 2 iNat), Santa Cruz (1 LACM), Santa Rosa (2 SBMNH)

Range: Also known from mainland (*Blaisdell, 1909*; *Triplehorn, 1996*).

Notes. This somewhat charismatic beetle is abundant in southern California. *LeConte (1876)*, *Fall (1897)*, and *Fall & Davis (1934)* recorded this species as *Cratidus osculans*.

### *Eleodes* (*Eleodes*) *acuticauda* LeConte, 1851

Nomenclatural Authority: *Bousquet et al. (2018)*

Literature Records: Anacapa (*Blaisdell, 1921*: 219; *Cockerell, 1940*: 284; *Miller, 1985a*: 21), San Clemente (*Blaisdell, 1909*: 283; *Blaisdell, 1921*: 219; *Doyen, 1974*: 87; *Miller, 1985a*: 21), San Miguel (*Blaisdell, 1921*: 219; *Cockerell, 1940*: 284; *Miller, 1985a*: 21), San Nicolas (*Blaisdell, 1921*: 219; *Cockerell, 1940*: 284; *Tanner, 1961*: 73; *Triplehorn, Thomas & Smith, 2015*: 161), Santa Barbara (*Blaisdell, 1921*: 219; *Cockerell, 1940*: 284; *Miller, 1985a*: 21; *Miller & Miller, 1985*: 128), Santa Cruz (*Miller, 1985a*: 21), Santa Rosa (*Miller, 1985a*: 21)

Digitized Records: Anacapa (3 LACM; 3 SBMNH), San Clemente (5 LACM; 6 MAJC; 5 SBMNH; 13 SDNHM), San Miguel (10 LACM; 19 SBMNH), San Nicolas (36 LACM; 24 SBMNH), Santa Barbara (16 LACM), Santa Catalina (5 SBMNH), Santa Cruz (7 LACM; 3 MAJC; 2 SBMNH), Santa Rosa (1 LACM; 17 SBMNH)

Range: Also known from mainland (*Triplehorn, Thomas & Smith, 2015*).

Notes. The recent revision by *Triplehorn, Thomas & Smith (2015)* only mentioned records from San Nicolas Island, but did not give specimen locality data other than types for any of the taxa in the revision; the accompanying distribution map shows a record for San Nicolas and no other islands. This species is abundant in southern California and is difficult to separate from *E. dentipes* in the vicinity of the city of Santa Barbara; see the remarks under that species. The name *Eleodes laticollis* LeConte, 1851 was first synonymized with *E. acuticauda* by *Horn (1870)* but subsequently used as a subspecies in the island records of *Blaisdell (1909)* and *Doyen (1974)*. The subspecies *E. laticollis apprima* Blaisdell, 1921 was erected for the Channel Islands populations and used by *Blaisdell (1921)*, *Cockerell (1940)*, *Tanner (1961)*, and *Miller (1985a)* before it was synonymized with *E. acuticauda* by *Triplehorn (1996)*.

### *Eleodes* (*Eleodes*) *dentipes* Eschscholtz, 1829

Nomenclatural Authority: *Bousquet et al. (2018)*

Literature Records: Anacapa (*Triplehorn, Thomas & Smith, 2015*: 165 [map]), San Clemente (*Fall, 1897*: 238), San Nicolas (*Fall, 1897*: 238), Santa Cruz (*Fall & Davis, 1934*: 144), Santa Rosa (*Fall, 1897*: 238)

Digitized Records: none

Range: Also known from mainland (*Triplehorn, Thomas & Smith, 2015*).

Notes. The recent revision of *Triplehorn, Thomas & Smith (2015)* gave no specimen locality data for this or any other treated taxa. However, their accompanying distribution map has a mark on Anacapa Island. This species and *E. acuticauda* are quite similar and can be difficult to distinguish in the region of Santa Barbara County where the two species distributions meet. These two species and island populations along with the mainland populations need to be carefully considered. The two species are distinguished by *E. acuticauda* having a much more transverse and strongly rounded pronotum than *E. dentipes* which reliably separates most species from the northern Central Valley and Bay Area of California from those around San Diego and Los Angeles. These literature records presumably overlap with the digitized records of *E. acuticauda*, but the species are retained as separate since the last revision (*Triplehorn, Thomas & Smith, 2015*) indicated that both species are known from the Channel Islands.

### *Eleodes* (*Melaneleodes*) *carbonaria* (Say, 1824)
Nomenclatural Authority: *Bousquet et al. (2018)*
Literature Records: Santa Catalina (*Fall, 1897*: 238; *Fall, 1901*: 167; *Blaisdell, 1909*: 75; *Cockerell, 1940*: 284; *Tanner, 1961*: 69)
Digitized Records: San Clemente (1 LACM), Santa Catalina (1 LACM; 8 SBMNH), Santa Rosa (3 LACM)
Range: Also known from mainland (*Triplehorn & Thomas, 2012*).
Notes. This species has been well documented from Santa Catalina island under several different species names. Originally referred to *Eleodes quadricollis* Eschscholtz, 1829 by *Fall (1897*, *1901)* (only known from around San Francisco as currently circumscribed), *Blaisdell (1909)* and *Tanner (1961)* recognized the island population as *Eleodes omissa* forma *catalinae* Blaisdell, 1909, which was later included within the subspecies concept of *Eleodes carbonaria omissa* LeConte, 1858 of *Triplehorn & Thomas (2012)*, which is the only subspecies distributed throughout southern California and Baja California. *Cockerell (1940*: 284) listed this species twice under the names *E. omissa catalinae* and *E. omissa pygmaea* Blaisdell, 1909, both of which are now considered synonyms of *E. carbonaria omissa* (*Triplehorn & Thomas, 2012*).

### *Eleodes* (*Steneleodes*) *gigantea* Mannerheim, 1843
Nomenclatural Authority: *Bousquet et al. (2018)*
Literature Records: none
Digitized Records: San Miguel (43 SBMNH; 4 SDNHM), Santa Rosa (6 LACM; 3 SBMNH)
Range: Also known from mainland (*Blaisdell, 1909*).
Notes. This species, abundant in the coastal ranges, is known from California and Baja California but has never been reported from the Channel Islands in the literature.

### Opatrini
Notes. Eleven genera and 31 species of Opatrini are known to occur in California (*Bousquet et al., 2018*; M. L. Gimmel, 2022, unpublished data).

**_Blapstinus_ Dejean, 1821**
Nomenclatural Authority: _Davis (1970)_, _Bousquet et al. (2018)_
Literature Records (genus-only): Santa Cruz (_Fall & Davis, 1934_: 144)
Digitized Records (genus-only): San Clemente (1 SBMNH), Santa Catalina (13 LACM), Santa Rosa (9 LACM; 2 SBMNH)
Notes. _Fall & Davis (1934)_ stated that their record was not _Blapstinus rufipes_ (= _B. discolor_) or _B. brevicollis_. The nomenclaturally unavailable dissertation of _Davis (1970_, _1976)_ provides keys and treatments for the genus but many specimens in collections are dubiously identified, particularly if done before 1970 (M. A. Johnston, 2022, personal observation). Fifteen species have been recorded from California (_Bousquet et al., 2018_).

**_Blapstinus angustus_ LeConte, 1851**
Nomenclatural Authority: _Bousquet et al. (2018)_
Literature Records: none
Digitized Records: San Clemente (1 SDNHM)
Range: Also known from mainland (_Bousquet et al., 2018_).
Notes. This is the type species of the genus _Mecysmus_ Horn, 1870, where it has been included since the genus description until recently, when _Lumen et al. (2019b)_ synonymized it with _Blapstinus_. The species is known from throughout southern California and western Arizona.

**_Blapstinus brevicollis_ LeConte, 1851**
Nomenclatural Authority: _Davis (1970)_, _Bousquet et al. (2018)_
Literature Records: Santa Cruz (_Davis, 1970_: 138), Santa Rosa (_Fall, 1897_: 238; in doubt, _Davis, 1970_: 138)
Digitized Records: Santa Catalina (1 SBMNH), Santa Cruz (4 SBMNH), Santa Rosa (9 SBMNH)
Range: Also known from mainland (_Davis, 1970_).
Notes. This species is common in southern California and Arizona and has been collected from driftwood (_Davis, 1970_: 136). The thorough yet nomenclaturally unavailable dissertation of _Davis (1970)_ clearly indicated two island records on the distribution map (see his figure 143), and in the material examined lists both "Santa Cruz" and "Santa Rosa Island" as localities from "Los Angeles County, California" which are interpreted as the two island records listed above.

**_Blapstinus discolor_ Horn, 1870**
Nomenclatural Authority: _Davis (1970)_, _Bousquet et al. (2018)_
Literature Records: Santa Catalina (_Fall, 1897_: 238; _Baker, 1905_: 59; _Davis, 1970_: 302)
Digitized Records: San Miguel (1 SBMNH), Santa Cruz (10 SBMNH), Santa Rosa (2 SBMNH)
Range: Also known from mainland (_Davis, 1970_).
Notes. This flightless and somewhat variable species is widely distributed throughout the western United States and is a common pest of crops along the southern California coast. _Davis (1970)_ reported this species from "Avalon" (Santa Catalina Island) in the material

examined but did not give a distinct island marker on the distribution map (his figure 163). *Fall (1897)* and *Baker (1905)* recorded this species as *Blapstinus rufipes* Casey, 1890 which was synonymized with *B. discolor* by *Davis (1982)*.

### *Conibius* LeConte, 1851
Nomenclatural Authority: *Bousquet et al. (2018)*
Notes. This genus of flightless beetles has two species recorded from California (*Bousquet et al., 2018*).

### *Conibius seriatus* LeConte, 1851
Nomenclatural Authority: *Bousquet et al. (2018)*
Literature Records: none
Digitized Records: San Clemente (1 SBMNH), Santa Catalina (2 SBMNH), Santa Cruz (2 SBMNH)
Range: Also known from mainland (*Bousquet et al., 2018*).
Notes. This species can be commonly found under stones or crawling on the ground at night.

### *Tonibius* Casey, 1895
Nomenclatural Authority: *Bousquet et al. (2018)*
Notes. This genus is monotypic with its single species known from California (*Bousquet et al., 2018*).

### *Tonibius sulcatus* (LeConte, 1851)
Nomenclatural Authority: *Bousquet et al. (2018)*
Literature Records: San Clemente (*Fall, 1897*: 238; *Fall, 1901*: 172)
Digitized Records: San Clemente (1 LACM; 3 SBMNH; 8 SDNHM)
Range: Also known from mainland (*Bousquet et al., 2018*).
Notes. This species is common in arid and coastal habitats in California and Baja California. *Fall (1897, 1901)* recorded this species as *Notibius sulcatus*, its original combination.

### *Ulus* Horn, 1870
Nomenclatural Authority: *Lumen et al. (2019a)*, *Bousquet et al. (2018)*
Notes. Two species of *Ulus* are known from California (*Lumen et al., 2019a*).

### *Ulus crassus* (LeConte, 1851)
Nomenclatural Authority: *Lumen et al. (2019a)*
Literature Records: Santa Cruz (*Fall, 1934*: 144)
Digitized Records: none
Range: Also known from mainland (*Lumen et al., 2019a*).
Notes. This species is fairly common throughout southern California but the recent revision of *Lumen et al. (2019a)* gave no Channel Island records and none have been found in museums. It seems unlikely that *Fall (1934)* would have confused this genus with anything else, but this record certainly requires verification.

### Diaperinae

Notes. Five tribes, 13 genera, and 29 species of Diaperinae have been recorded from California (*Bousquet et al., 2018*).

### Diaperini

Notes. Eight genera and 15 species of Diaperini have been recorded from California (*Bousquet et al., 2018*).

### *Platydema* Laporte & Brullé, 1831

Nomenclatural Authority: *Bousquet et al. (2018)*

Notes. Four species of *Platydema* are recorded from California (*Bousquet et al., 2018*).

### *Platydema oregonensis* LeConte, 1857

Nomenclatural Authority: *Bousquet et al. (2018)*

Literature Records: none

Digitized Records: Santa Cruz (4 SBMNH)

Range: Also known from mainland (*Bousquet et al., 2018*).

Notes. This species is widespread throughout the Pacific coastal region and has been recorded as *Platydema oregonense* since the time of its description until *Bousquet et al. (2018)* corrected the gender of the genus.

### Hypophlaeini

Notes. One genus and six species of Hypophlaeini have been recorded from California (*Bousquet et al., 2018*).

### *Corticeus* Piller & Mitterpacher, 1783

Nomenclatural Authority: *Bousquet et al. (2018)*

Notes. Six species of *Corticeus* have been recorded from California (*Bousquet et al., 2018*). The species were reviewed for North America by *Triplehorn (1990)*.

### *Corticeus opaculus* (LeConte, 1878)

Nomenclatural Authority: *Bousquet et al. (2018)*

Literature Records: Santa Cruz (*Triplehorn, 1990*: 294)

Digitized Records: Santa Cruz (1 OSUC)

Range: Also known from mainland (*Triplehorn, 1990*).

Notes. This subcortical species is not uncommon in coastal California habitats.

### Phaleriini

Notes. Two genera and six species of Phaleriini have been recorded from California (*Bousquet et al., 2018*).

### *Phaleria* Latreille, 1802

Nomenclatural Authority: *Bousquet et al. (2018)*

Notes. This genus inhabits coastal dunes around the world and can be readily found under beach wrack and dead fish. The New World components were revised by *Triplehorn & Watrous (1979)*. A single species is known from California (*Bousquet et al., 2018*).

*Phaleria rotundata* LeConte, 1851

Nomenclatural Authority: *Bousquet et al. (2018)*

Literature Records: Anacapa (*Triplehorn & Watrous, 1979*: 284), San Nicolas (*Triplehorn & Watrous, 1979*: 286), Santa Catalina (*Triplehorn & Watrous, 1979*: 286), Santa Cruz (*Triplehorn & Watrous, 1979*: 286)

Digitized Records: Anacapa (2 LACM), San Clemente (6 SBMNH), San Nicolas (1 LACM; 3 SBMNH), Santa Catalina (3 SBMNH), Santa Cruz (14 SBMNH), Santa Rosa (8 SBMNH)

Range: Also known from mainland (*Triplehorn & Watrous, 1979*).

Notes. This species has a somewhat restricted range, from northern Baja California through San Francisco, and is the only species of this genus known from the United States Pacific coastline.

**Pimeliinae**

Notes. Eleven tribes, 41 genera, and 248 species of Pimeliinae are known to occur in California (*Bousquet et al., 2018*; M. L. Gimmel, 2022, unpublished data).

**Anepsiini**

Notes. Five genera and nine species of Anepsiini have been recorded from California (*Bousquet et al., 2018*).

*Batuliodes* Casey, 1907

Nomenclatural Authority: *Doyen (1987)*

Notes. This genus, and tribe, was thoroughly revised by *Doyen (1987)* who provided reliable keys to genera and species. Four species of *Batuliodes* are known from California (*Doyen, 1987*).

*Batuliodes rotundicollis* (LeConte, 1851)

Nomenclatural Authority: *Bousquet et al. (2018)*

Literature Records: none

Digitized Records: San Clemente (1 SBMNH)

Range: Also known from mainland (*Doyen, 1987*).

Notes. This small and relatively infrequently collected species was not recorded from the Channel Islands in the revision by *Doyen (1987)*.

**Cnemeplatiini**

Notes. Two genera and eight species of Cnemeplatiini are known to occur in California (M. L. Gimmel, 2022, unpublished data).

*Alaudes* Horn, 1870

Nomenclatural Authority: *Aalbu, Caterino & Smith (2018)*

Notes. Individuals of *Alaudes* are the smallest of all known Tenebrionidae from the Channel Islands; they were revised recently by *Aalbu, Caterino & Smith (2018)*. Six species of *Alaudes* are recorded from California (*Aalbu, Caterino & Smith, 2018*).

*Alaudes singularis* Horn, 1870

Nomenclatural Authority: *Aalbu, Caterino & Smith (2018)*

Literature Records: San Clemente (*Aalbu, Caterino & Smith, 2018*: 265), San Nicolas (*Aalbu, Caterino & Smith, 2018*: 265)
Digitized Records: San Clemente (2 SBMNH)
Range: Also known from mainland (*Aalbu, Caterino & Smith, 2018*).

### *Lepidocnemeplatia* Bousquet & Bouchard, 2018
Nomenclatural Authority: *Bousquet et al. (2018)*
Notes. Only a single species of *Lepidocnemeplatia* is known from California (*Bousquet et al., 2018*).

### *Lepidocnemeplatia sericea* (Horn, 1870)
Nomenclatural Authority: *Bousquet et al. (2018)*
Literature Records: none
Digitized Records: Santa Cruz (10 SBMNH)
Range: Also known from mainland (*Bousquet et al., 2018*).
Notes. This species is widespread in the arid regions of western North America and is particularly prevalent in habitats with sandy substrate.

### Coniontini
Notes. Three genera and 63 species of Coniontini have been recorded from California (*Bousquet et al., 2018*; M. L. Gimmel, 2022, unpublished data).

### *Coelus* Eschscholtz, 1829
Nomenclatural Authority: *Doyen (1976)*
Digitized Records (genus-only): San Clemente (2 SBMNH), Santa Catalina (1 TAMU), Santa Cruz (1 TAMU)
Notes. *Doyen (1976)* revised *Coelus* and provided a reliable key. *Chatzimanolis, Norris & Caterino (2010)* explored the historical biogeography and phylogenetic relationships of Channel Island and mainland coastal populations of this genus. Four species are known from California, three of which occur on the islands (*Doyen, 1976*).

### *Coelus ciliatus* Eschscholtz, 1829
Nomenclatural Authority: *Doyen (1976)*, *Bousquet et al. (2018)*
Literature Records: Anacapa (*Doyen, 1976*: 616)
Digitized Records: San Nicolas (1 CASC), Santa Cruz (1 CASC), Santa Rosa (3 OSUC)
Range: Also known from mainland (*Doyen, 1976*).
Notes. This species is abundant and widely distributed along the mainland Pacific Coast.

### *Coelus globosus* LeConte, 1851
Nomenclatural Authority: *Doyen (1976)*, *Bousquet et al. (2018)*
Literature Records: Anacapa (*Doyen, 1976*: 618), San Miguel (*Doyen, 1976*: 618), San Nicolas (*Doyen, 1976*: 618; *Chatzimanolis, Norris & Caterino, 2010*: 787), Santa Barbara (*Doyen, 1976*: 618), Santa Catalina (*Doyen, 1976*: 618), Santa Cruz (*Doyen, 1976*: 618; *Chatzimanolis, Norris & Caterino, 2010*: 787), Santa Rosa (*Doyen, 1976*: 618)

Digitized Records: Anacapa (7 LACM), San Miguel (1 LACM; 4 SBMNH), San Nicolas (104 LACM; 1 SBMNH), Santa Cruz (2 MAJC; 10 SBMNH; 4 UCMC), Santa Rosa (6 LACM; 3 SBMNH)

Range: Also known from mainland (*Doyen, 1976*).

Notes. This coastal dune-inhabiting species is known from most of California's shoreline including all Channel Islands except San Clemente (*Doyen, 1976*: 618).

### *Coelus pacificus* Fall, 1897

Nomenclatural Authority: *Bousquet et al. (2018)*

Literature Records: Anacapa (*Doyen, 1976*: 623; *Miller, 1985a*: 21), San Clemente (*Fall, 1897*: 238; *Fall, 1901*: 166; *Casey, 1908*: 158; *Blaisdell, 1919*: 321; *Doyen, 1974*: 87; *Doyen, 1976*: 623; *Miller, 1985a*: 21; *Chatzimanolis, Norris & Caterino, 2010*: 787), San Miguel (*Blaisdell, 1919*: 321; *Doyen, 1976*: 623; *Miller, 1985a*: 21; *Chatzimanolis, Norris & Caterino, 2010*: 787), San Nicolas (*Fall, 1897*: 238; *Casey, 1908*: 158; *Blaisdell, 1919*: 321; *Doyen, 1976*: 623; *Miller, 1985a*: 21; *Chatzimanolis, Norris & Caterino, 2010*: 787), Santa Barbara (*Fall, 1901*: 166), Santa Catalina (*Miller, 1985a*: 21; *Chatzimanolis, Norris & Caterino, 2010*: 787), Santa Cruz (*Blaisdell, 1919*: 321; *Fall & Davis, 1934*: 144; *Doyen, 1976*: 623; *Miller, 1985a*: 21; *Chatzimanolis, Norris & Caterino, 2010*: 787), Santa Rosa (*Fall, 1897*: 238; *Fall, 1901*: 166; *Blaisdell, 1919*: 321; *Doyen, 1976*: 623; *Miller, 1985a*: 21; *Chatzimanolis, Norris & Caterino, 2010*: 787)

Digitized Records: San Clemente (292 LACM; 16 SBMNH; 5 SDNHM), San Miguel (39 CASC; 264 LACM; 144 SBMNH), San Nicolas (7 CASC; 103 LACM; 33 SBMNH; 3 iNat), Santa Barbara (5 LACM), Santa Catalina (3 LACM; 14 SBMNH), Santa Cruz (1 OSUC; 59 CASC; 10 LACM; 4 MAJC; 27 SBMNH), Santa Rosa (1 OSUC; 5 CASC; 69 LACM; 18 SBMNH; 1 USNM; 1 iNat)

Range: Endemic (*Fall, 1897*; *Doyen, 1976*).

Notes. A larva and pupa from San Clemente Island were studied by *Doyen (1976*: 611–612), who also synonymized *Coelus remotus* Fall, 1897 with *C. pacificus*. Unfortunately he merely stated that this species is distributed on the "California Channel Islands" (*Doyen, 1976*: 619) and beyond this only gave type specimen island records and several in the appendix. This species is recorded from all eight Channel Islands. The type locality of *C. pacificus* is San Nicolas Island (*Fall, 1897*). The name *C. remotus*, whose type locality is San Clemente Island, was used for the San Clemente population by *Fall (1897, 1901)*, *Casey (1908)*, *Blaisdell (1919)*, and *Doyen (1974)*.

### *Coniontis* Eschscholtz, 1829

Nomenclatural Authority: *Bousquet et al. (2018)*

Literature Records (genus-only): Santa Cruz (*Fall, 1934*: 144; *Naughton et al., 2014*: 304)

Digitized Records (genus-only): San Clemente (1 SDNHM; 4 iNat), San Nicolas (1 SBMNH; 1 iNat), Santa Catalina (15 SBMNH; 2 iNat), Santa Cruz (3 CSUC; 4 SBMNH), Santa Rosa (12 SBMNH; 2 SDNHM)

Notes. *Coniontis* is a problematic genus which requires a comprehensive revision, though one may start with the synonymy established by *Doyen (1977)*, which included no

identification resources, in conjunction with voluminous species descriptions provided by *Casey (1908)*, or instead attempt to use the outdated treatment by *Horn (1870)*. It seems clear that there is at least one endemic species (*C. lata* LeConte, 1866) and likely a second (*C. santarosae* Blaisdell, 1921) but the literature records and identified museum specimen records should otherwise all be considered dubious until the genus is revised. One Santa Cruz Island specimen labeled "*Coniontis musculus* ?" in the H.C. Fall collection in the Museum of Comparative Zoology, Harvard University was determined by H.C. Fall (S. Miller, 2022, personal communication) and it alludes to *C. muscula* Blaisdell, 1918; this questionable species record is probably the origin of the genus-only citation above. Forty-nine currently valid species are recorded from California (*Bousquet et al., 2018*; M. L. Gimmel, 2022, unpublished data).

### *Coniontis elliptica* Casey, 1884
Nomenclatural Authority: *Bousquet et al. (2018)*
Literature Records: Santa Catalina (*Fall, 1897*: 238; *Fall, 1901*: 165; *Baker, 1905*: 59; *Casey, 1908*: 88; *Doyen, 1977*: 2), Santa Rosa (*Fall, 1897*: 238)
Digitized Records: none
Range: Also known from mainland (*Casey, 1908*).
Notes. Casey erected the subspecies *C. elliptica catalinae* Casey, 1918 from Santa Catalina Island, which was later synonymized by *Doyen (1977)*.

### *Coniontis lamentabilis* Blaisdell, 1924
Nomenclatural Authority: *Bousquet et al. (2018)*
Literature Records: Santa Catalina (*Cockerell, 1940*: 284)
Digitized Records: none
Range: Also known from mainland (*Blaisdell, 1924b*).
Notes. *Cockerell (1940*: 284) listed this species from Santa Catalina based on the authority of Fall, who apparently concluded that the previous literature records of *C. subpubescens* belonged to this species. Without a synthetic revision or subsequent work explaining these conclusions, we have kept the records of these species distinct from each other.

### *Coniontis lata* LeConte, 1866
Nomenclatural Authority: *Bousquet et al. (2018)*
Literature Records: Anacapa (*Blaisdell, 1921*: 211; *Miller, 1985a*: 20), San Clemente (*LeConte, 1866a*: 113; *Horn, 1870*: 298; *Casey, 1890*: 377; *Fall, 1897*: 238; *Fall, 1901*: 165; *Casey, 1908*: 78; *Casey, 1908*: 80; *Cockerell, 1940*: 283; *Blaisdell, 1921*: 211; *Doyen, 1974*: 87; *Doyen, 1977*: 3; *Miller, 1985a*: 20), San Miguel (*Blaisdell, 1921*: 211; *Cockerell, 1940*: 283; *Miller, 1985a*: 20), San Nicolas (*Blaisdell, 1921*: 211), Santa Barbara (*Fall, 1897*: 238; *Fall, 1901*: 165; *Blaisdell, 1921*: 211; *Miller, 1985a*: 20; *Miller & Miller, 1985*: 128), Santa Cruz (*Casey, 1890*: 377; *Fall, 1897*: 238; *Fall, 1901*: 165; *Casey, 1908*: 79; *Fall & Davis, 1934*: 144; *Blaisdell, 1921*: 211; *Doyen, 1977*: 3; *Miller, 1985a*: 20), Santa Rosa (*Fall, 1897*: 238; *Fall, 1901*: 165; *Blaisdell, 1921*: 211; *Miller, 1985a*: 20)
Digitized Records: Anacapa (15 ASUHIC; 30 CASC; 5 LACM; 2 SBMNH), San Clemente (27 CASC; 2 LACM; 1 MAJC), San Miguel (7 CASC; 34 LACM), San Nicolas (6 CASC; 2

LACM; 1 SBMNH), Santa Barbara (1 OSUC; 14 CASC; 50 LACM; 6 SBMNH), Santa Cruz (8 OSUC; 17 CASC; 3 LACM)

Range: Endemic (*LeConte, 1866a*; *Fall, 1897*; *Casey, 1908*; *Doyen, 1977*).

Notes. Originally described from San Clemente (*LeConte, 1866a*) and subsequently found on Santa Cruz and described as a subspecies, this taxon was given its own genus, *Coniontides* Casey, 1908 by *Casey (1908)*, who recognized four species which were subsequently subsumed back under the present species: *Blaisdell (1921*, who reported this taxon simply as "*Coniontides*" across all the islands) synonymized *Coniontides clementinus* Casey, 1908 (described from San Clemente); *Doyen (1977)* synonymized *Coniontides finitimus* Casey, 1908 (uncertain type locality, likely Santa Rosa [*Casey, 1908*: 80]) and *Coniontis lata* var. *insularis* Casey, 1890 (described from Santa Cruz); *Coniontides* was synonymized with *Coniontis* by *Doyen (1972)*. *Fall (1901)* primarily recognized *Coniontis lata* but used *C. lata* var *insularis* for the Santa Cruz population. *Fall & Davis (1934)* used *Coniontides insularis* for the Santa Cruz and Santa Rosa populations. *Cockerell (1940)* variously used the names *Coniontides clementinus* and *Coniontides*.

### *Coniontis microsticta* Casey, 1908

Nomenclatural Authority: *Bousquet et al. (2018)*

Literature Records: none

Digitized Records: Santa Cruz (5 CASC)

Range: Also known from mainland (*Casey, 1908*).

Notes. The type and only given locality of this species is Alameda County, California (*Casey, 1908*), which makes these determinations dubious but impossible to discount until further revisionary works are undertaken.

### *Coniontis nemoralis* Eschscholtz, 1829

Nomenclatural Authority: *Bousquet et al. (2018)*

Literature Records: none

Digitized Records: Santa Cruz (5 CASC)

Range: Also known from mainland (*Casey, 1908*; *Bousquet et al., 2018*)

Notes. This species was described and reported from the vicinity of San Francisco and is currently separated into two putative subspecies from California and Oregon (*Bousquet et al., 2018*). The determinations of these specimens are dubious but impossible to discount until further revisionary works are undertaken.

### *Coniontis santarosae* Blaisdell, 1921

Nomenclatural Authority: *Bousquet et al. (2018)*

Literature Records: San Miguel (*Blaisdell, 1921*: 210; *Cockerell, 1940*: 284; *Miller, 1985a*: 20), Santa Cruz (*Cockerell, 1940*: 284), Santa Rosa (*Blaisdell, 1921*: 210; *Cockerell, 1940*: 284; *Miller, 1985a*: 20)

Digitized Records: San Miguel (2 CASC), Santa Cruz (3 CASC), Santa Rosa (75 CASC)

Range: Endemic (*Blaisdell, 1921*).

Notes. *Blaisdell (1921)* reported that this species clearly belonged to a different species group than *C. lata* as defined by *Casey (1908)*. Though these two taxa are likely distinct

from each other and are both putative island endemics, their status in relation to the mainland species remains to be critically examined.

### *Coniontis subpubescens* LeConte, 1851

Nomenclatural Authority: *Bousquet et al. (2018)*

Literature Records: Santa Catalina (*Fall, 1897*: 238), Santa Cruz (*LeConte, 1876*: 299; *Fall, 1897*: 238; *Fall & Davis, 1934*: 144)

Digitized Records: none

Range: Also known from mainland (*Bousquet et al., 2018*).

Notes. *Cockerell (1940*: 284) stated on the authority of Fall that the previous reports of this species should in fact refer to *C. lamentabilis*; this identification correction has not been seen in any other published works or museum records.

### *Coniontis viatica* Eschscholtz, 1829

Nomenclatural Authority: *Bousquet et al. (2018)*

Literature Records: Santa Cruz (*LeConte, 1876*: 299; *Fall, 1897*: 238; *Fall & Davis, 1934*: 144)

Digitized Records: none

Range: Also known from mainland.

### *Eusattus* LeConte, 1851

Nomenclatural Authority: *Bousquet et al. (2018)*

Notes. This genus was thoroughly revised by *Doyen (1984)* who provided reliable keys and distribution information. Ten species are reported from California (*Bousquet et al., 2018*).

### *Eusattus difficilis* LeConte, 1851

Nomenclatural Authority: *Bousquet et al. (2018)*

Literature Records: San Clemente (*Doyen, 1984*: 97).

Digitized Records: none

Range: Also known from mainland (*Doyen, 1984*).

Notes. This island record is taken from a distinct marker on a range map, though no further specimen data are given for this or any other distribution points on the map within the main text. This species is broadly distributed on coastal and mainland southern California.

### *Eusattus politus* Horn, 1883

Nomenclatural Authority: *Bousquet et al. (2018)*

Literature Records: San Miguel (*Blaisdell, 1921*: 215; *Cockerell, 1940*: 283; *Doyen, 1984*: 93; *Miller, 1985a*: 21), Santa Cruz (*Doyen, 1984*: 93), Santa Rosa (*Fall, 1897*: 238; *Fall, 1901*: 166; *Blaisdell, 1921*: 215; *Doyen, 1984*: 93; *Miller, 1985a*: 21)

Digitized Records: San Miguel (6 LACM; 20 SBMNH), Santa Rosa (1 LACM; 5 SBMNH)

Range: Endemic (*Doyen, 1984*).

Notes. This species was described from "Santa Barbara California" (*Horn, 1883*: 304) but no other specimens have ever been reported, definitively or putatively, from the mainland (*Doyen, 1984*). *Eusattus vanduzeei* Blaisdell, 1921 (type locality: Prince Island off San

Miguel Island) was synonymized by *Doyen (1984)*. Two subspecies were recognized in the last revision by *Doyen (1984)* which can be separated by the size of the punctures on the head and pronotal disc. *Eusattus politus politus Horn, 1883* is known from San Miguel, Prince, and Santa Rosa islands, while *E. politus cruzensis* Doyen, 1984 is known from Santa Cruz Island. The record from Santa Barbara Island by *Fall (1901*: 166) is problematic and here discounted. It likely resulted from Horn's original type locality, but the species was not reported from Santa Barbara Island by either *Doyen (1984)* or *Miller & Miller (1985)*. The populations from San Miguel and Santa Rosa islands were reported as *E. vanduzeei* by *Blaisdell (1921)* and *Cockerell (1940)*.

### *Eusattus robustus* LeConte, 1866
Nomenclatural Authority: *Bousquet et al. (2018)*
Literature Records: San Clemente (*LeConte, 1866a*: 112; *Horn, 1870*: 293; *Fall, 1897*: 238; *Fall, 1901*: 166; *Casey, 1908*: 59; *Doyen, 1974*: 87; *Tschinkel & Doyen, 1976*: 331; *Doyen, 1977*: 6; *Doyen, 1984*: 95; *Miller, 1985a*: 21), San Nicolas (*Doyen, 1984*: 95)
Digitized Records: San Clemente (1 BYUC; 1 OSUC; 36 LACM; 9 MAJC; 14 SBMNH; 2 SDNHM; 2 iNat), San Miguel (1 LACM), San Nicolas (13 LACM; 1 SBMNH), Santa Barbara (1 LACM), Santa Rosa (1 LACM)
Range: Endemic (*LeConte, 1866a*; *Fall, 1897*; *Doyen, 1977*).
Notes. *Casey (1908)* considered this species to form a distinct genus, *Nesostes* Casey, 1908, and recognized a subspecies, *E. robustus postremus* Casey, 1908, from a single specimen also from San Clemente Island. *Triplehorn (1968)* synonymized *Nesostes* under *Eusattus*, and *Doyen (1977)* synonymized the subspecies. *Doyen (1984*: 95) commented on what he believed were mislabeled specimens in the LACM from Santa Barbara, San Miguel, Anacapa, and Santa Rosa islands and reasoned that because the species was abundant on the other islands then singleton records must be faulty data. We include specimen records from LACM that likely overlap with those discounted by Doyen above. The specimens were collected by different people at different times so would all have to represent unique mislabeling events. This could be the case but we prefer to not discount these records at this time since we see no evidence to support them being erroneous other than the assumption that the species should be common throughout its range and across time.

### Edrotini
Notes. Fifteen genera and 78 species of Edrotini are known to occur in California (*Bousquet et al., 2018*; M. L. Gimmel, 2022, unpublished data). This tribe experienced a manyfold increase in described species from *Casey (1907)*, and groups that have not been revised subsequently have proved intractable for reliable identifications since. *Crypadius* was revised by *Thomas (1985)*. *Metoponium* and *Hylocrinus* are in need of a revision consisting primarily of synonymy (M. A. Johnston, K. Kanda, R. L. Aalbu, C. C. Wirth, 2023, unpublished data).

### *Cryptadius* LeConte, 1851
Nomenclatural Authority: *Bousquet et al. (2018)*

Notes. This genus inhabits coastal dunes and is distributed from the Channel Islands region south along the Baja California peninsula and coasts of the Gulf of California. The genus was revised by *Thomas (1985)*, who provided distributions and a reliable key to species. One species is known from California (*Bousquet et al., 2018*).

### *Cryptadius inflatus* LeConte, 1852
Nomenclatural Authority: *Bousquet et al. (2018)*
Literature Records: Santa Cruz (*Straughan & Hadley, 1980*: 392; *Thomas, 1985*: 197)
Digitized Records: Santa Cruz (1 MAJC; 14 SBMNH), Santa Rosa (4 SBMNH)
Range: Also known from mainland (*Thomas, 1985*).
Notes. All *Cryptadius* known from the United States, including the Channel Islands, belong to the nominate subspecies, *C. inflatus inflatus* LeConte, 1852.

### *Hylocrinus* Casey, 1907
Nomenclatural Authority: *Bousquet et al. (2018)*, M. A. Johnston, K. Kanda, R. L. Aalbu, C. C. Wirth, 2023, unpublished data
Notes. This genus is in great need of revision and seemingly many synonymies for the United States fauna, which was last treated by *Casey (1907)*. Seven putative species are known from California (*Bousquet et al., 2018*)

### *Hylocrinus longulus* (LeConte, 1851)
Nomenclatural Authority: *Bousquet et al. (2018)*, M. A. Johnston, K. Kanda, R. L. Aalbu, C. C. Wirth, 2023, unpublished data
Literature Records: none
Digitized Records: San Nicolas (1 SBMNH)
Range: Also known from mainland (*Casey, 1907*; M. A. Johnston, 2022, personal data).
Notes. This is one of two species recognized by M. A. Johnston, K. Kanda, R. L. Aalbu, C. C. Wirth, 2023, unpublished data that are fairly widespread throughout southern California. The single known specimen from San Nicolas is from a reliable collecting event and shows no observable difference from the mainland population.

### *Metoponium* Casey, 1907
Nomenclatural Authority: *Bousquet et al. (2018)*
Digitized Records (genus-only): San Clemente (1 LACM; 2 SBMNH), Santa Catalina (10 LACM; 23 SBMNH)
Notes. This genus, like the others in this tribe, requires extensive revision. It is highly probable that it only represents a small handful of valid species and is likely itself a synonym of the genus *Eurymetopon* Eschscholtz, 1829 (M. A. Johnston, 2022, personal observation). Literature and digitized records to the level of species are dubious until a proper revision can be undertaken. Twenty nominal species are currently known from California (*Bousquet et al., 2018*; M. L. Gimmel, 2022, unpublished data)

### *Metoponium convexicolle* (LeConte, 1851)
Nomenclatural Authority: *Bousquet et al. (2018)*
Literature Records: Santa Catalina (*Fall, 1897*: 238; *Baker, 1905*: 59)

Digitized Records: none

Range: Also known from mainland (LeConte, 1851; *Bousquet et al., 2018*).

Notes. The literature records for this species were under the name *Eurymetopon convexicolle*, its original combination.

### *Metoponium insulare* Casey, 1908

Nomenclatural Authority: *Bousquet et al. (2018)*

Literature Records: Santa Catalina (*Casey, 1907*: 308; *Miller, 1985a*: 21)

Digitized Records: Santa Catalina (1 USNM)

Range: Endemic (*Casey, 1907*).

Notes. The validity of this species is dubious (M. A. Johnston, 2022, personal observation).

### *Telabis* Casey, 1890

Nomenclatural Authority: *Bousquet et al. (2018)*

Notes. This genus is in great need of revision and many synonymies (M. A. Johnston, 2022, personal observation). Six currently valid species are recorded from California (*Bousquet et al., 2018*; M. L. Gimmel, 2022, unpublished data).

### *Telabis serratus* (LeConte, 1866)

Nomenclatural Authority: *Bousquet et al. (2018)*

Literature Records: none

Digitized Records: Santa Catalina (1 LACM)

Range: Also known from mainland (*Horn, 1870*).

Notes. This species is widespread across southern California and is common in sandy habitats.

### Nyctoporini

Notes. One genus and five species of Nyctoporini have been recorded from California (*Bousquet et al., 2018*).

### *Nyctoporis* Eschscholtz, 1829

Nomenclatural Authority: *Bousquet et al. (2018)*

Digitized Records (genus-only): Anacapa (6 LACM), San Miguel (3 LACM), Santa Catalina (5 LACM), Santa Cruz (5 LACM)

Notes. The catalog of *Bousquet et al. (2018)* included a number of synonymies for this genus which at present comprises four valid species, but there is no key or modern treatment to identify these taxa. This genus is restricted to California, with all five of its species known from the state (*Bousquet et al., 2018*).

### *Nyctoporis carinata* LeConte, 1851

Nomenclatural Authority: *Bousquet et al. (2018)*

Literature Records: San Miguel (*Caterino, Chatzimanolis & Richmond, 2015*: 278), Santa Catalina (*Fall, 1897*: 238; *Baker, 1905*: 59; *Caterino, Chatzimanolis & Richmond, 2015*: 278), Santa Cruz (*Fall & Davis, 1934*: 144; *Polihronakis & Caterino, 2010a*: 426; *Caterino,*

*Chatzimanolis & Richmond, 2015*: 278), Santa Rosa (*Caterino, Chatzimanolis & Richmond, 2015*: 278)

Digitized Records: Anacapa (2 SBMNH), San Miguel (1 LACM; 7 SBMNH), Santa Catalina (1 BYUC; 13 SBMNH; 1 iNat), Santa Cruz (8 LACM; 12 SBMNH; 5 TAMU), Santa Rosa (16 SBMNH)

Range: Also known from mainland (*Caterino, Chatzimanolis & Richmond, 2015*).

Notes. *Polihronakis & Caterino (2010a)* showed that this species has a high degree of genetic distance in mitochondrial loci between mainland populations, which seem to be the same species as the populations on the islands. *Caterino, Chatzimanolis & Richmond (2015)* provided a detailed molecular phylogeographic analysis of this and several other beetle species for the California Channel Islands. It seems clear that the island populations and those of southern mainland California are the same species, but the species boundaries become more uncertain in the northern half of the state (M. A. Johnston, 2022, personal observation).

**Stenochiinae**
Notes. Four genera and 15 species of Stenochiinae, all belonging to the tribe Cnodalonini, have been recorded from California (*Bousquet et al., 2018*).

**Cibdelis Mannerheim, 1843**
Nomenclatural Authority: *Bousquet et al. (2018)*
Notes. This genus is only known from California and needs a modern revision. All five currently valid species are recorded from California (*Bousquet et al., 2018*).

**Cibdelis bachei LeConte, 1861**
Nomenclatural Authority: *Bousquet et al. (2018)*
Literature Records: San Clemente (*Horn, 1870*: 341; *Fall, 1901*: 170), Santa Barbara (*LeConte, 1861*: 353; *Fall, 1897*: 238; *Fall, 1901*: 170; *Miller, 1985a*: 20; *Miller & Miller, 1985*: 128), Santa Catalina (*Fall, 1901*: 170; *Miller, 1985a*: 20), Santa Cruz (*Miller, 1985a*: 20)
Digitized Records: Santa Catalina (9 LACM; 2 MAJC; 5 SBMNH), Santa Cruz (4 MAJC; 18 SBMNH), Santa Rosa (8 SBMNH)
Range: Endemic (*LeConte, 1861*).
Notes. *Miller & Miller (1985)* suggested that the published type locality of Santa Barbara Island may be due to mislabeling and that the type might actually be from Santa Cruz Island. No other records are known from Santa Barbara Island, so this seems plausible.

**Coelocnemis Mannerheim, 1843**
Nomenclatural Authority: *Bousquet et al. (2018)*
Notes. This genus was revised by *Doyen (1973)*, who provided excellent keys and species treatments. Specimens of this genus are often misidentified or placed within unsorted *Eleodes* specimens in collections. Three specimens from Santa Cruz Island in the BYUC are determined as *Coelocnemis dilaticollis* Mannerheim, 1843 (= *Coelocnemis californica* Mannerheim, 1843) which are deemed likely misidentified or mis-georeferenced, but they may represent a new island record for this genus. Another unverified specimen from Santa

Catalina Island determined by H.C. Fall as *C. dilaticollis* exists in the H.C. Fall collection at the Museum of Comparative Zoology, Harvard (S. Miller, 2022, personal communication). Six species are recorded from California (*Bousquet et al., 2018*).

### *Coelocnemis magna* LeConte, 1851

Nomenclatural Authority: *Bousquet et al. (2018)*
Literature Records: Santa Catalina (*Doyen, 1973*: 90)
Digitized Records: Santa Catalina (9 LACM; 1 SBMNH)
Range: Also known from mainland (*Doyen, 1973*).
Notes. This species is relatively abundant in coastal habitats of southern California and can most frequently be found under bark and on dead logs.

### Tenebrioninae

Notes. Thirteen tribes, 25 genera, and 72 species of Tenebrioninae are known to occur in California (*Bousquet et al., 2018*; M. L. Gimmel, 2022, unpublished data).

### Apocryphini

Notes. One genus and three species of Apocryphini have been recorded from California (*Bousquet et al., 2018*).

### *Apocrypha* Eschscholtz, 1831

Nomenclatural Authority: *Bousquet et al. (2018)*
Notes. This genus putatively has species in both North America and South America. There are three species in North America, all of which are recorded from and restricted to California (*Doyen & Kitayama, 1980*).

### *Apocrypha anthicoides* Eschscholtz, 1831

Nomenclatural Authority: *Bousquet et al. (2018)*
Literature Records: none
Digitized Records: Anacapa (5 SBMNH), Santa Cruz (3 SBMNH)
Range: Also known from mainland (*Doyen & Kitayama, 1980*).
Notes. This species is widespread along coastal California, but was not recorded from the islands in the last review of the genus (*Doyen & Kitayama, 1980*).

### Eulabini

Notes. Three genera and eight species of Eulabini have been recorded from California (*Bousquet et al., 2018*).

### *Apsena* LeConte, 1851

Nomenclatural Authority: *Bousquet et al. (2018)*
Notes. This genus was revised by *Blaisdell (1932)*, who provided keys and species treatments. Determinations made since then seem fairly dubious and there is likely some synonymy that needs to be made within this group (M. A. Johnston, 2022, personal observation). Six species are reported from California (*Bousquet et al., 2018*).

*Apsena barbarae* Blaisdell, 1932

Nomenclatural Authority: *Bousquet et al. (2018)*

Literature Records: Santa Catalina (*Blaisdell, 1932*: 63), Santa Cruz (*Fall & Davis, 1934*: 144)

Digitized Records: none

Range: Also known from mainland (*Blaisdell, 1932*).

Notes. *Blaisdell (1932)* described this species in the "*pubescens*-group" with a type locality of Santa Barbara. Subsequently identified material may have been determined as *Apsena pubescens* (LeConte, 1851), listed below. *Berry (1970*: 304), in a nomenclaturally unavailable dissertation, suggested that this species should be a junior synonym of *A. pubescens*.

*Apsena grossa* (LeConte, 1866)

Nomenclatural Authority: *Bousquet et al. (2018)*

Literature Records: Anacapa (*Berry, 1970*: 299), San Clemente (*LeConte, 1866a*: 118, *Horn, 1870*: 324; *Fall, 1897*: 238; *Fall, 1901*: 168; *Blaisdell, 1932*: 75; *Berry, 1970*: 299; *Doyen, 1974*: 87; *Miller, 1985a*: 20), San Nicolas (*Fall, 1901*: 169; *Blaisdell, 1932*: 75; *Berry, 1970*: 299; *Miller, 1985a*: 20), Santa Barbara (*Fall, 1901*: 169; *Blaisdell, 1932*: 75; *Berry, 1970*: 299; *Miller, 1985a*: 20; *Miller & Miller, 1985*: 128), Santa Catalina (*Blaisdell, 1932*: 75; *Berry, 1970*: 299; *Miller, 1985a*: 20), Santa Rosa (*Berry, 1970*: 299)

Digitized Records: Anacapa (8 LACM), San Clemente (74 LACM; 2 MAJC; 15 SBMNH; 13 SDNHM), San Nicolas (103 LACM; 16 SBMNH), Santa Barbara (130 LACM; 2 MAJC; 17 SBMNH), Santa Catalina (13 LACM), Santa Rosa (4 LACM)

Range: Endemic (*LeConte, 1866a*; *Fall, 1897*; *Blaisdell, 1932*; *Berry, 1970*).

Notes. This is the largest species of the genus and is immediately recognizable by its rotund form. *Blaisdell (1932)* included this species in the "*pubescens*-group". It has only been recorded from the Channel Islands and was listed as its original combination *Eulabis grossa* by *LeConte (1866a)*, *Horn (1870)*, and *Fall (1897, 1901)*.

*Apsena pubescens* (LeConte, 1851)

Nomenclatural Authority: *Bousquet et al. (2018)*

Literature Records: Santa Catalina (*Fall, 1897*: 238; *Baker, 1905*: 57; *Blaisdell, 1932*: 55; *Berry, 1970*: 308)

Digitized Records: San Clemente (1 LACM), Santa Catalina (1 LACM; 13 SBMNH), Santa Cruz (3 LACM; 14 SBMNH), Santa Rosa (1 SBMNH)

Range: Also known from mainland (*Blaisdell, 1932*; *Berry, 1970*).

Notes. *Blaisdell (1932)* largely separated this species, namesake of the "*pubescens*-group" and type of the genus, from *A. barbarae* by means of more attenuate males and difference in setal length. All island records of these two taxa will need to be critically examined as part of a revision of this genus.

*Apsena rufipes* (Eschscholtz, 1829)

Nomenclatural Authority: *Bousquet et al. (2018)*

Literature Records: none

Digitized Records: Santa Cruz (1 CSUC; 1 SBMNH), Santa Rosa (1 SBMNH)

Range: Also known from mainland (*Blaisdell, 1932*; *Berry, 1970*).

Notes. *Blaisdell (1932)* placed this species in its own species group, the "*rufipes*-group", but provided no island records. The single specimen from CSUC is determined as "*Eulabis rufipes*".

### *Epantius* LeConte, 1851

Nomenclatural Authority: *Bousquet et al. (2018)*

Notes. This is a monotypic genus with its single species known from California (*Bousquet et al., 2018*).

### *Epantius obscurus* LeConte, 1851

Nomenclatural Authority: *Bousquet et al. (2018)*

Literature Records: Anacapa (*Blaisdell, 1943*: 238; *Berry, 1970*: 294), San Nicolas (*Blaisdell, 1932*: 94; *Blaisdell, 1943*: 238), Santa Catalina (*Berry, 1970*: 294), Santa Cruz (*Blaisdell, 1932*: 94; *Blaisdell, 1943*: 238; *Berry, 1970*: 294), Santa Rosa (*Fall, 1897*: 238; *Blaisdell, 1943*: 238)

Digitized Records: Anacapa (41 LACM), San Clemente (2 LACM; 4 SBMNH), San Miguel (2 LACM; 46 SBMNH; 1 SDNHM), San Nicolas (95 LACM; 7 SBMNH), Santa Catalina (4 SBMNH), Santa Cruz (98 LACM; 33 SBMNH), Santa Rosa (1 LACM; 5 SBMNH)

Range: Also known from mainland (*Blaisdell, 1932*).

Notes. This species is common along the California shoreline and can be found under wrack and dune vegetation near and above the high-tide line. *Fall (1897)* recorded this species as *Eulabis obscura*.

### Helopini

Notes. Two genera and 25 species of Helopini are known to occur in California (*Bousquet et al., 2018*; M. L. Gimmel, 2022, unpublished data).

### *Helops* Fabricius, 1775

Nomenclatural Authority: *Bousquet et al. (2018)*

Literature Records (genus-only): Santa Barbara (*Miller & Miller, 1985*: 128), Santa Catalina (*Fall, 1897*: 238)

Digitized Records (genus-only): San Clemente (6 SDNHM)

Notes. This genus, as currently circumscribed, is worldwide and the New World species are in great need of revision, all likely not being congeneric with the European type species. The *Helops* fauna of the western United States is particularly in need of revision, and both literature and digitized records should be reviewed. The most reliable key is that of *Horn (1870)*. *Miller & Miller (1985)* referred to a species of *Helops* other than *H. bachei* LeConte, 1861 from Santa Barbara Island on the authority of T.J. Spilman. There are 23 species known to occur in California (*Bousquet et al., 2018*; M. L. Gimmel, 2022, unpublished data).

### *Helops bachei* LeConte, 1861

Nomenclatural Authority: *Bousquet et al. (2018)*

Literature Records: San Clemente (*Doyen, 1974*: 87), Santa Barbara (*LeConte, 1861*: 353; *Horn, 1870*: 396; *Fall, 1897*: 238; *Fall, 1901*: 175; *Miller & Miller, 1985*: 128)

Digitized Records: Anacapa (1 LACM), San Clemente (9 LACM; 1 SBMNH), San Miguel (5 SBMNH), San Nicolas (13 LACM; 9 SBMNH), Santa Barbara (3 LACM), Santa Catalina (3 LACM; 15 SBMNH), Santa Cruz (1 CSUC; 1 MAJC; 3 SBMNH), Santa Rosa (1 LACM; 20 SBMNH)

Range: Endemic (*LeConte, 1861*; *Fall, 1901*); also known from mainland (*Horn, 1870*; *Fall, 1901*).

Notes. *Fall (1901*: 175) reported this species from the mainland but perhaps the "true form" is only found on the "Santa Barbara Islands". This species is very similar to the mainland species *Helops rugicollis* LeConte, 1866, but most noticeably differs by having rounded tubercles on the elytra. This complex and all the island records need to be closely examined. Whether this is an island endemic species is hard to know and is debatable according to the literature.

### *Helops blaisdelli* Casey, 1891

Nomenclatural Authority: *Bousquet et al. (2018)*

Literature Records: San Nicolas (*Cockerell, 1940*: 284)

Digitized Records: none

Range: Also known from mainland (*Casey, 1891*).

Notes. This identification by *Cockerell (1940*: 284) is somewhat dubious. *Casey (1891)* described this species from San Diego and likened it to *H. bachei*. It is possible this taxon corresponds to the mainland populations of the latter species reported by *Horn (1870)*, or it could be a valid species and, the *Cockerell (1940)* record notwithstanding, validates *H. bachei* as a true island endemic with a mainland sister species. Until this genus is revised, this record cannot be fully discounted for the Channel Islands.

### *Helops rugicollis* LeConte, 1866

Nomenclatural Authority: *Bousquet et al. (2018)*

Literature Records: none

Digitized Records: Santa Catalina (2 SBMNH)

Range: Also known from mainland (*Horn, 1870*).

Notes. See discussion on two species above; this taxon is dubious for the islands but is in an unrevised species complex with the other two species recorded from the Channel Islands. The true identity and number of species on the Channel Islands requires revision.

### Triboliini

Notes. Five genera and 13 species of Triboliini are known to occur in California (*Bousquet et al., 2018*; M. L. Gimmel, 2022, unpublished data).

### *Tribolium* MacLeay, 1825

Nomenclatural Authority: *Bousquet et al. (2018)*

Notes. This genus contains several cosmopolitan species that are strongly synanthropic. Six species are recorded from California (*Bousquet et al., 2018*; M. L. Gimmel, 2022, unpublished data)

### *Tribolium castaneum* (Herbst, 1797)
Nomenclatural Authority: *Bousquet et al. (2018)*
Literature Records: none
Digitized Records: Santa Catalina (1 BYUC)
Range: Also known from mainland (*Bousquet et al., 2018*).
Notes. This species, commonly referred to as the red flour beetle, is synanthropic and found throughout California wherever people live.

### Zopheridae
Notes. Two subfamilies, 18 genera, and 42 species of Zopheridae have been recorded from California (M. L. Gimmel, 2022, unpublished data).

### Colydiinae
Notes. Three tribes, 12 genera, and 28 species of Colydiinae have been recorded from California (M. L. Gimmel, 2022, unpublished data). The North American species of the subfamily were treated by *Stephan (1989)*; *Ivie et al. (2016)* provided an overview and key to the New World genera, as well as a checklist of New World species.

### Rhagoderini
Notes. One genus and three species of Rhagoderini have been recorded from California (*Stephan, 1989*; *Krinsky, 2015*).

### *Rhagodera* Mannerheim, 1843
Nomenclatural Authority: *Stephan (1989)*
Notes. Three species of *Rhagodera* have been recorded from California (*Stephan, 1989*; *Krinsky, 2015*).

### *Rhagodera costaefragmenta* Krinsky, 2015
Nomenclatural Authority: *Ivie et al. (2016)*
Literature Records: San Clemente (*Krinsky, 2015*: 294)
Digitized Records: none
Range: Endemic (*Krinsky, 2015*).
Notes. This species was described from five specimens collected on San Clemente Island which are reportedly deposited in the YPMC (*Krinsky, 2015*).

### *Rhagodera interrupta* Stephan, 1989
Nomenclatural Authority: *Ivie et al. (2016)*
Literature Records: San Nicolas (*Krinsky, 2015*: 293)
Digitized Records: San Nicolas (6 SBMNH)
Range: Unknown.
Notes. This species was described from five specimens located in the Ulke collection at the Carnegie Museum of Natural History which were only labeled as "California" (*Stephan,*

*1989*). No other localities have been recorded in print or in digitized specimen data besides the island record given here.

### *Rhagodera tuberculata* (Mannerheim, 1843)
Nomenclatural Authority: *Ivie et al. (2016)*
Literature Records: San Clemente (*Miller & Miller, 1985*: 127), Santa Barbara (*Miller & Miller, 1985*: 127; *Krinsky, 2015*: 293), Santa Cruz (*Krinsky, 2015*: 293)
Digitized Records: Santa Cruz (1 SBMNH)
Range: Also known from mainland (*Stephan, 1989*).
Notes. This is one of the more widespread and commonly collected species of this genus, with most of its known specimens originating from Los Angeles County, California (*Stephan, 1989*).

### Synchitini
Notes. Nine genera and 22 species of Synchitini have been recorded from California (*Stephan, 1989*; M. L. Gimmel, 2022, unpublished data).

### *Lasconotus* Erichson, 1845
Nomenclatural Authority: *Stephan (1989)*
Notes. Twelve species of *Lasconotus* have been recorded from California (*Stephan, 1989*).

### *Lasconotus linearis* Crotch, 1874
Nomenclatural Authority: *Ivie et al. (2016)*
Literature Records: none
Digitized Records: Santa Cruz (3 SBMNH), Santa Rosa (1 SBMNH)
Range: Also known from mainland (*Crotch, 1874*).

### *Megataphrus* Casey, 1890
Nomenclatural Authority: *Stephan (1989)*
Notes. One species of *Megataphrus* has been recorded from California (*Stephan, 1989*).

### *Megataphrus tenuicornis* Casey, 1890
Nomenclatural Authority: *Ivie et al. (2016)*
Literature Records: none
Digitized Records: Santa Rosa (16 SBMNH)
Range: Also known from mainland (*Stephan, 1989*).

### *Synchita* Hellwig, 1792
Nomenclatural Authority: *Stephan (1989)*
Notes. One species of *Synchita* has been recorded from California (*Stephan, 1989*).

### *Synchita lecontei* Ivie et al., 2016
Nomenclatural Authority: *Ivie et al. (2016)*
Literature Records: none
Digitized Records: Santa Cruz (2 SBMNH)
Range: Also known from mainland (*Stephan, 1989*).

Notes. This species was known as *Microsicus variegatus* (LeConte, 1858) in *Stephan (1989)*.

**Zopherinae: Zopherini**
Notes. Four tribes, six genera, and 14 species of Zopherinae, of which two genera and seven species belong to Zopherini, have been recorded from California (M. L. Gimmel, 2022, unpublished data).

***Phloeodes* LeConte, 1862**
Nomenclatural Authority: *Foley & Ivie (2008)*
Notes. Two species of *Phloeodes* have been recorded from California (*Foley & Ivie, 2008*). The species of this genus were revised by *Foley & Ivie (2008)*.

***Phloeodes diabolicus* (LeConte, 1851)**
Nomenclatural Authority: *Foley & Ivie (2008)*
Literature Records: Santa Cruz (*LeConte, 1876*: 299; *Fall, 1897*: 238; *Fall & Davis, 1934*: 144)
Digitized Records: none
Range: Also known from mainland (*García-París, Coca-Abia & Parra-Olea, 2006*; *Foley & Ivie, 2008*; *Polihronakis & Caterino, 2010b*).
Notes. No island records were given by *García-París, Coca-Abia & Parra-Olea (2006)*, *Foley & Ivie (2008)*, or *Polihronakis & Caterino (2010b)*. The literature records of this species are curious. This is a large and charismatic species which is readily separable from its congener *Phloeodes plicatus* (LeConte, 1859). We presume that *LeConte (1876)* and those that followed merely recorded the wrong species name, but these authors were familiar with these two species so it may yet prove to be a valid record.

***Phloeodes plicatus* (LeConte, 1859)**
Nomenclatural Authority: *Foley & Ivie (2008)*
Literature Records: Santa Catalina (*Polihronakis & Caterino, 2010b*: 3), Santa Cruz (*García-París, Coca-Abia & Parra-Olea, 2006*: 228; *Foley & Ivie, 2008*: 46)
Digitized Records: Santa Catalina (5 SBMNH), Santa Cruz (2 OSUC; 7 SBMNH)
Range: Also known from mainland (*García-París, Coca-Abia & Parra-Olea, 2006*; *Foley & Ivie, 2008*; *Polihronakis & Caterino, 2010b*).
Notes. We have examined specimens of this species and are confident of its occurrence on the islands, but see comments under *P. diabolicus*.

**COCCINELLOIDEA**

**Akalyptoischiidae**
Notes. One genus and 17 species of Akalyptoischiidae have been recorded from California (*Hartley, Andrews & McHugh, 2008*).

***Akalyptoischion* Andrews, 1976**
Nomenclatural Authority: *Hartley, Andrews & McHugh (2008)*

Notes. Seventeen species of *Akalyptoischion* have been recorded from California (*Hartley, Andrews & McHugh, 2008*). This genus was revised by *Hartley, Andrews & McHugh (2008)*.

***Akalyptoischion heterotrichos* Hartley, Andrews & McHugh, 2008**
Nomenclatural Authority: *Hartley, Andrews & McHugh (2008)*
Literature Records: none
Digitized Records: Santa Catalina (1 SBMNH)
Range: Also known from mainland (*Hartley, Andrews & McHugh, 2008*).

***Akalyptoischion hormathos* Andrews, 1976**
Nomenclatural Authority: *Hartley, Andrews & McHugh (2008)*
Literature Records: Santa Barbara (*Andrews, 1976a*: 9; *Miller & Miller, 1985*: 127; *Hartley, Andrews & McHugh, 2008*: 37)
Digitized Records: San Clemente (19 SBMNH), Santa Catalina (14 SBMNH), Santa Cruz (15 SBMNH), Santa Rosa (30 SBMNH)
Range: Also known from mainland (*Andrews, 1976a*; *Hartley, Andrews & McHugh, 2008*).

**Cerylonidae, NEW FAMILY RECORD**
Notes. Three genera and four species of Cerylonidae have been recorded from California (*Lawrence & Stephan, 1975*).

***Cerylon* Latreille, 1802**
Nomenclatural Authority: (*Lawrence & Stephan, 1975*).
Notes. The genus *Cerylon* contains two species recorded from California (*Lawrence & Stephan, 1975*). These were keyed by *Lawrence & Stephan (1975)*.

***Cerylon unicolor* (Ziegler, 1845)**
Nomenclatural Authority: *Lawrence & Stephan (1975)*
Literature Records: none
Digitized Records: Santa Cruz (2 SBMNH)
Range: Also known from mainland (*Lawrence & Stephan, 1975*).

**Coccinellidae**
Notes. The known North American species of Coccinellidae were fully treated by *Gordon (1985)*, with additions to the introduced fauna by *Gordon & Vandenberg (1991)*.
Two subfamilies, 42 genera, and 175 species are known from California (M. L. Gimmel, 2022, unpublished data).

**Coccinellinae**
Notes. Eight tribes, 38 genera, and 164 species of Coccinellinae have been recorded from California (*Gordon, 1985*; M. L. Gimmel, 2022, unpublished data).

**Cephaloscymnini**
Notes. One genus and species of Cephaloscymnini has been recorded from California (*Gordon, 1985*).

### *Cephaloscymnus* Crotch, 1873

Nomenclatural Authority: *Gordon (1985)*

Notes. One species of *Cephaloscymnus* is recorded from California (*Gordon, 1985*).

### *Cephaloscymnus occidentalis* Horn, 1895

Nomenclatural Authority: *Gordon (1985)*

Literature Records: Santa Catalina (*Fall, 1897*: 237; *Fall, 1901*: 87)

Digitized Records: none

Range: Also known from mainland (*Fall, 1901*; *Gordon, 1985*).

### Chilocorini

Notes. Five genera and 17 species of Chilocorini have been recorded from California (*Gordon, 1985*).

### *Axion* Mulsant, 1850

Nomenclatural Authority: *Gordon (1985)*

Notes. One species of *Axion* is recorded from California (*Gordon, 1985*).

### *Axion plagiatum* (Olivier, 1808)

Nomenclatural Authority: *Gordon (1985)*

Literature Records: none

Digitized Records: Santa Cruz (5 SBMNH)

Range: Also known from mainland (*Gordon, 1985*).

### *Chilocorus* Leach, 1815

Nomenclatural Authority: *Gordon (1985)*

Notes. Five species of *Chilocorus* have been recorded from California, two of which are adventive (*Gordon, 1985*). These were keyed by *Gordon (1985)*.

### *Chilocorus* undetermined species

Literature Records: Santa Catalina (*Seavey, 1892*: 263; *Fall, 1897*: 237)

Digitized Records: Santa Cruz (1 UCSB)

Notes. The records from *Seavey (1892)* and *Fall (1897)* were reported as *Chilocorus bivulnerus* Mulsant, 1850, a current junior synonym of *Chilocorus stigma* (Say, 1835), a species that does not occur in California. Early California records of *C. bivulnerus*, therefore, refer to either *Chilocorus fraternus* LeConte, 1860 or *Chilocorus orbus* Casey, 1899, two species which can only be reliably distinguished by examination of male genitalia (*Gordon, 1985*).

### Coccidulini

Notes. Nine genera and 51 species of Coccidulini have been recorded from California (*Gordon, 1985*; M. L. Gimmel, 2022, unpublished data).

### *Nephus* Mulsant, 1846

Nomenclatural Authority: *Gordon (1985)*

Digitized Records (genus-only): Anacapa (1 LACM), Santa Barbara (1 LACM), Santa Catalina (1 LACM)

Notes. Six species of *Nephus* in three subgenera (*Scymnobius* Casey, 1899; *Sidis* Mulsant, 1850; *Turboscymnus* Gordon, 1976) have been reported from California (*Gordon, 1985*).

### Nephus (Scymnobius) guttulatus (LeConte, 1852)

Nomenclatural Authority: *Gordon (1985)*

Literature Records: Santa Catalina (*Fall, 1897*: 237; *Fall, 1901*: 86)

Digitized Records: Anacapa (4 SBMNH), Santa Barbara (1 SBMNH), Santa Cruz (3 SBMNH)

Range: Also known from mainland (*Fall, 1901*; *Gordon, 1985*).

Notes. *Fall (1897*, *1901)* recorded this species as *Scymnus guttulatus*.

### Nephus (Scymnobius) sordidus (Horn, 1895)

Nomenclatural Authority: *Gordon (1985)*

Literature Records: none

Digitized Records: Anacapa (1 SBMNH), San Nicolas (9 SBMNH), Santa Catalina (2 SBMNH), Santa Rosa (1 SBMNH)

Range: Also known from mainland (*Gordon, 1985*).

### Nephus (Sidis) binaevatus (Mulsant, 1850)

Nomenclatural Authority: *Gordon (1985)*

Literature Records: Santa Catalina (*Cockerell, 1940*: 286; *Gordon, 1985*: 293 [map])

Digitized Records: Santa Catalina (1 iNat)

Range: Also known from mainland (*Gordon, 1985*).

Notes. This species was recorded as *Scymnus binaevatus* by *Cockerell (1940)*. It was introduced to California from South Africa in 1921 for mealybug control (*Gordon, 1985*).

### Rhyzobius Stephens, 1829

Nomenclatural Authority: *Gordon (1985)*

Notes. Two introduced species of *Rhyzobius* are recorded from California (*Gordon, 1985*).

### Rhyzobius forestieri (Mulsant, 1853)

Nomenclatural Authority: *Gordon (1985)*

Literature Records: Santa Catalina (*Pope, 1981*: 26), Santa Cruz (*Naughton et al., 2014*: 303)

Digitized Records: Santa Cruz (1 SBMNH)

Range: Also known from mainland (*Gordon, 1985*).

Notes. This species was introduced to North America (*Gordon, 1985*).

### Rhyzobius lophanthae (Blaisdell, 1892)

Nomenclatural Authority: *Gordon (1985)*

Literature Records: San Clemente (*Fall, 1897*: 237; *Cockerell, 1940*: 286)

Digitized Records: San Nicolas (1 SBMNH), Santa Cruz (2 SBMNH), Santa Rosa (1 SBMNH)

Range: Also known from mainland (*Gordon, 1985*).

Notes. *Fall (1897)* reported this species as "*Rhizobius lophanthae*", and *Cockerell (1940)* reported it as "*Lindorus lophantae*". This species was introduced to North America (*Gordon, 1985*).

**Scymnus Kugelann, 1794**
Nomenclatural Authority: *Gordon (1985)*
Digitized Records (genus-only): Santa Barbara (1 SBMNH)
Notes. Thirty-six species of *Scymnus* have been reported from California, all but four belonging to the subgenus *Pullus* Mulsant, 1846 and the remainder to *Scymnus* (*Scymnus*) (*Gordon, 1985*). These were keyed out by *Gordon (1976)* and *Gordon (1985)*. The above Santa Barbara Island record is based on a single female specimen of *Pullus* that keys out to couplet 29 of *Gordon's (1985)* key to *Pullus* of "Region IV", which relies on male genitalia to separate the species.

**Scymnus (Pullus) ardelio Horn, 1895**
Nomenclatural Authority: *Gordon (1985)*
Literature Records: San Clemente (*Fall, 1897*: 237), Santa Catalina (*Fall, 1897*: 237)
Digitized Records: none
Range: Also known from mainland (*Gordon, 1985*).

**Scymnus (Pullus) cervicalis Mulsant, 1850**
Nomenclatural Authority: *Gordon (1985)*
Literature Records: Santa Catalina (*Fall, 1897*: 237; *Fall, 1901*: 86)
Digitized Records: Santa Cruz (3 SBMNH), Santa Rosa (3 SBMNH)
Range: Also known from mainland (*Fall, 1901*; *Gordon, 1985*).

**Scymnus (Pullus) coniferarum Crotch, 1874**
Nomenclatural Authority: *Gordon (1985)*
Literature Records: none
Digitized Records: Santa Cruz (1 SBMNH)
Range: Also known from mainland (*Gordon, 1985*).

**Scymnus (Pullus) falli Gordon, 1976**
Nomenclatural Authority: *Gordon (1985)*
Literature Records: Santa Barbara (*Gordon, 1985*: 199; *Miller, 1985a*: 20; *Miller & Miller, 1985*: 127), Santa Cruz (*Gordon, 1976*: 140; *1985*: 199 [map]; *Miller, 1985a*: 20; *Miller & Miller, 1985*: 127; *Naughton et al., 2014*: 303), Santa Rosa (*Miller, 1985a*: 20; *Miller & Miller, 1985*: 127)
Digitized Records: San Miguel (1 SBMNH), Santa Cruz (8 SBMNH), Santa Rosa (3 SBMNH)
Range: Endemic (*Gordon, 1976*; *Gordon, 1985*; *Miller, 1985a*; *Miller & Miller, 1985*; *Naughton et al., 2014*).

**Scymnus (Pullus) jacobianus Casey, 1899**
Nomenclatural Authority: *Gordon (1985)*

Literature Records: none
Digitized Records: San Clemente (2 SBMNH), San Miguel (2 SBMNH), Santa Barbara (6 SBMNH)
Range: Also known from mainland (*Gordon, 1985*).

### *Scymnus* (*Pullus*) *loewii* Mulsant, 1850
Nomenclatural Authority: *Gordon (1985)*
Literature Records: Santa Cruz (*Fall & Davis, 1934*: 143; *Gordon, 1976*: 124)
Digitized Records: San Clemente (2 SBMNH), Santa Cruz (1 SBMNH)
Range: Also known from mainland (*Gordon, 1976*, *1985*).
Notes. *Fall & Davis (1934)* recorded this species as *Scymnus cinctus* LeConte, 1852, which is now a junior synonym of *S. loewii* (see *Gordon, 1985*).

### *Scymnus* (*Pullus*) *marginicollis* Mannerheim, 1843
Nomenclatural Authority: *Gordon (1985)*
Literature Records: Santa Catalina (*Horn, 1895*: 105; *Fall, 1897*: 237), Santa Cruz (*Gordon, 1976*: 128)
Digitized Records: Anacapa (1 SBMNH), San Nicolas (5 SBMNH), Santa Catalina (4 SBMNH), Santa Cruz (10 LACM; 1 SBMNH), Santa Rosa (2 LACM; 1 SBMNH)
Range: Also known from mainland (*Gordon, 1976*, *1985*).

### *Scymnus* (*Pullus*) *pallens* LeConte, 1852
Nomenclatural Authority: *Gordon (1985)*
Literature Records: Santa Catalina (*Cockerell, 1940*: 286), Santa Cruz (*Gordon, 1976*: 86; *Naughton et al., 2014*: 303)
Digitized Records: Santa Cruz (4 SBMNH), Santa Rosa (2 SBMNH)
Range: Also known from mainland (*Gordon, 1976*, *1985*).

### *Scymnus* (*Scymnus*) *difficilis* Casey, 1899
Nomenclatural Authority: *Gordon (1985)*
Literature Records: none
Digitized Records: San Miguel (10 SBMNH), Santa Rosa (1 SBMNH)
Range: Also known from mainland (*Gordon, 1985*).

### *Scymnus* (*Scymnus*) *fenderi* Malkin, 1943
Nomenclatural Authority: *Gordon (1985)*
Literature Records: none
Digitized Records: Santa Rosa (3 SBMNH)
Range: Also known from mainland (*Gordon, 1985*).

### *Scymnus* (*Scymnus*) *nebulosus* LeConte, 1852
Nomenclatural Authority: *Gordon (1985)*
Literature Records: San Miguel (*Miller & Davis, 1986*: 550), Santa Catalina (*Fall, 1897*: 237; *Fall, 1901*: 86; *Cockerell, 1940*: 286)

Digitized Records: Santa Catalina (1 OSUC; 5 SBMNH; 1 iNat), Santa Cruz (28 LACM; 10 SBMNH), Santa Rosa (2 SBMNH)

Range: Also known from mainland (*Fall, 1901*; *Gordon, 1985*).

### *Stethorus* Weise, 1885

Nomenclatural Authority: *Gordon (1985)*

Notes. One species of *Stethorus* has been reported from California (*Gordon, 1985*).

### *Stethorus punctum* (LeConte, 1852)

Nomenclatural Authority: *Gordon (1985)*

Literature Records: none

Digitized Records: Santa Catalina (3 SBMNH), Santa Cruz (1 SBMNH)

Range: Also known from mainland (*Gordon, 1985*).

Notes. The subspecies of *S. punctum* occurring in California is *S. p. picipes* Casey, 1899 (*Gordon, 1985*).

### *Zagloba* Casey, 1899

Nomenclatural Authority: *Gordon (1985)*

Notes. One species of *Zagloba* has been reported from California (*Gordon, 1985*).

### *Zagloba ornata* (Horn, 1895)

Nomenclatural Authority: *Gordon (1985)*

Literature Records: Santa Catalina (*Horn, 1895*: 112; *Fall, 1897*: 237; *Fall, 1901*: 87)

Digitized Records: San Miguel (2 SBMNH), Santa Cruz (1 SBMNH)

Range: Also known from mainland (*Fall, 1901*; *Gordon, 1985*).

Notes. *Horn (1895)* and *Fall (1897, 1901)* recorded this species as *Cephaloscymnus ornatus*.

### Coccinellini

Notes. Sixteen genera and 41 species of Coccinellini have been recorded from California (*Gordon, 1985*; M. L. Gimmel, 2022, unpublished data).

### *Coccinella* Linnaeus, 1758

Nomenclatural Authority: *Gordon (1985)*

Digitized Records (genus-only): San Clemente (4 LACM), San Miguel (12 LACM), San Nicolas (29 LACM), Santa Catalina (1 LACM), Santa Cruz (2 LACM)

Notes. Eleven species of *Coccinella* have been recorded from California (*Gordon, 1985*; *Gordon & Vandenberg, 1991*). This genus was revised for North America by *Brown (1962)*, but an updated key was provided by *Gordon (1985)*. *Fall (1901*: 84) reported the taxon "*Coccinella transversoguttata* var. *transversalis*" from "the islands". *Dobzhansky (1931*: 16), however, noted that "This form [*C. transversoguttata*] is apparently lacking in southern California". We did not attempt to include this record below.

### *Coccinella californica* Mannerheim, 1843

Nomenclatural Authority: *Gordon (1985)*

Literature Records: Anacapa (*Miller & Miller, 1985*: 126), San Clemente (*Fall, 1897*: 237; *Dobzhansky, 1931*: 13; *Miller & Miller, 1985*: 126), San Nicolas (*Fall, 1897*: 237;

*Dobzhansky, 1931*: 13; *Miller & Miller, 1985*: 126), Santa Barbara (*Miller & Miller, 1985*: 126), Santa Catalina (*Fall, 1897*: 237; *Miller & Miller, 1985*: 126), Santa Cruz (*Dobzhansky, 1931*: 13; *Miller & Miller, 1985*: 126), Santa Rosa (*Fall, 1897*: 237; *Dobzhansky, 1931*: 13; *Miller & Miller, 1985*: 126)

Digitized Records: Anacapa (3 LACM; 7 SBMNH), San Clemente (2 LACM; 1 SBMNH; 25 iNat), San Miguel (2 LACM; 1 SBMNH), San Nicolas (11 LACM; 2 SBMNH; 9 iNat), Santa Barbara (18 LACM; 1 SBMNH), Santa Catalina (8 LACM; 8 iNat), Santa Cruz (2 LACM; 7 SBMNH; 3 UCSB; 4 iNat), Santa Rosa (3 LACM; 14 SBMNH; 5 iNat)

Range: Also known from mainland (*Gordon, 1985*).

### *Coccinella johnsoni* Casey, 1908

Nomenclatural Authority: *Gordon (1985)*

Literature Records: San Clemente (*Dobzhansky, 1931*: 14; *Brown, 1962*: 794; *Miller & Miller, 1985*: 127), San Nicolas (*Dobzhansky, 1931*: 14; *Brown, 1962*: 794; *Miller & Miller, 1985*: 127), Santa Barbara (*Miller & Miller, 1985*: 127)

Digitized Records: San Clemente (8 SBMNH), San Nicolas (10 SBMNH), Santa Barbara (3 LACM)

Range: Also known from mainland (*Dobzhansky, 1931*; *Brown, 1962*; *Gordon, 1985*).

### *Coccinella novemnotata* Herbst, 1793

Nomenclatural Authority: *Gordon (1985)*

Literature Records: none

Digitized Records: San Miguel (1 SBMNH)

Range: Also known from mainland (*Gordon, 1985*).

### *Coccinella septempunctata* (Linnaeus, 1758)

Nomenclatural Authority: *Gordon (1985)*

Literature Records: none

Digitized Records: Anacapa (1 iNat), San Clemente (2 SBMNH; 13 iNat), San Miguel (2 SBMNH), San Nicolas (1 SBMNH; 3 iNat), Santa Barbara (1 iNat), Santa Catalina (4 SBMNH; 3 iNat), Santa Cruz (4 SBMNH; 7 iNat), Santa Rosa (4 iNat)

Range: Also known from mainland (*Gordon, 1985*).

Notes. This species was introduced to North America for control of aphids (*Gordon, 1985*).

### *Cycloneda* Crotch, 1871

Nomenclatural Authority: *Gordon (1985)*

Digitized Records (genus-only): Santa Cruz (1 iNat)

Notes. Two species of *Cycloneda* have been recorded from California (*Gordon, 1985*).

### *Cycloneda polita* Casey, 1899

Nomenclatural Authority: *Gordon (1985)*

Literature Records: Santa Catalina (*Fall, 1897*: 237), Santa Cruz (*Naughton et al., 2014*: 303)

Digitized Records: Santa Cruz (3 LACM; 5 SBMNH; 5 UCSB), Santa Rosa (1 LACM)

Range: Also known from mainland (*Gordon, 1985*).

Notes. *Fall (1897)* recorded this species as "*Cycloneda oculata* Fabricius".

### *Cycloneda sanguinea* (Linnaeus, 1763)

Nomenclatural Authority: *Gordon (1985)*

Literature Records: Santa Catalina (*Seavey, 1892*: 263; *Fall, 1897*: 237)

Digitized Records: Anacapa (1 SBMNH), Santa Catalina (1 LACM; 5 SBMNH; 2 iNat), Santa Cruz (12 SBMNH; 2 UCSB; 6 iNat)

Range: Also known from mainland (*Gordon, 1985*).

Notes. *Seavey (1892)* recorded this species as *Coccinella sanguinea*. The subspecies occurring in California is the nominate subspecies, *C. s. sanguinea* (Linnaeus, 1763).

### *Hippodamia* Dejean, 1837

Nomenclatural Authority: *Gordon (1985)*

Notes. Ten species of *Hippodamia* have been reported from California (*Gordon, 1985*).

### *Hippodamia convergens* Guérin-Méneville, 1842

Nomenclatural Authority: *Gordon (1985)*

Literature Records: Anacapa (*Miller & Miller, 1985*: 127), San Miguel (*Miller & Miller, 1985*: 127), Santa Barbara (*Miller & Miller, 1985*: 127), Santa Catalina (*Seavey, 1892*: 263; *Fall, 1897*: 237; *Cockerell, 1940*: 286), Santa Cruz (*Miller & Miller, 1985*: 127), Santa Rosa (*Miller & Miller, 1985*: 127)

Digitized Records: Anacapa (8 SBMNH), San Clemente (1 SBMNH; 6 iNat), San Miguel (1 SBMNH), San Nicolas (1 LACM; 1 SBMNH; 1 iNat), Santa Barbara (1 SBMNH), Santa Catalina (1 SBMNH; 3 iNat), Santa Cruz (27 LACM; 15 SBMNH; 3 UCSB; 2 iNat), Santa Rosa (1 LACM; 5 SBMNH)

Range: Also known from mainland (*Chapin, 1946*; *Gordon, 1985*).

Notes. The map in *Chapin (1946*: plate 21) shows the presence of this species on at least two islands, which were not mentioned. *Cockerell (1940)* reported this species as "*Hippodamia obsoleta* LeConte" (= *Hippodamia convergens* var. *obsoleta* Crotch, 1873), a current synonym of *H. convergens* (see *Gordon, 1985*).

### *Hippodamia quinquesignata* (Kirby, 1837)

Nomenclatural Authority: *Gordon (1985)*

Literature Records: San Clemente (*Fall, 1897*: 237), San Nicolas (*Cockerell, 1940*: 286), Santa Catalina (*Seavey, 1892*: 263; *Fall, 1897*: 237), Santa Cruz (*Fall & Davis, 1934*: 143), Santa Rosa (*Fall, 1897*: 237)

Digitized Records: Anacapa (5 SBMNH), San Clemente (10 SBMNH; 1 iNat), San Miguel (55 LACM; 16 SBMNH), San Nicolas (12 LACM; 7 SBMNH), Santa Catalina (3 LACM; 4 SBMNH), Santa Cruz (12 LACM; 15 SBMNH; 18 iNat), Santa Rosa (29 LACM; 37 SBMNH; 1 iNat)

Range: Also known from mainland (*Gordon, 1985*).

Notes. *Seavey (1892)*, *Fall (1897)*, and *Fall & Davis (1934)* recorded this species as *Hippodamia ambigua* LeConte, 1852. *Cockerell (1940)* recorded it as "*H. quinquesignata*

Kirby, variety". *Chapin (1946*: 16) recorded this taxon as "Abundant in western California and the Channel Islands" as the subspecies *H. q. punctulata* LeConte, 1852, now considered a synonym of *H. q. ambigua* LeConte, 1852, the only subspecies occurring in coastal California (see *Gordon, 1985*). *Chapin's (1946*: plate 20) map shows its presence on multiple islands. *Fall (1897*: 239) noted the Santa Rosa specimens might well be *Hippodamia convergens*. Based on the virtual absence of elytral dark maculation (except for scutellar spot), all *Hippodamia quinquesignata* (Kirby, 1837) represented on the Channel Islands belong to *H. q. ambigua*. Members of this subspecies with the white convergent lines present on the pronotum can be difficult to distinguish from immaculate members of *H. convergens* without examination of male genitalia (*Gordon, 1985*: 727).

***Olla* Casey, 1899**
Nomenclatural Authority: *Gordon (1985)*
Notes. One species of *Olla* has been recorded from California (*Gordon, 1985*).

***Olla v-nigrum* (Mulsant, 1866)**
Nomenclatural Authority: *Gordon (1985)*
Literature Records: none
Digitized Records: Anacapa (1 LACM), Santa Catalina (1 LACM)
Range: Also known from mainland (*Gordon, 1985*).

***Paranaemia* Casey, 1899**
Nomenclatural Authority: *Gordon (1985)*
Notes. One species of *Paranaemia* has been recorded from California (*Gordon, 1985*).

***Paranaemia vittigera* (Mannerheim, 1843)**
Nomenclatural Authority: *Gordon (1985)*
Literature Records: Santa Cruz (*LeConte, 1876*: 298; *Fall, 1897*: 237; *Fall & Davis, 1934*: 143)
Digitized Records: none
Range: Also known from mainland (*Gordon, 1985*).
Notes. This species was recorded as *Hippodamia vittigera* by *LeConte (1876)* and *Fall (1897)*, and as *Ceratomegilla vittigera* by *Fall & Davis (1934)*.

***Psyllobora* Dejean, 1836**
Nomenclatural Authority: *Gordon (1985)*
Digitized Records (genus-only): Santa Cruz (10 EMEC; 17 LACM; 9 UCSB)
Notes. Three species of *Psyllobora* have been recorded from California (*Gordon, 1985*).

***Psyllobora renifer* Casey, 1899**
Nomenclatural Authority: *Gordon (1985)*
Literature Records: none
Digitized Records: Santa Cruz (1 UCSB)
Range: Also known from mainland (*Gordon, 1985*).

*Psyllobora vigintimaculata* (Say, 1824)

Nomenclatural Authority: *Gordon (1985)*

Literature Records: Santa Catalina (*Seavey, 1892*: 263; *Fall, 1897*: 237), Santa Cruz (*Fall & Davis, 1934*: 143; *Naughton et al., 2014*: 303)

Digitized Records: San Miguel (6 SBMNH), San Nicolas (4 SBMNH), Santa Catalina (12 SBMNH; 1 iNat), Santa Cruz (21 SBMNH), Santa Rosa (2 SBMNH)

Range: Also known from mainland (*Gordon, 1985*).

Notes. This species was recorded as *Psyllobora taedata* LeConte, 1860 by *Seavey (1892)* and *Fall (1897)*, and as *Psyllobora 20-maculata* var. *taedata* by *Fall & Davis (1934)*. *Seavey (1892)* reported it from *Artemisia californica*.

**Diomini**

Notes. One genus and three species of Diomini have been recorded from California (*Gordon, 1985*).

*Diomus* Mulsant, 1850

Nomenclatural Authority: *Gordon (1985)*

Notes. Three species of *Diomus* have been recorded from California (*Gordon, 1985*).

*Diomus debilis* (LeConte, 1852)

Nomenclatural Authority: *Gordon (1985)*

Literature Records: none

Digitized Records: Anacapa (3 SBMNH), Santa Cruz (1 SBMNH)

Range: Also known from mainland (*Gordon, 1985*).

**Hyperaspidini**

Notes. Four genera and 47 species of Hyperaspidini have been recorded from California (*Gordon, 1985*).

*Hyperaspidius* Crotch, 1873

Nomenclatural Authority: *Gordon (1985)*

Digitized Records (genus-only): San Miguel (1 SBMNH), San Nicolas (1 LACM; 1 SBMNH)

Notes. Eight species of *Hyperaspidius* have been recorded from California (*Gordon, 1985*).

*Hyperaspidius comparatus* Casey, 1899

Nomenclatural Authority: *Gordon (1985)*

Literature Records: San Miguel (*Cockerell, 1940*: 286; *Gordon, 1985*: 361)

Digitized Records: none

Range: Also known from mainland (*Gordon, 1985*).

Notes. *Cockerell's (1940)* San Miguel Island record was referred to *Hyperaspidius vittigerus* (LeConte, 1852), which was indicated as previously bearing the name *Hyperaspidius trimaculatus* (Linnaeus, 1767). However, *Hyperaspidius vittigerus* was not shown to occur west of the Rocky Mountains by *Gordon (1985)*, so *Cockerell's (1940)* record probably refers to *H. comparatus*.

*Hyperaspis* **Redtenbacher, 1844**
Nomenclatural Authority: *Gordon (1985)*
Literature Records (genus-only): Santa Cruz (*Naughton et al., 2014*: 303)
Digitized Records (genus-only): San Clemente (1 SBMNH), Santa Rosa (4 SBMNH)
Notes. Thirty-six species of *Hyperaspis* have been recorded from California (*Gordon, 1985*). The *Naughton et al. (2014)* record above is based on a record of "*Hyperaspis* sp." in addition to the three taxa listed below.

*Hyperaspis lateralis* **Mulsant, 1850**
Nomenclatural Authority: *Gordon (1985)*
Literature Records: Santa Catalina (*Seavey, 1892*: 262; *Fall, 1897*: 237), Santa Cruz (*Naughton et al., 2014*: 303)
Digitized Records: Santa Catalina (1 SBMNH), Santa Cruz (1 SBMNH)
Range: Also known from mainland (*Gordon, 1985*).
Notes. This species was reported from *Artemisia californica* by *Seavey (1892)*.

*Hyperaspis* **species near** *annexa* **LeConte, 1852**
Nomenclatural Authority: *Gordon (1985)*
Literature Records: Santa Cruz (*Naughton et al., 2014*: 303)
Digitized Records: Santa Cruz (3 SBMNH)
Range: Unknown.
Notes. *Naughton et al. (2014)* recorded this species as "*Hyperaspis* nr. *annexa*". MLG observed two of these vouchers and one additional specimen from SBMNH, and they do appear quite similar to mainland *H. annexa*, but with much less yellow and more extensive black coloration. These specimens require more detailed study.

*Hyperaspis taeniata* **LeConte, 1852**
Nomenclatural Authority: *Gordon (1985)*
Literature Records: Santa Cruz (*Naughton et al., 2014*: 303)
Digitized Records: Santa Cruz (3 LACM; 1 SBMNH)
Range: Also known from mainland (*Gordon, 1985*).

**Microweiseinae**
Notes. Three tribes, five genera, and 11 species of Microweiseinae are known to occur in California (M. L. Gimmel, 2022, unpublished data). *Escalona & Ślipiński (2011)* provided a generic revision and reclassification of this subfamily.

**Carinodulini**
Notes. One genus and species of Carinodulini is known to occur in California (*Escalona & Ślipiński, 2011*).

*Carinodulinka* **Ślipiński & Tomaszewska, 2002**
Nomenclatural Authority: *Escalona & Ślipiński (2011)*
Notes. No described species of *Carinodulinka* have yet been recorded from California (see below).

***Carinodulinka* undescribed species near *baja* Ślipiński & Tomaszewska, 2002**
Nomenclatural Authority: *Escalona & Ślipiński (2011)*
Literature Records: none
Digitized Records: San Clemente (1 SBMNH)
Range: Also known from mainland (*Escalona & Ślipiński, 2011*).
Notes. According to *Escalona & Ślipiński (2011*: 13), California specimens of the genus *Carinodulinka* Ślipiński & Tomaszewska, 2002 are an unnamed species.

**Microweiseini**
Notes. Three genera and eight species of Microweiseini are known to occur in California (*Gordon, 1985*; M. L. Gimmel, 2022, unpublished data).

***Coccidophilus* Brèthes, 1905**
Nomenclatural Authority: *Escalona & Ślipiński (2011)*
Notes. One species of *Coccidophilus* has been recorded from California (*Gordon, 1985*).

***Coccidophilus atronitens* (Casey, 1899)**
Nomenclatural Authority: *Gordon (1985)*
Literature Records: none
Digitized Records: Santa Cruz (16 SBMNH), Santa Rosa (6 SBMNH)
Range: Also known from mainland (*Gordon, 1985*).

***Microweisea* Cockerell, 1903**
Nomenclatural Authority: *Escalona & Ślipiński (2011)*
Notes. Certain species in this genus were until recently known as *Gnathoweisea* Gordon, 1970, a genus synonymized with *Microweisea* by *Escalona & Ślipiński (2011)*.

***Microweisea* undetermined species**
Literature Records: none
Digitized Records: Santa Catalina (1 iNat)

***Nipus* Casey, 1899**
Nomenclatural Authority: *Escalona & Ślipiński (2011)*
Notes. Three species of the genus *Nipus* are known from California (M. L. Gimmel, 2022, unpublished data).

***Nipus niger* Casey, 1899**
Nomenclatural Authority: *Gordon (1985)*
Literature Records: none
Digitized Records: Santa Rosa (1 SBMNH)
Range: Also known from mainland (*Gordon, 1985*).

**Serangiini**
Notes. One genus and two species of Serangiini have been recorded from California (*Gordon, 1985*).

***Delphastus* Casey, 1899**
Nomenclatural Authority: *Escalona & Ślipiński (2011)*
Notes. Two species of *Delphastus* have been recorded from California (*Gordon, 1985*).

***Delphastus catalinae* (Horn, 1895)**
Nomenclatural Authority: *Gordon (1985)*
Literature Records: San Clemente (*Fall, 1897*: 237), Santa Catalina (*Horn, 1895*: 83; *Fall, 1897*: 237; *Casey, 1899*: 112; *Fall, 1901*: 85; *Gordon, 1970*: 367; *Gordon, 1985*: 64), Santa Cruz (*Naughton et al., 2014*: 303)
Digitized Records: Santa Catalina (1 LACM; 1 SBMNH), Santa Cruz (3 SBMNH)
Range: Also known from mainland (*Gordon, 1985*).
Notes. *Horn (1895)* and *Fall (1897, 1901)* recorded this species as *Cryptognatha catalinae*. The species was originally (*Horn, 1895*; *Casey, 1899*) considered endemic to Santa Catalina Island. *Fall (1897*: 239) doubted its taxonomic validity.

**Corylophidae**
Notes. This family contains five tribes, six genera, and 14 species known from California (M. L. Gimmel, 2022, unpublished data), all of which belong to the subfamily Corylophinae. The classification used here follows *Robertson et al. (2013)*. This family is very poorly understood in North America, and modern keys to species do not exist for most genera.

**Aenigmaticini**
Notes. One genus and species of Aenigmaticini occurs in California (*Pakaluk, 1985*).

***Aenigmaticum* Matthews, 1888**
Nomenclatural Authority: *Robertson et al. (2013)*
Notes. One species of *Aenigmaticum* occurs in California (*Pakaluk, 1985*). This genus was reviewed by *Pakaluk (1985)*.

***Aenigmaticum californicum* Casey, 1889**
Nomenclatural Authority: *Pakaluk (1985)*
Literature Records: Anacapa (*Miller & Miller, 1985*: 126), Santa Barbara (*Miller & Miller, 1985*: 126)
Digitized Records: Anacapa (7 SBMNH), San Miguel (3 SBMNH), San Nicolas (8 SBMNH), Santa Barbara (2 SBMNH)
Range: Also known from mainland (*Pakaluk, 1985*).
Notes. Reported from *Erophyllum*, *Hemizonia* (both Asteraceae) and *Frankenia* (Frankeniaceae) on Santa Barbara Island by *Miller & Miller (1985)*.

**Orthoperini**
Notes. One genus and four species of Orthoperini have been recorded from California (M. L. Gimmel, 2022, unpublished data).

***Orthoperus* Stephens, 1829**
Nomenclatural Authority: *Bowestead & Leschen (2002)*, *Robertson et al. (2013)*

Notes. Four species of *Orthoperus* have been recorded from California (M. L. Gimmel, 2022, unpublished data).

### *Orthoperus* undetermined species
Literature Records: none
Digitized Records: Santa Cruz (9 SBMNH)

### Sericoderini
Notes. One genus and three species of Sericoderini have been recorded from California (M. L. Gimmel, 2022, unpublished data).

### *Sericoderus* Stephens, 1829
Nomenclatural Authority: *Bowestead & Leschen (2002)*, *Robertson et al. (2013)*
Notes. Three species of *Sericoderus* have been recorded from California (M. L. Gimmel, 2022, unpublished data).

### *Sericoderus* undetermined species
Literature Records: none
Digitized Records: San Nicolas (3 SBMNH), Santa Cruz (3 SBMNH), Santa Rosa (1 SBMNH)

### Endomychidae
Notes. Six subfamilies, eight genera, and 13 species of Endomychidae have been recorded from California (*Shockley, Tomaszewska & McHugh, 2009*).

### Lycoperdininae
Notes. Two genera and four species of Lycoperdininae have been recorded from California (*Shockley, Tomaszewska & McHugh, 2009*).

### *Aphorista* Gorham, 1873
Nomenclatural Authority: *Shockley, Tomaszewska & McHugh (2009)*
Notes. Two species of *Aphorista* have been recorded from California (*Shockley, Tomaszewska & McHugh, 2009*).

### *Aphorista morosa* (LeConte, 1859)
Nomenclatural Authority: *Shockley, Tomaszewska & McHugh (2009)*
Literature Records: Santa Rosa (*Fall, 1897*: 237)
Digitized Records: Santa Catalina (2 LACM), Santa Cruz (2 SBMNH; 1 UCSB), Santa Rosa (4 SBMNH)
Range: Also known from mainland (*Shockley, Tomaszewska & McHugh, 2009*).

### Latridiidae
Notes. Two subfamilies, 13 genera, and 61 species of Latridiidae are known to occur in California (M. L. Gimmel, 2022, unpublished data).

**Corticariinae**

Notes. Five genera and 31 species of Corticariinae are known to occur in California (M. L. Gimmel, 2022, unpublished data).

*Corticaria* **Marsham, 1802**

Nomenclatural Authority: *Rücker (2021)*

Notes. Ten species of *Corticaria* have been reported from California (M. L. Gimmel, 2022, unpublished data).

*Corticaria* **undetermined species**

Literature Records: Santa Catalina (*Fall, 1897*: 237)

Digitized Records: Santa Rosa (2 SBMNH)

Notes. *Fall's (1897)* above record refers to the genus only.

*Corticarina* **Reitter, 1881**

Nomenclatural Authority: *Rücker (2021)*

Literature Records (genus-only): Santa Cruz (*Naughton et al., 2014*: 303)

Digitized Records (genus-only): Anacapa (5 SBMNH), San Clemente (5 SBMNH), San Miguel (25 SBMNH), San Nicolas (7 SBMNH), Santa Barbara (15 SBMNH), Santa Catalina (1 SBMNH), Santa Cruz (51 SBMNH), Santa Rosa (14 SBMNH)

Notes. The record from *Naughton et al. (2014)* refers to the genus only, and probably represents one of the four species below. Nine species of *Corticarina* have been recorded from California (M. L. Gimmel, 2022, unpublished data).

*Corticarina cavicollis* **(Mannerheim, 1844)**

Nomenclatural Authority: *Rücker (2021)*

Literature Records: none

Digitized Records: Santa Cruz (1 SBMNH)

Range: Also known from mainland (*Rücker, 2021*).

*Corticarina herbivagans* **(LeConte, 1855)**

Nomenclatural Authority: *Rücker (2021)*

Literature Records: San Miguel (*Miller & Miller, 1985*: 127), Santa Barbara (*Miller & Miller, 1985*: 127)

Digitized Records: Santa Barbara (1 SBMNH)

Range: Also known from mainland (*Rücker, 2021*).

*Corticarina milleri* **Andrews, 1992**

Nomenclatural Authority: *Andrews (1992)*, *Rücker (2021)*

Literature Records: San Miguel (*Andrews, 1992*: 278), San Nicolas (*Andrews, 1992*: 278), Santa Barbara (*Andrews, 1992*: 277), Santa Rosa (*Andrews, 1992*: 278)

Digitized Records: Anacapa (1 SBMNH), Santa Cruz (1 SBMNH)

Range: Endemic (*Andrews, 1992*).

*Corticarina minuta* **(Fabricius, 1792)**

Nomenclatural Authority: *Rücker (2021)*

Literature Records: none
Digitized Records: Anacapa (1 SBMNH), Santa Cruz (4 SBMNH)
Range: Also known from mainland (*Rücker, 2021*).

### *Fuchsina* Fall, 1899
Nomenclatural Authority: *Rücker (2021)*
Notes. Two described species of *Fuchsina* have been reported from California (*Andrews, 1976c*). These were revised by *Andrews (1976c)*, but an apparently undescribed species occurs on the Channel Islands.

### *Fuchsina* undescribed species
Literature Records: Santa Cruz (*Naughton et al., 2014*: 303)
Digitized Records: San Clemente (15 SBMNH), Santa Catalina (47 SBMNH), Santa Cruz (29 SBMNH), Santa Rosa (16 SBMNH)
Range: Endemic (M. L. Gimmel, 2021, personal observation).
Notes. The record from *Naughton et al. (2014)* refers to the genus only. The SBMNH specimens from the Channel Islands are morphologically different from either of the two described species of *Fuchsina* as circumscribed by *Andrews (1976c)*. The Channel Island specimens are similar to *Fuchsina occulta* Fall, 1899 in the lack of eye facets and antenna with 10 antennomeres, but have a shorter, broader pronotum and shorter elytra (M. L. Gimmel, 2021, personal observation).

### *Melanophthalma* Motschulsky, 1866
Nomenclatural Authority: *Rücker (2021)*
Literature Records (genus-only): Santa Cruz (*Naughton et al., 2014*: 303)
Digitized Records (genus-only): Anacapa (1 SBMNH), San Clemente (2 SBMNH), San Nicolas (2 SBMNH), Santa Catalina (9 SBMNH), Santa Cruz (5 SBMNH), Santa Rosa (1 SBMNH)
Notes. *Naughton et al. (2014)* mistakenly reported this genus as "*Melanophthalmus*". Eight species of *Melanophthalma* have been reported from California in two subgenera, *Cortilena* Motschulsky, 1867 and *Melanophthalma* (*s.str.*) (M. L. Gimmel, 2022, unpublished data).

### *Melanophthalma* (*Cortilena*) *casta* Fall, 1899
Nomenclatural Authority: *Rücker (2021)*
Literature Records: San Nicolas (*Miller & Miller, 1985*: 127), Santa Barbara (*Miller & Miller, 1985*: 127)
Digitized Records: Santa Barbara (1 SBMNH)
Range: Also known from mainland (*Fall, 1899*).
Notes. Recorded by *Miller & Miller (1985)* as *Cortilena casta*.

### *Melanophthalma* (*Melanophthalma*) *americana* (Mannerheim, 1844)
Nomenclatural Authority: *Rücker (2021)*
Literature Records: San Clemente (*Fall, 1897*: 237), San Miguel (*Cockerell, 1940*: 286), Santa Catalina (*Fall, 1897*: 237)
Digitized Records: Santa Cruz (3 SBMNH)

Range: Also known from mainland (*Rücker, 2021*).

Notes. Reported by *Fall (1897)* as *Corticaria distinguenda* Comolli, 1837, but specimens in North American identified as this species are presently known as *M. americana*.

### *Melanophthalma* (*Melanophthalma*) *insularis* Fall, 1899
Nomenclatural Authority: *Rücker (2021)*
Literature Records: San Clemente (*Fall, 1899*: 174; *Fall, 1901*: 102; *Miller, 1985a*: 20)
Digitized Records: none
Range: Endemic (*Fall, 1899*, *1901*; *Miller, 1985a*).
Notes. The status of this purportedly endemic taxon has not been reviewed since *Fall's (1899)* original description.

### Latridiinae
Notes. Eight genera and 30 species of Latridiinae are known to occur in California (M. L. Gimmel, 2022, unpublished data).

### *Cartodere* Thomson, 1859
Nomenclatural Authority: *Rücker (2021)*
Notes. Four species of *Cartodere* in two subgenera, *Aridius* Motschulsky, 1866 and *Cartodere* (*s.str.*), occur in California (M. L. Gimmel, 2022, unpublished data).

### *Cartodere* (*Aridius*) *australica* (Belon, 1887)
Nomenclatural Authority: *Rücker (2021)*
Literature Records: none
Digitized Records: Santa Cruz (25 SBMNH)
Range: Also known from mainland (*Rücker, 2021*).
Notes. This species is presumably adventive in California.

### *Dienerella* Reitter, 1911
Nomenclatural Authority: *Rücker (2021)*
Notes. Three species of *Dienerella* are known to occur in California (M. L. Gimmel, 2022, unpublished data).

### *Dienerella* undetermined species
Literature Records: Santa Catalina (*Caterino & Chandler, 2010*: 191)
Digitized Records: Santa Catalina (1 SBMNH)
Notes. More morphospecies exist in the SBMNH collection than there are named species known to occur in California (M. L. Gimmel, 2021, personal observation). Consequently, we have not attempted to identify the single Channel Islands specimen known to us.

### *Enicmus* Thomson, 1859
Nomenclatural Authority: *Rücker (2021)*
Notes. Six species of *Enicmus* have been recorded from California (M. L. Gimmel, 2022, unpublished data).

### *Enicmus aterrimus* Motschulsky, 1866

Nomenclatural Authority: *Rücker (2021)*

Literature Records: none

Digitized Records: Santa Cruz (1 SBMNH), Santa Rosa (1 SBMNH)

Range: Also known from mainland (*Rücker, 2021*).

### *Metophthalmus* Motschulsky, 1850

Nomenclatural Authority: *Rücker (2021)*

Literature Records (genus-only): Santa Catalina (*Caterino & Chandler, 2010*: 191), Santa Cruz (*Naughton et al., 2014*: 303)

Notes. This genus was revised by *Andrews (1976b)*. Six species of this genus have been reported from California (*Andrews, 1976b*) belonging to two subgenera, *Metatypus* Belon, 1897 and *Metophthalmus* (*s.str.*), three of which are reported from the Channel Islands below. *Caterino & Chandler (2010*: 191) were first to mention the presence of this genus on Santa Catalina. Five of the 12 specimens of *Metophthalmus* reported from Santa Cruz by *Naughton et al. (2014)* were not determined to species; vouchers for these are not located in SBMNH (M. L. Gimmel, 2021, personal observation). Almost certainly the latter represent one or more of the three species recorded below, though two additional species, *Metophthalmus kanei* Andrews, 1976 and *Metophthalmus septemstriatus* Hatch, 1962, may occur on the Channel Islands.

### *Metophthalmus* (*Metatypus*) *haigi* Andrews, 1976

Nomenclatural Authority: *Rücker (2021)*

Literature Records: Santa Cruz (*Naughton et al., 2014*: 303)

Digitized Records: San Clemente (3 SBMNH), Santa Catalina (3 SBMNH), Santa Cruz (7 SBMNH)

Range: Also known from mainland (*Andrews, 1976b*; *Rücker, 2021*).

### *Metophthalmus* (*Metatypus*) *rudis* Fall, 1899

Nomenclatural Authority: *Rücker (2021)*

Literature Records: Santa Cruz (*Andrews, 1976b*: 53; *Naughton et al., 2014*: 303)

Digitized Records: San Clemente (26 SBMNH), Santa Catalina (42 SBMNH), Santa Cruz (14 SBMNH), Santa Rosa (24 SBMNH)

Range: Also known from mainland (*Andrews, 1976b*; *Rücker, 2021*).

### *Metophthalmus* (*Metatypus*) *trux* Fall, 1899

Nomenclatural Authority: *Rücker (2021)*

Literature Records: Santa Cruz (*Naughton et al., 2014*: 303)

Digitized Records: San Clemente (5 SBMNH), Santa Catalina (3 SBMNH), Santa Cruz (12 SBMNH), Santa Rosa (2 SBMNH)

Range: Also known from mainland (*Andrews, 1976b*; *Rücker, 2021*).

### *Revelieria* Perris, 1869

Nomenclatural Authority: *Rücker (2021)*

Notes. One species of *Revelieria* has been reported from California (*Rücker, 2021*).

*Revelieria californica* **Fall, 1899**
Nomenclatural Authority: *Rücker (2021)*
Literature Records: none
Digitized Records: Santa Cruz (8 SBMNH), Santa Rosa (3 SBMNH)
Range: Also known from mainland (*Fall, 1899*).

*Stephostethus* **LeConte, 1878**
Nomenclatural Authority: *Rücker (2021)*
Notes. Four species of *Stephostethus* have been reported from California (M. L. Gimmel, 2022, unpublished data).

*Stephostethus armatulus* **(Fall, 1899)**
Nomenclatural Authority: *Rücker (2021)*
Literature Records: Santa Catalina (*Fall, 1899*: 118)
Digitized Records: none
Range: Also known from mainland (*Fall, 1899*; *Rücker, 2021*).
Notes. *Fall (1899)* recorded this species as *Lathridius armatulus*.

*Stephostethus costicollis* **(LeConte, 1855)**
Nomenclatural Authority: *Rücker (2021)*
Literature Records: Santa Catalina (*Fall, 1897*: 237; *Cockerell, 1940*: 286)
Digitized Records: San Clemente (1 SBMNH), Santa Catalina (1 SBMNH)
Range: Also known from mainland (*Rücker, 2021*).
Notes. This species was recorded as *Coninomus fulvipennis* Mannerheim by *Fall (1897)*, and corrected to *Latridius costicollis* by *Cockerell (1940)*. It has since been placed in the genus *Stephostethus*.

*Stephostethus liratus* **(LeConte, 1863)**
Nomenclatural Authority: *Rücker (2021)*
Literature Records: none
Digitized Records: Santa Rosa (2 SBMNH)
Range: Also known from mainland (*Rücker, 2021*).

**EROTYLOIDEA**

**Erotylidae**
Notes. Three subfamilies, six genera, and 11 species of Erotylidae have been recorded from California (M. L. Gimmel, 2022, unpublished data).

**Cryptophilinae**
Notes. One genus and species of Cryptophilinae has been recorded from California (M. L. Gimmel, 2022, unpublished data).

*Cryptophilus* **Reitter, 1874**
Nomenclatural Authority: *Gimmel, Leschen & Esser (2019)*

Notes. One species of the genus *Cryptophilus* has been recorded from California (M. L. Gimmel, 2022, unpublished data).

### *Cryptophilus angustus* (Rosenhauer, 1856)
Nomenclatural Authority: *Esser (2017)*
Literature Records: none
Digitized Records: Santa Cruz (3 SBMNH)
Range: Also known from mainland (*Esser, 2017*).
Notes. This species was long known as *Cryptophilus integer* (Heer, 1841), but the type of that species was discovered to belong to Cryptophagidae. *Cryptophilus angustus* is the proper name for this species, which was introduced from the Palearctic realm (*Esser, 2017*; *Gimmel, Leschen & Esser, 2019*).

### Erotylinae
Notes. Four genera and eight species of Erotylinae have been recorded from California (*Boyle, 1956*; M. L. Gimmel, 2022, unpublished data).

### *Dacne* Latreille, 1796
Nomenclatural Authority: *Boyle (1956)*
Notes. Four species of *Dacne* have been recorded from California (*Boyle, 1956*).

### *Dacne californica* (Horn, 1870)
Nomenclatural Authority: *Boyle (1956)*
Literature Records: Santa Catalina (*Cockerell, 1940*: 286; *Boyle, 1956*: 142), Santa Cruz (*Naughton et al., 2014*: 303)
Digitized Records: San Clemente (3 SBMNH), Santa Catalina (5 SBMNH), Santa Cruz (6 SBMNH), Santa Rosa (13 SBMNH)
Range: Also known from mainland (*Boyle, 1956*).
Notes. The island vouchers of this species housed in SBMNH are certainly *Dacne* (*s.str.*), which has just one described species in California (*D. californica*). However, they are morphologically different from mainland exemplars of that species, being narrower and more setose, and the prosternal lines are differently shaped. These may prove to be a distinct, undescribed species.

### NITIDULOIDEA

### Kateretidae
Notes. Four genera and six species of Kateretidae have been recorded from California (M. L. Gimmel, 2022, unpublished data).

### *Amartus* LeConte, 1861
Nomenclatural Authority: *Cline & Audisio (2010)*
Notes. Two species of *Amartus* have been reported from California (*Savage & Seeno, 1981*). *Savage & Seeno (1981)* reviewed the genus *Amartus* for North America.

***Amartus tinctus*** (Mannerheim, 1843)
Nomenclatural Authority: *Savage & Seeno (1981)*
Literature Records: San Clemente (*Savage & Seeno, 1981*: 80), San Miguel (*Cockerell, 1940*: 286), Santa Rosa (*Savage & Seeno, 1981*: 80)
Digitized Records: San Clemente (6 LACM), Santa Rosa (9 SBMNH)
Range: Also known from mainland (*Savage & Seeno, 1981*).

***Heterhelus*** Jacquelin du Val, 1858
Nomenclatural Authority: *Cline & Audisio (2010)*
Notes. One species of *Heterhelus* has been reported from California (*Parsons, 1943*).

***Heterhelus sericans*** (LeConte, 1869)
Nomenclatural Authority: *Habeck (2002)*
Literature Records: Santa Catalina (*Fall, 1897*: 237)
Digitized Records: none
Range: Also known from mainland (*Fall, 1901*; *Habeck, 2002*).
Notes. *Fall (1897, 1901)* recorded this species as *Cercus sericans*, and *Fall (1901*: 98) listed it as occurring "throughout Southern California and the adjacent islands".

**Monotomidae**
Notes. Two subfamilies, eight genera, and 19 species of Monotomidae are known to occur in California (M. L. Gimmel, 2022, unpublished data).

**Monotominae**
Notes. Seven genera and 14 species of Monotominae are known to occur in California (M. L. Gimmel, 2022, unpublished data).

***Hesperobaenus*** LeConte, 1861
Nomenclatural Authority: *McElrath, Gimmel & Powell (2021)*
Notes. The genus *Hesperobaenus* contains two species occurring in California (*Bousquet, 2002b*). The genus was revised for North America by *Bousquet (2002b)*.

***Hesperobaenus abbreviatus*** (Motschulsky, 1845)
Nomenclatural Authority: *Bousquet (2002b)*
Literature Records: Santa Cruz (*Bousquet, 2002b*: 210)
Digitized Records: Anacapa (1 SBMNH), Santa Cruz (9 SBMNH), Santa Rosa (4 SBMNH)
Range: Also known from mainland (*Bousquet, 2002b*).

***Macreurops*** Casey, 1916
Nomenclatural Authority: *McElrath, Gimmel & Powell (2021)*
Notes. One species of *Macreurops* is known from California (*Bousquet, 2002a*).

***Macreurops longicollis*** (Horn, 1879)
Nomenclatural Authority: *Bousquet (2002a)*
Literature Records: none

Digitized Records: Santa Cruz (15 SBMNH)
Range: Also known from mainland (*Bousquet, 2002a*).

### *Phyconomus* LeConte, 1861
Nomenclatural Authority: *McElrath, Gimmel & Powell (2021)*
Notes. One species of *Phyconomus* is known from California (*Bousquet, 2002a*).

### *Phyconomus marinus* (LeConte, 1858)
Nomenclatural Authority: *Bousquet (2002a)*
Literature Records: none
Digitized Records: San Miguel (10 SBMNH), Santa Cruz (1 SBMNH)
Range: Also known from mainland (*Bousquet, 2002a*).

### Nitidulidae
Notes. Eight subfamilies, 22 genera, and 63 species of Nitidulidae are known to occur in California (M. L. Gimmel, 2022, unpublished data).

### Carpophilinae
Notes. Four genera and 17 species of Carpophilinae are known to occur in California (M. L. Gimmel, 2022, unpublished data).

### *Carpophilus* Stephens, 1829
Nomenclatural Authority: *McElrath, Gimmel & Powell (2021)*
Digitized Records (genus-only): Santa Cruz (1 LACM; 3 SBMNH), Santa Rosa (4 LACM; 5 SBMNH)
Notes. Eleven species of *Carpophilus* (*sensu* *Powell et al., 2020*) are known from California (M. L. Gimmel, 2022, unpublished data).

### *Carpophilus* (*Ecnomorphus*) *discoideus* LeConte, 1858
Nomenclatural Authority: *McElrath, Gimmel & Powell (2021)*
Literature Records: none
Digitized Records: Santa Cruz (2 SBMNH)
Range: Also known from mainland (*Parsons, 1943*).

### *Carpophilus* (*Ecnomorphus*) *ligneus* Murray, 1864
Nomenclatural Authority: *McElrath, Gimmel & Powell (2021)*
Literature Records: none
Digitized Records: Anacapa (1 SBMNH), San Miguel (1 SBMNH), San Nicolas (1 SBMNH)
Range: Also known from mainland (*McElrath, Gimmel & Powell, 2021*).

### *Nitops* Murray, 1864
Nomenclatural Authority: *McElrath, Gimmel & Powell (2021)*
Notes. This genus contains a single species in California (M. L. Gimmel, 2022, unpublished data).

*Nitops pallipennis* (Say, 1823)

Nomenclatural Authority: *McElrath, Gimmel & Powell (2021)*

Literature Records: San Clemente (*Fall, 1897*: 237), Santa Catalina (*Seavey, 1892*: 262; *Fall, 1897*: 237; *Grant & Connell, 1979*: 100; *Grant & Grant, 1979*: 323), Santa Cruz (*LeConte, 1876*: 299; *Fall, 1897*: 237; *Fall & Davis, 1934*: 143)

Digitized Records: San Clemente (18 SBMNH), San Miguel (3 SBMNH), San Nicolas (3 SBMNH), Santa Barbara (9 SBMNH), Santa Catalina (1 LACM; 3 SBMNH), Santa Cruz (66 LACM; 3 SBMNH), Santa Rosa (29 LACM; 7 SBMNH)

Range: Also known from mainland (*Parsons, 1943*; *Grant & Connell, 1979*; *Grant & Grant, 1979*).

Notes. Recorded from flowers of *Opuntia littoralis* var. *littoralis* (Engelm.) Cockerell on Santa Catalina (*Grant & Connell, 1979*; *Grant & Grant, 1979*). This species was recorded by all authors cited above prior to *Powell et al. (2020)* as *Carpophilus pallipennis*.

**Cryptarchinae**

Notes. Two genera and seven species of Cryptarchinae have been recorded from California (*Parsons, 1943*; *McCoshum, 2012*).

*Cryptarcha* Shuckard, 1839

Nomenclatural Authority: *McElrath, Gimmel & Powell (2021)*

Notes. Three species of *Cryptarcha* have been recorded from California (*Parsons, 1943*).

*Cryptarcha gila* Parsons, 1938

Nomenclatural Authority: *McElrath, Gimmel & Powell (2021)*

Literature Records: Santa Cruz (*Naughton et al., 2014*: 304)

Digitized Records: Santa Catalina (2 SBMNH), Santa Rosa (2 SBMNH)

Range: Also known from mainland (*Parsons, 1943*).

*Glischrochilus* Reitter, 1873

Nomenclatural Authority: *McElrath, Gimmel & Powell (2021)*

Notes. Four species of *Glischrochilus* are known from California (M. L. Gimmel, 2022, unpublished data).

*Glischrochilus quadrisignatus* (Say, 1835)

Nomenclatural Authority: *McElrath, Gimmel & Powell (2021)*

Literature Records: Santa Catalina (*McCoshum, 2012*: 348)

Digitized Records: Santa Catalina (1 SBMNH)

Range: Also known from mainland (*McCoshum, 2012*).

Notes. This species is adventive in California (*McCoshum, 2012*).

*Glischrochilus sanguinolentus* (Olivier, 1790)

Nomenclatural Authority: *McElrath, Gimmel & Powell (2021)*

Literature Records: Santa Catalina (*McCoshum, 2012*: 348)

Digitized Records: Santa Catalina (1 SBMNH)

Range: Also known from mainland (*McCoshum, 2012*).

Notes. This species is adventive in California (*McCoshum, 2012*).

### Meligethinae

Notes. Two genera and two species of Meligethinae have been recorded from California (*Parsons, 1943*; *Easton, 1955*). *Easton (1955)* revised the species in this subfamily for North America. Generic concepts and some species concepts, however, have changed since then (*Audisio et al., 2009*).

### *Brassicogethes* Audisio & Cline, 2009

Nomenclatural Authority: *Audisio et al. (2009)*

Notes. This genus contains one species in California (*Parsons, 1943*; *Audisio et al., 2009*).

### *Brassicogethes aeneus* (Fabricius, 1775)

Nomenclatural Authority: *Audisio et al. (2009)*

Literature Records: none

Digitized Records: Santa Catalina (1 SBMNH)

Range: Also known from mainland (*Parsons, 1943*; *Easton, 1955*).

Notes. This species is Holarctic in distribution (*Audisio et al., 2009*).

### Nitidulinae

Notes. Eight genera and 19 species of Nitidulinae are known to occur in California (M. L. Gimmel, 2022, unpublished data).

### *Nitidula* Fabricius, 1775

Nomenclatural Authority: *Parsons (1943)*, *McElrath, Gimmel & Powell (2021)*

Notes. This genus contains three species in California (*Parsons, 1943*). *Parsons (1943)* provided a key to species in North America.

### *Nitidula flavomaculata* Rossi, 1790

Nomenclatural Authority: *McElrath, Gimmel & Powell (2021)*

Literature Records: none

Digitized Records: Santa Catalina (1 SBMNH)

Range: Also known from mainland (*Parsons, 1943*).

Notes. This species was introduced into North America from the Mediterranean region (*Parsons, 1943*).

### *Thalycra* Erichson, 1843

Nomenclatural Authority: *McElrath, Gimmel & Powell (2021)*

Notes. This genus contains eight species in California (*Howden, 1961*; M. L. Gimmel, 2022, unpublished data). *Howden (1961)* provided a key to New World species.

### *Thalycra* undetermined species

Literature Records: none

Digitized Records: Santa Rosa (8 SBMNH)

**CUCUJOIDEA**

**Cryptophagidae**
Notes. Two subfamilies, 11 genera, and 60 species of Cryptophagidae are known to occur in California (M. L. Gimmel, 2022, unpublished data). *Pelletier & Hébert (2019)* provided a revision and identification guide to the species of Cryptophagidae in the northern US and Canada, which is helpful for making identifications in the southern US as well.

**Atomariinae**
Notes. Two genera and 28 species of Atomariinae, all belonging to Atomariini, are known to occur in California (M. L. Gimmel, 2022, unpublished data).

*Atomaria* **Stephens, 1829**
Nomenclatural Authority: *Pelletier & Hébert (2019)*
Literature Records (genus-only): Santa Rosa (*Fall, 1897*: 237)
Notes. *Fall (1897)* did not specify subgenus when he cited the record of "*Atomaria* sp.".
*Pelletier & Hébert (2019)* provided a key to all North American species of the genus.
Twenty-seven species of *Atomaria* have been reported from California, 11 from the subgenus *Anchicera* Thomson, 1863 and 16 from the subgenus *Atomaria* (*s.str.*) (M. L. Gimmel, 2022, unpublished data).

*Atomaria* (*Anchicera*) *lewisi* **Reitter, 1877**
Nomenclatural Authority: *Pelletier & Hébert (2019)*
Literature Records: none
Digitized Records: Santa Cruz (6 SBMNH)
Range: Also known from mainland (*Pelletier & Hébert, 2019*).
Notes. This species is adventive from Europe (*Pelletier & Hébert, 2019*).

*Atomaria* (*Anchicera*) *nubipennis* **Casey, 1900**
Nomenclatural Authority: *Pelletier & Hébert (2019)*
Literature Records: none
Digitized Records: San Clemente (20 SBMNH)
Range: Also known from mainland (*Pelletier & Hébert, 2019*).

*Atomaria* (*Atomaria*) *puella* **(Casey, 1900)**
Nomenclatural Authority: *Pelletier & Hébert (2019)*
Literature Records: none
Digitized Records: Santa Cruz (1 SBMNH)
Range: Also known from mainland (*Pelletier & Hébert, 2019*).

**Cryptophaginae: Cryptophagini**
Notes. Two tribes, nine genera, and 32 species of Cryptophaginae, of which seven genera and 29 species belong to Cryptophagini, are known to occur in California (M. L. Gimmel, 2022, unpublished data).

*Cryptophagus* **Herbst, 1792**
Nomenclatural Authority: *Pelletier & Hébert (2019)*
Notes. *Woodroffe & Coombs (1961)* revised the genus *Cryptophagus* for North America. *Pelletier & Hébert (2019)* provided an updated key to and illustrations of most North American species. In California, 22 species have been reported (M. L. Gimmel, 2022, unpublished data).

*Cryptophagus tuberculosus* **Mäklin, 1853**
Nomenclatural Authority: *Woodroffe & Coombs (1961)*, *Pelletier & Hébert (2019)*
Literature Records: San Clemente (*Fall, 1897*: 237; *Cockerell, 1940*: 286), Santa Catalina (*Fall, 1897*: 237; *Cockerell, 1940*: 286)
Digitized Records: San Clemente (5 SBMNH), Santa Cruz (5 SBMNH)
Range: Also known from mainland (*Woodroffe & Coombs, 1961*).
Notes. *Fall (1897)* recorded this species as "*Cryptophagus* sp.", which *Cockerell (1940)* indicated as *Cryptophagus debilis* LeConte, 1858 based on communication from H.C. Fall. *Cryptophagus debilis* was synonymized with *C. tuberculosus* by *Woodroffe & Coombs (1961)*.

**Laemophloeidae, NEW FAMILY RECORD**
Notes. Eight genera and 16 species of Laemophloeidae are known to occur in California (M. L. Gimmel, 2022, unpublished data).

*Narthecius* **LeConte, 1861**
Nomenclatural Authority: *McElrath, Gimmel & Powell (2021)*
Notes. Two species of *Narthecius* are known to occur in California (M. L. Gimmel, 2022, unpublished data).

*Narthecius striaticeps* **Fall, 1907**
Nomenclatural Authority: *McElrath, Gimmel & Powell (2021)*
Literature Records: none
Digitized Records: Santa Cruz (1 SBMNH)
Range: Also known from mainland (*Fall & Cockerell, 1907*).

**Phalacridae**
Notes. Five genera and 21 species of Phalacridae are known to occur in California (M. L. Gimmel, 2022, unpublished data).

*Phalacrus* **Paykull, 1800**
Nomenclatural Authority: *Gimmel (2013)*
Notes. This genus needs revision; the species in North America are not currently identifiable (*Gimmel, 2013*).

*Phalacrus* **undetermined species 1**
Literature Records: none
Digitized Records: San Nicolas (6 SBMNH)

Notes. This species has microsculpture on the elytra, the left mandible with a ventral tooth, and a prominent metaventral process that exceeds the mesocoxae; this possibly represents *Phalacrus conjunctus* Casey, 1890 (M. L. Gimmel, 2021, personal observation).

### *Phalacrus* undetermined species 2
Literature Records: Santa Cruz (*Naughton et al., 2014*: 304)
Digitized Records: Santa Cruz (3 SBMNH)
Notes. This species has no microsculpture on the elytra, the left mandible with a ventral tooth, and a short metaventral process not exceeding the mesocoxae; this possibly represents *Phalacrus ovalis* LeConte, 1856 (M. L. Gimmel, 2021, personal observation). Santa Cruz Island vouchers from the *Naughton et al. (2014)* study citing "*Phalacrus* sp." were examined by MLG and belong to this morphospecies.

### Silvanidae, NEW FAMILY RECORD
Notes. Two subfamilies, nine genera, and 14 species of Silvanidae are known to occur in California (M. L. Gimmel, 2022, unpublished data).

### *Silvanoprus* Reitter, 1911
Nomenclatural Authority: *McElrath, Gimmel & Powell (2021)*
Notes. One introduced species of *Silvanoprus* is now known from California; the record below represents a **new state record** for the genus.

### *Silvanoprus angusticollis* (Reitter, 1876)
Nomenclatural Authority: *McElrath, Gimmel & Powell (2021)*
Literature Records: none
Digitized Records: Santa Cruz (6 SBMNH)
Range: Also known from mainland (*McElrath, Gimmel & Powell, 2021*).
Notes. This represents a **new state record** for California. This species is adventive in North America.

### CHRYSOMELOIDEA

### Cerambycidae
Notes. Seven subfamilies, 143 genera, and 317 species of Cerambycidae are known to occur in California (M. L. Gimmel, 2022, unpublished data). *Bezark & Monné (2013)* provided a nomenclatural checklist of all New World Cerambycidae. *Linsley (1962*, *1963*, *1964)*, *Linsley & Chemsak (1984*, *1995)*, and *Chemsak (1996*, *2005)* monographed the North American fauna. The subfamily Parandrinae occurs on the nearby mainland but is not known from the Channel Islands.

### Cerambycinae
Notes. Twenty-three tribes, 72 genera, and 161 species of Cerambycinae are known to occur in California (M. L. Gimmel, 2022, unpublished data).

**Callidiini**

Notes. Six genera and 35 species of Callidiini are known to occur in California (M. L. Gimmel, 2022, unpublished data).

***Callidiellum* Linsley, 1940**

Nomenclatural Authority: *Bezark & Monné (2013)*

Notes. Two species of *Callidiellum* are known to occur in California, plus one species (*C. rufipenne* below) recorded only as an interception (M. L. Gimmel, 2022, unpublished data).

***Callidiellum rufipenne* (Motschulsky, 1860)**

Nomenclatural Authority: *Linsley (1964)*, *Miller & Miller (1985)*

Literature Records: Santa Barbara (*Miller & Miller, 1985*: 130)

Digitized Records: none

Range: Also known from mainland (*Linsley, 1964*).

Notes. Introduced to North America from eastern Asia; the Santa Barbara Island record is probably an interception and does not represent a breeding population (see *Miller & Miller, 1985*).

***Phymatodes* Mulsant, 1839**

Nomenclatural Authority: *Bezark & Monné (2013)*

Notes. Fifteen species of *Phymatodes* are known to occur in California (M. L. Gimmel, 2022, unpublished data). *Swift & Ray (2010)* presented a revised key to the North American species.

***Phymatodes decussatus* (LeConte, 1857)**

Nomenclatural Authority: *Bezark & Monné (2013)*

Literature Records: Santa Rosa (*Fall, 1897*: 238)

Digitized Records: Santa Cruz (2 SBMNH), Santa Rosa (3 SBMNH)

Range: Also known from mainland (*Linsley, 1964*).

Notes. According to *Linsley (1964*: 50), the subspecies occurring in coastal California is *P. d. decussatus*. *Fall (1897)* reported this species as *Phymatodes juglandis* Leng, 1890, and listed the record with a question mark; this record was included in the species' synonymy in *Linsley (1964*: 52). However, *P. juglandis* was recently made a junior synonym of *P. decussatus* by *Swift & Ray (2010*: 42).

***Phymatodes grandis* Casey, 1912**

Nomenclatural Authority: *Bezark & Monné (2013)*

Literature Records: none

Digitized Records: Santa Catalina (20 LACM; 8 SBMNH; 2 iNat), Santa Cruz (4 LACM; 7 SBMNH)

Range: Also known from mainland (*Linsley, 1964*).

Notes. The species *Phymatodes obscurus* (LeConte, 1859) was given the unnecessary replacement name of *Phymatodes lecontei* Linsley, 1938, but the proper name for this species is *P. grandis* (see *Swift & Ray, 2010*).

**Clytini**

Notes. Eight genera and 30 species of Clytini have been recorded from California (*Linsley, 1964*; M. L. Gimmel, 2022, unpublished data).

***Xylotrechus* Chevrolat, 1860**

Nomenclatural Authority: *Bezark & Monné (2013)*

Notes. Eight species of *Xylotrechus* have been recorded from California (M. L. Gimmel, 2022, unpublished data).

***Xylotrechus insignis* LeConte, 1873**

Nomenclatural Authority: *Bezark & Monné (2013)*

Literature Records: Santa Catalina (*Fall, 1897*: 238; *Fall, 1901*: 147; *Hopping, 1932*: 542; *Cockerell, 1940*: 286)

Digitized Records: Santa Catalina (1 CASC; 4 LACM)

Range: Also known from mainland (*Fall, 1901*; *Linsley, 1964*).

Notes. *Fall (1897, 1901)* recorded this species as *Xylotrechus obliteratus* LeConte, 1873, which was amended to *X. insignis* by communication of E.G. Linsley to *Cockerell (1940)* and in *Hopping (1932)*. *Fall (1897)* reported that it occurred on willows.

***Xylotrechus nauticus* (Mannerheim, 1843)**

Nomenclatural Authority: *Bezark & Monné (2013)*

Literature Records: Santa Cruz (*Linsley, 1964*: 109 [map])

Digitized Records: Santa Catalina (2 CASC), Santa Cruz (7 CASC; 5 SBMNH)

Range: Also known from mainland (*Linsley, 1964*).

**Eburiini**

Notes. Two genera and three species of Eburiini have been recorded from California (*Linsley, 1962*, *1963*).

***Enaphalodes* Haldeman, 1847**

Nomenclatural Authority: *Bezark & Monné (2013)*

Notes. Two species of *Enaphalodes* have been recorded from California (*Linsley, 1963*).

***Enaphalodes hispicornis* (Linnaeus, 1767)**

Nomenclatural Authority: *Bezark & Monné (2013)*

Literature Records: Santa Catalina (*Fall, 1897*: 238; *Fall, 1901*: 144; *Garnett, 1918*: 177; *Linsley, 1963*: 66 [map])

Digitized Records: Santa Catalina (6 LACM)

Range: Also known from mainland (*Fall, 1901*; *Garnett, 1918*; *Linsley, 1963*).

Notes. This species was reported by *Fall (1897, 1901)* and *Garnett (1918)* as *Romaleum simplicicolle* (Haldeman, 1847), which is now recognized as a synonym of the variable species *E. hispicornis* (see *Linsley, 1963*: 64).

**Hesperophanini**

Notes. Six genera and eight species of Hesperophanini have been recorded from California (*Linsley, 1962*).

**Brothylus LeConte, 1859**

Nomenclatural Authority: *Bezark & Monné (2013)*

Notes. Two species of *Brothylus* have been recorded from California (*Linsley, 1962*).

**Brothylus gemmulatus LeConte, 1859**

Nomenclatural Authority: *Bezark & Monné (2013)*

Literature Records: none

Digitized Records: Santa Catalina (22 LACM; 1 iNat)

Range: Also known from mainland (*Linsley, 1962*).

**Holopleurini**

Notes. One species of Holopleurini has been recorded from California (*Linsley, 1962*).

**Holopleura LeConte, 1873**

Nomenclatural Authority: *Bezark & Monné (2013)*

Notes. One species of *Holopleura* has been recorded from California (*Linsley, 1962*).

**Holopleura marginata LeConte, 1873**

Nomenclatural Authority: *Bezark & Monné (2013)*

Literature Records: none

Digitized Records: Santa Catalina (3 LACM)

Range: Also known from mainland (*Linsley, 1962*).

**Hyboderini**

Notes. Four genera and six species of Hyboderini have been recorded from California (*Linsley, 1963*).

**Callimus Mulsant, 1864**

Nomenclatural Authority: *Bezark & Monné (2013)*

Notes. Two species of *Callimus* have been recorded from California (*Linsley, 1963*, as *Lampropterus* Mulsant, 1863).

**Callimus ruficollis (LeConte, 1873)**

Nomenclatural Authority: *Bezark & Monné (2013)*

Literature Records: none

Digitized Records: Santa Catalina (3 LACM), Santa Cruz (29 LACM; 1 SBMNH)

Range: Also known from mainland (*Linsley, 1963*).

Notes. *Linsley (1963)* reported this species as *Lampropterus ruficollis*.

**Megobrium LeConte, 1873**

Nomenclatural Authority: *Bezark & Monné (2013)*

Notes. One species of *Megobrium* has been recorded from California (*Linsley, 1963*).

**Megobrium edwardsi LeConte, 1873**

Nomenclatural Authority: *Bezark & Monné (2013)*

Literature Records: Santa Rosa (*LeConte, 1873*: 193; *Fall, 1897*: 238; *Fall, 1901*: 145; *Garnett, 1918*: 206; *Linsley, 1963*: 150)

Digitized Records: Santa Catalina (2 SBMNH; 4 LACM)

Range: Also known from mainland (*Fall, 1901*; *Garnett, 1918*; *Linsley, 1963*).

Notes. Often misspelled *M. edwardsii*, this species was considered endemic at the time of its description (*LeConte, 1873*).

### Methiini

Notes. Two genera and eight species of Methiini are known to occur in California (*Linsley, 1962*; M. L. Gimmel, 2022, unpublished data).

### *Styloxus* LeConte, 1873

Nomenclatural Authority: *Bezark & Monné (2013)*

Notes. Two species of *Styloxus* have been recorded from California (*Linsley, 1962*).

### *Styloxus fulleri* (Horn, 1880)

Nomenclatural Authority: *Bezark & Monné (2013)*

Literature Records: none

Digitized Records: Santa Cruz (1 iNat)

Range: Also known from mainland (*Linsley, 1962*).

Notes. According to *Linsley (1962*: 40), the subspecies occurring in California is *S. f. californicus* (Fall, 1901).

### Oemini

Notes. Six genera and six species of Oemini are known to occur in California (*Linsley, 1962*; M. L. Gimmel, 2022, unpublished data).

### *Paranoplium* Casey, 1924

Nomenclatural Authority: *Bezark & Monné (2013)*

Notes. One species of *Paranoplium* has been recorded from California (*Linsley, 1962*).

### *Paranoplium gracile* (LeConte, 1881)

Nomenclatural Authority: *Bezark & Monné (2013)*

Literature Records: Santa Catalina (*Fall, 1897*: 238; *Fall, 1901*: 144; *Garnett, 1918*: 176)

Digitized Records: Santa Catalina (1 LACM)

Range: Also known from mainland (*Fall, 1901*; *Garnett, 1918*; *Linsley, 1962*).

Notes. This species was reported by *Fall (1897*, *1901)* and *Garnett (1918)* as *Oeme gracilis*. According to *Linsley (1962*: 20), the coastal California subspecies is *P. g. gracile*.

### Psebiini

Notes. One species of Psebiini has been recorded from California (*Linsley, 1963*).

### *Nathrius* Brèthes, 1916

Nomenclatural Authority: *Bezark & Monné (2013)*

Notes. One species of *Nathrius* is known to occur in California (*Linsley, 1963*).

***Nathrius brevipennis*** (Mulsant, 1839)
Nomenclatural Authority: *Bezark & Monné (2013)*
Literature Records: none
Digitized Records: Santa Cruz (2 SBMNH)
Range: Also known from mainland (*Linsley, 1963*).
Notes. This species was introduced to North America from southern Europe (*Linsley, 1963*: 155; *Bezark & Monné, 2013*).

**Phoracanthini**
Notes. One genus and two species of Phoracanthini are known to occur in California (*Bezark & Monné, 2013*).

***Phoracantha*** Newman, 1840
Nomenclatural Authority: *Bezark & Monné (2013)*
Notes. Two species of *Phoracantha* are now known to occur in California (*Bezark & Monné, 2013*).

***Phoracantha recurva*** Newman, 1840
Nomenclatural Authority: *Bezark & Monné (2013)*
Literature Records: none
Digitized Records: Santa Catalina (1 SBMNH; 1 iNat), Santa Cruz (2 SBMNH)
Range: Also known from mainland (*Bezark & Monné, 2013*).
Notes. This species was introduced to North America from Australia (*Bezark & Monné, 2013*).

***Phoracantha semipunctata*** (Fabricius, 1775)
Nomenclatural Authority: *Bezark & Monné (2013)*
Literature Records: none
Digitized Records: Santa Catalina (2 LACM), Santa Cruz (3 SBMNH)
Range: Also known from mainland (*Bezark & Monné, 2013*).
Notes. This species was introduced to North America from Australia (*Bezark & Monné, 2013*).

**Lamiinae**
Notes. Nine tribes, 20 genera, and 31 species of Lamiinae have been recorded from California (*Linsley & Chemsak, 1984*, *1995*).

**Acanthocinini**
Notes. Seven genera and eight species of Acanthocinini have been recorded from California (*Linsley & Chemsak, 1995*).

***Sternidocinus*** Dillon, 1956
Nomenclatural Authority: *Bezark & Monné (2013)*
Notes. One species of *Sternidocinus* occurs in California (*Linsley & Chemsak, 1995*).

*Sternidocinus barbarus* (Van Dyke, 1920)
Nomenclatural Authority: *Bezark & Monné (2013)*
Literature Records: Santa Cruz (*Dillon, 1956*: 167; *Linsley & Chemsak, 1995*: 57)
Digitized Records: Santa Cruz (11 LACM; 5 SBMNH)
Range: Also known from mainland (*Dillon, 1956*; *Linsley & Chemsak, 1995*).

**Parmenini**
Notes. Two genera and two species of Parmenini have been recorded from California (*Linsley & Chemsak, 1984*).

*Ipochus* LeConte, 1852
Nomenclatural Authority: *Bezark & Monné (2013)*
Notes. One species of *Ipochus* occurs in California (*Linsley & Chemsak, 1984*).

*Ipochus fasciatus* LeConte, 1852
Nomenclatural Authority: *Bezark & Monné (2013)*
Literature Records: Anacapa (*Linsley & Chemsak, 1984*: 11 [map]; *Miller & Miller, 1985*: 130), San Clemente (*Linsley & Chemsak, 1984*: 11 [map]), San Miguel (*Cockerell, 1940*: 287; *Linsley & Chemsak, 1984*: 11 [map]; *Miller & Miller, 1985*: 130), Santa Barbara (*Miller & Miller, 1985*: 130), Santa Catalina (*Fall, 1897*: 238; *Casey, 1913b*: 281; *Cockerell, 1940*: 287; *Miller & Miller, 1985*: 130), Santa Cruz (*Linsley & Chemsak, 1984*: 11 [map]; *Miller & Miller, 1985*: 130; *Naughton et al., 2014*: 303), Santa Rosa (*Linsley & Chemsak, 1984*: 11 [map]; *Miller & Miller, 1985*: 130)
Digitized Records: Anacapa (3 CASC; 11 LACM; 2 SBMNH), San Miguel (5 LACM; 11 SBMNH), San Nicolas (1 SBMNH), Santa Barbara (1 LACM), Santa Catalina (3 CASC; 12 LACM; 6 SBMNH; 1 SDNHM; 1 USNM; 2 iNat), Santa Cruz (1 CASC; 1 LACM; 7 SBMNH; 1 iNat), Santa Rosa (1 CASC; 6 LACM; 1 SBMNH)
Range: Also known from mainland (*Linsley & Chemsak, 1984*).
Notes. *Fall (1897)* reported this species occurring "rather plentifully… under bark and on the branches of dead *Rhus laurina* (or *R. integrifolia*)." *Casey (1913b)* reported the presumed endemic Santa Catalina Island population as *Ipochus catalinae* Casey, 1913. This species was synonymized with *I. fasciatus* by *Linsley & Chemsak (1984*: 12), a finding already suggested by *Cockerell (1940*: 287).

**Phytoeciini**
Notes. Two genera and two species of Phytoeciini have been recorded from California (*Linsley & Chemsak, 1995*).

*Oberea* Mulsant, 1839
Nomenclatural Authority: *Bezark & Monné (2013)*
Notes. One species of *Oberea* has been recorded from California (*Linsley & Chemsak, 1995*).

*Oberea quadricallosa* LeConte, 1874
Nomenclatural Authority: *Bezark & Monné (2013)*

Literature Records: none
Digitized Records: Santa Cruz (2 SBMNH)
Range: Also known from mainland (*Linsley & Chemsak, 1995*).

**Pogonocherini**
Notes. Four genera and eight species of Pognocherini have been recorded from California (*Linsley & Chemsak, 1984*).

***Lophopogonius* Linsley, 1935**
Nomenclatural Authority: *Bezark & Monné (2013)*
Notes. One species of *Lophopogonius* has been recorded from California (*Linsley & Chemsak, 1984*).

***Lophopogonius crinitus* (LeConte, 1873)**
Nomenclatural Authority: *Bezark & Monné (2013)*
Literature Records: none
Digitized Records: Santa Cruz (1 SBMNH), Santa Rosa (3 SBMNH)
Range: Also known from mainland (*Linsley & Chemsak, 1984*).

**Saperdini**
Notes. One genus and three species of Saperdini have been recorded from California (*Linsley & Chemsak, 1984*).

***Saperda* Fabricius, 1775**
Nomenclatural Authority: *Bezark & Monné (2013)*
Notes. Three species of *Saperda* have been recorded from California (*Linsley & Chemsak, 1995*).

***Saperda horni* Joutel, 1902**
Nomenclatural Authority: *Bezark & Monné (2013)*
Literature Records: none
Digitized Records: Santa Cruz (1 SBMNH)
Range: Also known from mainland (*Linsley & Chemsak, 1995*).

**Lepturinae**
Notes. Six tribes, 35 genera, and 89 species of Lepturinae have been recorded from California (*Chemsak, 2005*). *Chemsak (2005)* provided a guide to the genera and species of the subfamily in North America.

**Desmocerini**
Notes. One genus and two species of Desmocerini have been recorded from California (*Chemsak, 2005*).

***Desmocerus* Dejean, 1821**
Nomenclatural Authority: *Bezark & Monné (2013)*
Notes. Two species of *Desmocerus* have been recorded from California (*Chemsak, 2005*).

***Desmocerus californicus*** **Horn, 1881**

Nomenclatural Authority: *Bezark & Monné (2013)*

Literature Records: none

Digitized Records: Santa Cruz (3 LACM), Santa Rosa (1 LACM)

Range: Also known from mainland (*Chemsak, 2005*).

Notes. All island records of this species belong to *D. c. californicus* Horn, 1881. The other subspecies, *D. c. dimorphus* Fisher, 1921, is restricted to the Central Valley of California and is federally protected.

**Lepturini**

Notes. Eighteen genera and 48 species of Lepturini have been recorded from California (*Chemsak, 2005*).

***Anastrangalia*** **Casey, 1924**

Nomenclatural Authority: *Bezark & Monné (2013)*

Notes. Two species of *Anastrangalia* have been recorded from California (*Chemsak, 2005*).

***Anastrangalia laetifica*** **(LeConte, 1859)**

Nomenclatural Authority: *Bezark & Monné (2013)*

Literature Records: none

Digitized Records: Santa Cruz (1 CASC; 1 LACM; 4 SBMNH; 1 iNat)

Range: Also known from mainland (*Chemsak, 2005*).

***Strophiona*** **Casey, 1913**

Nomenclatural Authority: *Bezark & Monné (2013)*

Notes. Two species of *Strophiona* have been recorded from California (*Chemsak, 2005*).

***Strophiona tigrina*** **Casey, 1913**

Nomenclatural Authority: *Bezark & Monné (2013)*

Literature Records: none

Digitized Records: Santa Catalina (2 SBMNH), Santa Cruz (1 LACM; 2 SBMNH)

Range: Also known from mainland (*Chemsak, 2005*).

***Xestoleptura*** **Casey, 1913**

Nomenclatural Authority: *Bezark & Monné (2013)*

Notes. Four species of *Xestoleptura* have been recorded from California (*Chemsak, 2005*).

***Xestoleptura crassipes*** **(LeConte, 1857)**

Nomenclatural Authority: *Bezark & Monné (2013)*

Literature Records: none

Digitized Records: Santa Cruz (1 SBMNH)

Range: Also known from mainland (*Chemsak, 2005*).

**Necydalini**

Notes. Two genera and seven species of Necydalini have been recorded from California (*Chemsak, 2005*). The tribe is sometimes recognized as a full subfamily (*e.g.*, *Bezark & Monné, 2013*).

**_Necydalis_ Linnaeus, 1758**

Nomenclatural Authority: *Bezark & Monné (2013)*

Notes. There are six California species of the Holarctic genus *Necydalis* (*Chemsak, 2005*).

**_Necydalis laevicollis_ LeConte, 1869**

Nomenclatural Authority: *Bezark & Monné (2013)*

Literature Records: none

Digitized Records: Santa Rosa (1 USNM)

Range: Also known from mainland (*Chemsak, 2005*).

Notes. The Santa Rosa Island record represents a significant southerly range extension for this species; the nearest recorded specimens are known from the San Francisco Bay area (*Chemsak, 2005*).

**Rhagiini**

Notes. Twelve genera and 35 species of Rhagiini have been recorded from California (*Chemsak, 2005*).

**_Brachysomida_ Casey, 1913**

Nomenclatural Authority: *Bezark & Monné (2013)*

Notes. Two species of *Brachysomida* have been recorded from California (*Chemsak, 2005*).

**_Brachysomida californica_ (LeConte, 1851)**

Nomenclatural Authority: *Bezark & Monné (2013)*

Literature Records: none

Digitized Records: Santa Rosa (1 SBMNH)

Range: Also known from mainland (*Chemsak, 2005*).

**_Centrodera_ LeConte, 1850**

Nomenclatural Authority: *Bezark & Monné (2013)*

Notes. Seven species of *Centrodera* have been recorded from California (*Chemsak, 2005*).

**_Centrodera autumnata_ Leech, 1963**

Nomenclatural Authority: *Bezark & Monné (2013)*

Literature Records: none

Digitized Records: Santa Cruz (2 LACM; 1 SBMNH)

Range: Also known from mainland (*Chemsak, 2005*).

**_Centrodera spurca_ (LeConte, 1857)**

Nomenclatural Authority: *Bezark & Monné (2013)*

Literature Records: none

Digitized Records: Santa Cruz (2 iNat)
Range: Also known from mainland (*Chemsak, 2005*).

### *Stenocorus* Geoffroy, 1762
Nomenclatural Authority: *Bezark & Monné (2013)*
Notes. Four species of *Stenocorus* have been recorded from California (*Chemsak, 2005*).

### *Stenocorus* (*Stenocorus*) *vestitus* (Haldeman, 1847)
Nomenclatural Authority: *Bezark & Monné (2013)*
Literature Records: none
Digitized Records: Santa Cruz (2 LACM; 11 SBMNH; 1 SDNHM)
Range: Also known from mainland (*Chemsak, 2005*).

### Prioninae
Notes. Five tribes, six genera, and 12 species of Prioninae have been recorded from California (M. L. Gimmel, 2022, unpublished data).

### Callipogonini
Notes. One genus and two species of Callipogonini have been recorded from California (*Chemsak, 1996*; *Swift, Santos-Silva & Nearns, 2010*).

### *Trichocnemis* LeConte, 1851
Nomenclatural Authority: *Bezark & Monné (2013)*
Notes. This genus was reviewed by *Swift, Santos-Silva & Nearns (2010)*. Two species occur in California (*Chemsak, 1996*; *Swift, Santos-Silva & Nearns, 2010*).

### *Trichocnemis spiculatus* LeConte, 1851
Nomenclatural Authority: *Bezark & Monné (2013)*
Literature Records: none
Digitized Records: Santa Rosa (1 SBMNH)
Range: Also known from mainland (*Linsley, 1962*; *Chemsak, 1996*).
Notes. This species was indicated as *Ergates spiculatus* in *Linsley (1962)* and *Chemsak (1996)*, but was returned to *Trichocnemis* in *Swift, Santos-Silva & Nearns (2010)*.

### Prionini
Notes. Two genera and five species of Prionini have been recorded from California (M. L. Gimmel, 2022, unpublished data).

### *Prionus* Geoffroy, 1762
Nomenclatural Authority: *Santos-Silva, Nearns & Swift (2016)*
Notes. This genus was revised for the New World by *Santos-Silva, Nearns & Swift (2016)*. Two species occur in California (*Chemsak, 1996*; *Santos-Silva, Nearns & Swift, 2016*).

### *Prionus* (*Prionus*) *californicus* Motschulsky, 1845
Nomenclatural Authority: *Santos-Silva, Nearns & Swift (2016)*
Literature Records: none

Digitized Records: Santa Cruz (3 CASC; 19 LACM; 15 SBMNH; 2 UCSB; 2 iNat)

Range: Also known from mainland (*Linsley, 1962*; *Chemsak, 1996*; *Santos-Silva, Nearns & Swift, 2016*).

Notes. It is somewhat surprising that this large and readily recognizable species has not been reported in the literature from Santa Cruz Island before now.

### Spondylidinae: Asemini

Notes. Three tribes, seven genera, and 16 species of Spondylidinae, of which four genera and nine species belong to Asemini, have been recorded from California (*Chemsak, 1996*).

### *Arhopalus* Audinet-Serville, 1834

Nomenclatural Authority: *Bezark & Monné (2013)*

Notes. Three species of this genus are known to occur in California (*Chemsak, 1996*). A key to the North American species was provided by *Chemsak (1996)*.

### *Arhopalus asperatus* (LeConte, 1859)

Nomenclatural Authority: *Bezark & Monné (2013)*

Literature Records: Santa Catalina (*Linsley, 1962*: 71 [map])

Digitized Records: none

Range: Also known from mainland (*Linsley, 1962*; *Chemsak, 1996*).

### *Arhopalus productus* (LeConte, 1850)

Nomenclatural Authority: *Bezark & Monné (2013)*

Literature Records: Santa Catalina (*Linsley, 1962*: 74 [map])

Digitized Records: none

Range: Also known from mainland (*Linsley, 1962*; *Chemsak, 1996*).

### *Asemum* Eschscholtz, 1830

Nomenclatural Authority: *Bezark & Monné (2013)*

Notes. Three species of this genus are known to occur in California (*Chemsak, 1996*). A key to the North American species was provided by *Chemsak (1996)*.

### *Asemum nitidum* LeConte, 1873

Nomenclatural Authority: *Bezark & Monné (2013)*

Literature Records: none

Digitized Records: Santa Cruz (1 SBMNH)

Range: Also known from mainland (*Linsley, 1962*; *Chemsak, 1996*).

### Chrysomelidae

Notes. Nine subfamilies, 106 genera, and 436 species of Chrysomelidae have been recorded from California (*Riley, Clark & Seeno, 2003*; *Kingsolver, 2004*; M. L. Gimmel, 2022, unpublished data). The subfamilies Donaciinae and Synetinae have not been reported from the Channel Islands.

**Bruchinae: Bruchini**

Notes. Two tribes, 12 genera, and 43 species of Bruchinae, of which 10 genera and 35 species are in the tribe Bruchini, have been recorded from California (*Kingsolver, 2004*; M. L. Gimmel, 2022, unpublished data). *Kingsolver (2004)* reviewed the North American fauna of the subfamily.

***Acanthoscelides* Schilsky, 1905**

Nomenclatural Authority: *Kingsolver (2004)*

Notes. Seventeen species of *Acanthoscelides* have been recorded from California (*Kingsolver, 2004*).

***Acanthoscelides margaretae* Johnson, 1970**

Nomenclatural Authority: *Kingsolver (2004)*

Literature Records: none

Digitized Records: San Miguel (12 SBMNH), Santa Catalina (3 SBMNH), Santa Cruz (2 SBMNH), Santa Rosa (7 SBMNH)

Range: Also known from mainland (*Kingsolver, 2004*).

***Acanthoscelides napensis* Johnson, 1970**

Nomenclatural Authority: *Kingsolver (2004)*

Literature Records: San Miguel (*Miller & Davis, 1986*: 550)

Digitized Records: San Clemente (1 LACM; 15 SBMNH), Santa Cruz (20 SBMNH), Santa Rosa (125 LACM; 16 SBMNH)

Range: Also known from mainland (*Kingsolver, 2004*).

***Acanthoscelides pauperculus* (LeConte, 1857)**

Nomenclatural Authority: *Kingsolver (2004)*

Literature Records: San Miguel (*Cockerell, 1940*: 287), Santa Catalina (*Fall, 1897*: 238)

Digitized Records: none

Range: Also known from mainland (*Kingsolver, 2004*).

Notes. Recorded as *Bruchus pauperculus* by *Fall (1897)* and *Cockerell (1940)*.

***Acanthoscelides pullus* (Fall, 1910)**

Nomenclatural Authority: *Kingsolver (2004)*

Literature Records: Santa Catalina (*Fall, 1910*: 180)

Digitized Records: Anacapa (76 LACM; 1 SBMNH), San Clemente (10 LACM; 1 SBMNH), San Miguel (5 LACM; 1 SBMNH), San Nicolas (8 LACM; 1 SBMNH), Santa Cruz (2 LACM), Santa Rosa (57 LACM; 1 SBMNH)

Range: Also known from mainland (*Kingsolver, 2004*).

Notes. *Fall (1910)* recorded this species as *Bruchus pullus*.

***Megacerus* Fåhraeus, 1839**

Nomenclatural Authority: *Kingsolver (2004)*

Notes. One species of *Megacerus* has been recorded from California (*Kingsolver, 2004*).

*Megacerus* (*Megacerus*) *impiger* (Horn, 1873)
Nomenclatural Authority: *Kingsolver (2004)*
Literature Records: Santa Cruz (*Schlising, 1980*: 6 [map])
Digitized Records: San Nicolas (1 SBMNH), Santa Cruz (1 SBMNH)
Range: Also known from mainland (*Schlising, 1980*; *Kingsolver, 2004*).

*Stator* Bridwell, 1946
Nomenclatural Authority: *Kingsolver (2004)*
Notes. Two species of *Stator* have been recorded from California (*Kingsolver, 2004*).

*Stator limbatus* (Horn, 1873)
Nomenclatural Authority: *Kingsolver (2004)*
Literature Records: Santa Catalina (*Cockerell, 1940*: 287; *Johnson, 1963*: 861)
Digitized Records: Santa Catalina (1 iNat)
Range: Also known from mainland (*Johnson, 1963*; *Kingsolver, 2004*).
Notes. Recorded as *Bruchus limbatus* by *Cockerell (1940)*.

**Cassidinae: Cassidini**
Notes. Three tribes, 13 genera, and 25 species of Cassidinae are known to occur in California (*Riley, Clark & Seeno, 2003*; M. L. Gimmel, 2022, unpublished data).

*Charidotella* Weise, 1896
Nomenclatural Authority: *Riley, Clark & Seeno (2003)*
Notes. One species of *Charidotella* has been recorded from California (*Riley, Clark & Seeno, 2003*).

*Charidotella sexpunctata* (Fabricius, 1781)
Nomenclatural Authority: *Riley, Clark & Seeno (2003)*
Literature Records: none
Digitized Records: Anacapa (68 LACM; 15 SBMNH), Santa Cruz (5 SBMNH; 1 UCSB; 6 iNat)
Range: Also known from mainland (*Riley, Clark & Seeno, 2003*).
Notes. The only subspecies of *C. sexpunctata* occurring in California is *C. s. bicolor* (Fabricius, 1798) (*Riley, Clark & Seeno, 2003*).

**Chrysomelinae: Chrysomelini**
Notes. Two tribes, 14 genera, and 32 species of Chrysomelinae, of which 12 genera and 30 species belong to Chrysomelini, have been recorded from California (*Riley, Clark & Seeno, 2003*; M. L. Gimmel, 2022, unpublished data). *Wilcox (1972)* provided keys to all North American genera and species of Chrysomelinae known at the time.

*Calligrapha* Chevrolat, 1836
Nomenclatural Authority: *Riley, Clark & Seeno (2003)*
Notes. Six species of *Calligrapha* have been reported from California (*Riley, Clark & Seeno, 2003*).

***Calligrapha* (*Calligramma*) *sigmoidea* (LeConte, 1859)**
Nomenclatural Authority: *Riley, Clark & Seeno (2003)*
Literature Records: none
Digitized Records: Santa Rosa (1 SBMNH)
Range: Also known from mainland (*Riley, Clark & Seeno, 2003*).

***Gastrophysa* Chevrolat, 1836**
Nomenclatural Authority: *Riley, Clark & Seeno (2003)*
Notes. One species of *Gastrophysa* has been recorded from California (*Riley, Clark & Seeno, 2003*).

***Gastrophysa cyanea* Melsheimer, 1847**
Nomenclatural Authority: *Riley, Clark & Seeno (2003)*
Literature Records: none
Digitized Records: San Nicolas (6 SBMNH), Santa Cruz (8 SBMNH), Santa Rosa (4 LACM)
Range: Also known from mainland (*Riley, Clark & Seeno, 2003*).

***Phaedon* Megerle von Mühlfeld, 1823**
Nomenclatural Authority: *Riley, Clark & Seeno (2003)*
Notes. Five species of *Phaedon* have been recorded from California (*Riley, Clark & Seeno, 2003*).

***Phaedon* (*Allophaedon*) *prasinellus* (LeConte, 1861)**
Nomenclatural Authority: *Riley, Clark & Seeno (2003)*
Literature Records: Santa Barbara (*Miller & Miller, 1985*: 131)
Digitized Records: Santa Barbara (1 SBMNH)
Range: Also known from mainland (*Riley, Clark & Seeno, 2003*).

***Plagiodera* Chevrolat, 1836**
Nomenclatural Authority: *Riley, Clark & Seeno (2003)*
Notes. One species of *Plagiodera* has been recorded from California (*Riley, Clark & Seeno, 2003*).

***Plagiodera* (*Plagiomorpha*) *californica* (Rogers, 1856)**
Nomenclatural Authority: *Riley, Clark & Seeno (2003)*
Literature Records: Santa Cruz (*Cockerell, 1940*: 287)
Digitized Records: Santa Cruz (14 LACM; 24 SBMNH)
Range: Also known from mainland (*Fall, 1901*).
Notes. Recorded as *Lina californica* by *Cockerell (1940)*.

***Trachymela* Weise, 1908**
Nomenclatural Authority: *Riley, Clark & Seeno (2003)*
Notes. One species of *Trachymela* is known from California (*Riley, Clark & Seeno, 2003*).

*Trachymela sloanei* (Blackburn, 1896)

Nomenclatural Authority: *Riley, Clark & Seeno (2003)*

Literature Records: none

Digitized Records: Santa Cruz (2 UCSB; 1 iNat)

Range: Also known from mainland (*Riley, Clark & Seeno, 2003*).

Notes. This species was introduced from Australia (*Riley, Clark & Seeno, 2003*).

**Criocerinae: Lemiini**

Notes. Two tribes, three genera, and six species of Criocerinae, of which two genera and four species belong to Lemiini, have been recorded from California (*Riley, Clark & Seeno, 2003*; M. L. Gimmel, 2022, unpublished data).

*Lema* Fabricius, 1798

Nomenclatural Authority: *Riley, Clark & Seeno (2003)*

Notes. Three species of *Lema* have been recorded from California (*Riley, Clark & Seeno, 2003*).

*Lema daturaphila* Kogan & Goeden, 1970

Nomenclatural Authority: *Riley, Clark & Seeno (2003)*

Literature Records: Santa Cruz (*Fall & Davis, 1934*: 144)

Digitized Records: Santa Cruz (74 LACM; 5 SBMNH), Santa Rosa (25 LACM; 2 SBMNH)

Range: Also known from mainland (*Riley, Clark & Seeno, 2003*).

Notes. This species was recorded as *Lema trilineata* var. *californica* Schaeffer, 1933 by *Fall & Davis (1934)*.

**Cryptocephalinae: Cryptocephalini**

Notes. Three tribes, 10 genera, and 79 species of Cryptocephalinae, of which three genera and 61 species belong to Cryptocephalini, have been recorded from California (*Riley, Clark & Seeno, 2003*; M. L. Gimmel, 2022, unpublished data).

*Cryptocephalus* Geoffroy, 1762

Nomenclatural Authority: *Riley, Clark & Seeno (2003)*

Notes: Twelve species of *Cryptocephalus* have been reported from California (*Riley, Clark & Seeno, 2003*). The species were revised for North America by *White (1968)*.

*Cryptocephalus sanguinicollis* Suffrian, 1852

Nomenclatural Authority: *Riley, Clark & Seeno (2003)*

Literature Records: none

Digitized Records: Santa Catalina (1 SBMNH)

Range: The two subspecies of *C. sanguinicollis* together span most of western North America. The subspecies reported from the Channel Islands is *C. s. nigerrimus* Crotch, 1874.

*Diachus* LeConte, 1880

Nomenclatural Authority: *Riley, Clark & Seeno (2003)*

Notes. Two species of *Diachus* have been recorded from California (*Riley, Clark & Seeno, 2003*).

*Diachus auratus* (Fabricius, 1801)

Nomenclatural Authority: *Riley, Clark & Seeno (2003)*

Literature Records: San Clemente (*Fall, 1897*: 238), San Miguel (*Miller & Davis, 1986*: 550), Santa Catalina (*Fall, 1897*: 238), Santa Cruz (*Naughton et al., 2014*: 303), Santa Rosa (*Fall, 1897*: 238)

Digitized Records: Anacapa (5 SBMNH), San Clemente (21 SBMNH), San Miguel (19 SBMNH), San Nicolas (9 SBMNH), Santa Catalina (15 SBMNH), Santa Cruz (16 SBMNH), Santa Rosa (4 SBMNH)

Range: Also known from mainland (*Riley, Clark & Seeno, 2003*).

Notes. This species was reported from flowers of *Malacothrix* by *Miller & Davis (1986)*.

*Pachybrachis* Chevrolat, 1836

Nomenclatural Authority: *Riley, Clark & Seeno (2003)*

Digitized Records (genus-only): Santa Cruz (1 UCSB)

Notes. Forty-seven species of *Pachybrachis* have been recorded from California (M. L. Gimmel, 2022, unpublished data).

*Pachybrachis melanostictus* Suffrian, 1852

Nomenclatural Authority: *Riley, Clark & Seeno (2003)*

Literature Records: none

Digitized Records: Santa Cruz (16 SBMNH)

Range: Also known from mainland (*Riley, Clark & Seeno, 2003*).

*Pachybrachis mobilis* Fall, 1915

Nomenclatural Authority: *Riley, Clark & Seeno (2003)*

Literature Records: none

Digitized Records: Santa Catalina (3 SBMNH)

Range: Also known from mainland (*Riley, Clark & Seeno, 2003*).

*Pachybrachis pluripunctatus* Fall, 1915

Nomenclatural Authority: *Riley, Clark & Seeno (2003)*

Literature Records: none

Digitized Records: Santa Cruz (1 SBMNH)

Range: Also known from mainland (*Riley, Clark & Seeno, 2003*).

*Pachybrachis punctatus* Bowditch, 1909

Nomenclatural Authority: *Riley, Clark & Seeno (2003)*

Literature Records: Santa Catalina (*Fall, 1915*: 343)

Digitized Records: Santa Cruz (1 SBMNH)

Range: Also known from mainland (*Fall, 1915*).

Notes. *Fall (1915)* reported this species as *Pachybrachys punctatus*. Earlier, *Fall (1897)*: 238) listed two separate undetermined "*Pachybrachys*" species from Santa Catalina; this probably represents one of them.

### *Pachybrachis quadratus* Fall, 1915
Nomenclatural Authority: *Riley, Clark & Seeno (2003)*
Literature Records: Santa Catalina (*Fall, 1915*: 406)
Digitized Records: none
Range: Also known from mainland (*Fall, 1915*).
Notes. *Fall (1915)* reported this species as *Pachybrachys punctatus*. Earlier, *Fall (1897)*: 238) listed two separate undetermined "*Pachybrachys*" species from Santa Catalina; this probably represents one of them.

### Eumolpinae
Notes. Three tribes, 10 genera, and 36 species of Eumolpinae have been recorded from California (*Riley, Clark & Seeno, 2003*; M. L. Gimmel, 2022, unpublished data). *Straughan & Hadley (1980*: 392) recorded "Eumolpinae" from Catalina Harbor, Santa Catalina Island.

### Adoxini
Notes. Two genera and four species of Adoxini have been recorded from California (*Riley, Clark & Seeno, 2003*).

### *Colaspidea* Laporte, 1833
Nomenclatural Authority: *Riley, Clark & Seeno (2003)*
Notes. Three species of *Colaspidea* have been recorded from California (*Riley, Clark & Seeno, 2003*).

### *Colaspidea smaragdula* (LeConte, 1857)
Nomenclatural Authority: *Riley, Clark & Seeno (2003)*
Literature Records: San Clemente (*Fall, 1897*: 238; *Fall, 1901*: 154; *Fall, 1933*: 232; *Miller, 1985a*: 21), Santa Catalina (*Fall, 1897*: 238; *Linell, 1898*: 481; *Fall, 1901*: 154; *Fall, 1933*: 232; *Miller, 1985a*: 21)
Digitized Records: San Clemente (8 SBMNH), Santa Catalina (2 SBMNH), Santa Cruz (6 SBMNH)
Range: Also known from mainland (*Riley, Clark & Seeno, 2003*).
Notes. A supposed endemic, *Colaspidea subvittata* Fall, 1897, was described from San Clemente and Santa Catalina and reported in the works of *Fall (1897*, *1901*, *1933)*, *Linell (1898)*, and *Miller (1985a)*. The species was later synonymized with the widespread *C. smaragdula* by *Schultz (1970)*, reflected in *Riley, Clark & Seeno (2003)*.

### Eumolpini
Notes. Five genera and 22 species of Eumolpini have been recorded from California (*Riley, Clark & Seeno, 2003*; M. L. Gimmel, 2022, unpublished data).

### *Spintherophyta* Dejean, 1836
Nomenclatural Authority: *Riley, Clark & Seeno (2003)*

Notes. Three species of *Spintherophyta* have been recorded from California (*Riley, Clark & Seeno, 2003*; *Gilbert & Clark, 2020*). These were keyed by *Gilbert & Clark (2020)*.

### *Spintherophyta punctum* Gilbert & Clark, 2020
Nomenclatural Authority: *Riley, Clark & Seeno (2003)*
Literature Records: Santa Rosa (*Gilbert & Clark, 2020*: 558)
Digitized Records: none
Range: Endemic (*Gilbert & Clark, 2020*).
Notes. The holotype and paratypes of this species were collected "on leaves of willow" (*Gilbert & Clark, 2020*).

### Galerucinae
Notes. Four tribes, 39 genera, and 199 species have been recorded from California (*Riley, Clark & Seeno, 2003*; M. L. Gimmel, 2022, unpublished data). *Wilcox (1965)* provided keys to the non-Alticini taxa for North America.

### Alticini
Notes. Twenty-one genera and 114 species of Alticini have been recorded from California (*Riley, Clark & Seeno, 2003*; M. L. Gimmel, 2022, unpublished data).

### *Altica* Geoffroy, 1762
Nomenclatural Authority: *Riley, Clark & Seeno (2003)*
Notes. Twenty species of *Altica* have been recorded from California (*Riley, Clark & Seeno, 2003*).

### *Altica* undetermined species
Literature Records: none
Digitized Records: Anacapa (4 SBMNH), Santa Cruz (1 SBMNH)

### *Aulacothorax* Boheman, 1858
Nomenclatural Authority: *Bezděk & Konstantinov (2017)*.
Notes. One species of *Aulacothorax* has been recorded from California (*Riley, Clark & Seeno, 2003*). This genus was, until recently, known as *Orthaltica* Crotch, 1873 (see *Bezděk & Konstantinov, 2017*).

### *Aulacothorax recticollis* (LeConte, 1861)
Nomenclatural Authority: *Bezděk & Konstantinov (2017)*
Literature Records: none
Digitized Records: Santa Catalina (3 SBMNH), Santa Cruz (1 SBMNH)
Range: Also known from mainland (*Riley, Clark & Seeno, 2003*).
Notes. This species was until recently known as *Orthaltica recticollis* (see *Bezděk & Konstantinov, 2017*).

### *Dibolia* Latreille, 1829
Nomenclatural Authority: *Riley, Clark & Seeno (2003)*

Notes. Three species of *Dibolia* have been recorded from California (*Riley, Clark & Seeno, 2003*). The genus was revised for North America by *Parry (1974)*.

### *Dibolia californica* Parry, 1974
Nomenclatural Authority: *Riley, Clark & Seeno (2003)*
Literature Records: none
Digitized Records: Santa Cruz (4 SBMNH)
Range: Also known from mainland (*Parry, 1974*).

### *Disonycha* Chevrolat, 1836
Nomenclatural Authority: *Riley, Clark & Seeno (2003)*
Notes. Nine species of *Disonycha* have been recorded from California (*Riley, Clark & Seeno, 2003*). *Blake (1933)* keyed out the species.

### *Disonycha latiovittata* Hatch, 1932
Nomenclatural Authority: *Riley, Clark & Seeno (2003)*
Literature Records: none
Digitized Records: Santa Cruz (27 LACM), Santa Rosa (11 LACM)
Range: Also known from mainland (*Blake, 1933*).

### *Epitrix* Foudras, 1859
Nomenclatural Authority: *Riley, Clark & Seeno (2003)*
Notes. Four species of *Epitrix* have been recorded from California (*Riley, Clark & Seeno, 2003*). *Seeno & Andrews (1972)* provided keys to the California species.

### *Epitrix similaris* Gentner, 1944
Nomenclatural Authority: *Riley, Clark & Seeno (2003)*
Literature Records: Santa Catalina (*Gentner, 1944*: 142; *Seeno & Andrews, 1972*: 59 [map])
Digitized Records: none
Range: Also known from mainland (*Gentner, 1944*; *Seeno & Andrews, 1972*).

### *Epitrix subcrinita* (LeConte, 1857)
Nomenclatural Authority: *Riley, Clark & Seeno (2003)*
Literature Records: Santa Cruz (*Naughton et al., 2014*: 303)
Digitized Records: Santa Cruz (6 SBMNH)
Range: Also known from mainland (*Seeno & Andrews, 1972*).
Notes. MLG examined vouchers from the *Naughton et al. (2014)* study, which were previously identified only to genus; they belong to *E. subcrinita*.

### *Longitarsus* Berthold, 1827
Nomenclatural Authority: *Riley, Clark & Seeno (2003)*
Notes. Ten species of *Longitarsus* have been recorded from California (*Riley, Clark & Seeno, 2003*).

### *Longitarsus* undetermined species 1
Literature Records: none

Digitized Records: San Clemente (1 SBMNH)

Notes. This is a pale, brachypterous species with complete elytra and long antennae, and less than 1.5 mm in total body length (M. L. Gimmel, 2021, personal observation).

### *Longitarsus* undetermined species 2

Literature Records: none

Digitized Records: San Nicolas (2 SBMNH)

Notes. This is a pale, brachypterous species with complete elytra and short antennae, and at least 2.0 mm in total body length (M. L. Gimmel, 2021, personal observation).

### *Phyllotreta* Chevrolat, 1836

Nomenclatural Authority: *Riley, Clark & Seeno (2003)*

Digitized Records (genus-only): Santa Cruz (1 SBMNH)

Notes. Twenty-one species of *Phyllotreta* have been recorded from California (*Riley, Clark & Seeno, 2003*). The specimen from Santa Cruz Island housed in the SBMNH has been tentatively identified as *Phyllotreta inconspicua* Chittenden, 1927, pending examination of the type of that species (H. Douglas, 2022, personal communication).

### *Phyllotreta pusilla* Horn, 1889

Nomenclatural Authority: *Riley, Clark & Seeno (2003)*

Literature Records: Santa Catalina (*Fall, 1897*: 238)

Digitized Records: none

Range: Also known from mainland (*Riley, Clark & Seeno, 2003*).

### Galerucini

Notes. Ten genera and 36 species of Galerucini have been recorded from California (*Riley, Clark & Seeno, 2003*; *Viswajyothi & Clark, 2022*).

### *Erynephala* Blake, 1936

Nomenclatural Authority: *Riley, Clark & Seeno (2003)*

Notes. Two species of *Erynephala* have been recorded from California (*Riley, Clark & Seeno, 2003*).

### *Erynephala morosa* (LeConte, 1857)

Nomenclatural Authority: *Riley, Clark & Seeno (2003)*

Literature Records: Santa Rosa (*Fall, 1897*: 238)

Digitized Records: none

Range: Also known from mainland (*Riley, Clark & Seeno, 2003*).

Notes. Recorded as *Monoxia puncticollis* (Say, 1824) by *Fall (1897)*. Based on information in *Blake (1936)*, this record almost certainly applies to *E. morosa* and not to the more easterly-occurring species currently known as *Erynephala puncticollis* (Say, 1824).

### *Monoxia* LeConte, 1865

Nomenclatural Authority: *Riley, Clark & Seeno (2003)*

Notes. Six species of *Monoxia* have been recorded from California (*Riley, Clark & Seeno, 2003*; *Viswajyothi & Clark, 2022*).

***Monoxia* undetermined species**
Literature Records: none
Digitized Records: Anacapa (2 SBMNH)
Notes. This pair of *Monoxia* from Anacapa Island did not readily match any species presented in *Blake (1939)* (M. L. Gimmel, 2021, personal observation).

***Trirhabda* LeConte, 1865**
Nomenclatural Authority: *Riley, Clark & Seeno (2003)*
Digitized Records (genus-only): Santa Cruz (9 UCSB)
Notes. Sixteen species of *Trirhabda* have been recorded from California (*Riley, Clark & Seeno, 2003*).

***Trirhabda confusa* Blake, 1931**
Nomenclatural Authority: *Riley, Clark & Seeno (2003)*
Literature Records: none
Digitized Records: Santa Cruz (1 SBMNH)
Range: Also known from mainland (*Riley, Clark & Seeno, 2003*).

***Trirhabda sericotrachyla* Blake, 1931**
Nomenclatural Authority: *Riley, Clark & Seeno (2003)*
Literature Records: none
Digitized Records: Santa Cruz (10 SBMNH), Santa Rosa (6 SBMNH)
Range: Also known from mainland (*Riley, Clark & Seeno, 2003*).

***Yingabruxia* Viswajyothi & Clark, 2022**
Nomenclatural Authority: *Viswajyothi & Clark (2022)*
Notes. Three species of *Yingabruxia* have been recorded from California (*Riley, Clark & Seeno, 2003*; *Viswajyothi & Clark, 2022*).

***Yingabruxia sordida* (LeConte, 1858)**
Nomenclatural Authority: *Viswajyothi & Clark (2022)*
Literature Records: San Clemente (*Miller & Miller, 1985*: 130), Santa Barbara (*Miller & Miller, 1985*: 130)
Digitized Records: San Nicolas (1 SBMNH), Santa Barbara (3 SBMNH)
Range: Also known from mainland (*Riley, Clark & Seeno, 2003*). *Miller & Miller (1985)* reported this species as *Monoxia sordida*; this species was recently transferred to the genus *Yingabruxia*.

**Luperini**
Notes. Nine genera and 49 species of Luperini have been recorded from California (*Riley, Clark & Seeno, 2003*; M. L. Gimmel, 2022, unpublished data).

***Diabrotica* Chevrolat, 1836**
Nomenclatural Authority: *Riley, Clark & Seeno (2003)*
Notes. Two species of *Diabrotica* have been recorded from California (*Riley, Clark & Seeno, 2003*).

*Diabrotica undecimpunctata* **Mannerheim, 1843**
Nomenclatural Authority: *Riley, Clark & Seeno (2003)*
Literature Records: Santa Barbara (*Miller & Miller, 1985*: 130), Santa Catalina (*Seavey, 1892*: 263; *Fall, 1897*: 238), Santa Cruz (*Smith, 1966*: 109 [map]; *Miller & Miller, 1985*: 130)
Digitized Records: Santa Barbara (1 SBMNH), Santa Cruz (8 LACM; 3 SBMNH; 6 UCSB), Santa Rosa (7 LACM)
Range: Also known from mainland (*Smith, 1966*; *Riley, Clark & Seeno, 2003*).
Notes. This species was recorded as *Diabrotica soror* LeConte, 1865 by *Seavey (1892)* and *Fall (1897)*.

*Scelolyperus* **Crotch, 1874**
Nomenclatural Authority: *Riley, Clark & Seeno (2003)*
Notes. Twenty species of *Scelolyperus* have been recorded from California (*Riley, Clark & Seeno, 2003*). *Clark (1996)* revised the genus for North America.

*Scelolyperus torquatus* **(LeConte, 1884)**
Nomenclatural Authority: *Riley, Clark & Seeno (2003)*
Literature Records: Santa Catalina (*Wilcox, 1965*: 140)
Digitized Records: none
Range: Also known from mainland (*Wilcox, 1965*; *Clark, 1996*; *Riley, Clark & Seeno, 2003*).

**CURCULIONOIDEA**

**Attelabidae**
Notes. One subfamily (Rhynchitinae), six genera, and 18 species of Attelabidae have been recorded from California (*O'Brien & Wibmer, 1982*).

*Deporaus* **Samouelle, 1819**
Nomenclatural Authority: *O'Brien & Wibmer (1982)*
Notes. One species of *Deporaus* has been recorded from California (*O'Brien & Wibmer, 1982*).

*Deporaus glastinus* **(LeConte, 1857)**
Nomenclatural Authority: *O'Brien & Wibmer (1982)*
Literature Records: Santa Cruz (*Hamilton, 1969*: 394)
Digitized Records: Santa Catalina (9 LACM), Santa Cruz (33 LACM; 17 SBMNH)
Range: Also known from mainland (*O'Brien & Wibmer, 1982*; *Hamilton, 1969*, *2002*).

*Temnocerus* **Thunberg, 1815**
Nomenclatural Authority: *Hamilton (2002)*
Notes. Six species of *Temnocerus* have been recorded from California (*O'Brien & Wibmer, 1982*). This genus was until recently known as *Pselaphorhynchites* Schilsky, 1903.

*Temnocerus aeratoides* **(Fall, 1901)**
Nomenclatural Authority: *Hamilton (1971)*, *Hamilton (2002)*
Literature Records: none

Digitized Records: Santa Cruz (2 SBMNH)

Range: Also known from mainland (*Hamilton, 1971*).

Notes. *Hamilton (1971)* recorded this species as *Pselaphorhynchites aeratoides*.

### *Temnocerus aureus* (LeConte, 1876)

Nomenclatural Authority: *Hamilton (1971)*, *Hamilton (2002)*

Literature Records: San Clemente (*Fall, 1897*: 239; *Fall, 1901*: 186)

Digitized Records: Santa Cruz (21 SBMNH)

Range: Also known from mainland (*Fall, 1901*).

Notes. *Fall (1897, 1901)* recorded this species as *Rhynchites aureus*, and *Hamilton (1971)* as *Pselaphorhynchites aureus*.

### *Temnocerus insularis* (Fall, 1929)

Nomenclatural Authority: *Hamilton (1971)*, *Hamilton (2002)*

Literature Records: San Clemente (*Fall, 1929b*: 294; *Hamilton, 1969*: 211), Santa Catalina (*Fall, 1897*: 239; *Fall, 1929b*: 294; *Hamilton, 1969*: 211; *Hamilton, 1971*: 986)

Digitized Records: none

Range: Also known from mainland (*Hamilton, 1969, 1971*).

Notes. *Fall (1897*: 239) reported this species as "*Rhynchites* sp. nov.?"; *Fall (1929b)* recorded this species as *Rhynchites insularis*, and *Hamilton (1969, 1971)* recorded it as *Pselaphorhynchites insularis*. The species was thought to be endemic at the time of its description (*Fall, 1929b*).

### *Temnocerus naso* (Casey, 1885)

Nomenclatural Authority: *Hamilton (1971)*, *Hamilton (2002)*

Literature Records: none

Digitized Records: Santa Cruz (1 UCRC)

Range: Also known from mainland (*Hamilton, 1971*).

Notes. *Hamilton (1971)* recorded this species as *Pselaphorhynchites naso*. This species is distributed through much of mainland southern California (*Hamilton, 1971*).

### Brentidae

Notes. One subfamily, 13 genera, and 35 species of Brentidae have been recorded from California (M. L. Gimmel, 2022, unpublished data).

### Apioninae

Literature Records (subfamily-only): Santa Cruz (*Naughton et al., 2014*: 303)

Digitized Records (subfamily-only): San Clemente (3 SBMNH), San Miguel (1 SBMNH), Santa Cruz (51 SBMNH), Santa Rosa (3 SBMNH)

Notes. This subfamily is fairly diverse in California, with 13 genera and 35 species recorded (*O'Brien & Wibmer, 1982*; M. L. Gimmel, 2022, unpublished data). Although they were monographed for North America by *Kissinger (1968)*, they are exceedingly challenging to identify. Because of this challenge, we have included a "Digitized Records" section above for this subfamily to indicate the amount of material in dire need of expert determination.

***Coelocephalapion* Wagner, 1914**
Nomenclatural Authority: *Anderson & Kissinger (2002)*
Notes. Six species of *Coelocephalapion* have been recorded from California (*O'Brien & Wibmer, 1982*).

***Coelocephalapion antennatum* (Smith, 1884)**
Nomenclatural Authority: *Kissinger (1968)*
Literature Records: Santa Catalina (*Fall, 1897*: 239), Santa Cruz (*Kissinger, 1968*: 249)
Digitized Records: none
Range: Also known from mainland (*Kissinger, 1968*).
Notes. Recorded as *Apion antennatum* by *Fall (1897)* and *Kissinger (1968)*, with the latter work not including the species in a subgenus. A new classification of former *Apion* Herbst, 1797 was developed by *Alonso-Zarazaga (1990)*, but not fully implemented for North American species (*Anderson & Kissinger, 2002*); however, this species has been informally moved to the genus *Coelocephalapion* in, *e.g.*, bugguide.net.

***Coelocephalapion californicum* (Smith, 1884)**
Nomenclatural Authority: *Kissinger (1968)*
Literature Records: Santa Cruz (*Cockerell, 1940*: 287)
Digitized Records: none
Range: Also known from mainland (*Kissinger, 1968*).
Notes. Reported by *Cockerell (1940)* as *Apion californicum. Kissinger (1968)* included this species in the subgenus *Apion* (*Coelocephalapion*), which was upgraded to genus by *Kissinger (1992)*.

***Coelocephalapion oedorhynchum* (LeConte, 1858)**
Nomenclatural Authority: *Kissinger (1968)*
Literature Records: Santa Catalina (*Fall, 1897*: 239; *Fall, 1898a*: 130)
Digitized Records: none
Range: Also known from mainland (*Fall, 1898a*; *Kissinger, 1968*).
Notes. Recorded as *Apion oedorhynchum* by *Fall (1897*, *1898a)* and *Kissinger (1968)*, with the latter work not including the species in a subgenus. A new classification of former *Apion* was developed by *Alonso-Zarazaga (1990)*, but not fully implemented for North American species (*Anderson & Kissinger, 2002*); however, this species has been informally moved to the genus *Coelocephalapion* in, *e.g.*, bugguide.net.

**Curculionidae**
Notes. Nineteen subfamilies, 221 genera, and 821 species of Curculionidae are known to occur in California (M. L. Gimmel, 2022, unpublished data). Subfamilies occurring on the nearby mainland but not known from the Channel Islands include: Bagoinae, Conoderinae, Cryptorhynchinae, Gonipterinae, Mesoptiliinae, Platypodinae.

**Baridinae: Baridini**

Notes. Three tribes, 15 genera, and 48 species of Baridinae, including seven genera and 24 species of Baridini, are known to occur in California (*O'Brien & Wibmer, 1982*; M. L. Gimmel, 2022, unpublished data).

*Trichobaris* **LeConte, 1876**

Nomenclatural Authority: *Anderson (2002)*

Digitized Records (genus-only): Santa Cruz (2 UCSB)

Notes. Three species of *Trichobaris* have been recorded from California (*O'Brien & Wibmer, 1982*).

*Trichobaris compacta* **Casey, 1892**

Nomenclatural Authority: *O'Brien & Wibmer (1982)*

Literature Records: none

Digitized Records: Santa Cruz (5 SBMNH)

Range: Also known from mainland (*O'Brien & Wibmer, 1982*).

**Ceutorhynchinae: Ceutorhynchini**

Notes. Five tribes, 14 genera, and 42 species of Ceutorhynchinae, of which six genera and 31 species belong to Ceutorhynchini, are known to occur in California (*O'Brien & Wibmer, 1982*; M. L. Gimmel, 2022, unpublished data).

*Ceutorhynchus* **Germar, 1824**

Nomenclatural Authority: *Anderson (2002)*

Notes. Twenty-four species of *Ceutorhynchus* are known to occur in California (M. L. Gimmel, 2022, unpublished data).

*Ceutorhynchus assimilis* **(Paykull, 1792)**

Nomenclatural Authority: *O'Brien & Wibmer (1982)*

Literature Records: none

Digitized Records: Santa Cruz (2 SBMNH)

Range: Also known from mainland (*O'Brien & Wibmer, 1982*).

Notes. This species is introduced from Europe (*O'Brien & Wibmer, 1982*).

**Cossoninae**

Notes. Five tribes, 12 genera, and 29 species of Cossoninae are known to occur in California (*O'Brien & Wibmer, 1982*; M. L. Gimmel, 2022, unpublished data).

**Onycholipini**

Notes. Two genera and two species of Onycholipini have been recorded from California (M. L. Gimmel, 2022, unpublished data).

*Pselactus* **Broun, 1886**

Nomenclatural Authority: *Anderson (2002)*

Notes. A single, adventive species of *Pselactus* is known from North America (*Anderson, 2002*).

***Pselactus spadix*** (Herbst, 1795)
Nomenclatural Authority: *O'Brien & Wibmer (1982)*
Literature Records: none
Digitized Records: Santa Rosa (2 SBMNH)
Range: Also known from mainland (*O'Brien & Wibmer, 1982*).
Notes. This beach-dwelling and driftwood-inhabiting species is introduced to North America (*Anderson, 2002*).

**Rhyncolini**
Notes. Five genera and 14 species of Rhyncolini are known to occur in California (M. L. Gimmel, 2022, unpublished data).

***Elassoptes*** **Horn, 1873**
Nomenclatural Authority: *Anderson (2002)*
Notes. A single species of *Elassoptes* is known from North America (*Anderson, 2002*).

***Elassoptes marinus*** **Horn, 1873**
Nomenclatural Authority: *O'Brien & Wibmer (1982)*
Literature Records: none
Digitized Records: San Clemente (2 SBMNH), San Miguel (26 SBMNH), San Nicolas (5 SBMNH), Santa Cruz (16 SBMNH), Santa Rosa (21 SBMNH)
Range: Also known from mainland (*O'Brien & Wibmer, 1982*).
Notes. This species is a beach inhabitant associated with driftwood (*Anderson, 2002*).

***Rhyncolus*** **Germar, 1817**
Nomenclatural Authority: *Anderson (2002)*
Literature Records (genus-only): Santa Barbara (*Miller & Miller, 1985*: 131)
Digitized Records (genus-only): San Miguel (4 SBMNH)
Notes. *Miller & Miller (1985)* recorded the genus only, noting that it needs revision. Nine nominal species of *Rhyncolus* are known to occur in California (M. L. Gimmel, 2022, unpublished data).

***Rhyncolus cylindricollis*** **Wollaston, 1873**
Nomenclatural Authority: *O'Brien & Wibmer (1982)*
Literature Records: none
Digitized Records: Santa Cruz (6 SBMNH)
Range: Also known from mainland (*O'Brien & Wibmer, 1982*).

**Curculioninae**
Notes. Nine tribes, 23 genera, and 131 species of Curculioninae are known to occur in California (M. L. Gimmel, 2022, unpublished data).

**Anthonomini**
Notes. Six genera and 60 species of Anthonomini are known to occur in California (M. L. Gimmel, 2022, unpublished data).

### Anthonomus Germar, 1817

Nomenclatural Authority: *Anderson (2002)*

Digitized Records (genus-only): Anacapa (2 LACM; 2 SBMNH), Santa Catalina (5 SBMNH)

Notes. Forty-five species of *Anthonomus* belonging to five subgenera (*Anthomorphus* Weise, 1883, *Anthonomochaeta* Dietz, 1891, *Anthonomorphus* Dietz, 1891, *Anthonomus* (*s.str.*), and *Cnemocyllus* Dietz, 1891) are known from California (M. L. Gimmel, 2022, unpublished data). The subgenus *Cnemocyllus* was revised by *Clark & Burke (2005)*. All SBMNH genus-only records belong to the subgenus *Cnemocyllus*.

### Anthonomus (Anthonomus) pauperculus LeConte, 1876

Nomenclatural Authority: *Clark et al. (2019)*

Literature Records: Santa Catalina (*Seavey, 1892*: 262; *Fall, 1897*: 239; *Cockerell, 1940*: 287; *Clark et al., 2019*: 796)

Digitized Records: none

Range: Also known from mainland (*Clark et al., 2019*).

Notes. The record of *Anthonomus canus* LeConte, 1876 (now a junior synonym of *Anthonomus* (*Cnemocyllus*) *decipiens* LeConte, 1876) by *Seavey (1892)* was considered to be erroneous according to *Fall (1897*: 235) and refers to *A. pauperculus* (see *Cockerell, 1940*: 287). Members of the *Anthonomus squamosus* LeConte, 1876 species-group, to which *A. pauperculus* belongs, were revised by *Clark et al. (2019)*.

### Anthonomus (Cnemocyllus) inermis Boheman, 1859

Nomenclatural Authority: *Clark & Burke (2005)*

Literature Records: San Miguel (*Miller & Miller, 1985*: 131), Santa Barbara (*Miller & Miller, 1985*: 131), Santa Rosa (*Clark & Burke, 2005*: 45)

Digitized Records: Santa Barbara (1 LACM)

Range: Also known from mainland (*Clark & Burke, 2005*).

Notes. Recorded as *Anthonomus subvittatus* LeConte, 1876 by *Miller & Miller (1985)*, which is a junior synonym of *A. inermis* (see *Clark & Burke, 2005*). *Miller & Miller (1985)* reported it from *Hemizonia clementina* (Asteraceae) on Santa Barbara Island.

### Anthonomus (Cnemocyllus) undescribed species

Literature Records: none

Digitized Records: Santa Barbara (2 SBMNH)

Range: ?Endemic.

Notes. These seven SBMNH specimens (on two pins) from Santa Barbara Island, one of which is a dissected male, were marked as "*Anthonomus* n. sp. #2, *Cnemocyllus* gp." by Horace R. Burke in 2009. Based on the funicle with seven antennomeres, it belongs to the *A. inermis* group of *Clark & Burke (2005)* (M. L. Gimmel, 2021, personal observation). It is unknown whether specimens of this morphospecies exist from other localities.

## Curculionini

Notes. One genus and three species of Curculionini have been recorded from California (*Gibson, 1969*).

### *Curculio* Linnaeus, 1758

Nomenclatural Authority: *Anderson (2002)*
Literature Records (genus-only): Santa Cruz (*Naughton et al., 2014*: 303)
Digitized Records (genus-only): Santa Catalina (1 SBMNH), Santa Cruz (1 SBMNH)
Notes. Three species of *Curculio* have been recorded from California (*Gibson, 1969*). The *Naughton et al. (2014)* record refers to a single specimen of "*Curculio* sp." in addition to records of *C. uniformis*. *Seavey (1892*: 262) recorded *Balaninus obtusus* [=*Curculio obtusus* (Blanchard, 1884)] from Santa Catalina Island. *Gibson (1969)* used an unnecessary replacement name, *Curculio neocorylus* Gibson, 1969 to refer to this species. *Fall (1897*: 235) doubted the validity of this identification; we agree that this record must be in error, since this species has not subsequently been reported from California and is otherwise only known from east of the Rocky Mountains (*Gibson, 1969*). This record almost certainly represents one of the two species of *Curcuilo* listed below. The North American species of *Curculio* were revised by *Gibson (1969)*.

### *Curculio aurivestis* Chittenden, 1927

Nomenclatural Authority: *O'Brien & Wibmer (1982)*
Literature Records: Santa Catalina (*Chittenden, 1927*: 186, 191)
Digitized Records: Santa Catalina (1 USNM)
Range: Also known from mainland (*Gibson, 1969*).
Notes. *Chittenden (1927)* recognized two species, one from the mainland (*C. aurivestis*) and one insular and presumed endemic (*Curculio brevinasus* Chittenden, 1927), the latter representing the Santa Catalina record above. *Gibson (1969)* synonymized the two and recognized a single species ranging from British Columbia to southern California where it breeds in various species of *Quercus*.

### *Curculio uniformis* (LeConte, 1857)

Nomenclatural Authority: *O'Brien & Wibmer (1982)*
Literature Records: Santa Catalina (*Fall, 1897*: 239; *Fall, 1901*: 199; *Chittenden, 1908*: 22), Santa Cruz (*Naughton et al., 2014*: 303)
Digitized Records: Santa Cruz (4 SBMNH)
Range: Also known from mainland (*Fall, 1901*; *Chittenden, 1908*; *Gibson, 1969*).
Notes. *Fall (1897)* recorded this species as *Balaninus occidentis* Casey, 1897 and noted this species had previously been confused with *Balaninus uniformis*, but is distinct. *Fall (1901)* recorded this species as *B. uniformis*. *Chittenden (1908)* synonymized *B. occidentis* with *B. uniformis*, and *Gibson (1969)* incorrectly used *C. occidentis* as the valid name.

## Smicronychini

Notes. Two genera and 21 species of Smicronychini have been recorded from California (*O'Brien & Anderson, 1996*).

*Smicronyx* Schoenherr, 1843
Nomenclatural Authority: *Anderson (2002)*
Literature Records (genus-only): Santa Rosa (*Fall, 1897*: 239)
Digitized Records (genus-only): San Clemente (1 SBMNH)
Notes. Nineteen species of *Smicronyx* have been recorded from California (*O'Brien & Anderson, 1996*). *Fall (1897)* recorded only "*Smicronyx*, sp." from Santa Rosa Island. *Anderson (1962)* revised the species for North America.

*Smicronyx cinereus* (Motschulsky, 1845)
Nomenclatural Authority: *O'Brien & Anderson (1996)*
Literature Records: Santa Rosa (*Anderson, 1962*: 208)
Digitized Records: none
Range: Also known from mainland (*Anderson, 1962*).
Notes. The record from *Fall (1897)* may refer to this species, but because there is at least one new island record for the genus (San Clemente) the genus should be reexamined from the Channel Islands.

**Tychiini**
Notes. Three genera and 14 species of Tychiini have been recorded from California (*O'Brien & Wibmer, 1982*).

*Sibinia* Germar, 1817
Nomenclatural Authority: *Anderson (2002)*
Notes. Five species of *Sibinia* are known to occur in California (*O'Brien & Wibmer, 1982*). This genus was revised for the New World by *Clark (1978)*.

*Sibinia maculata* (LeConte, 1876)
Nomenclatural Authority: *O'Brien & Wibmer (1982)*
Literature Records: San Miguel (*Clark, 1978*: 363; *Miller & Miller, 1985*: 131), San Nicolas (*Fall, 1901*: 197), Santa Barbara (*Miller & Miller, 1985*: 131)
Digitized Records: Santa Barbara (1 LACM)
Range: Also known from mainland (*Fall, 1901*; *Clark, 1978*).
Notes. *Fall (1901)* recorded this species as *Paragoges maculatus* LeConte. Reported from "sage brush" on Santa Barbara Island by *Miller & Miller (1985)*.

*Tychius* Germar, 1817
Nomenclatural Authority: *Anderson (2002)*
Literature Records (genus-only): San Nicolas (*Fall, 1897*: 239)
Notes. Six species of *Tychius* have been recorded from California (*O'Brien & Wibmer, 1982*). The genus was revised for North America by *Clark (1971)*. The status of the San Nicolas Island record referred to by *Fall (1897)* as "*Tychius*, n. sp." is unknown.

*Tychius lineellus* LeConte, 1876
Nomenclatural Authority: *O'Brien & Wibmer (1982)*
Literature Records: Santa Cruz (*Clark, 1971*: 18), Santa Rosa (*Clark, 1971*: 18)

Digitized Records: Santa Cruz (4 LACM), Santa Rosa (5 LACM; 2 SBMNH)
Range: Also known from mainland (*Clark, 1971*).

### Cyclominae
Notes. Two genera and 17 species of Cyclominae have been recorded from California (*O'Brien, 1997*).

### *Listroderes* Schoenherr, 1826
Nomenclatural Authority: *Anderson (2002)*
Digitized Records (genus-only): Santa Cruz (1 UCSB)
Notes. Two introduced species of *Listroderes* have been recorded from California (*O'Brien, 1997*). This genus was partially revised by *Morrone (1993)*.

### *Listroderes costirostris* Schoenherr, 1826
Nomenclatural Authority: *Morrone (1993)*
Literature Records: none
Digitized Records: San Miguel (14 SBMNH), San Nicolas (11 LACM), Santa Catalina (1 LACM), Santa Rosa (1 LACM)
Range: Also known from mainland (*Morrone, 1993*).
Notes. This species was accidentally introduced to North America from South America (*Morrone, 1993*).

### *Listronotus* Jekel, 1865
Nomenclatural Authority: *Anderson (2002)*
Digitized Records (genus-only): Santa Catalina (3 SBMNH)
Notes. Fifteen species of *Listronotus* have been recorded from California (*O'Brien, 1997*). The larger species of this genus in North America were taxonomically treated by *O'Brien (1981)*.

### *Listronotus sordidus* (Gyllenhal, 1834)
Nomenclatural Authority: *O'Brien & Wibmer (1982)*
Literature Records: San Nicolas (*Cockerell, 1940*: 287)
Digitized Records: none
Range: Also known from mainland (*O'Brien & Wibmer, 1982*).
Notes. This species was reported by *Cockerell (1940)* as *Listronotus obliquus* LeConte, 1876. This species is now considered a junior synonym of *L. sordidus* (see *O'Brien & Wibmer, 1982*: 70), but this species' reported range does not include California (*O'Brien & Wibmer, 1982*). The specific identity of the San Nicolas Island record is therefore still in question.

### Dryophthorinae: Rhynchophorini
Notes. Two tribes, seven genera, and 27 species of Dryophthorinae, of which all but one genus and species belong to Rhynchophorini, have been recorded from California (M. L. Gimmel, 2022, unpublished data).

### *Scyphophorus* Schoenherr, 1838
Nomenclatural Authority: *Anderson (2002)*

Notes. Two species of *Scyphophorus* have been recorded from California (*Vaurie, 1971*). The species of the genus were treated by *Vaurie (1971)*.

### *Scyphophorus yuccae* Horn, 1873
Nomenclatural Authority: *Vaurie (1971)*, *O'Brien & Wibmer (1982)*
Literature Records: none
Digitized Records: Santa Cruz (2 SBMNH)
Range: Also known from mainland (*Vaurie, 1971*; *O'Brien & Wibmer, 1982*).
Notes. This species breeds in the stems of *Hesperoyucca whipplei* (Torr.) Trel. (Agavaceae), a common plant on the mainland that does not occur natively in the Channel Islands. However, a variety of yucca species have been planted there.

### *Sphenophorus* Schoenherr, 1838
Nomenclatural Authority: *Anderson (2002)*
Notes. Eighteen species of *Sphenophorus* have been recorded from California (M. L. Gimmel, 2022, unpublished data). This genus was treated for North America by *Vaurie (1951)* as the genus *Calendra* Clairville & Schellenberg, 1798.

### *Sphenophorus graminis* Chittenden, 1905
Nomenclatural Authority: *O'Brien & Wibmer (1982)*
Literature Records: none
Digitized Records: Santa Rosa (1 SBMNH)
Range: Also known from mainland (*Vaurie, 1951*; *O'Brien & Wibmer, 1982*).

### *Sphenophorus phoeniciensis* Chittenden, 1904
Nomenclatural Authority: *O'Brien & Wibmer (1982)*
Literature Records: none
Digitized Records: Santa Cruz (1 SBMNH)
Range: Also known from mainland (*Vaurie, 1951*; *O'Brien & Wibmer, 1982*).

### *Sphenophorus simplex* LeConte, 1860
Nomenclatural Authority: *O'Brien & Wibmer (1982)*
Literature Records: none
Digitized Records: San Nicolas (1 SBMNH), Santa Rosa (1 SBMNH)
Range: Also known from mainland (*Vaurie, 1951*; *O'Brien & Wibmer, 1982*).

### *Sphenophorus vomerinus* LeConte, 1858
Nomenclatural Authority: *O'Brien & Wibmer (1982)*
Literature Records: Santa Rosa (*Fall, 1897*: 239)
Digitized Records: none
Range: Also known from mainland (*Vaurie, 1951*; *O'Brien & Wibmer, 1982*).

### Entiminae
Notes. Fifteen tribes, 53 genera, and 205 species of Entiminae are known to occur in California (M. L. Gimmel, 2022, unpublished data).

**Geonemini**

Notes. Four genera and 64 species of Geonemini are known to occur in California (*O'Brien & Wibmer, 1982*).

***Trigonoscuta* Motschulsky, 1853**

Nomenclatural Authority: *Anderson (2002)*

Literature Records (genus-only): San Clemente (*Doyen, 1974*: 87), San Miguel (*Van Dam & Matzke, 2016*: 1527), San Nicolas (*Van Dam & Matzke, 2016*: 1527), Santa Catalina (*Van Dam & Matzke, 2016*: 1527), Santa Cruz (*Van Dam & Matzke, 2016*: 1527), Santa Rosa (*Van Dam & Matzke, 2016*: 1527)

Digitized Records (genus-only): San Clemente (2 LACM; 7 SBMNH), San Miguel (14 SBMNH), San Nicolas (32 LACM; 24 SBMNH), Santa Catalina (9 LACM), Santa Cruz (5 SBMNH), Santa Rosa (25 SBMNH)

Notes. This genus was revised in elaborate detail by *Pierce (1975)*, who diagnosed and named minute variation in these sand-dwelling beetles, recognizing over 150 species and subspecies, most of which were limited to a single locality or even collecting event. *Miller (1985a*: 21), in a list of endemic species of the Channel Islands, did not bother to individually list all of the purported endemic *Trigonoscuta* described by *Pierce (1975)*. A robust molecular phylogeny was published by *Van Dam & Matzke (2016)* which sampled specimens from five of the Channel Islands. This phylogeny seems to indicate that at least two species are present on the islands and each of those are found on multiple islands. This contradicts *Pierce (1975)* who enumerated different species or subspecies for nearly every beach locality collected on the islands and mainland California. It seems likely that coastal dune *Trigonoscuta* may have similar diversity to the tenebrionid genus *Coelus* which has a nearly identical distribution and habitat (see account above for this genus). It actually may be the case that the two "subgenera" identified by Pierce as inhabiting the islands align more appropriately with true species diversity than the numerous "species" described. This genus is in great need of a modern revision and likely tells a very interesting story of dune colonization between the islands and mainland California. *Van Dam & Matzke (2016*, see their fig. 5 and supplemental Information) found two well supported clades which one might consider species: one from the northern islands of Santa Rosa, Santa Cruz, and San Miguel sister to a specimen from the mainland, and a second from Santa Catalina and San Nicolas islands. Specimens of this genus are known from all eight Channel Islands.

The records presented under each species largely reflect the work of *Pierce (1975)* and the specimens included within that study.

***Trigonoscuta anacapensis* Pierce, 1975**

Nomenclatural Authority: *O'Brien & Wibmer (1982)*

Literature Records: Anacapa (*Pierce, 1975*: 48)

Digitized Records: Anacapa (18 LACM)

Range: Endemic (*Pierce, 1975*).

Notes. Described in the subgenus *Trigonoscuta* (*s.str.*), which *Pierce (1975)* understood as otherwise restricted to the mainland Pacific Coast. This species was described from West

Anacapa Island and considered similar to populations described from the coast of Ventura County, California (*Pierce, 1975*: 49).

### *Trigonoscuta catalina* Pierce, 1975
Nomenclatural Authority: *O'Brien & Wibmer (1982)*
Literature Records: Santa Catalina (*Pierce, 1975*: 53)
Digitized Records: Santa Catalina (24 LACM)
Range: Endemic (*Pierce, 1975*).
Notes. Described in the subgenus *Nesocatoecus* Pierce, 1975, which *Pierce (1975)* understood as restricted to the Channel Islands.

### *Trigonoscuta clemente* Pierce, 1975
Nomenclatural Authority: *O'Brien & Wibmer (1982)*
Literature Records: San Clemente (*Pierce, 1975*: 53)
Digitized Records: San Clemente (2031 LACM), Santa Barbara (2 LACM)
Range: Endemic (*Pierce, 1975*).
Notes. The correct spelling of this species is *T. clemente* (*Pierce, 1975*), though in the same paper the incorrect original spelling of *T. sanclemente* was also used (nec. *T. sanclementis Pierce, 1975*; see *O'Brien & Wibmer, 1982*: 33). Described in the subgenus *Nesocatoecus*, which *Pierce (1975)* understood as restricted to the Channel Islands. *Pierce (1975)* described five subspecies largely based off of dune host plants the beetles were collected from, which seem highly unlikely to represent different taxa in an evolutionary context. The subspecies are: *T. c. clemente* Pierce, 1975; *T. c. isola* Pierce, 1975; *T. c. excavata* Pierce, 1975; *T c. latesecula* Pierce, 1975; *T. c. traskiae* Pierce, 1975.

### *Trigonoscuta curviscroba* Pierce, 1975
Nomenclatural Authority: *O'Brien & Wibmer (1982)*
Literature Records: Santa Barbara (*Pierce, 1975*: 53; *Miller & Miller, 1985*: 131)
Digitized Records: none
Range: Endemic (*Pierce, 1975*).
Notes. Described in the subgenus *Nesocatoecus*, which *Pierce (1975)* understood as restricted to the Channel Islands. *Miller & Miller (1985)* merely listed *T. curviscroba* as being described from Santa Barbara Island, and doubted the taxonomic validity of this species.

### *Trigonoscuta miguelensis* Pierce, 1975
Nomenclatural Authority: *O'Brien & Wibmer (1982)*
Literature Records: San Miguel (*Pierce, 1975*: 46)
Digitized Records: San Miguel (21 LACM; 10 SBMNH)
Range: Endemic (*Pierce, 1975*).
Notes. Described in the subgenus *Nesocatoecus*, which *Pierce (1975)* understood as restricted to the Channel Islands.

### *Trigonoscuta nesiotis* Pierce, 1975
Nomenclatural Authority: *O'Brien & Wibmer (1982)*

Literature Records: Anacapa (*Pierce, 1975*: 48)
Digitized Records: Anacapa (35 LACM)
Range: Endemic (*Pierce, 1975*).
Notes. Described in the subgenus *Nesocatoecus*, which *Pierce (1975)* understood as restricted to the Channel Islands. This species was described from West Anacapa Island.

### *Trigonoscuta nicolana* Pierce, 1975
Nomenclatural Authority: *O'Brien & Wibmer (1982)*
Literature Records: San Nicolas (*Pierce, 1975*: 49)
Digitized Records: San Nicolas (264 LACM), Santa Barbara (2 LACM)
Range: Endemic (*Pierce, 1975*).
Notes. Described in the subgenus *Nesocatoecus*, which *Pierce (1975)* understood as restricted to the Channel Islands. Eight subspecies were described by *Pierce (1975)*, which corresponded to the dune host plant the beetles were found associated with. It seems highly unlikely that these names represent different taxa in an evolutionary context.
The subspecies are: *T. n. nicolana* Pierce, 1975; *T. n. longinoda* Pierce, 1975; *T. n. latelobata* Pierce, 1975; *T. n. nonmarginata* Pierce, 1975; *T. n. latespiculum* Pierce, 1975; *T. n. lateconjuncta* Pierce, 1975; *T. n. sulcata* Pierce, 1975; *T. n. breviconjuncta* Pierce, 1975.

### *Trigonoscuta pilosa* Motschulsky, 1953
Nomenclatural Authority: *O'Brien & Wibmer (1982)*
Literature Records: San Clemente (*Fall, 1897*: 239), Santa Rosa (*Fall, 1897*: 239)
Digitized Records: none
Range: Also known from mainland (*Pierce, 1975*).
Notes. In the genus revision by *Pierce (1975)*, this species was interpreted to occur from Washington, Oregon, and California north of the San Andreas fault, from which were described eight subspecies. The literature records for this taxon predate Pierce's revision and likely correspond to other taxa named therein. This species belongs to the subgenus *Trigonoscuta* (*s.str.*), which *Pierce (1975)* recognized as restricted to the mainland except for *T. anacapensis* from Anacapa island.

### *Trigonoscuta sanctabarbarae* Pierce, 1975
Nomenclatural Authority: *O'Brien & Wibmer (1982)*
Literature Records: Santa Barbara (*Pierce, 1975*: 52; *Miller & Miller, 1985*: 131).
Digitized Records: Santa Barbara (46 LACM)
Range: Endemic (*Pierce, 1975*).
Notes. The correct spelling of this name is *T. sanctabarbarae*, but in the same paper (*Pierce, 1975*) the incorrect original spelling *T. santabarbarae* was also used (see *O'Brien & Wibmer, 1982*: 35). Described in the subgenus *Nesocatoecus*, which *Pierce (1975)* understood as restricted to the Channel Islands. *Pierce (1975)* included three subspecies, all from Santa Barbara Island: *T. s. sanctabarbarae* Pierce, 1975; *T. s. mesembryanthemi* Pierce, 1975; *T. s. lycii* Pierce, 1975. *Miller & Miller (1985)* merely listed these three subspecies (as "*Trigonoscuta santabarbarae*") as being described from Santa Barbara Island. They doubted the taxonomic validity of these taxa.
***Trigonoscuta sanctarosae* Pierce, 1975**
Nomenclatural Authority: *O'Brien & Wibmer (1982)*
Literature Records: Santa Rosa (*Pierce, 1975*: 47, 48)
Digitized Records: Santa Rosa (138 LACM)
Range: Endemic (*Pierce, 1975*).
Notes. Described in the subgenus *Nesocatoecus*, which *Pierce (1975)* understood as restricted to the Channel Islands. *Pierce (1975*: 47–48) described two subspecies, both from Santa Rosa Island: *T. s. sanctarosae* Pierce, 1975 and *T. s. astragalensis* Pierce, 1975.

***Trigonoscuta stantoni* Sleeper, 1975**
Nomenclatural Authority: *O'Brien & Wibmer (1982)*
Literature Records: Santa Cruz (*Pierce, 1975*: 77)
Digitized Records: Santa Cruz (35 SBMNH; 1 iNat)
Range: Endemic (*Pierce, 1975*).
Notes. Described in the subgenus *Nesocatoecus*, which *Pierce (1975)* understood as restricted to the Channel Islands. This species was described by the editor of the revision (*Pierce, 1975*) to fill in an island gap that Pierce had postulated should have a species present.

**Naupactini**
Notes. Four genera and seven species of Naupactini have been recorded from California (M. L. Gimmel, 2022, unpublished data).

***Naupactus* Dejean, 1821**
Nomenclatural Authority: *Anderson (2002)*
Notes. One species of *Naupactus* is known from California (M. L. Gimmel, 2022, unpublished data).

***Naupactus cervinus* Boheman, 1840**
Nomenclatural Authority: *O'Brien & Wibmer (1982)*
Literature Records: Santa Catalina (*Cockerell, 1940*: 287)
Digitized Records: Santa Catalina (1 LACM; 1 SBMNH; 1 iNat), Santa Cruz (89 LACM)
Range: Also known from mainland (*O'Brien & Wibmer, 1982*).
Notes. Recorded as *Pantomorus fulleri* (Horn) by *Cockerell (1940)*, and often referred to as *Pantomorus cervinus* in the literature. This species is introduced in North America (*O'Brien & Wibmer, 1982*).

**Otiorhynchini**
Notes. Three genera and 13 species of Otiorhynchini have been recorded from California (*O'Brien & Wibmer, 1982*).

***Otiorhynchus* Germar, 1822**
Nomenclatural Authority: *Anderson (2002)*

Notes. Five species of *Otiorhynchus* have been recorded from California (*O'Brien & Wibmer, 1982*). A key and distributional summary of the genus in North America was provided by *Warner & Negley (1976)*.

### *Otiorhynchus cribricollis* Gyllenhal, 1834
Nomenclatural Authority: *O'Brien & Wibmer (1982)*
Literature Records: none
Digitized Records: San Nicolas (1 SBMNH)
Range: Also known from mainland (*Warner & Negley, 1976*; *O'Brien & Wibmer, 1982*).
Notes. This species is adventive in North America from Europe (*Warner & Negley, 1976*).

### *Sciopithes* Horn, 1876
Nomenclatural Authority: *Anderson (2002)*
Notes. Six species of *Sciopithes* have been recorded from California (*O'Brien & Wibmer, 1982*).

### *Sciopithes insularis* Van Dyke, 1935
Nomenclatural Authority: *O'Brien & Wibmer (1982)*
Literature Records: San Clemente (*Van Dyke, 1935*: 91; *Miller, 1985a*: 21; *Miller & Miller, 1985*: 131)
Digitized Records: none
Range: Endemic (*Van Dyke, 1935*; *Miller, 1985a*).

### *Sciopithes setosus* Casey, 1888
Nomenclatural Authority: *O'Brien & Wibmer (1982)*
Literature Records: San Clemente (*Fall, 1897*: 239; *Fall, 1901*: 188), Santa Barbara (*Miller & Miller, 1985*: 131)
Digitized Records: none
Range: Also known from mainland (*Fall, 1901*).
Notes. *Fall (1897)* recorded this species as "var.". *Miller & Miller (1985)* reported this species from *Coreopsis gigantea* on Santa Barbara Island.

### Peritelini
Notes. Sixteen genera and 37 species of Peritelini have been recorded from California (*O'Brien & Wibmer, 1982*).

### *Geodercodes* Casey, 1888
Nomenclatural Authority: *Anderson (2002)*
Notes. One species of *Geodercodes* occurs in North America (*Anderson, 2002*).

### *Geodercodes latipennis* Casey, 1888
Nomenclatural Authority: *O'Brien & Wibmer (1982)*, *Anderson (2002)*
Literature Records: Santa Cruz (*Polihronakis, Caterino & Chatzimanolis, 2010*: 940; *Naughton et al., 2014*: 303), Santa Rosa (*Polihronakis, Caterino & Chatzimanolis, 2010*: 940)

Digitized Records: San Clemente (8 SBMNH), Santa Cruz (11 SBMNH), Santa Rosa (16 SBMNH)

Range: Also known from mainland (*O'Brien & Wibmer, 1982*; *Anderson, 2002*; *Polihronakis, Caterino & Chatzimanolis, 2010*).

Notes. *Polihronakis, Caterino & Chatzimanolis (2010)* showed that this species is made up of sexual and asexual populations across the Coast and Transverse ranges of California; the populations on the northern Channel Islands are asexual.

### *Nemocestes* Van Dyke, 1936
Nomenclatural Authority: *Anderson (2002)*

Notes. This genus contains nine species in California (*O'Brien & Wibmer, 1982*, as *Geoderces* Horn, 1876; M. L. Gimmel, 2022, unpublished data). Eight of these species were keyed by *Van Dyke (1936)*.

### *Nemocestes* undetermined species
Literature Records: Santa Cruz (*Naughton et al., 2014*: 303)

Digitized Records: Santa Catalina (8 SBMNH), Santa Cruz (6 SBMNH), Santa Rosa (3 SBMNH)

### *Peritelinus* Casey, 1888
Nomenclatural Authority: *Anderson (2002)*

Notes. This genus contains two species in California (*O'Brien & Wibmer, 1982*). These were distinguished by *Van Dyke (1936)*.

### *Peritelinus* undetermined species
Literature Records: none

Digitized Records: Anacapa (6 SBMNH)

### *Stenoptochus* Casey, 1888
Nomenclatural Authority: *O'Brien & Wibmer (1982)*; *Anderson (2002)*

Notes. This genus contains two species in California (*O'Brien & Wibmer, 1982*; *Anderson, 2002*). These were distinguished by *Van Dyke (1935)*.

### *Stenoptochus* undetermined species
Literature Records: none

Digitized Records: Santa Cruz (1 SBMNH)

### Sitonini
Notes. One genus and nine species of Sitonini have been recorded from California (*Bright, 1994*).

### *Sitona* Germar, 1817
Nomenclatural Authority: *Bright (1994)*

Notes. This genus contains nine described species in California (*Bright, 1994*). These were revised for North America by *Bright (1994)*.

***Sitona californius* Fåhraeus, 1840**
Nomenclatural Authority: *Bright (1994)*
Literature Records: San Miguel (*Blaisdell, 1938*: 33; *Miller, 1985a*: 21; *Bright, 1994*: 294 [map]), Santa Cruz (*Bright, 1994*: 294 [map])
Digitized Records: Anacapa (16 LACM), San Miguel (10 LACM; 1 SBMNH), Santa Catalina (2 SBMNH), Santa Cruz (5 SBMNH), Santa Rosa (1 SBMNH)
Range: Also known from mainland (*Bright, 1994*).
Notes. This species was described and recorded by *Blaisdell (1938)* as the purported endemic *Sitona cockerelli* Blaisdell, 1938, listed as such by *Miller (1985a*: 21). *Bright (1989*: 77), however, synonymized this species with *S. californicus*, later recognized with the spelling *S. californius*.

**Erirhininae**
Notes. Six genera and 10 species of Erirhininae have been recorded from California (*O'Brien & Anderson, 1996*; M. L. Gimmel, 2022, unpublished data).

***Notiodes* Schoenherr, 1838**
Nomenclatural Authority: *Anderson (2002)*
Notes. Five species of *Notiodes* have been recorded from California (*O'Brien & Anderson, 1996*).

***Notiodes aeratus* (LeConte, 1876)**
Nomenclatural Authority: *O'Brien & Anderson (1996)*
Literature Records: none
Digitized Records: San Clemente (1 SBMNH)
Range: Also known from mainland (*O'Brien & Anderson, 1996*).

**Hyperinae**
Notes. Two genera and five species of Hyperinae have been recorded from California (*O'Brien & Wibmer, 1982*; M. L. Gimmel, 2022, unpublished data).

***Hypera* Germar, 1817**
Nomenclatural Authority: *Anderson (2002)*
Notes. Four introduced species of *Hypera* have been recorded from California (*O'Brien & Wibmer, 1982*).

***Hypera postica* (Gyllenhal, 1813)**
Nomenclatural Authority: *O'Brien & Wibmer (1982)*
Literature Records: none
Digitized Records: San Clemente (2 SBMNH), San Nicolas (1 SBMNH), Santa Cruz (16 SBMNH), Santa Rosa (1 SBMNH)
Range: Also known from mainland (*O'Brien & Wibmer, 1982*).
Notes. This species is introduced in North America (*O'Brien & Wibmer, 1982*).

**Lixinae**

Notes. Three tribes, seven genera, and 30 species of Lixinae are known to occur in California (M. L. Gimmel, 2022, unpublished data).

**Cleonini**

Notes. Two genera and 17 species of Cleonini have been recorded from California (*Anderson, 1988*). The species were revised for the New World by *Anderson (1988)*.

***Apleurus* Chevrolat, 1873**

Nomenclatural Authority: *Anderson (2002)*

Notes. Six species of *Apleurus* are known to occur in California (*Anderson, 1988*).

***Apleurus* (*Apleurus*) *jacobinus* (Casey, 1891)**

Nomenclatural Authority: *Anderson (1988)*

Literature Records: San Miguel (*Anderson, 1988*: 646 [map])

Digitized Records: San Miguel (1 SBMNH)

Range: Also known from mainland (*Anderson, 1988*).

***Scaphomorphus* Motschulsky, 1860**

Nomenclatural Authority: *Anderson (2002)*

Notes. This genus contains 12 species in California (*Anderson, 1988*).

***Scaphomorphus americanus* (Csiki, 1934)**

Nomenclatural Authority: *Anderson (1988)*, *Alonso-Zarazaga & Lyal (1999)*, *Anderson (2002)*

Literature Records: San Clemente (*Fall, 1897*: 239; *Anderson, 1988*: 649 [map])

Digitized Records: none

Range: Also recorded from mainland (*Anderson, 1988*).

Notes. This species was recorded by *Fall (1897)* as *Cleonus basalis* Fall, 1897, which is a junior homonym that was replaced by *Csiki (1934)* with *Cleonus* (*Cleonidius*) *americanus* Csiki, 1934. *O'Brien & Wibmer (1982)* corrected the name to *Cleonis americanus* (Csiki, 1934); later, *Anderson (1988)* moved the species to *Cleonidius* Casey, 1891. *Cleonidius* was more recently synonymized under the resurrected *Scaphomorphus* in *Alonso-Zarazaga & Lyal (1999*: 192). The species was considered endemic to San Clemente Island at the time of its original description (*Fall, 1897*).

**Rhinocyllini**

Notes. Two genera and three species of Rhinocyllini are known to occur in California (M. L. Gimmel, 2022, unpublished data).

***Rhinocyllus* Germar, 1817**

Nomenclatural Authority: *Anderson (2002)*

Notes. This genus contains a single species in California (*O'Brien & Wibmer, 1982*).

***Rhinocyllus conicus* (Froelich, 1792)**

Nomenclatural Authority: *Anderson (2002)*

Literature Records: Santa Cruz (*Hawkes, Andres & Dunn, 1972*: 14; *Goeden & Ricker, 1974*: 9; *Goeden & Ricker, 1977*: 289), Santa Rosa (*Miller, 1996*: 197)

Digitized Records: Santa Cruz (1 iNat)

Range: Also known from mainland (*O'Brien & Wibmer, 1982*; *Anderson, 2002*).

Notes. This species was introduced into North America for biological control of *Carduus nutans* L. (Asteraceae) (*Goeden & Ricker, 1974*, *1977*; *Anderson, 2002*). Although *Hawkes, Andres & Dunn (1972)* and *Goeden & Ricker (1974*, *1977)* reported that the 1971 Santa Cruz Island introductions were unsuccessful, the 2017 iNaturalist record above provides evidence that the species may be established there. *Miller (1996)* reported on an attempted 1979 introduction to Santa Rosa Island, the results of which are unknown.

**Molytinae**

Notes. Eight tribes, 13 genera, and 24 species of Molytinae have been recorded from California (M. L. Gimmel, 2022, unpublished data).

**Conotrachelini**

Notes. Two genera and three species of Conotrachelini have been recorded from California (*O'Brien & Wibmer, 1982*).

***Micromastus* LeConte, 1876**

Nomenclatural Authority: *Anderson (2002)*

Notes. This genus contains a single species in California (*O'Brien & Wibmer, 1982*; *Anderson, 2002*).

***Micromastus gracilis* (Boheman, 1859)**

Nomenclatural Authority: *Anderson (2002)*

Literature Records: none

Digitized Records: Santa Catalina (12 SBMNH)

Range: Also known from mainland (*O'Brien & Wibmer, 1982*).

**Emphyastini**

Notes. Two genera and two species of Emphyastini have been recorded from California (*O'Brien, 1997*).

***Emphyastes* Mannerheim, 1852**

Nomenclatural Authority: *Anderson (2002)*

Notes. This genus contains a single species in California (*O'Brien & Wibmer, 1982*; *Anderson, 2002*). It was treated in the subfamily Cyclominae by *Anderson (2002)*.

***Emphyastes fucicola* Mannerheim, 1852**

Nomenclatural Authority: *Anderson (2002)*

Literature Records: none

Digitized Records: San Clemente (7 SBMNH), San Nicolas (14 LACM; 5 SBMNH), Santa Cruz (2 SBMNH), Santa Rosa (1 iNat)

Range: Also known from mainland (*O'Brien & Wibmer, 1982*; *Anderson, 2002*).

Notes. Adults and larvae are associated with buried, decaying seaweed on sandy beaches (*Anderson, 2002*).

### *Thalasselephas* Egorov & Korotyaev, 1976
Nomenclatural Authority: *Anderson (2002)*
Notes. This genus contains a single species in California (*Anderson, 2002*). The genus was renamed from *Phycocoetes* LeConte, 1876 as *Neophycocoetes* O'Brien & Wibmer, 1982 in *O'Brien & Wibmer (1982)*.

### *Thalasselephas testaceus* (LeConte, 1876)
Nomenclatural Authority: *Anderson (2002)*
Literature Records: none
Digitized Records: San Clemente (4 SBMNH), San Nicolas (11 SBMNH), Santa Rosa (2 SBMNH)
Range: Also known from mainland (*O'Brien & Wibmer, 1982*; *Anderson, 2002*).
Notes. Adults of this species are found under seaweed on sandy beaches (*Anderson, 2002*).

### Raymondionyminae
Notes. Three genera and five species of Raymondionyminae have been recorded from California (*O'Brien & Wibmer, 1982*).

### *Gilbertiola* Osella, 1982
Nomenclatural Authority: *Anderson (2002)*
Notes. This genus contains two described species from California (*Anderson, 2002*).

### *Gilbertiola* undetermined species
Literature Records: Santa Catalina (*Caterino & Chandler, 2010*: 191)
Digitized Records: Santa Catalina (1 SBMNH)
Notes. This species, represented by a single SBMNH specimen collected on Santa Catalina Island, was reported only as the genus *Gilbertiola* by *Caterino & Chandler (2010)*.

### Scolytinae
Notes. Two tribes, 49 genera, and 196 species of Scolytinae are known to occur in California (*Atkinson, 2021*; M. L. Gimmel, 2022, unpublished data). *Bright & Stark (1973)* reviewed the California fauna of Scolytinae, and *Wood (1982)* reviewed the entire North American fauna known at the time. *Atkinson (2021)* is an excellent online reference for staying up-to-date with North American scolytine nomenclature.

### Hylesinini
Notes. Sixteen genera and 69 species of Hylesinini are known to occur in California (*Atkinson, 2021*; M. L. Gimmel, 2022, unpublished data).

### *Carphobius* Blackman, 1943
Nomenclatural Authority: *Wood (1982)*
Notes. The genus *Carphobius* has not been previously reported from California (*Atkinson, 2021*), but is known from two species occurring from Arizona to Guatemala (*Wood, 1982*).

### *Carphobius* undescribed species

Literature Records: none

Digitized Records: San Clemente (1 SBMNH)

Range: Endemic (A. Cognato & S. Smith, 2022, unpublished data)

Notes. The San Clemente Island specimen cited above belongs to a new species that is currently under description (A. Cognato & S. Smith, 2022, personal communication).

### *Carphoborus* Eichhoff, 1864

Nomenclatural Authority: *Wood (1982)*

Notes. Nine species of this genus are reported from California (*Atkinson, 2021*).

### *Carphoborus declivis* Wood, 1954

Nomenclatural Authority: *Wood (1982)*

Literature Records: none

Digitized Records: Santa Rosa (1 SBMNH)

Range: Also known from mainland (*Wood, 1982*).

### *Dendroctonus* Erichson, 1836

Nomenclatural Authority: *Wood (1982)*

Notes. Six species of this genus are reported to occur in California (*Atkinson, 2021*).

### *Dendroctonus valens* LeConte, 1859

Nomenclatural Authority: *Atkinson (2021)*

Literature Records: none

Digitized Records: Santa Cruz (5 SBMNH; 1 UCSB)

Range: Also known from mainland (*Wood, 1982*).

### Scolytini

Notes. Thirty-three genera and 127 species of Scolytini are known to occur in California (*Atkinson, 2021*; M. L. Gimmel, 2022, unpublished data).

### *Coccotrypes* Eichhoff, 1878

Nomenclatural Authority: *Wood (1982)*

Notes. Three adventive species of *Coccotrypes* have been recorded from California (*Atkinson, 2021*).

### *Coccotrypes dactyliperda* (Fabricius, 1801)

Nomenclatural Authority: *Atkinson (2021)*

Literature Records: Santa Catalina (*Holzman et al., 2009*: 1078)

Digitized Records: none

Range: Also known from mainland (*Wood, 1982*).

Notes. This species was introduced to North America (*Wood, 1982*).

### *Dendrocranulus* Schedl, 1937

Nomenclatural Authority: *Wood (1982)*

Notes. One species of *Dendrocranulus* has been recorded from California (*Atkinson, 2021*).

**Dendrocranulus cucurbitae** (LeConte, 1879)

Nomenclatural Authority: *Atkinson (2021)*

Literature Records: Santa Cruz (*Naughton et al., 2014*: 303)

Digitized Records: Anacapa (2 SBMNH), San Clemente (3 SBMNH), Santa Catalina (5 SBMNH), Santa Cruz (7 SBMNH), Santa Rosa (10 SBMNH)

Range: Also known from mainland (*Wood, 1982*).

Notes. This species was reported by *Naughton et al. (2014)* as *Dendrocranulus californicus* (Hopkins, 1915), now considered a junior synonym of *D. cucurbitae* (*Wood, 1982*; *Atkinson, 2021*).

**Gnathotrichus** Eichhoff, 1869

Nomenclatural Authority: *Wood (1982)*

Notes. This genus is represented in California by five species (*Atkinson, 2021*).

**Gnathotrichus pilosus** (LeConte, 1868)

Nomenclatural Authority: *Atkinson (2021)*

Literature Records: none

Digitized Records: Santa Cruz (4 SBMNH)

Range: Also known from mainland (*Wood, 1982*).

**Hypothenemus** Westwood, 1836

Nomenclatural Authority: *Wood (1982)*

Notes. Only three species of this large genus have been recorded from California (*Atkinson, 2021*).

**Hypothenemus eruditus** Westwood, 1836

Nomenclatural Authority: *Atkinson (2021)*

Literature Records: none

Digitized Records: San Clemente (1 SBMNH), Santa Catalina (1 SBMNH), Santa Rosa (11 SBMNH)

Range: Also known from mainland (*Wood, 1982*).

**Ips** DeGeer, 1775

Nomenclatural Authority: *Wood (1982)*

Notes. *Lanier (1970*: 1418) hypothesized that *Ips plastographus maritimus* Lanier, 1970 probably occurred on pines "on the islands off the coast of southern California and Mexico", but this species has not yet been recorded from the Channel Islands.

**Ips paraconfusus** Lanier, 1970

Nomenclatural Authority: *Atkinson (2021)*

Literature Records: Santa Cruz (*Bright & Stark, 1973*: 92 [map])

Digitized Records: Santa Cruz (1 SBMNH)

Range: Also known from mainland (*Bright & Stark, 1973*; *Wood, 1982*).

**Monarthrum** Kirsch, 1866

Nomenclatural Authority: *Wood (1982)*

Notes. Three species of this genus are known to occur in California (*Atkinson, 2021*).

**Monarthrum scutellare (LeConte, 1857)**
Nomenclatural Authority: *Atkinson (2021)*
Literature Records: none
Digitized Records: Santa Cruz (6 SBMNH)
Range: Also known from mainland (*Wood, 1982*).

**Pityophthorus Eichhoff, 1864**
Nomenclatural Authority: *Wood (1982)*
Notes. Thirty-three species of *Pityophthorus* are known to occur in California (*Atkinson, 2021*).

**Pityophthorus carmeli Swaine, 1918**
Nomenclatural Authority: *Atkinson (2021)*
Literature Records: none
Digitized Records: Santa Cruz (10 SBMNH), Santa Rosa (1 SBMNH)
Range: Also known from mainland (*Wood, 1982*).

**Procryphalus Hopkins, 1915**
Nomenclatural Authority: *Wood (1982)*
Notes. One species of this genus occurs in California (*Atkinson, 2021*).

**Procryphalus utahensis Hopkins, 1915**
Nomenclatural Authority: *Atkinson (2021)*
Literature Records: none
Digitized Records: San Miguel (1 SBMNH)
Range: Also known from mainland (*Wood, 1982*).

**Pseudips Cognato, 2000**
Nomenclatural Authority: *Cognato (2000)*
Notes. This genus contains two species occurring in California (*Atkinson, 2021*).

**Pseudips mexicanus (Hopkins, 1905)**
Nomenclatural Authority: *Atkinson (2021)*
Literature Records: Santa Cruz (*Bright & Stark, 1973*: 86)
Digitized Records: none
Range: Also known from mainland (*Bright & Stark, 1973*).
Notes. *Bright & Stark (1973)* and *Wood (1982)* reported this species as *Ips mexicanus*; however, *Cognato (2000)* placed this species in the new genus *Pseudips*.

**Pseudopityophthorus Swaine, 1918**
Nomenclatural Authority: *Wood (1982)*
Literature Records (genus-only): Santa Barbara (*Miller & Miller, 1985*: 132)

Notes. Three species of this genus are known to occur in California (*Atkinson, 2021*). *Miller & Miller (1985)* reported an unidentified species of this genus from Santa Barbara Island that was collected from *Eriophyllum* (Asteraceae).

**Pseudopityophthorus agrifoliae** Blackman, 1931
Nomenclatural Authority: *Atkinson (2021)*
Literature Records: none
Digitized Records: Santa Cruz (1 SBMNH)
Range: Also known from mainland (*Wood, 1982*).

**Pseudopityophthorus pubipennis** (LeConte, 1857)
Nomenclatural Authority: *Atkinson (2021)*
Literature Records: none
Digitized Records: Santa Cruz (1 SBMNH), Santa Rosa (1 SBMNH)
Range: Also known from mainland (*Wood, 1982*).

**Stenoclyptus** Blackman, 1943
Nomenclatural Authority: *Wood (1982)*, *Atkinson (2021)*
Notes. One species of this genus is known from California (*Atkinson, 2021*). The name of this genus was misspelled "*Stenocleptus*" in *Wood (1982)*.

**Stenoclyptus sulcatus** (Bruck, 1936)
Nomenclatural Authority: *Atkinson (2021)*
Literature Records: none
Digitized Records: Santa Rosa (11 SBMNH)
Range: Also known from mainland (*Wood, 1982*).

**Xyleborinus** Reitter, 1913
Nomenclatural Authority: *Wood (1982)*
Notes. One species of this genus is known from California (*Atkinson, 2021*).

**Xyleborinus saxesenii** (Ratzeburg, 1837)
Nomenclatural Authority: *Atkinson (2021)*
Literature Records: none
Digitized Records: Santa Cruz (9 SBMNH), Santa Rosa (1 SBMNH)
Range: Also known from mainland (*Wood, 1982*; *Rabaglia, Dole & Cognato, 2006*).
Notes. This species was probably introduced to North America from Asia (*Rabaglia, Dole & Cognato, 2006*).

## ACKNOWLEDGEMENTS

Norm Woodley (Hereford, AZ), Paul Johnson (South Dakota State University, Brookings, SD), Anthony Cognato and Sarah Smith (Michigan State University, East Lansing, MI), Robert Barney (West Virginia State University, Institute, WV), Hume Douglas (Agriculture and Agri-Food Canada, Ottawa), and Art Evans (Richmond, VA) assisted with IDs during the course of this project. Giar-Ann Kung (LACM) was extremely helpful

in providing access to and sharing information about LACM specimens. Doug Yanega (UCRC) helped with locating and confirming identifications for UCRC specimens, and Alexey Tishechkin (CDFA) helped with locating CDFA specimens. Kojun Kanda (USDA-ARS) helped with locating and identifying specimens in USNM. Al Newton (Field Museum of Natural History, Chicago, IL) sent critical pieces of literature. An unpublished checklist of CA Channel Islands Carabidae from Kip Will (University of California, Berkeley, CA) was valuable in highlighting digitized records not georeferenced prior to this project. We give a huge thanks to Scott Miller (USNM) for providing extensive notes on Channel Islands specimens and a thorough manuscript review. Kip Will and an anonymous reviewer also provided valuable comments that improved the manuscript.

### Funding

This project was supported by The Nature Conservancy grant number 02052021-14766 (Channel Islands Beetle Diversity & Distribution) to Matthew L. Gimmel and M. Andrew Johnston. The funders had no role in study design, data collection and analysis, decision to publish, or preparation of the manuscript.

### Grant Disclosures

The following grant information was disclosed by the authors:
The Nature Conservancy: 02052021-14766.

### Competing Interests

The authors declare that they have no competing interests.

### Author Contributions

- Matthew L. Gimmel conceived and designed the experiments, performed the experiments, analyzed the data, prepared figures and/or tables, authored or reviewed drafts of the article, and approved the final draft.
- M. Andrew Johnston conceived and designed the experiments, performed the experiments, analyzed the data, prepared figures and/or tables, authored or reviewed drafts of the article, and approved the final draft.
- Michael S. Caterino performed the experiments, authored or reviewed drafts of the article, and approved the final draft.

### Data Availability

The specimen dataset is available at Zenodo: Johnston, M. Andrew, & Gimmel, Matthew L. (2022). Coleoptera of the California Channel Islands—digitized specimen records (1.1) [Data set]. Zenodo. DOI 10.5281/zenodo.6795735.

The analytical scripts and figure generation code are available at Andrew Johnston. (2022). California Channel Islands Coleoptera—analyses and figure code (v1.0.0). Zenodo. DOI 10.5281/zenodo.7121512.

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
