# Peer review of "Annotated checklist of the beetles (Coleoptera) of the California Channel Islands"

_PeerJ, doi:10.7717/peerj.14793_

## Round 0.1 · original submission · Minor Revisions

This is an outstanding manuscript documenting beetle diversity in the CA channel islands. The additional island area and rarefaction analyses are nice additions to an important, preliminary checklist. The three reviewers are all very complimentary but also made worthwhile suggestions which I would ask that you address in your revised manuscript.

Reviewer 1 ·

Basic reporting

Line 213 “not include general biological information except in special cases and except as it relates”

Line 220 “anticipate that taxonomic experts will come to different and novel conclusions than historical” to “anticipate that taxonomic experts may come to conclusions different than those of historical”

Lines 238-239 “to generate both rarefaction curves and an estimate of actual species diversity given the observed data?. An annotated R script with raw data is available via Zenodo (Johnston 2022).”

Line 526 “Notes. This genus was widely known in the earlier literature as Bembidium. Fully 115”

Figure 1. It is pretty interesting that island area has a correlation to number of species so much higher than that of the distance to mainland!


I like the inclusion of Table 3. Certainly the number of taxa present on the islands made it difficult to keep up with the listings in the actual text, so I think the abbreviated Table 3 is a great addition.

Structure of the manuscript worked well, and I feel the authors did a good job of explaining their goals and methods in the introduction. Raw data was shared.

Experimental design

I believe the authors did a thorough job investigating and reporting on museum, digitized, and iNaturalist records for beetles of the Channel Islands.

Validity of the findings

Gimmel et al. clearly explained why they chose to include the records they did and exclude those they did not. I believe it would be straightforward for another researcher to replicate their search and reporting methods. The authors did not seek to include unconfirmed identifications to increase the taxa counts.

I am not a coleopterist so cannot venture a comment about the accuracy of these findings, but the authors did an excellent job explaining which records they chose to include. As they stated in the text, modern beetle taxonomists may have differing conclusions than those historically working on the islands, and I believe the authors did a good job of reporting where records are located so they can be confirmed by experts.

·

Basic reporting

This is a well written, well documented, compilation of the existing knowledge of beetles of the California Channel Islands, including many new identifications and clarifications of the status of various species. It is a major contribution to understanding the biodiversity of these islands, and indeed, the adjacent southern California mainland. Having done similar projects before, I know how much work goes into the basic documentation and identification checking for a faunal review like this.

Experimental design

The design of data gathering is clearly documented, as are the analyses they have conducted. For some future analysis (not suggesting for this one now), it would be interesting in the biogeographic analysis to take into account the very different geographical histories of the northern versus southern islands, e.g., that the northern islands were united in fairly recent geological time, thus making them historically one large island. Nested sets analysis or path analysis might allow a model that incorporates this?

Validity of the findings

The findings are clearly documented, including areas that need further research.

Additional comments

I have reviewed the manuscript against the bibliography that I have continued to somewhat haphazardly assemble on California Islands insects, as well as my compilation of endemic species. The manuscript is comprehensive! I congratulate the authors on a job well done.

I append a few additional literature citations below, but they do not materially change the compilation.

Table 3 is very hard to use as currently formatted. The published version needs a header on the top of each page to allow the columns to be clearly identified. It would also be useful to use an abbreviation rather than "yes" under duplicate, endemic or adventive (e.g., "end" rather than "yes").

Table 3 comments:

If Bembidion palosverdes is (presumed) extinct on the mainland, that does NOT make it endemic on the island.

Eusattus robustus -- the authors only record one specimen each from the four islands that were also questioned by Doyen. Given the species is common where it occurs, as shown by larger numbers of records from the other islands, I think it should not be considered established on San Miguel, Santa Barbara, Anacapa, and Santa Rosa Islands (as per the text).

Here are some additional records from USNM in the BOLD database, also available from the USNM collections search portal at https://collections.nmnh.si.edu/search/ento/ -- but nothing new.

Trichochrous CCDB-30443-D11 SICOA522-18
Trichochrous CCDB-30443-D12 SICOA523-18
Dermestes frischii CCDB-30443-E01 SICOA524-18
Dermestes frischii CCDB-30443-E02 SICOA525-18
Phobetus CCDB-32961-B09 SICOC211-18
Phobetus CCDB-32961-B10 SICOC212-18
Coenonycha clementina CCDB-32961-B11 SICOC213-18
Coenonycha fulva CCDB-32961-B12 SICOC214-18
Coenonycha santacruzae CCDB-32961-C01 SICOC215-18

Here are some additional literature citations if useful:

Berry, R. L. 1970. A revision of the tribes Cerenopini and Eulabini (formerly Scaurini, in part) of North America (Coleoptera: Tenebrionidae). University Microfilms 71-17958. Ohio State University, Columbus. Epantius obscurus LeConte 1851 from Santa Cruz, Anacapa, and Santa Catalina Is., Apsena grossa (LeConte) 1866 endemic to Santa Rosa, Anacapa, Santa Catalina, San Nicolas, San Clemente, and Santa Barbara Is., Apsena pubescens (LeConte) 1851 from Santa Catalina I. (Coleoptera: Tenebrionidae).

Blaisdell, F. E., Sr. 1924. Studies in the Melyridae (Coleoptera) number four. Transactions of the American Entomological Society 50(4):313-318. Descriptive comments on Listrus anacapensis Blaisdell 1924, "an insular species" (Coleoptera: Melyridae). (p. 316).

d'Orchymont, A. 1942. Revision des Laccobius Américans (Coleoptera Hydrophilinae Hydrobiini). Bulletin du Musée royal d'Histoire naturelle de Belgique 18(30):1-18. Laccobius ellipticus LeConte, 1855 from Santa Cruz I. (Coleoptera: Hydrophilidae) (p. 3).

Garcia-Paris, M., M. M. Coca-Abia, and G. Parra-Olea. 2006. Re-evaluation of the genera Phloeodes, Noserus and Nosoderma (Coleoptera: Zopheridae) with description of a new species of Nosoderma from northern Mexico. Ann. Soc. Ent. Fr. (n.s.) 42(2):215-230. Nosoderma diabolicum (LeConte) 1851 from Santa Cruz I. (Coleoptera: Zopheridae). See Foley and Ivie 2008.

Goeden, R. D., and D. W. Ricker. 1974. Imported seed weevils attack Italian and Milk Thistles in Southern California. California Agriculture 28(1):8-9. Reports introductions of Rhinocyllus conicus (Froelich), 1792 (Coleoptera: Curculionidae) to Santa Cruz I. in 1971 and 1972 for biological control of Silybum marianum (L.) Gaertn. (Asteraceae). See also Goeden and Ricker, 1977; Hawkes et al., 1972.

Goeden, R. D., and D. W. Ricker. 1977. Establishment of Rhinocyllus conicus on Milk Thistle in Southern California. Weed Science 25:288-292. Reports introductions of Rhinocyllus conicus (Froelich), 1792 (Coleoptera: Curculionidae) to Santa Cruz I. in 1971 and 1972 for biological control of Silybum marianum (L.) Gaertn. (Asteraceae). See also Goeden and Ricker, 1974; Hawkes et al., 1972.

Graves, R. C. 1988. Geographic distribution of the North American tiger beetle Cicindela hirticollis Say. Cicindela 20(1):1-21. Cicindela hirticollis Say from Santa Rosa and Santa Catalina Is. [following Nagano, 1982] (Coleoptera: Carabidae) (p. 4).

Greenwalt, L. A. 1977. Determination that seven Californa Channel Island animals and plants are either endangered species or threatened species. Federal Register 42(155):40682-40685. Available data do not support endangered or threatened status for Coenonycha clementina from San Clemente I. (Coleoptera: Scarabaeidae).

Hamilton, R. W. 1969. Studies of the Rhynchophorous families Nemonychidae, Attelabidae, and Rhynchitidae, with a revision of North American species of Attelabus Linnaeus, Rhynchites Schneider and Eugnamptus Schoenherr (Coleoptera: Curculionidae). Unpublished Ph.D. thesis. 492 pp. Ohio State University. Pselaphorhynchites insularis (Fall), 1929, from Santa Catalina I. (TL) and mainland (p. 209); Deporaus glastinus (LeConte), 1857, from Santa Cruz I. (p. 394) (Coleoptera: Rhynchitidae).

Hawkes, R. B., L. A. Andres, and P. H. Dunn. 1972. Seed weevil released to control Milk Thistle. California Agriculture 26(12):14. Reports introductions of Rhinocyllus conicus (Froelich), 1792 (Coleoptera: Curculionidae) to Santa Cruz I. in 1971 and 1972 for biological control of Silybum marianum (L.) Gaertn. (Asteraceae). See also Goeden and Ricker, 1974, 1977.

Holzman, J. P., A. J. Bohonak, L. R. Kirkendall, D. Gottlieb, A. R. Harari, and S. T. Kelley. 2009. Inbreeding variability and population structure in the invasive haplodiploid palm-seed borer (Coccotrypes dactyliperda). Journal of Evolutionary Biology 22:1076-1087. Coccotrypes dactyliperda Fabricius from Avalon, Santa Catalina I. (Coleoptera: Curculionidae: Scolytinae).

Hopping, G. R. 1932. A revision of the Clytini of Boreal America (Cerambycidae, Coleoptera). Part I. Annals of the Entomological Society of America 25(3):529-577. Xylotrechus insignis LeConte 1873 from Santa Catalina I. [p. 542]

Huether, J. P., and M. K. Huether. 2015. Distributional and biological notes on recently collected North American Meloidae (Coleoptera). Pan-Pacific Entomologist 91(2):157-167. Meloe barbarus LeConte, 1861, recorded from San Clemente, Santa Barbara, and Santa Catalina Is. (Coleoptera: Meloidae).

Liebherr, J. K., and E. C. Zimmerman. 2000. Hawaiian Carabidae (Coleoptera), Part 1: Introduction and Tribe Platynini. Insects of Hawaii 16:1-494. Mentions Channel Islands populations of Calathus ruficollis ruficollis Dejean (Coleoptera: Carabidae) (page 466).

Linell, M. L. 1898. New species of Coleoptera of the family Chrysomelidae, with a short review of the tribe Chylamydini. Proceedings of the United States National Museum 20(1130):473-485. Redescribes Colaspidea subvittata Fall, 1897, from Santa Catalina I. See Schultz, 1970; Riley et al., 2003. (Coleoptera: Chrysomelidae). (p. 481)

Linsley, E. G. 1940. The origin and distribution of the Cerambycidae of North America, with special reference to the fauna of the Pacific Slope. Proc. Sixth Pac. Sci. Cong. 4:269-282. Ipochus fasciatus LeConte, 1852, Atimia dorsalis LeConte, 1869, Xylotrechus insignis LeConte, 1873, and Romaleum hispicorne Linnaeus, 1767 [now Enaphalodes hispicornis (Linnaeus), 1767] from unstated "southern California islands" (p. 272 and fig. 1). Suggests last three are "recent immigrants". See Linsley, 1958. (Coleoptera: Cerambycidae).

Linsley, E. G. 1958. Geographical origins and phylogenetic affinities of the cerambycid beetle fauna of western North America. Pages 299-320 in C. L. Hubbs, editor. Zoogeography. Amer. Assoc. Advan. Sci., Washington, D.C. Ipochus fasciatus LeConte, 1852, Atimia maritima Linsley, 1939 [now Atimia confusa maritima Linsley, 1939], Xylotrechus insignis LeConte, 1873, and Romaleum hispicorne Linnaeus, 1767 from unstated "southern California Islands". Suggests last two are "recent immigrants". See Linsley, 1940. (Coleoptera: Cerambycidae).

Miller, S. E. 1996. A history of entomological exploration on Santa Rosa Island. Pages 193-200 in K. B. Allen, editor. Island of the cowboys: Santa Rosa Island. Santa Cruz Island Foundation, Santa Barbara, California. Reviews the study of arthropods on Santa Rosa I. from 1872 to 1995. Includes the first published record of Rhinocyllus conicus (Froelich), 1792 (Coleoptera: Curculionidae) introduced to Santa Rosa I. in 1979 for biological control of Silybum marianum (L.) Gaertn. and Carduus pycnocephalus L. (Asteraceae).

Pope, R. D. 1981. 'Rhyzobius ventralis' (Coleoptera: Coccinellidae), its constituent species, and their taxonomy and historical roles in biological control. Bull. Ent. Res. 71:19-31. Rhyzobius forestieri (Mulsant), 1853 from Santa Catalina I. (as "Avalon"); introduced from Australia (p. 27).

Short, A. E. Z., and M. S. Caterino. 2009. On the validity of habitat as a predictor of genetic structure in aquatic systems: a comparative study using California water beetles. Molecular Ecology 18(3):403-414. COI haplotype diversity, phylogeographical concordance, population connectivity and population isolation of lotic water beetle species Anacaena signaticollis Fall 1924 from Santa Cruz I. (Coleoptera: Hydrophilidae) and Stictotarsus striatellus LeConte 1852 from Santa Rosa and Santa Cruz Is. (Coleoptera: Dytiscidae).

Smith, H. S., and E. J. Branigan. 1916. On Delphastus catalinae, a valuable ladybird enemy of the white flies. California Department of Agriculture Monthly Bulletin 5:448-450. First mainland record of Delphastus catalinae (Horn), 1895, originally described from Santa Catalina I. (Coleoptera: Coccinellidae).

Tschinkel, W. R., and J. T. Doyen. 1976. Sound production by substratal tapping in beetles of the genus Eusattus (Tentyriidae: Coniontini). Coleopterists Bulletin 30(4):331-335. Behavior of Eusattus robustus LeConte from San Clemente I. (Coleoptera: Tenebrionidae).

·

Basic reporting

The manuscript is informative, well-structured, clearly written, and easy to follow and use. I found only one sentence I thought could be better worded (marked in PDF). The literature coverage is excellent.

Experimental design

The approach is well-thought-out and described. The linkages to the data sources and specimens are very well documented. The purpose of the paper is fully explained.

Validity of the findings

The authors are very realistic about the preliminary nature of the list and do not over extend hypotheses based on what data they have. The analyses presented are descriptive and probably don't tell us too much about the factors leading to the current beetle diversity on the islands. Some discussion of the historical and contemporary land use of the islands (which varies massively island to island) might be appropriate here.

Additional comments

This is a very important contribution to understanding the beetle fauna of California. While it is a way to go from this to an aspiration goal of complete knowledge of the Islands' beetle fauna, it is a big, important step in that direction.

There are some very minor suggestions marked in the PDF, mostly in carabids.

---

## Round 0.2 · accepted · Accept

Thank you so much for carefully addressing all of the reviewers' comments in your revised manuscript. I have assessed the manuscript and the responses to the reviews and am happy to recommend the manuscript for publication.